# Non-Asymptotic Analysis Of Data Augmentation For Precision Matrix Estimation

**Lucas Morisset**$^*$
Qube Research and Technologies
& École Polytechnique

**Adrien Hardy**
Qube Research and Technologies

**Alain Durmus**
École Polytechnique

## Abstract

This paper addresses the problem of inverse covariance (also known as precision matrix) estimation in high-dimensional settings. Specifically, we focus on two classes of estimators: linear shrinkage estimators with a target proportional to the identity matrix, and estimators derived from data augmentation (DA). Here, DA refers to the common practice of enriching a dataset with artificial samples—typically generated via a generative model or through random transformations of the original data—prior to model fitting. For both classes of estimators, we derive estimators and provide concentration bounds for their quadratic error. This allows for both method comparison and hyperparameter tuning, such as selecting the optimal proportion of artificial samples. On the technical side, our analysis relies on tools from random matrix theory. We introduce a novel deterministic equivalent for generalized resolvent matrices, accommodating dependent samples with specific structure. We support our theoretical results with numerical experiments.

## 1 Introduction

In this work, we consider the problem of estimating the inverse covariance matrix, also known as the precision matrix, of a random vector from i.i.d. zero-mean samples $[X_1, \cdots, X_n] \in \mathbb{R}^{d \times n}$ with true covariance $\Sigma_X = \mathbb{E}\left[X_1 X_1^\top\right]$. Here, $n$ denotes the number of samples, and $d$ is the dimensionality of the data. This problem has important applications in statistics and signal processing (see, e.g., [FLL16, Car88]).

We are particularly interested in the high-dimensional regimes, where the data dimension $d$ and the number of samples $n$ are of the same order. In this setting, the sample covariance matrix $C_X = n^{-1} X X^\top$ may be non-invertible or poorly-conditionned. As a result, using its inverse as an estimator can lead to numerical instability and high estimation error. To address this issue, shrinkage estimators for the covariance matrix have been proposed as a regularization method [BGP16, LW04, SS05, LSW03], which involve adding a target matrix to $C_X$. The simplest and most common choice for the target is a multiple of the identity matrix, which effectively shifts the eigenvalues above a threshold $\lambda > 0$, improving stability. In addition to linear shrinkage and even more importantly, this paper also explores the use of data augmentation (DA) as an alternative strategy.

Data augmentation (DA) involves increasing the size of a dataset by incorporating additional artificial samples. The underlying intuition is that, in many cases, it is possible to artificially replicate the

---

$^*$Correspondence to: lucas.morisset@polytechnique.edu.

39th Conference on Neural Information Processing Systems (NeurIPS 2025).

data distribution, thereby reducing the variance of the model while maintaining relatively low bias. Due to its effectiveness in low-data regimes and its ability to mitigate overfitting, DA has become increasingly popular and is now widely used in machine learning and data science [SK19, GSK18, CKNH20, GSA+20]. It finds applications across a variety of fields, including computer vision [SK19], natural language processing [FGW+21], and neuroscience [LLM20].

Two main types of data augmentation (DA) can be distinguished. The first is Transformative Data Augmentation (TDA), where original samples are transformed through a random mapping—for example, by adding Gaussian noise or applying a random mask. The second is Generative Data Augmentation (GDA), in which artificial samples are generated using a generative model and added to the training dataset. In the case of GDA, we assume that the generative model has been pre-trained on the original samples $X$.

In both cases, the artificial samples are dependent on the original data. Although TDA and GDA differ conceptually, our framework and results encompass both approaches. More precisely, we consider the inverse of the empirical covariance matrix associated with the augmented dataset $\tilde{X} = [X_1, \ldots, X_n, G_1, \ldots, G_m]$ as an estimator of $\Sigma_X^{-1}$, where each $G_i \in \mathbb{R}^d$ is an artificial sample. That is, we study the estimator which consists of the inverse of $(n+m)^{-1}\tilde{X}\tilde{X}^\top + \lambda \, \mathrm{I}_d$, for some regularization parameter $\lambda > 0$.

Although there is extensive empirical evidence that DA improves the performance of machine learning models, the theoretical literature on the subject remains relatively limited. In this work, our goal is to establish performance guarantees that enable meaningful comparisons between different DA strategies and, as a by-product, allow for the optimization of certain associated hyperparameters. To this end, we leverage tools from random matrix theory to construct estimators of the quadratic error for DA-based estimators, which, under mild assumptions, satisfy exponential concentration inequalities. Our analysis can also be adapted—and in fact simplifies—to cover linear shrinkage estimators with a target proportional to the identity matrix. To summarize, our main contributions are as follows:

● In Section 2, we focus on the estimator of $\Sigma_X^{-1}$ given by the inverse of a linear shrinkage estimator, where the shrinkage target is a scalar multiple of the identity matrix. Specifically, we derive estimators for the quadratic error and show that they satisfy non-asymptotic exponential concentration bounds. Our results hold under standard assumptions from random matrix theory—namely, the Lipschitz concentration property for $X$.

● In Section 3, we extend our analysis to data-augmented estimators under appropriate conditions on the DA procedure, which we show hold true for common DA.

● Finally, for both scenarios, we show how our estimators for the quadratic error can be used to compare and tune their corresponding methods with respect to key hyper-parameters such as the number of additional samples $m$ for DA. These conclusions are illustrated numerically on real data in Section 4.

**Notation and convention.**    Motivated by high-dimensional statistics, we will consider that $n$, $d$ and $m$ are variables, yet for notation simplicity, we will most often not reflect dependancies in $n$, $m$ or $d$ in our notations. Additionally, we write, $x \lesssim y$ (resp $x \gtrsim y$) whenever $x \leq Cy$ (resp $x \geq Cy$) for a universal constant $C$ that neither depends on the model's parameters, nor on the parameters $n, m, d$. For any matrix $\mathbf{H} \in \mathbb{R}^{d \times k}$, we denoted by $C_{\mathbf{H}} = k^{-1}\mathbf{H}\mathbf{H}^\top$ the corresponding covariance matrix. For any symmetric matrix $\Sigma \in \mathbb{R}^{d \times d}$, we denote by $\lambda_d(\Sigma) \leq \ldots \leq \lambda_1(\Sigma)$ its eigenvalues. The Frobenius and operator norms are denoted $\|\cdot\|_{\mathrm{F}}$ and $\|\cdot\|_{\mathrm{op}}$ respectively. Random variables will be referred to by capital letters $X$, $G$, $Z$, and we will denote by $\mathrm{Ber}(p)$, $\mathrm{N}(\mathbf{m}, \boldsymbol{\Sigma})$ and $\mathrm{Unif}(\mathsf{E})$ the Bernoulli distribution of parameter $p \in [0, 1]$, the Gaussian distribution of mean $\mathbf{m} \in \mathbb{R}^d$ and covariance $\boldsymbol{\Sigma} \in \mathbb{R}^{d \times d}$, and the uniform distribution over a discrete set $\mathsf{E}$. Furthermore, we introduce the $p$-Wasserstein metric, $W_p^p(\nu_1, \nu_2) = \inf_{\gamma \in \Gamma(\nu_1, \nu_2)} \int \|x - y\|_{\mathrm{F}}^p \, \mathrm{d}\gamma(x, y)$ for any distributions $\nu_1$ and $\nu_2$ on $\mathbb{R}^{d \times k}$, and where $\gamma \in \Gamma(\nu_1, \nu_2)$ if and only if for any $\mathsf{E} \in \mathbb{R}^{d \times k}$, $\gamma(\mathbb{R}^{d \times n}, \mathsf{E}) = \nu_1(\mathsf{E})$, and $\gamma(\mathsf{E}, \mathbb{R}^{d \times n}) = \nu_2(\mathsf{E})$.

## 1.1 Related work

**Data Augmentation.** Numerous empirical studies have demonstrated the benefits of using DA when training machine learning models [MM22b, MMN22, vDM01]. Among popular DA schemes, we can mention AutoAugment [CZM$^+$19, LKK$^+$19, ZWZZ20], which aims to learn an optimal augmentation policy from data by combining a set of sub-policies. In addition, significant works have also explored the incorporation of knowledge about the distribution invariances directly into the training procedures [CDL20].

However, DA does not always lead to a systematic improvement in test error [KIB$^+$23, HGK20, CL25], and very little theoretical understanding backs-up the improvement observed empirically. An early analysis by [Bis95] showed that adding Gaussian noise to data points is equivalent to applying Tikhonov regularization. Building on this seminal work—and given the practical importance of DA for machine learning practitioners—a few studies have sought to develop a theoretical understanding of its effects.

In the context of kernel methods, [DGR$^+$19] showed that DA can be approximated by a combination of first-order feature averaging and second-order variance regularization. Similarly, in the context of linear and logistic regressions, [LKDM22] revealed that DA induces implicit spectral regularization in two ways: first, by adjusting the relative proportions of the eigenvalues of the data covariance matrix in a training-dependent way; and second, by uniformly shifting the entire spectrum via ridge regression.

Taking a different perspective, [WZVR20] consider a family of linear transformations and study their effects on the ridge estimator in an over-parametrized linear regression setting. First, they show that transformations that preserve the labels of the data can improve estimation by increasing the span of the training data. Second, they show that transformations that mix data can improve estimation by playing a regularization effect. They proposed an augmentation scheme that searches over the linear span of a set of transformations, aiming to maximize model uncertainty on the transformed data.

Recently, studies have shown that even small amounts of artificial data can lead to *model collapse* [SSZ$^+$24, DFK24, DFSK25], a phenomenon where the performance of generative models deteriorates when recursively trained on synthetic data.

**Precision Matrix Estimation.** In a high-dimensional settings—where the number of covariates is comparable to or exceeds the number of observations—traditional covariance estimation methods often suffer from poor conditioning, making the estimation of the precision matrix particularly challenging. To address this, [LW04] introduced linear shrinkage methods, which involve forming a convex combination $\varpi C_X + (1 - \varpi)\mathbf{T}$ of the sample covariance matrix $C_X$ with a shrinkage target $\mathbf{T}$, where $\varpi \in [0, 1]$. [LW03, LW04] derived the optimal value of $\varpi$ that minimizes the mean squared error between the shrinkage estimator and the true covariance matrix $\Sigma_X$.

Extending this work, [LW12, LW22, BGBP23] proposed and analyzed non-linear shrinkage estimators of the form $U f(D)U^\top$, where $C_X = UDU^\top$ is the eigenvalue decomposition of $C_X$ and $f$ is a suitably chosen function applied to the eigenvalues.

Fewer works have addressed shrinkage methods specifically designed for precision matrix estimation. Among them, [BGP16] studied estimators of the form $(1 - \varpi)C_X^{-1} + \varpi\Pi$, where $\Pi$ is a deterministic shrinkage target, and derived the optimal shrinkage intensity $\varpi$. In addition, [WPTZ15] considered estimators of the form $\Omega_X(\varpi) = ((1 - \varpi)C_X + \varpi \operatorname{I}_d)^{-1}$ and derived the optimal $\varpi$ to minimize the objective function $\|\Omega_X(\varpi)\Sigma_X - \operatorname{I}_d\|_{\mathrm{F}}$ in the high-dimensional regime where $d/n \to \gamma > 0$.

An independent line of research, motivated by Gaussian graphical models [YL07], has focused on estimating sparse precision matrices. In particular, [MH12, CLL11] introduced the Graphical Lasso method, which has become widely used for this purpose.

**Random Matrix Theory.** Since the pioneering work of [Wis28], numerous studies have investigated the behavior of the eigenvalues and eigenvectors of the sample covariance matrix $C_X$ in the high-dimensional regime where $d/n \to \gamma > 0$; see, for example, [MP67, Sil89]. More recently, [AEK$^+$14] first demonstrated that, in the isotropic setting, the resolvent $(C_X + \lambda \operatorname{I}_d)^{-1}$ converges weakly to a scalar multiple of the identity matrix as $d/n \to \gamma > 0$. This was later extended to the anisotropic case by [KY17], who established so-called deterministic equivalent results for $(C_X + \lambda \operatorname{I}_d)^{-1}$. These results have been further generalized to settings with more complex dependency structures. In

particular, [Cho22, LC23] showed that deterministic equivalents continue to hold under weaker assumptions.

Building on these foundational results, several studies have established connections between classical random matrix theory—particularly the results of [MP67, Sil89]—and the behavior of modern machine learning models. In particular, random matrix theory has proven instrumental in explaining the *double-descent* phenomenon, initially observed in linear models [HMRT22, DLM20, MVSS20, BLLT20, DKT21], and later extended to certain classes of shallow models, such as random feature models [MM22a, LCM21, GLK$^+$21, DRBK20]. Complementing these works, [SDCL24, SCDL24] provided a sharp asymptotic characterization of the test error in deep random feature models—representing a significant step toward understanding generalization in deeper architectures. Finally, [ITL$^+$24] leveraged random matrix theory to develop precise performance estimates for multi-task learning across a variety of statistical models.

## 2  Inverse covariance estimation using shrinkage method

We consider here the following estimator $R_X(\lambda)$ of the precision matrix, and its squared error:

$$R_X(\lambda) = (C_X + \lambda\,I_d)^{-1}\,, \qquad \mathcal{E}_X(\lambda) := \frac{1}{d}\|R_X(\lambda) - \Sigma_X^{-1}\|_F^2\,. \qquad (1)$$

Here, $\lambda > 0$ is a hyperparameter that controls the strength of the regularization. Note that this estimator is not per se a shrinkage estimator, but it is the inverse of a shrinkage estimator of the covariance matrix. In addition, $R_X(\lambda)$ is also referred to as the diagonal loading estimator in the signal processing community; see e.g., [LSW03]. Furthermore, note that our result can be readily applied to an estimators of the form $((1-\alpha)C_X + \alpha\sigma\,I_d)^{-1} = (1-\alpha)^{-1}(C_X + \alpha\sigma\,I_d/(1-\alpha))^{-1}$, for any $\sigma \in \mathbb{R}_+$ and $\alpha \in (0,1)$.

This estimator does not rely on any data augmentation procedure. However, as we will see, applying data augmentation leads to results that are closely related to this regularization approach. This is not surprising, as it is relatively well known that data augmentation induces an implicit regularization effect [Bis95, LKDM22]. For this reason, we present this simple case in detail as a preliminary step, which will allow us to transition more smoothly to the data-augmented case in Section 3. As already emphasized in the introduction, our main goal is to derive a data-centric estimate for the error $\mathcal{E}_X(\lambda)$. To this end, we introduce two assumptions.

**H1 (Concentration of $X$).** *The random matrix $X \in \mathbb{R}^{d\times n}$ writes $\Sigma^{1/2}Z$, where $Z \in \mathbb{R}^{d\times n}$ has independant sub-Gaussian entries with parameter $\sigma_X$.*

**H**1 is standard in the random matrix theory literature, yet one can employ a more general framework, as in [Cho22, ITL$^+$24, LC18] by introducing the notion of Lipschitz concentration (**H**6), we detail this generalization throughout the appendix and stick to this simpler framework in the main body for the sake of simplicity. In addition, we expect our results to remain valid under a finite-moment assumption on the entries of $X$, albeit with weaker—typically polynomial—concentration bounds, as done in [LP11]. We leave a detailed investigation of this extension for future work.

We next suppose that with high probability the leave one-out covariance matrix $C_X^-$ is well-conditionned. This matrix is defined for any $\mathbf{X} \in \mathbb{R}^{d\times n}$ as the covariance matrix

$$C_{\mathbf{X}}^- = C_{\mathbf{X}^-}\,, \qquad \mathbf{X}^- = [0, \mathbf{X}_2, \ldots, \mathbf{X}_n]\,.$$

More formally, for any $\eta > 0$ define

$$\mathsf{A}_\eta = \{\mathbf{X} \in \mathbb{R}^{d\times n} : \lambda_d(C_{\mathbf{X}}^-) \geq \eta\}\,. \qquad (2)$$

**H2 (Model conditionning).** *There exist $\eta > 0$ and $c_X > 0$ such that $\mathbb{P}\left(X \notin \mathsf{A}_\eta\right) \lesssim e^{-c_X n}$.*

We highlight in Appendix A that **H**2 holds provided **H**1 holds and $n \geq K_X(d + \eta + 1)$ for some constant $K_X$ depending only on $\sigma_X$.

We are now ready to present our estimator for $\mathcal{E}_X(\lambda)$ and to state its concentration properties. The estimator is given by:

$$\hat{\mathcal{E}}_X(\lambda) := \frac{1}{d}\left(\operatorname{tr}\left(\mathrm{R}_X(\lambda)^2\right) - \frac{2(1-d/n)\operatorname{tr}\left(\mathrm{R}_X(0)\right)}{\lambda}\mathbb{1}_{\mathsf{A}_\eta}(X) + \frac{2tr\left(\mathrm{R}_X(\lambda)\right)}{\lambda\mathfrak{b}(\lambda)} + \operatorname{tr}\left(\Sigma_X^{-2}\right)\right) \quad (3)$$

$$\mathfrak{b}(\lambda) := \frac{1}{1 - d/n + (\lambda/n)\operatorname{tr}\left(\mathrm{R}_X(\lambda)\right)} \ .$$

Note that for a fixed $\eta > 0$, $\hat{\mathcal{E}}_X(\lambda)$ is computable from the data $X$ only, up to an additive constant.

**Theorem 1.** *Assume **H1** and **H2**. Then, it holds for all $t \geq 0$ and $\lambda > 0$,*

$$\mathbb{P}\left(\left|\mathcal{E}_X(\lambda) - \hat{\mathcal{E}}_X(\lambda)\right| \geq t + \Delta_X(\lambda)\right) \lesssim \exp\left(-c\lambda_d(\Sigma_X)^2\sigma_X^2 nd\eta^3 t^2\right)$$

*for a universal constant $c > 0$ and where*

$$\Delta_X(\lambda) := \frac{C_1\sigma_X^2\sqrt{d}\|\Sigma_X\|_{\mathrm{op}}^3}{n\lambda_d(\Sigma_X)\eta^6} + C_2\mathrm{e}^{-c_X n} + \frac{1}{\lambda^3 nd} \ .$$

*Here $C_1, C_2 > 0$ are explicit polynomial functions of $\|\Sigma_X\|_{\mathrm{op}}^{-1}$, $\lambda_d(\Sigma_X)$, $(\eta + \lambda)$ and $c_X^{-1}$, see* (B).

From a practical standpoint, the previous result can be used to optimize the hyperparameter $\lambda$ by minimizing the function $\lambda \mapsto \mathcal{E}_X(\lambda)$, using $\hat{\mathcal{E}}_X$ as a proxy. Moreover, it is worth noting that the derivative of $\lambda \mapsto \hat{\mathcal{E}}_X$ depends only on the data matrix $X$, and not on the true covariance $\Sigma_X$. As a result, $\hat{\mathcal{E}}_X$ can be minimized using a gradient descent scheme, provided that the parameter $\eta > 0$ satisfies **H2**. We illustrate these applications on real data in Section 4.

Although the full proof of Theorem 1 is deferred to Appendix B, we provide here a sketch of its derivation. We highlight the main ideas and technical challenges, and note that the proof of our result on estimation using data augmentation, Theorem 2, shares several of these steps.

First, expanding the Frobenius norm in (1), we have,

$$\mathcal{E}_X(\lambda) = (1/d)\operatorname{tr}\left(\mathrm{R}_X(\lambda)^2\right) - (2/d)\operatorname{tr}\left(\Sigma_X^{-1}\mathrm{R}_X(\lambda)\right) + (1/d)\operatorname{tr}\left(\Sigma_X^{-2}\right) \ .$$

The first term in the previous expansion is directly computable from the data matrix $X$, while the last term is constant with respect to $\lambda$ and can be ignored when the goal is to optimize $\lambda$. Therefore, it suffices to establish a deterministic equivalent for $\mathrm{R}_X(\lambda)$ and thus of $\operatorname{tr}\left(\Sigma_X^{-1}\mathrm{R}_X(\lambda)\right)$. To this end, we rely on the following result, whose proof is postponed to appendix B.

**Proposition 1.** *Assume **H1** and **H2**. Let $\mathbf{B} \in \mathbb{R}^{d \times d}$ be a deterministic matrix, then we have for all $\lambda \geq 0$,*

$$\mathbb{P}\left(\left|\frac{1}{d}\operatorname{tr}\left(\mathbf{B}\left\{\mathrm{R}_X(\lambda)\mathbb{1}_{\mathsf{A}_\eta}(X) - \mathbb{E}\left[\mathrm{R}_X(\lambda)\mathbb{1}_{\mathsf{A}_\eta}(X)\right]\right\}\right)\right| \geq t\right) \lesssim \exp\left(-c(\eta + \lambda)^3\sigma_X^2 ndt^2\right) \ .$$

*Furthermore, defining $f_\lambda(\mathfrak{b}) = 1 + n^{-1}\operatorname{tr}\left(\Sigma_X(\Sigma_X/\mathfrak{b} + \lambda\mathrm{I}_d)^{-1}\right)$ and $\mathfrak{b}^* := \mathfrak{b}^*(\lambda)$ as the unique fixed point of $f_\lambda$ on $[1, \infty)$, we have*

$$\left\|\mathbb{E}\left[\left\{\mathrm{R}_X(\lambda) - \bar{\mathrm{R}}_X^{\mathfrak{b}^*}(\lambda)\right\}\mathbb{1}_{\mathsf{A}_\eta}(X)\right]\right\|_{\mathrm{F}} \lesssim \frac{C_1\sigma_X^2\sqrt{d}\|\Sigma_X\|_{\mathrm{op}}^3}{n\lambda_1(\Sigma_X)(\eta + \lambda)^6} + C_2\mathrm{e}^{-c_X n} \ ,$$

$$\bar{\mathrm{R}}_X^{\mathfrak{b}^*}(\lambda) = \left(\frac{\Sigma_X}{\mathfrak{b}^*} + \lambda\mathrm{I}_d\right)^{-1} \ ,$$

*where $C_1, C_2$ are defined as in Theorem 1.*

Proposition 1 shows that $\mathrm{R}_X(\lambda)$ concentrates around $\bar{\mathrm{R}}_X^{\mathfrak{b}^*}(\lambda)$, which we refer to as a deterministic equivalent of $\mathrm{R}_X(\lambda)$. Our result extends that of [Cho22] by covering the case of vanishing regularization ($\lambda \to 0$). This extension constitutes the main technical innovation required for the proof of Theorem 1.

We can now leverage the deterministic equivalent of Proposition 1 to rewrite $(1/d)\operatorname{tr}\left(\Sigma_X^{-1}\operatorname{R}_X(\lambda)\right)$. Informally, it holds with high probability that

$$\frac{1}{d}\operatorname{tr}\left(\Sigma_X^{-1}\operatorname{R}_X(\lambda)\right) \approx \frac{1}{d}\operatorname{tr}\left(\Sigma_X^{-1}\left(\frac{\Sigma_X}{\mathfrak{b}^*(\lambda)}+\lambda\operatorname{I}_d\right)^{-1}\right)\mathbb{1}_{\mathsf{A}_\eta}(X)$$

$$= \frac{1}{\lambda d\mathfrak{b}^*(0)}\operatorname{tr}\left(\bar{\operatorname{R}}_X^{\mathfrak{b}^*(0)}(0)\right)\mathbb{1}_{\mathsf{A}_\eta}(X) - \frac{1}{\lambda d\mathfrak{b}^*(\lambda)}\operatorname{tr}\left(\bar{\operatorname{R}}_X^{\mathfrak{b}^*(\lambda)}(\lambda)\right)\mathbb{1}_{\mathsf{A}_\eta}(X)$$

where the last equality follows from the identity $\mathbf{A}^{-1}-\mathbf{B}^{-1}=\mathbf{A}^{-1}\{\mathbf{B}-\mathbf{A}\}\mathbf{B}^{-1}$. Finally, using Proposition 1 again, and that $\mathfrak{b}^*(0)=(1-d/n)^{-1}$ is the fixed point of $f_0:\mathfrak{b}\mapsto 1+\mathfrak{b}d/n$, we get

$$\frac{1}{d}\operatorname{tr}\left(\Sigma_X^{-1}\operatorname{R}_X(\lambda)\right) \approx \frac{1-d/n}{\lambda d}\operatorname{tr}\left(\operatorname{R}_X(0)\right)\mathbb{1}_{\mathsf{A}_\eta}(X) - \frac{1}{\lambda d\mathfrak{b}^*(\lambda)}\operatorname{tr}\left(\operatorname{R}_X(\lambda)\right) . \tag{4}$$

Finally, by the definition of $\mathfrak{b}^*(\lambda)$ and through straightforward algebraic manipulations, we obtain

$$\mathfrak{b}^*(\lambda) = 1 + \frac{1}{n}\operatorname{tr}\left(\Sigma_X\bar{\operatorname{R}}_X^{\mathfrak{b}^*(\lambda)}(\lambda)\right) = 1 + \mathfrak{b}^*(\lambda)\left\{\frac{d}{n}-\frac{\lambda}{n}\operatorname{tr}\left(\bar{\operatorname{R}}_X^{\mathfrak{b}^*(\lambda)}(\lambda)\right)\right\} .$$

Therefore, applying Proposition 1 again yields

$$\mathfrak{b}^*(\lambda) = \frac{1}{1-(d/n)+(\lambda/n)\operatorname{tr}\left(\bar{\operatorname{R}}_X^{\mathfrak{b}^*(\lambda)}(\lambda)\right)} \approx \frac{1}{1-(d/n)+(\lambda/n)\operatorname{tr}\left(\operatorname{R}_X(\lambda)\right)} = \mathfrak{b}(\lambda) ,$$

Plugging this estimate in (4), we identify $\hat{\mathcal{E}}_X(\lambda)$ (3) and it completes the proof of Theorem 1. The formal proof is postponed to Appendix B in the supplement.

## 3    Precision matrix estimation using generic data augmentation

In this section, we investigate a data augmentation strategy to improve the estimation of the inverse covariance matrix of $X$. Specifically, we consider an additional set of artificial samples $G = [G_1,\cdots,G_m]$, which may depend on $X$ and are typically generated using either TDA or GDA techniques; see Table 1. More precisely, given $X$, we assume that $G$ is drawn from a known regular conditional distribution $(\mathbf{X},\mathsf{A})\mapsto\nu_\mathbf{X}(\mathsf{A})$, meaning that for any measurable set $\mathsf{E}\subset\mathbb{R}^d$, we have $\mathbb{P}\left(G_i\in\mathsf{E}\mid X\right)=\nu_X(\mathsf{E})$. Then, we consider the following new estimator and define its quadratic error as:

$$\operatorname{R}_{\mathrm{Aug}}(\lambda) := \left((n+m)^{-1}\{XX^\top+GG^\top\}+\lambda\operatorname{I}_d\right)^{-1} , \quad \mathcal{E}_{\mathrm{Aug}}(\lambda) := (1/d)\|\operatorname{R}_{\mathrm{Aug}}(\lambda)-\Sigma_X^{-1}\|_{\mathrm{F}}^2 .$$

In the following, in addition to **H**1 and **H**2, which pertain to $X$, we introduce further assumptions on $G$. These are organized into three categories: a concentration assumption on $G$, a smoothness assumption on $\nu_X$, and a stability assumption on $\nu_X$.

**H3** (**Concentration of** $G$). *The random matrix $G\in\mathbb{R}^{d\times m}$ has i.i.d centered columns conditionally on $X$, i.e., $\mathbb{E}[G_j|X]=0$ for any $j\in\{1,\ldots,m\}$. In addition,*

*(i) The columns of $G$ are sub-Gaussian, with parameter $\sigma_G$*

*(ii) There exist $0\le\beta\le 1$, and $\Lambda_G:\mathbb{R}^{d\times n}\to\mathbb{R}^{d\times d}$ such that almost surely*

$$\mathbb{E}\left[C_G\mid X\right] = \beta C_X + \Lambda_G(X) ,$$

*and $\Lambda_G(X)$ is a positive definite matrix satisfying for some $\kappa>0$, $\kappa^{-1}\le\lambda_d(\Lambda_G(X))\le\lambda_1(\Lambda_G(X))\le\kappa$ almost surely on $\mathsf{A}_\eta$ defined in (2).*

Part (i) of **H**3 is a concentration assumption on $G$ conditional on $X$, similar to **H**1.

Regarding the second part (ii), it can be interpreted as a structural assumption. In most cases, the parameters $\beta$ and $\Lambda_G$ can be directly derived from the definition of the augmentation process. Table 1 provides values of $\beta$ and $\Lambda_G$ for a range of common DA schemes. As an example, consider the case where $G$ is drawn from a TDA procedure of the form $G_j=g(X_{I_j},Z_j)$, where $\{Z_j\}_{j=1}^m$ are i.i.d. and 1-Lipschitz concentrated (Definition 1), and $(I_j)_{j=1}^m$ are i.i.d, with $I_1\sim\operatorname{Unif}(\{1,\cdots,n\})$, we also

| | Augmentation Name | Description | $\Lambda_G$ | $\beta$ |
|---|---|---|---|---|
| **GDA** | Fixed Gaussian GDA | $G_j \sim \mathrm{N}(0, \Lambda)$ | $\Lambda$ | 0 |
| | Gaussian mixture GDA | $G_j \sim \sum_{i=1}^k w_i \mathrm{N}(\mu_i, \Lambda_i)$ | $\sum_{i=1}^k w_i\{\Lambda_i + \mu_i\mu_i^\top\}$ | 0 |
| **TDA** | Fixed Gaussian TDA | $X_{I_j} + Z_j \ , \ Z_j \sim \mathrm{N}(0, \Lambda)$ | $\Lambda$ | 1 |
| | Random mask TDA | $X_{I_j} \odot Z_j \ , \ Z_j \sim \mathrm{Ber}(\rho)^{\otimes d}$ | $\rho(1-\rho)\,\mathrm{diag}(C_X)$ | $\rho$ |
| | Salt & Pepper TDA | $X_{I_j} \odot Z_j + (1 - Z_j) \odot \mathrm{N}(0, \sigma^2)$ | $\rho(1-\rho)\,\mathrm{diag}(C_X) + (1-\rho)\sigma^2\,\mathrm{I}_d$ | $\rho$ |

Table 1: Various augmentation procedures and corresponding $\beta$ and $\Lambda_G$. We used the notation $I_j \sim \mathrm{Unif}(\{1, \cdots, n\})$. For more details, we refer to Appendix A.

assume that for any $x \in \mathbb{R}^d$, $\mathbb{E}_Z[g(Z_1, x)] = \sqrt{\beta^{(\mathrm{e})}}x$ for some $\beta^{(\mathrm{e})} \geq 0$. Then, it is straightforward to verify that **H**3-(ii) is satisfied with $\beta \leftarrow \beta^{(\mathrm{e})}$ and $\Lambda(X) \leftarrow \Lambda^{(\mathrm{e})}(X)$ where

$$\Lambda^{(\mathrm{e})}(X) = \frac{1}{n}\sum_{i=1}^n \mathbb{E}\left[\{g(Z_1, X_i) - \sqrt{\beta^{(\mathrm{e})}}X_i\}\{g(Z_1, X_i) - \sqrt{\beta^{(\mathrm{e})}}X_i\}^\top \ \Big| \ X_i\right] . \tag{5}$$

Table 1 below, shows the value of $\beta$ and $\Lambda_G(X)$ for a variety of common data-augmentation scheme.

Our second assumption on $G$ suppose that $\mathbf{X} \mapsto \nu_{\mathbf{X}}$ and $\mathbf{X} \mapsto \Lambda_G(\mathbf{X})$ are Lipschitz. More precisely:

**H4 (Smoothness of the artificial distribution).** *There exist* $\mathsf{L}_G \geq 0$ *and* $\mathsf{L}_\Lambda \geq 0$ *such that for any* $\mathbf{X}, \mathbf{Y} \in \mathbb{R}^{d \times n}$, *and* $m \in \mathbb{N}$,

$$W_1(\nu_{\mathbf{X}}^{\otimes m}, \nu_{\mathbf{Y}}^{\otimes m}) \leq \sqrt{m}\mathsf{L}_G\|\mathbf{X} - \mathbf{Y}\|_{\mathrm{F}} \ , \qquad \|\Lambda_G(\mathbf{X}) - \Lambda_G(\mathbf{Y})\|_{\mathrm{F}} \leq \mathsf{L}_\Lambda\|\mathbf{X} - \mathbf{Y}\|_{\mathrm{F}} \ .$$

Note that the DA examples Table 1 all satisfy this assumption provided $X$ has compact support. Otherwise, we believe that our results are robust enough to hold only when $\Lambda_G$ and $\mathbf{X} \mapsto \nu_{\mathbf{X}}$ are locally Lipschitz, albeit with slightly weaker convergence guarantees.

**H5 (Stability of the artificial distribution).** *(i) The map* $\mathbf{X} \mapsto \nu_{\mathbf{X}}$ *is invariant under permutation of the columns of* $\mathbf{X}$, *i.e., for any permutation* $\varsigma : \{1, \ldots, n\} \to \{1, \ldots, n\}$, $\nu_{\mathbf{X}} = \nu_{\mathbf{X}_\varsigma}$ *where* $\mathbf{X}_\varsigma = [\mathbf{X}_{\varsigma(1)}, \ldots, \mathbf{X}_{\varsigma(n)}]$.

*(ii) Furthermore, we assume that there exists* $\mathsf{K} \geq 0$, *such that for any* $m \in \mathbb{N}$,

$$W_1(\nu_X^{\otimes m}, \nu_{X^-}^{\otimes m}) \leq \sqrt{m}\mathsf{K} \ , \quad a.s.$$

Typically, $\mathsf{K}$ should remain bounded with respect to both $n$ and $d$. **H**5 can be interpreted as a condition ensuring that the data augmentation procedure used to generate the $\{G_j\}_{j=1}^m$ does not depend on any specific individual sample. It is met by various data augmentation procedures found in the literature. We provide in our next result a condition on $\nu_X$ and $\nu_{X^-}$ only, which implies **H**5-(ii). It proof is postponed to Appendix A.

**Proposition 2.** *Suppose that* $W_2(\nu_X, \nu_{X^-}) \leq \mathsf{K}$. *Then,* **H**5-*(ii) holds.*

**Remark 1.** *As a non-trivial example of a DA scheme that satisfies* **H**4 *and* **H**5, *let us consider the Random mask TDA, described in Table 1. We further illustrate our assumptions on other DA strategies in Appendix A. Let* $\mathbf{X}, \mathbf{Y} \in \mathbb{R}^{d \times n}$, *and consider the coupling of* $\nu_{\mathbf{X}}$ *and* $\nu_{\mathbf{Y}}$, *defined as for* $j \in \{1, \ldots, m\}$, $G_j = Z_j \odot \mathbf{X}_{I_j}$, $G'_j = Z_j \odot \mathbf{Y}_{I_j}$, *where* $I_j \sim \mathrm{Unif}(\{1, \cdots, n\})$, $Z_j \sim \mathrm{Ber}(\rho)^{\otimes d}$, *and* $\odot$ *is the elementwise multiplication. Then we have by the Cauchy-Schwarz inequality,*

$$W_1(\nu_{\mathbf{X}}^{\otimes m}, \nu_{\mathbf{Y}}^{\otimes m}) \leq \sqrt{m}W_2(\nu_{\mathbf{X}}, \nu_{\mathbf{Y}}) \leq \sqrt{m}\sqrt{\mathbb{E}\left[\|(\mathbf{X}_{I_1} - \mathbf{Y}_{I_1}) \odot Z_1\|_2^2\right]} \leq \sqrt{m}\rho\|\mathbf{X} - \mathbf{Y}\|_{\mathrm{F}} \ .$$

*Furthermore, from Table 1, we know that* $\Lambda_G(\mathbf{X}) = \frac{1-\rho}{\rho}\mathrm{diag}(C_{\mathbf{X}})$, *therefore it is locally-Lipschitz only. However, assuming that* $X$ *is bounded, we can always find another function* $\tilde{\Lambda}_G$ *satisfying* **H**3-*(ii) and which is Lipschitz.*

*We show through similar computations and using Proposition 2 that* **H**5 *is satisfied*

$$W_2(\nu_{\mathbf{X}}, \nu_{\mathbf{X}^-}) \leq \sqrt{\mathbb{E}\left[\|(X_{I_1} - X_{I_1}^-) \odot Z_1\|_2^2\right]} \leq \rho\sqrt{n^{-1}\mathbb{E}\left[\|X_1\|_2^2\right]} = \rho\sqrt{n^{-1}\,\mathrm{tr}\,(\Sigma_X)} \ .$$

We are now ready to introduce our estimate of $\mathcal{E}_{\text{Aug}}(\lambda)$. To this end, for any $\mathfrak{a} \geq 1$,

$$\bar{\text{R}}_{G|X}^{\mathfrak{a}}(\lambda) := \left( (1-\alpha) C_X + \alpha \frac{\mathbb{E}[C_G \mid X]}{\mathfrak{a}} + \lambda \text{I}_d \right)^{-1} . \tag{6}$$

where $\alpha = m/(n+m)$. In addition, we also consider the quantities

$$\mathfrak{a}_x(X) = 1 + \frac{1 - (1 - \beta/\mathfrak{a}_g(X))\alpha}{n} X_1^\top \mathbb{E}[R_{X^-\sqcup G}(\lambda) \mid X] X_1 , \tag{7}$$

$$\mathfrak{a}_g(X) = 1 + \frac{\alpha}{m} \text{tr}\left( \mathbb{E}[C_G \mid X] \mathbb{E}[R_{X\sqcup G}(\lambda) \mid X] \right) ,$$

and the two functions

$$\Phi_1(X) = \frac{(1 - d/n)}{d} \text{tr}\left( R_X(0) \left( \frac{\alpha \Lambda_G(X)}{\mathfrak{a}_g(X)} + \lambda \text{I}_d \right)^{-1} \right) \mathbb{1}_{\text{A}_\eta}(X) , \tag{8}$$

$$\Phi_2(X) = \frac{1 - (1 - \beta/\mathfrak{a}_g(X))\alpha}{d\mathfrak{a}_x(X)} \text{tr}\left( \bar{\text{R}}_{G|X}^{\mathfrak{a}_g(X)}(\lambda) \left( \frac{\alpha \Lambda_G(X)}{\mathfrak{a}_g(X)} + \lambda \text{I}_d \right)^{-1} \right) ,$$

Finally, we set

$$\hat{\mathcal{E}}_{\text{Aug}}(\lambda) := \frac{1}{d} \text{tr}\left( \text{R}_{\text{Aug}}(\lambda)^2 \right) - 2(\Phi_1(X) - \Phi_2(X)) + \frac{1}{d} \text{tr}\left( \Sigma_X^{-2} \right) , \tag{9}$$

and we emphasize that $\hat{\mathcal{E}}_{\text{Aug}}(\lambda)$ is computable from $X$ alone, provided we can sample from the distribution of $G$ conditionally on $X$. Our main result below states conditions under which $\hat{\mathcal{E}}_{\text{Aug}}(\lambda)$ concentrates around $\mathcal{E}_{\text{Aug}}(\lambda)$, for $\lambda$ arbitrarily small.

**Theorem 2.** *Assume* **H**1 *to* **H**5. *Let* $\hat{\mathcal{E}}_{\text{Aug}}(\lambda)$ *be defined in* (9). *Denoting* $\varepsilon = \min\{\eta, \lambda\}$, *for two scalars* $\tau_1$ *and* $\tau_2$, *(also independant of* $n$, $d$ *and* $m$, *and depending polynomially on* $\varepsilon$) *defined in* (78), *it holds*

$$\mathbb{P}\left( \left| \hat{\mathcal{E}}_{\text{Aug}}(\lambda) - \mathcal{E}_{\text{Aug}}(\lambda) \right| \geq t + \Delta_{\text{Aug}} \right) \lesssim n \exp\left( -k(n+m) \min\{\varepsilon^9 t^2/\tau_2, \varepsilon^7 t/\tau_1\} \right) ,$$

*where*

$$\Delta_{\text{Aug}} := \tilde{C}_1 \frac{(\sigma_X^2 + \sigma_G^2)(1 + c_X^{-1})(\|\Sigma_X\|_{\text{op}}^4 \kappa + \|\Sigma_X\|_{\text{op}} \kappa^4)}{(1-\alpha)n\lambda_d(\Sigma_X)^2 \varepsilon^7} + \tilde{C}_2 \frac{\mathbb{E}[\|\Lambda_G(X) - \mathbb{E}[\Lambda_G(X)]\|_{\text{F}}]}{\varepsilon^3 \sqrt{d}}$$

$$+ \frac{\|\Sigma_X\|_{\text{op}}^2}{\sqrt{d}\varepsilon^2} \|\Sigma_X \mathbb{E}[\Lambda_G(X)] - \mathbb{E}[\Lambda_G(X)] \Sigma_X\|_{\text{F}} .$$

*and the constants* $\tilde{C}_1$ *and* $\tilde{C}_2$ *depend polynomially on* $\lambda_d(\Sigma_X)$, $\|\Sigma_X\|_{\text{op}}^{-1}$, $\kappa^{-1}$, $n/m$, $\text{K}$, $\text{L}_G$ *and* $\varepsilon$.

In the statement above, the three contributions to $\Delta_{\text{Aug}}$ are small under natural conditions. The first term decays like $n^{-1}$ provided the covariance matrices $\Sigma_X$ and $\Lambda_G(X)$ remain well-conditioned and the fraction of artificial samples stays bounded away from one. The second term vanishes if the fluctuations of $\Lambda_G(X)$ are adequately controlled. Finally, the third term is negligible only when $\mathbb{E}[\Lambda_G(X)]$ approximately commutes with $\Sigma_X$, for instance, when the eigenvectors of $\Sigma_X$ are known, when the augmentation is isotropic on average (so that $\mathbb{E}[\Lambda_G(X)]$ is a scalar matrix), or more generally when $\mathbb{E}[\Lambda_G(X)]$ splits into a low-rank component plus a multiple of the identity (as in Gaussian mixture augmentations with few components relative to $d$, c.f. table 1).

## 4  Numerical experiments

In this section, we illustrate Theorem 1 and Theorem 2 on real datasets. We use MNIST and CIFAR10, consisting of 70,000 labeled $28 \times 28$ images and 60,000 labeled $32 \times 32$ images, respectively, with the following preprocessing:

**MNIST.** We discard the labels, normalize pixel values to $[0, 1]$, and add pixel-level Gaussian noise with standard deviation $\sigma = 0.1$ to ensure that the covariance matrix $\Sigma_X$ is well-conditioned.

**CIFAR10.** We discard the labels and convert images to grayscale.

For both datasets, we denote by $X = [X_1, \ldots, X_n] \in \mathbb{R}^{d \times n}$ the matrix formed by the first $n$ samples, for varying $n > 0$. To approximate $\mathcal{E}_X(\lambda)$ and $\mathcal{E}_{\text{Aug}}(\lambda)$, we use the sample covariance matrix $\hat{\Sigma}_X$ computed from all available samples (70,000 for MNIST, 60,000 for CIFAR10), and consider the proxies

$$\mathcal{E}_X^{\mathcal{D}}(\lambda) := \frac{1}{d} \big\| \mathrm{R}_X(\lambda) - \hat{\Sigma}_X^{-1} \big\|_{\mathrm{F}}^2 \quad \text{and} \quad \mathcal{E}_{\text{Aug}}^{\mathcal{D}}(\lambda) := \frac{1}{d} \big\| \mathrm{R}_{\text{Aug}}(\lambda) - \hat{\Sigma}_X^{-1} \big\|_{\mathrm{F}}^2, \tag{10}$$

which are expected to closely approximate $\mathcal{E}_X(\lambda)$ and $\mathcal{E}_{\text{Aug}}(\lambda)$ since the sample size greatly exceeds the data dimension.

Figure 1 summarizes our results for MNIST. In particular, figure 1a reports $\lambda \mapsto \hat{\mathcal{E}}_X(\lambda)$ for various $\gamma = 784/n$ over $\lambda \in [10^{-3}, 1]$, and compares it with the proxy above. Figure 1b and Figure 1c present $\hat{\mathcal{E}}_{\text{Aug}}(0)$ as a function of $\alpha = m/(n+m)$ under two data-augmentation schemes. The first is a $k$-centroid Gaussian-mixture GDA,

$$G_j = m_{I_j}(X) + \sigma \mathrm{N}(0, \mathrm{I}_d),$$

where the centroids $\{m_i\}_{i=1}^k$ are estimated via EM on $X$ and $I_j \sim \text{Unif}(\{1, \ldots, k\})$. The second is a Gaussian-noise TDA,

$$G_j = X_{I_j} + \sigma \mathrm{N}(0, \mathrm{I}_d),$$

with $I_j \sim \text{Unif}(\{1, \ldots, n\})$. In both cases, the minimizers of $\lambda \mapsto \hat{\mathcal{E}}_X(\lambda)$ and $\lambda \mapsto \hat{\mathcal{E}}_{\text{Aug}}(\lambda)$ are consistently close to those of the proxies $\mathcal{E}_X^{\mathcal{D}}(\lambda)$ and $\mathcal{E}_{\text{Aug}}^{\mathcal{D}}(\lambda)$, which should very closely approximate the true errors.

Symmetrically, for CIFAR10 (after grayscale conversion, so $d = 1024$), figure 2a reports $\lambda \mapsto \hat{\mathcal{E}}_X(\lambda)$ for various $\gamma = 1024/n$ over $\lambda \in [10^{-3}, 1]$ and compares it with the proxy in (10). Figures 2b and 2c present $\hat{\mathcal{E}}_{\text{Aug}}(0)$ as a function of $\alpha = m/(n+m)$ under the same $k$-centroid Gaussian-mixture GDA and Gaussian-noise TDA schemes as above. In all cases, the minimizers of $\lambda \mapsto \hat{\mathcal{E}}_X(\lambda)$ and $\lambda \mapsto \hat{\mathcal{E}}_{\text{Aug}}(\lambda)$ closely match those of the proxies $\mathcal{E}_X^{\mathcal{D}}(\lambda)$ and $\mathcal{E}_{\text{Aug}}^{\mathcal{D}}(\lambda)$.

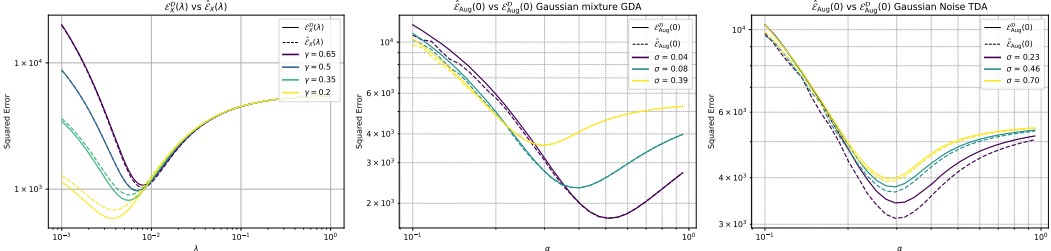

(a) Non-augmented, Ridge-like esti-  (b) 10-centroid Gaussian mixture  (c) Gaussian noise TDA.
mator.                               GDA.

Figure 1: Numerical results on MNIST for $\hat{\mathcal{E}}_X(\lambda)$ and $\hat{\mathcal{E}}_{\text{Aug}}(\lambda)$, compared with (10).

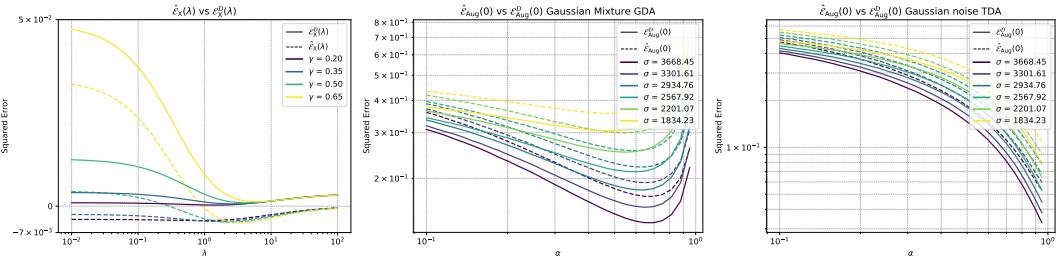

(a) Non-augmented, Ridge-like esti-  (b) 10-centroid Gaussian mixture  (c) Gaussian noise TDA.
mator.                               GDA.

Figure 2: Numerical results on CIFAR-10 for $\hat{\mathcal{E}}_X(\lambda)$ and $\hat{\mathcal{E}}_{\text{Aug}}(\lambda)$, compared with (10).

# 5  Conclusion

In this paper, we established new results based on random matrix theory that allow one to quantify from data only the impact of the regularization effect induced by data augmentation on a common class of precision matrix estimates. In the meantime, we presented a formula that allows one to compute from data only the error of a non-augmented "Ridgelike" precision matrix estimator. From a practical point of view, our results might allow one to optimally tune the hyperparameters of a data augmentation scheme for estimating the bottom eigenvalues and eigenvectors of the covariance matrix of the data, provided the data augmentation scheme satisfies a strict commutativity condition. Furthermore, it is well understood that the precision matrix is a fundamental object in many statistical models; hence, a natural extension of this work would be to study the generalization error of various machine learning models, such as linear regression, kernel regression, or some class of shallow networks.

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

# A In-depth justification of the hypothesis

This appendix provides detailed justifications for the technical assumptions introduced in the main text. In Appendix A.1, we analyze the concentration of the smallest eigenvalues of empirical covariance matrices under mild conditions, thereby establishing **H2** for standard random matrix models commonly studied in the literature. Subsequently, in Appendix A.2, we focus on data augmentation schemes and identify natural conditions for TDA and GDA under which Assumptions **H 3**–**H5** are satisfied.

## A.1 Discussions on H2

In this subsection, we establish explicit conditions under which **H2** holds and provide closed-form expressions for the parameters $\eta$ and $c_X$. These expressions are not directly estimable from data, as they depend on structural properties of the population covariance $\Sigma_X$, in particular its smallest eigenvalue $\lambda_d(\Sigma_X)$. Nonetheless, they yield useful theoretical insight into the regimes where our results are applicable. Formally, we obtain the following result:

**Proposition 3.** *Assume that $X$ satisfies **H1**, and that $\lambda_d(\Sigma_X) > 0$. There exists a universal constant $c$ such that whenever $n > d > 0$, **H2** is guaranteed to hold for any choice of $\eta$ and $c_X$ satisfying:*

$$\eta < \lambda_d(\Sigma_X)\Big(\sqrt{\tfrac{n-1}{n}} - \sqrt{\tfrac{d}{n}}\Big), \quad \text{and} \quad c_X = c\Big(\sqrt{\tfrac{n-1}{n}} - \sqrt{\tfrac{d}{n}} - \sqrt{\tfrac{\eta}{\lambda_d(\Sigma_X)}}\Big)^2 .$$

To support the previous claim, we introduce a standard non-asymptotic result from random matrix theory. For a rectangular matrix $\mathbf{A} \in \mathbb{R}^{d \times n}$, we denote by $s_{\min}(\mathbf{A})$ its smallest singular value. The following theorem, due to Rudelson and Vershynin [Ver11, Theorem 5.39], provides a sharp lower bound on $s_{\min}$ for random sub-Gaussian matrices.

**Theorem 3** (Rudelson–Vershynin [Ver11, Theorem 5.39]). *Let $Z$ be a $d \times n$ random matrix with $n \geq d$, whose columns are independent, identically distributed, mean-zero, isotropic sub-Gaussian random vectors in $\mathbb{R}^d$. Then there exist absolute constants $c > 0$ such that, for all $t \geq 0$,*

$$\mathbb{P}\Big(s_{\min}(Z) \geq \sqrt{n} - \sqrt{d} - t\Big) \geq 1 - 2e^{-ct^2}.$$

Observing that $X^- = \Sigma_X^{1/2} Z^-$ where $Z$ has isotropic and independant columns (under **H1**) and applying Theorem 3 to $Z$, we obtain the following bound on the probability of encountering small eigenvalues in the leave-one-out covariance matrix $C_X^-$.

**Corollary 1.** *Assume that $X$ satisfies **H1**. Then, for every $\epsilon > 0$,*

$$\mathbb{P}\big(\lambda_d(C_X^-) \leq \eta\big) \lesssim \exp\left(-c\left(\sqrt{\tfrac{n-1}{n}} - \sqrt{\tfrac{d}{n}} - \sqrt{\tfrac{\eta}{\lambda_d(\Sigma_X)}}\right)^2 n\right),$$

*where $c > 0$ is the same absolute constants as in Theorem 3. In particular, Proposition 3 follows directly.*

*Proof.* Let $Z = \Sigma_X^{-1/2} X$. Then $Z$ is a random matrix with i.i.d. isotropic sub-Gaussian columns, since $X$ satisfies **H1** and,

$$\mathbb{E}\big[\tfrac{1}{n} Z Z^\top\big] = \Sigma_X^{-1/2} \mathbb{E}[C_X] \Sigma_X^{-1/2} = \mathrm{I}_d .$$

Using the inequality $s_{\min}(AB) \geq s_{\min}(A) s_{\min}(B)$, we obtain

$$\lambda_d(C_X^-) = \tfrac{1}{n} s_{\min}(X^-)^2 \geq \tfrac{1}{n} \lambda_d(\Sigma_X) s_{\min}(Z^-)^2 , \quad \text{where} \quad Z^- = [Z_2, \dots, Z_n]$$

Hence, for any $0 \leq t \leq \sqrt{n-1} - \sqrt{d}$, we have

$$\mathbb{P}\left(\lambda_d(C_X^-) \geq \lambda_d(\Sigma_X) \frac{(\sqrt{n-1} - \sqrt{d} - t)^2}{n}\right) \geq \mathbb{P}\Big(s_{\min}(Z^-) \geq \sqrt{n-1} - \sqrt{d} - t\Big) .$$

Applying Theorem 3, we deduce that for all $t \geq 0$, we have,

$$\mathbb{P}\left(\lambda_d(C_X^-) \geq \lambda_d(\Sigma_X)\left(\sqrt{\tfrac{n-1}{n}} - \sqrt{\tfrac{d}{n}} - \tfrac{t}{\sqrt{n}}\right)^2\right) \geq 1 - 2\mathrm{e}^{-ct^2} . \tag{11}$$

Now fix any $0 < \eta \leq \lambda_d(\Sigma_X)\left(\sqrt{(n-1)/n} - \sqrt{d/n}\right)^2$ and define

$$t_\eta = \left( \sqrt{\tfrac{n-1}{n}} - \sqrt{\tfrac{d}{n}} - \sqrt{\tfrac{\eta}{\lambda_d(\Sigma_X)}} \right) \sqrt{n} \ .$$

By construction, $t_\eta \geq 0$. And substituting $t = t_\eta$ into (11) yields

$$\mathbb{P}\big(\lambda_d(C_X^-) \geq \eta\big) \ \geq \ 1 - 2\mathrm{e}^{-ct_\eta^2} \ ,$$

which is the desired bound. $\qquad\square$

## A.2  Discussions on H3, H4, H5

In this section, we demonstrate that several common data augmentation (DA) schemes satisfy Assumptions **H3**–**H5**. We also discuss the limitations of these assumptions and identify scenarios in which they hold exactly, thereby clarifying the regimes where our results apply. We begin by introducing a generalization of **H1** which will help us achieve more general statements, as well as simplify the proofs. To this end, we introduce the following definition of Lipschitz concentrated random vectors:

**Definition 1** (Lispchitz concentration)**.** *We say that,*

(i)  *The random vector $X_1 \in \mathbb{R}^d$ is Lispchitz concentrated with parameter $\sigma$ if and only if for any 1-Lipschitz function $f$, and any $s \geq 0$, we have,*

$$\mathbb{E}\left[\exp\left(s\{f(X_1) - \mathbb{E}\left[f(X_1)\right]\}\right)\right] \leq \exp\left(\sigma^2 s^2\right)$$

(ii)  *The probability distribution $\mu \in \mathcal{P}(\mathbb{R}^d)$ has the Lispchitz concentration property of paramet $\sigma$ if and only if for $X_1 \sim \mu$, $X_1$ is Lipschitz concentrated with parameter $\sigma$.*

and we replace **H1** by the following assumption:

**H6** (Lipschitz concentration of the data)**.** *The columns $X_1, \dots, X_n \in \mathbb{R}^d$ of the data matrix $X \in \mathbb{R}^{d \times n}$ are independent random vectors, each of which is Lipschitz concentrated with parameter $\sigma_X > 0$ in the sense of Definition 1. Equivalently, for every 1-Lipschitz function $f : \mathbb{R}^d \to \mathbb{R}$ and every $s \geq 0$,*

$$\mathbb{E}[\exp(s\{f(X_i) - \mathbb{E}[f(X_i)]\})] \ \leq \ 2\exp\left(\tfrac{\sigma_X^2 s^2}{2}\right), \qquad \textit{for all } i \in \{1, \dots, n\}.$$

One can easily check **H1** implies **H6**, furthermore the class of matrix satifying **H6** being stable by Lispchitz transformations (up to a rescaling of a concentration parameter), will turn out very convenient for the proofs of our main results.

We now provide a set of simple sufficient conditions under which **H3** is satisfied, we believe that the vast majority of common data augmentation scheme satify this condition. First, in the case of GDA schemes, we show that under an almost sure smoothness property of the sample generation process Item (i) is satisfied:

**Proposition 4.** *Let $X \in \mathbb{R}^{d \times n}$ be a random matrix. Assume that for each $j \in \{1, \dots, m\}$, $G_j = f(Z_j, X)$, where $Z_j$ are i.i.d. random vectors with the $\sigma_Z$-Lipschitz concentration property, and where $f(\cdot, X) : \mathbb{R}^d \to \mathbb{R}^d$ is almost surely $\mathsf{L}_f$-Lipschitz. Then $G = [G_1, \dots, G_m]$ satisfies Item (i) of **H3**, with parameter*

$$\sigma_G \leftarrow \mathsf{L}_f \sigma_Z.$$

*Proof.* Let $\mu$ denote the distribution of $Z$, so that for any measurable set $\mathsf{E} \subset \mathbb{R}^d$, $\mu(\mathsf{E}) = \mathbb{P}(Z \in \mathsf{E})$. Since $G_j = f(Z_j, X)$, the conditional law of $G_j$ given $X$ is the pushforward measure of $\mu$ under $f(\cdot, X)$:

$$\nu_X = f(\cdot, X)^\# \mu \ ,$$

where for a measurable map $\varphi : \mathsf{A} \to \mathsf{A}$ and a measure $\mu$ on $\mathsf{A}$, we recall the notation $\varphi^\# \mu(\mathsf{E}) = \mu(\varphi^{-1}(\mathsf{E}))$.

To show that $G$ satisfies Item (i), we set $h : \mathbb{R}^d \to \mathbb{R}$ to be any 1-Lipschitz function such that $\mathbb{E}[h(G_1)] = 0$. Consider, for $s \geq 0$,

$$\mathbb{E}[\exp(sh(G_1)) \mid X] = \mathbb{E}[\exp(sh(f(Z_1, X))) \mid X] .$$

The mapping

$$z_1 \mapsto h\big(f(z_1, X)\big)$$

is centered with respect to $\mu^{\otimes m}$ by the assumption on $h$, and it is $\mathsf{L}_f$-Lipschitz almost surely, since it is the composition of a 1-Lipschitz map and an $\mathsf{L}_f$-Lipschitz map. Because $Z \sim \mu$ has the $\sigma_Z$-Lipschitz concentration property we thus obtain

$$\mathbb{E}[\exp(sh(G_1)) \mid X] \leq \exp\left(s^2 \mathsf{L}_f^2 \sigma_Z^2\right) .$$

This establishes that $\nu_X$ has the $\sigma_G$-Lipschitz concentration property with $\sigma_G = \mathsf{L}_f \sigma_Z$, and completes the proof. □

Similarly, in the case of TDA schemes, we have highlight the following sufficient condition for Item (i) of **H**3:

**Proposition 5.** *Let $X \in \mathbb{R}^{d \times n}$ be a random matrix. Assume that $G_j = f(Z_j, X_{I_j})$ where:*

- *$f$ is a $\mathsf{L}_f$-Lipschitz function w.r.t its first argument.*

- *$I_j \sim \mathrm{Unif}(\{1, \ldots, n\})$, and $Z_j$ has the $\sigma_Z$-Lipschitz concentration property.*

- *The augmented samples lie in a compact, for all $i$, $\|\mathbb{E}[f(Z, X_i) \mid X]\|_2 \leq K$.*

*Then $G = [G_1, \cdots, G_m]$ satisfies Item (i) of **H**3 for*

$$\sigma_G^2 \leftarrow \mathsf{L}_f^2 + cK^2 + c\mathsf{L}_f^2 \sigma_Z^2 ,$$

*where $c > 0$ is a universal constant.*

*Proof.* Note that under the assumptions of Proposition 5, we have,

$$\nu_X = \frac{1}{n} \sum_{i=1}^{n} f(\cdot, X_i)^{\#} \mu ,$$

where $\mu$ is the distribution of $Z$, such that $\mathbb{P}(Z \in \mathsf{E}) = \mu(\mathsf{E})$, for any $\mathsf{E} \subset \mathbb{R}^d$, and we used the notation $\varphi^{\#}\mu$ for the pushforward measure, $\varphi^{\#}\mu(\mathsf{E}) = \mu(\varphi^{-1}(\mathsf{E}))$, for any $\mathsf{E} \subset \mathbb{R}^d$.

We show that $\nu_X$ has the Lipschitz concentration property, to this end, let $h : \mathbb{R}^d \to \mathbb{R}$ be a Lipschitz function ($X$-measureable) such that $h(0) = 0$ (note that this can be assumed without loss of generality). For notation simplicity, we further define $\bar{h} = h - \mathbb{E}[h(G_1) \mid X]$, then we have for any $s \geq 0$,

$$\mathbb{E}[\exp(s\{h(G_1) - \mathbb{E}[h(G_1) \mid X]\}) \mid X] = \mathbb{E}\left[\exp\left(s\bar{h}(G_1)\right) \mid X\right]$$

$$= \frac{1}{n} \sum_{i=1}^{n} \mathbb{E}\left[\exp\left(s\bar{h}(f(Z_i, X_i))\right) \mid X\right]$$

Denote by $m_i = \mathbb{E}\left[\bar{h}(f(Z_i, X_i)) \mid X\right] = \mathbb{E}[h(f(Z_i, X_i)) \mid X] - \mathbb{E}[h(G_1) \mid X]$, we further write,

$$\mathbb{E}[\exp(s\{h(G_1) - \mathbb{E}[h(G_1) \mid X]\}) \mid X] = \frac{1}{n} \sum_{i=1}^{n} \exp(sm_i) \mathbb{E}\left[\exp\left(s\left\{\bar{h}(f(Z_i, X_i)) - m_i\right\}\right)\right]$$

$$\leq \exp\left(s^2 \mathsf{L}_f^2\right) \frac{1}{n} \sum_{i=1}^{n} \exp(sm_i) ,$$

where we have used the Lipschitz concentration of $\mu$, and the Lipschitz property of $h$ in the last bound. We now denote $\pi$ as the following measure,

$$\pi = \frac{1}{n} \sum_{i=1}^{n} \delta_{m_i} ,$$

where $\delta_{m_i}$ is the Dirac measure at $m_i$. Remarking that $n^{-1}\sum_{i=1}^{n} m_i = 0$, and that $\pi$ has bounded support (because the $X_i$'s are boudned and the maps $f$ and $h$ are Lipschitz). We further write,

$$\mathbb{E}\left[\exp\left(s\{h(G_1) - \mathbb{E}\left[h(G_1)\mid X\right]\}\right)\mid X\right] \leq \exp\left(s^2 \mathsf{L}_f^2\right)\mathbb{E}_\pi\left[\exp(s\{M - \mathbb{E}\left[M\right]\})\right]. \quad (12)$$

to conclude the proof, note that $M \sim \pi$ has bounded support in $\mathbb{R}$, as so it is necessarily sub-Gaussian, with sub-Gaussian norm,

$$\|M\|_{\Psi_2} \leq \frac{1}{\ln(2)}\sup_i |m_i|,$$

which follows from [Ver09], Example 2.5.8. Thus, we have for a universal constant $c > 0$,

$$\mathbb{E}_\pi\left[\exp\left(s\left\{M - \mathbb{E}\left[M\right]\right\}\right)\right] \leq \exp\left(cs^2 \sup_i |m_i|^2\right). \quad (13)$$

Finally, we bound $|m_i|$ independantly of $i$, leveraging the boundedness of $X$. We have,

$$\begin{aligned}
|m_i| &= |\mathbb{E}\left[h(f(Z_i, X_i))\mid X\right]| \\
&= |h(f(0, X_i))| + |\mathbb{E}\left[h(f(Z_i, X_i)) - h(f(0, X_i))\mid X\right]| \\
&\leq |h(f(0, X_i))| + \mathsf{L}_f \mathbb{E}\left[\|Z_i\|_2\right] \\
&\leq \sup_{i \leq n}|h(f(0, X_i))| + \mathsf{L}_f\sqrt{\mathbb{E}\left[Z_i^\top Z_i\right]}
\end{aligned}$$

$$\begin{aligned}
\sup_i |m_i| &= \sup_i |\mathbb{E}\left[h(f(Z_i, X_i))\mid X\right] - \mathbb{E}\left[f(G_1)\mid X\right]| \\
&\leq 2\sup_i |\mathbb{E}\left[h(f(Z_i, X_i))\mid X\right]| \\
&\leq 2|h(0)| + 2\sup_i \mathbb{E}\left[|h(f(Z_i, X_i)) - h(0)|\mid X\right] \\
&\leq 2|h(0)| + 2\sup_i \mathbb{E}\left[\|f(Z_i, X_i)\|_2 \mid X\right] \\
&\leq 2|h(0)| + 2\sup_i \|\mathbb{E}\left[f(Z_i, X_i)\mid X\right]\|_2 + 2\sup_i \sqrt{\mathbb{E}\left[\|f(Z_i, X_i) - \mathbb{E}\left[f(Z_i, X_i)\mid X\right]\|_2 \mid X\right]} \\
&\leq 2\sup_i \|\mathbb{E}\left[f(Z_i, X_i)\mid X\right]\|_2 + 2\mathsf{L}_f \sigma_Z.
\end{aligned}$$

Where in the last line, we have used the Lipschitz concentration property of $Z_i$ (as well as $f(\cdot, X_i)$ being $\mathsf{L}_f$ Lispchitz), and the fact that $h(0) = 0$. We conclude the proof by using the boundedness assumption on $\mathbb{E}\left[f(Z, X_i)\mid X\right]$, which yields,

$$\sup_i |m_i|^2 \leq (2K + 2\mathsf{L}_f \sigma_Z)^2 \leq 4K^2 + 4\mathsf{L}_f^2 \sigma_Z^2,$$

plugging this back into (12) and (13), we obtain,

$$\mathbb{E}\left[\exp\left(s\left\{h(G_1) - \mathbb{E}\left[f(G_1)\mid X\right]\right\}\right)\mid X\right] \leq \exp\left(s^2\left\{\mathsf{L}_f^2 + cK^2 + c\mathsf{L}_f^2 \sigma_Z^2\right\}\right)$$

$\square$

As consequences of Propositions 4 and 5, a broad class of commonly used data-augmentation (DA) schemes satisfy Item (i) from **H**3. In particular:

**(1) Deep generative models.** Consider a generative mapping

$$f(z, X) = \theta_X^{(L)}\sigma_L\left(\cdots \sigma_1\left(\theta_X^{(1)} z\right)\right),$$

where $L \geq 1$. Let $d_\ell$ denote the width of layer $\ell$ (so $d_0 = d_Z$ and $d_L = d$). For each $\ell = 1, \ldots, L$, assume $\sigma_\ell : \mathbb{R}^{d_\ell} \to \mathbb{R}^{d_\ell}$ is a non-linear, 1-Lipschitz activation and $\theta_X^{(\ell)} \in \mathbb{R}^{d_\ell \times d_{\ell-1}}$ is a (possibly $X$-dependent) weight matrix. Suppose further that the operator norms are a.s. bounded by a constant $K$, i.e., $\|\theta_X^{(\ell)}\|_{\mathrm{op}} \leq K$ for all $\ell$. Then, by Proposition 4, the matrix

$$G = [G_1, \ldots, G_m], \qquad G_j = f(Z_j, X), \quad Z_j \sim \mathrm{N}(0, \mathrm{I}_{d_Z}),$$

satisfies Item (i). Indeed, the network $f(\cdot, X)$ is $\prod_{\ell=1}^{L}\|\theta_X^{(\ell)}\|_{\mathrm{op}}$-Lipschitz, hence $K^L$-Lipschitz a.s., from Proposition 4 it results that $G$ satisfies Item (i) of **H**3 with concentration parameter $K^L$.

**(2) Transformative data augmentation.** Likewise, Proposition 5 provides mild conditions under which Item (i) holds for transformative DA schemes. Consider

$$G = [G_1, \ldots, G_m], \qquad G_j = f(X_{I_j}, Z_j), \quad I_j \sim \mathrm{Unif}(\{1, \ldots, n\}), \quad Z_j \sim \mu \,,$$

i.e. we randomly select the sample to be deformed, and the deformation is smooth w.r.t. some parameter $Z$. Numerous standard transformative DA mechanisms use smooth, small-amplitude perturbations; later in this section we detail the cases of Gaussian noise and random masking.

The second part Item (ii) of **H3** is not always theoretically guaranteed, yet in the case of an unbiased TDA it is an immediate consequence of the law of total variance. Indeed, assuming $G_j = f(X_{I_j}, Z_j)$ as in Proposition 5, and that $\forall \, \mathbf{x}, \; \mathbb{E}[f(\mathbf{x}, Z)] = \mathbf{x}$, one can write

$$\mathbb{E}[C_G \mid X] = \frac{1}{n} \sum_{i=1}^{n} \mathbb{E}[f(X_i, Z) \mid X] \, \mathbb{E}[f(X_i, Z) \mid X]^\top$$

$$+ \frac{1}{n} \sum_{i=1}^{n} \mathbb{E}\Big[ \big\{ f(X_i, Z) - \mathbb{E}[f(X_i, Z) \mid X] \big\} \big\{ f(X_i, Z) - \mathbb{E}[f(X_i, Z) \mid X] \big\}^\top \Big| X \Big]$$

$$= C_X + \frac{1}{n} \sum_{i=1}^{n} \mathbb{E}\Big[ \big\{ f(X_i, Z) - X_i \big\} \big\{ f(X_i, Z) - X_i \big\}^\top \Big| X \Big] \,,$$

where $I_j \sim \mathrm{Unif}(\{1, \ldots, n\})$ and $Z_j \sim \mu$ are independent.

In the case of GDA, the decomposition in Item (ii) of **H3** holds trivially, yet no simple expression of $\Lambda_G(X)$ exists.

We now spell out conditions under which **H5** holds. To this end, we introduce the following upper bound:

**Lemma 1.** *Let $\mu_1$ and $\mu_2$ be two probability measures on $\mathbb{R}^d$. Then, for any $m \geq 1$,*

$$W_1\big(\mu_1^{\otimes m}, \mu_2^{\otimes m}\big) \;\leq\; \sqrt{m}\, W_2(\mu_1, \mu_2) \,.$$

*Proof.* Recall that

$$W_1\big(\mu_1^{\otimes m}, \mu_2^{\otimes m}\big) \;=\; \inf_{\gamma_m \in \Gamma(\mu_1^{\otimes m}, \mu_2^{\otimes m})} \int \|x - x'\|_{\mathrm{F}} \, \mathrm{d}\gamma_m(x, x') \,,$$

where $\Gamma(\cdot, \cdot)$ denotes the set of all couplings and $\|\cdot\|_{\mathrm{F}}$ is the Euclidean/Frobenius norm on $(\mathbb{R}^d)^m$. Let $\gamma_* \in \Gamma(\mu_1, \mu_2)$ be an optimal coupling for $W_2$, so that

$$W_2(\mu_1, \mu_2)^2 \;=\; \int \|u - v\|_2^2 \, \mathrm{d}\gamma_*(u, v) \,.$$

Consider $\gamma_m := \gamma_*^{\otimes m} \in \Gamma(\mu_1^{\otimes m}, \mu_2^{\otimes m})$. Then, by the definition of $W_1$ and Cauchy–Schwarz,

$$W_1\big(\mu_1^{\otimes m}, \mu_2^{\otimes m}\big) \leq \int \|x - x'\|_{\mathrm{F}} \, \mathrm{d}\gamma_*^{\otimes m}(x, x') \;\leq\; \left( \int \|x - x'\|_{\mathrm{F}}^2 \, \mathrm{d}\gamma_*^{\otimes m}(x, x') \right)^{1/2}$$

$$= \left( \sum_{i=1}^{m} \int \|x_i - x_i'\|_2^2 \, \mathrm{d}\gamma_*(x_i, x_i') \right)^{1/2} \;=\; \sqrt{m}\, W_2(\mu_1, \mu_2) \,.$$

$\square$

The previous result is particularly convenient for demonstrating that **H4** and **H5** hold, which will be done in full detail for all DA scheme presented in Table 1. Towards a full justification of Table 1, we show that (5) holds. In particular, we establish that the following is true:

**Lemma 2.** *Assume that*

$$G_j = f(X_{I_j}, Z_j), \qquad j = 1, \ldots, m,$$

*where $I_j \sim \mathrm{Unif}\{1, \ldots, n\}$ and $Z_1, \ldots, Z_m$ are i.i.d. random variables. Further suppose that for each $i \leq n$,*

$$\mathbb{E}\big[ f(X_i, Z_1) \mid X_i \big] = \sqrt{\beta}\, X_i, \qquad \beta \in [0, 1].$$

*Then*

$$\mathbb{E}\big[C_G \mid X\big] = \beta\, C_X + \frac{1}{n}\sum_{i=1}^{n}\mathbb{E}\Big[\big(f(X_i, Z_1) - \sqrt{\beta}\, X_i\big)\big(f(X_i, Z_1) - \sqrt{\beta}\, X_i\big)^{\top} \,\Big|\, X\Big],$$

*and Item (ii) of **H3** holds.*

*Proof.* We use the notation

$$\mathbb{E}[G \mid X] = \big[\mathbb{E}_Z[f(X_1, Z) \mid X],\, \dots,\, \mathbb{E}_Z[f(X_n, Z) \mid X]\big] \in \mathbb{R}^{d \times n}.$$

By the law of total variance,

$$\mathbb{E}[C_G \mid X] = \mathbb{E}\big[G_1 G_1^{\top} \mid X\big] = \frac{1}{n}\sum_{i=1}^{n}\mathbb{E}\big[f(X_i, Z)f(X_i, Z)^{\top} \mid X\big]$$

$$= C_{\mathbb{E}[G\mid X]} + \frac{1}{n}\sum_{i=1}^{n}\mathbb{E}\Big[\big(f(X_i, Z) - \mathbb{E}[f(X_i, Z) \mid X]\big)\big(f(X_i, Z) - \mathbb{E}[f(X_i, Z) \mid X]\big)^{\top} \,\Big|\, X\Big]$$

$$= \beta\, C_X + \frac{1}{n}\sum_{i=1}^{n}\mathbb{E}\Big[\big(f(X_i, Z) - \sqrt{\beta}\, X_i\big)\big(f(X_i, Z) - \sqrt{\beta}\, X_i\big)^{\top} \,\Big|\, X\Big],$$

which concludes the proof. $\qquad\qquad\square$

Relying on the above results Lemma 1 and Lemma 2, we now justify the results presented in Table 1.

**Fixed Gaussian GDA:** Consider the Gaussian GDA scheme where, for all $j \in \{1, \dots, m\}$, we have $G_j \sim \mathrm{N}(0, \Lambda)$, for some fixed positive semi-definite matrix $\Lambda$.

We recall from [LC18, Theorem 2.19] that the standard Gaussian distribution $\mathrm{N}(0, \mathrm{I}_d)$ in $\mathbb{R}^d$ satisfies the 1-Lipschitz concentration property. Moreover, the mapping $\mathbf{Z} \mapsto \Lambda^{1/2}\mathbf{Z}$ is $\|\Lambda^{1/2}\|_{\mathrm{op}}$-Lipschitz (with respect to the Frobenius norm), which, by the same result, implies that $G$ is $\|\Lambda^{1/2}\|_{\mathrm{op}}$-Lipschitz concentrated.

Furthermore, we have

$$\mathbb{E}[C_G \mid X] = \mathbb{E}\big[G_1 G_1^{\top}\big] = \Lambda\,,$$

which shows that the Gaussian GDA scheme satisfies **H3**, with $\beta \leftarrow 0$, $\Lambda_G \leftarrow \Lambda$, and $\sigma_G \leftarrow \|\Lambda^{1/2}\|_{\mathrm{op}}$.

Finally, note that $\nu_{\mathbf{X}} = \mathrm{N}(0, \Lambda)$ is constant (i.e., independent of $X$), and therefore trivially satisfies both **H4** and **H5**.

**Gaussian GDA:** In the more general and realistic case where the covariance matrix of the artificial distribution depends on $X$, we assume that $G_j \sim \mathrm{N}(0, \Lambda(X))$ for all $j \leq m$, such that $\Lambda$ is $\mathrm{L}_\Lambda$-Lipschitz (with respect to the Frobenius norm), and that $K^{-1} \leq \lambda_d(\Lambda(\mathbf{X})) \leq \dots \leq \|\Lambda(X)\|_{\mathrm{op}} \leq K$ almost surely. These assumptions directly ensure that both parts of **H3** are satisfied: indeed $G$ is guaranteed to be $K$-Lipschitz concentrated conditionally on $X$ for the same reason as in the fixed Gaussian case, and Item (ii) holds with $\Lambda_G(X) \leftarrow \Lambda(X)$ by definition. Similarly, the second part of **H4** is satisfied by the hypothesis on $\Lambda(X)$.

To show the first part of **H4**, we use an equivalent "Procrustes" form of 2-Wasserstein for zero-mean Gaussians with covariances:

$$W_2\big(\nu_{\mathbf{X}}, \nu_{\mathbf{Y}}\big) = \min_{U \in \mathcal{O}(d)}\big\|\Lambda(X)^{1/2} - \Lambda(Y)^{1/2}U\big\|_{\mathrm{F}}$$

$$= \mathrm{tr}\left(\Lambda(X) + \Lambda(Y) - 2(\Lambda(X)^{1/2}\Lambda(Y)\Lambda(X)^{1/2})^{1/2}\right),$$

which follows by expanding $\|A^{1/2} - B^{1/2}U\|_{\mathrm{F}}^2$ and maximizing $\mathrm{tr}(A^{1/2}B^{1/2}U)$ over orthogonal $U$ via von Neumann's trace inequality. This yields

$$W_2(\nu_{\mathbf{X}}, \nu_{\mathbf{Y}}) \leq \|\Lambda(X)^{1/2} - \Lambda(Y)^{1/2}\|_{\mathrm{F}}. \tag{14}$$

To further bound the above $W_2$ metric, we need to prove that spectral transformations of large symmetric matrices are Lipschitz. To this end, we introduce the following lemma:

**Lemma 3.** *Let $\mathbf{A}$ and $\mathbf{B}$ be two symmetric matrices in $\mathbb{R}^{d \times d}$, with respective eigenvalues $\lambda_1(\mathbf{A}) \geq \cdots \geq \lambda_d(\mathbf{A})$ and $\lambda_1(\mathbf{B}) \geq \cdots \geq \lambda_d(\mathbf{B})$. Let $f : \mathbb{R} \to \mathbb{R}$ be $\mathsf{L}_f$-Lipschitz on an interval containing $[\lambda_d(\mathbf{A}), \lambda_1(\mathbf{A})] \cup [\lambda_d(\mathbf{B}), \lambda_1(\mathbf{B})]$. Then*

$$\|f(\mathbf{A}) - f(\mathbf{B})\|_{\mathrm{F}} \leq \mathsf{L}_f \|\mathbf{A} - \mathbf{B}\|_{\mathrm{F}} ,$$

*where for any symmetric matrix $\mathbf{M} = \mathbf{P} \operatorname{diag}(d_1, \dots, d_d) \mathbf{P}^{\top}$, we define $f(\mathbf{M}) = \mathbf{P} \operatorname{diag}(f(d_1), \dots, f(d_d)) \mathbf{P}^{\top}$.*

*Proof.* Define the path $W(t) = \mathbf{B} + t(\mathbf{A} - \mathbf{B})$ for $t \in [0, 1]$. Note that each $W(t)$ is symmetric, and

$$f(\mathbf{A}) - f(\mathbf{B}) = \int_0^1 \frac{\mathrm{d}}{\mathrm{d}t} f(W(t)) \, \mathrm{d}t .$$

By the triangle inequality,

$$\|f(\mathbf{A}) - f(\mathbf{B})\|_{\mathrm{F}} \leq \int_0^1 \left\| \frac{\mathrm{d}}{\mathrm{d}t} f(W(t)) \right\|_{\mathrm{F}} \mathrm{d}t = \int_0^1 \left\| f'(W(t)) (\mathbf{A} - \mathbf{B}) \right\|_{\mathrm{F}} \mathrm{d}t ,$$

where $f'(W(t))$ denotes the (matrix) derivative coming from the spectral calculus. Since $f$ is $\mathsf{L}_f$-Lipschitz on an interval containing the spectrum of each $W(t)$, we have $\|f'(W(t))\|_{\mathrm{op}} \leq \mathsf{L}_f$ for a.e. $t$, hence

$$\|f(\mathbf{A}) - f(\mathbf{B})\|_{\mathrm{F}} \leq \int_0^1 \mathsf{L}_f \|\mathbf{A} - \mathbf{B}\|_{\mathrm{F}} \, \mathrm{d}t = \mathsf{L}_f \|\mathbf{A} - \mathbf{B}\|_{\mathrm{F}} .$$

This proves the claim. $\qquad\square$

To conclude on the Lispchitz bound, apply Lemma 3 to $f(t) = \sqrt{t}$ on the spectral interval of $\Lambda(X)$ and $\Lambda(Y)$. Recall that we have $\lambda_d(\Lambda(\cdot)) \geq K^{-1}$, then $\|f'\|_{\infty} = \sup_{t \geq K^{-1}} \frac{1}{2\sqrt{t}} \leq \frac{\sqrt{K}}{2}$, so

$$\|\Lambda(\mathbf{X})^{1/2} - \Lambda(\mathbf{Y})^{1/2}\|_{\mathrm{F}} \leq \frac{\sqrt{K}}{2} \|\Lambda(\mathbf{X}) - \Lambda(\mathbf{Y})\|_{\mathrm{F}} \leq \frac{\sqrt{K}}{2} \mathsf{L}_\Lambda \|\mathbf{X} - \mathbf{Y}\|_{\mathrm{F}}.$$

Combining the previous with (14) yields

$$W_2(\nu_{\mathbf{X}}, \nu_{\mathbf{Y}}) \leq \frac{\sqrt{K}}{2} \mathsf{L}_\Lambda \|\mathbf{X} - \mathbf{Y}\|_{\mathrm{F}} .$$

**Mixture GDA (concise).** We consider a DA scheme that, conditionally on $X$, samples from a $N$-component mixture with

$$G_j = \Lambda_{I_j}(\mathbf{X})^{1/2} Z_j + m_{I_j}(\mathbf{X}), \qquad Z_j \sim \mu, \ I_j \sim \operatorname{Unif}\{1, \dots, N\},$$

where each $\Lambda_k(\mathbf{X}) \succeq 0$, the mixture is centered $\sum_{k=1}^N m_k(\mathbf{X}) = 0$, and $\mu$ is bounded and isotropic with $\mathbb{E}[ZZ^{\top}] = \sigma^2 \mathrm{I}_d$. Assume $\mu$ is $\sigma_Z$–Lipschitz concentrated and, for every $k$, $\Lambda_k(\cdot)$ and $m_k(\cdot)$ are bounded Lipschitz functions (with constants $L_{\Lambda_k}, L_{m_k}$). Let $u$ be the uniform measure on $\{1, \dots, N\}$ and define

$$f_{\mathbf{X}}(z, k) = \Lambda_k(\mathbf{X})^{1/2} z + m_k(\mathbf{X}), \qquad \nu_{\mathbf{X}} = (f_{\mathbf{X}})^{\#}(\mu \otimes u).$$

*Concentration and conditional covariance.* Since $\mu \otimes u$ is Lipschitz concentrated and $f_{\mathbf{X}}$ is Lipschitz on $\operatorname{Supp}(\mu) \times \{1, \dots, N\}$, the pushforward $\nu_{\mathbf{X}}$ is Lipschitz concentrated, so Item (i) holds. Moreover,

$$\mathbb{E}[C_G \mid X] = \mathbb{E}[G_1 G_1^{\top} \mid X] = \frac{1}{N} \sum_{k=1}^N \left( \Lambda_k(\mathbf{X})^{1/2} \mathbb{E}[ZZ^{\top}] \Lambda_k(\mathbf{X})^{1/2} + m_k(\mathbf{X}) m_k(\mathbf{X})^{\top} \right) =: \Lambda_G(\mathbf{X}),$$

and since $\Lambda_k(\cdot)$, $m_k(\cdot)$ are Lipschitz and bounded, so is $\Lambda_G(\cdot)$; hence the second part of **H**4 and Item (ii) of **H**3 follow.

*First part of H4.* By Kantorovich–Rubinstein and i.i.d. structure,

$$W_1\big(\nu_{\mathbf{X}}^{\otimes m}, \nu_{\mathbf{Y}}^{\otimes m}\big) \leq \mathbb{E}\Big\|\big(f_{\mathbf{X}}(Z_j, I_j) - f_{\mathbf{Y}}(Z_j, I_j)\big)_{j=1}^m\Big\|_{\mathrm{F}} \leq \sqrt{m}\left(\mathbb{E}\|f_{\mathbf{X}}(Z, I) - f_{\mathbf{Y}}(Z, I)\|_2^2\right)^{1/2}$$

$$= \sqrt{m}\left(\frac{1}{N}\sum_{k=1}^N \mathbb{E}\big\|(\Lambda_k(\mathbf{X})^{1/2} - \Lambda_k(\mathbf{Y})^{1/2})Z + (m_k(\mathbf{X}) - m_k(\mathbf{Y}))\big\|_2^2\right)^{1/2}$$

$$\leq \sqrt{m}\left(\frac{\sigma_Z^2}{N}\sum_{k=1}^N \|\Lambda_k(\mathbf{X})^{1/2} - \Lambda_k(\mathbf{Y})^{1/2}\|_{\mathrm{F}}^2 + \frac{1}{N}\sum_{k=1}^N \|m_k(\mathbf{X}) - m_k(\mathbf{Y})\|_2^2\right)^{1/2}.$$

If the spectra of $\Lambda_k(\cdot)$ are uniformly bounded below by $K^{-1} > 0$, then by Lemma 3 with $f(t) = \sqrt{t}$,

$$\|\Lambda_k(\mathbf{X})^{1/2} - \Lambda_k(\mathbf{Y})^{1/2}\|_{\mathrm{F}} \leq \frac{\sqrt{K}}{2}\|\Lambda_k(\mathbf{X}) - \Lambda_k(\mathbf{Y})\|_{\mathrm{F}} \leq \frac{L_{\Lambda_k}\sqrt{K}}{2}\|\mathbf{X} - \mathbf{Y}\|_{\mathrm{F}},$$

and $\|m_k(\mathbf{X}) - m_k(\mathbf{Y})\|_2 \leq L_{m_k}\|\mathbf{X} - \mathbf{Y}\|_{\mathrm{F}}$. Thus

$$W_1\big(\nu_{\mathbf{X}}^{\otimes m}, \nu_{\mathbf{Y}}^{\otimes m}\big) \leq \sqrt{m}\left(\frac{\sigma_Z^2 K}{4N}\sum_{k=1}^N L_{\Lambda_k}^2 + \frac{1}{N}\sum_{k=1}^N L_{m_k}^2\right)^{1/2}\|\mathbf{X} - \mathbf{Y}\|_{\mathrm{F}},$$

which proves the first part of **H**4. Stability follows similarly.

**Fixed Gaussian TDA.** Consider the TDA scheme

$$G_j = X_{I_j} + \Lambda^{1/2}Z_j, \qquad Z_j \sim \mathrm{N}(0, \mathrm{I}_d), \quad I_j \sim \mathrm{Unif}\{1, \dots, n\}.$$

By Lemma 2 with $\beta = 1$ (unbiasedness), we obtain

$$\mathbb{E}[C_G \mid X] = C_X + \Lambda.$$

Moreover, the augmentation noise law *is fixed*:

$$\nu_{\mathbf{X}} = \mathrm{N}(0, \Lambda),$$

hence it does not depend on $X$ and therefore **H**3, **H**4, and **H**5 are satisfied trivially in this setting. (Equivalently, since the standard Gaussian is 1-Lipschitz concentrated and the map $z \mapsto \Lambda^{1/2}z$ is $\|\Lambda^{1/2}\|_{\mathrm{op}}$-Lipschitz, $G$ is $\|\Lambda^{1/2}\|_{\mathrm{op}}$-Lipschitz concentrated conditionally on $X$.)

**Random masking TDA.** Consider the augmentation

$$G_j = b_j \odot X_{I_j}, \qquad I_j \sim \mathrm{Unif}\{1, \dots, n\}, \ \ b_j \sim \mathrm{Bernoulli}(1-\rho)^{\otimes d} \ (\text{i.i.d.}),$$

where $\odot$ denotes elementwise product. Assume $X$ is bounded, i.e., $\|X_i\|_2 \leq K$ a.s.

*Concentration and conditional covariance.* Writing

$$\nu_{\mathbf{X}} = (f_{\mathbf{X}})^{\#}\big(\mathrm{Bernoulli}(1-\rho)^{\otimes d} \otimes \mathrm{Unif}\{1, \dots, n\}\big)$$

with $f_{\mathbf{X}}(b, i) = b \odot X_i$, the map $f_{\mathbf{X}}$ is Lipschitz on the compact domain (with a constant independent of $\mathbf{X}$ by boundedness of $X$). Hence, by Proposition 5, $G$ is Lipschitz concentrated conditionally on $X$, i.e. Item (i) holds. Moreover, Lemma 2 with $\beta = 1 - \rho$ yields

$$\mathbb{E}[C_G \mid X] = (1-\rho)C_X + \Lambda_G(\mathbf{X}), \qquad \Lambda_G(\mathbf{X}) = \rho(1-\rho)\,\mathrm{diag}(C_{\mathbf{X}}),$$

so Item (ii) also holds.

*Smoothness.* Since $\Lambda_G(\mathbf{X})$ is a composition of Lipschitz maps on the bounded set $[-K, K]^{d \times n}$, it is Lipschitz; this proves the second part of **H**4. For the first part, couple $(\nu_{\mathbf{X}}^{\otimes m}, \nu_{\mathbf{Y}}^{\otimes m})$ by using the same $(I_j, b_j)$ on both sides. Then

$$W_1\big(\nu_{\mathbf{X}}^{\otimes m}, \nu_{\mathbf{Y}}^{\otimes m}\big) \leq \mathbb{E}\Big\|\big(b_j \odot X_{I_j} - b_j \odot Y_{I_j}\big)_{j=1}^m\Big\|_{\mathrm{F}}$$

$$\leq \sqrt{m}\left(\mathbb{E}\|b_1 \odot (X_{I_1} - Y_{I_1})\|_2^2\right)^{1/2}$$

$$= \sqrt{m}\left(\frac{1}{n}\sum_{i=1}^n \mathbb{E}\|b_1 \odot (X_i - Y_i)\|_2^2\right)^{1/2} = \sqrt{m(1-p)}\left(\frac{1}{n}\sum_{i=1}^n \|X_i - Y_i\|_2^2\right)^{1/2}$$

$$\leq \sqrt{m(1-p)}\,\|\mathbf{X} - \mathbf{Y}\|_{\mathrm{F}},$$

since $\mathbb{E}[b_{1,k}^2] = \mathbb{E}[b_{1,k}] = 1 - p$ for each coordinate $k$. This proves the first part of **H**4. Stability follows by the same argument.

# B   Proof of theorem 1

This appendix provides the proof of Theorem 1. Along the way, we introduce several auxiliary lemmas on concentration for transformations of $X$ under **H** 6. appendix B.1 establishes concentration bounds for random variables of the form $f(X)\, \mathbb{1}_{\mathsf{E}}(X)$ when $f$ is Lipschitz only on a subset $\mathsf{E} \subset \mathbb{R}^{d \times n}$ (not necessarily on all of $\mathbb{R}^{d \times n}$). We show that the Lipschitz concentration of $X$ still yields sharp control of $f(X)\, \mathbb{1}_{\mathsf{E}}(X)$. appendix B.2 then analyzes quadratic forms $X_1^\top M(X^-)\, X_1$, where $M : \mathbb{R}^{d \times n} \to \mathbb{R}^{d \times d}$ and $X$ satisfies **H** 6. The derivations rely on the Hanson–Wright inequality (see [LC18, Remark 2.31]). Building on these results, appendix B.3 derives a deterministic equivalent, in the spirit of [Cho22, Thm. 6.16], under **H** 2, which allows regularizations arbitrarily close to $0$. Finally, appendix B.4 combines the above ingredients to complete the proof of theorem 1.

For simplicity, throughout this section we set $\sigma_X = 1$.

## B.1   Some sub Gaussian concentration bounds

In this section we study random variables of the form $f(X)\, \mathbb{1}_{\mathsf{E}}(X)$, where $f$ is Lipschitz only on a subset $\mathsf{E} \subset \mathbb{R}^{d \times n}$ (and not necessarily on all of $\mathbb{R}^{d \times n}$). We show that $f(X)\, \mathbb{1}_{\mathsf{E}}(X)$ still admits sub-Gaussian tails and we derive a tight upper bound on its sub-Gaussian norm in proposition 6.

We begin with a standard Lipschitz extension lemma (see [Kir34]); for completeness, we include a short proof.

**Lemma 4.** *Let $f : \mathsf{E} \to \mathbb{R}$ be $\mathsf{L}_f$-Lipschitz on $\mathsf{E} \subset \mathbb{R}^{d \times n}$. Define*

$$\tilde{f}(\mathbf{X}) \ := \ \inf_{\mathbf{Y} \in \mathsf{E}} \left\{ f(\mathbf{Y}) + \mathsf{L}_f\, \|\mathbf{X} - \mathbf{Y}\|_{\mathrm{F}} \right\}, \qquad \mathbf{X} \in \mathbb{R}^{d \times n}.$$

*Then $\tilde{f}$ is $\mathsf{L}_f$-Lipschitz on $\mathbb{R}^{d \times n}$ and $\tilde{f}(\mathbf{X}) = f(\mathbf{X})$ for all $\mathbf{X} \in \mathsf{E}$.*

*Proof.* Fix $\mathbf{X}, \mathbf{X}' \in \mathbb{R}^{d \times n}$ and any $\mathbf{Y} \in \mathsf{E}$. By the triangle inequality,

$$f(\mathbf{Y}) + \mathsf{L}_f \|\mathbf{X} - \mathbf{Y}\|_{\mathrm{F}} \ \le \ f(\mathbf{Y}) + \mathsf{L}_f \|\mathbf{X}' - \mathbf{Y}\|_{\mathrm{F}} + \mathsf{L}_f \|\mathbf{X} - \mathbf{X}'\|_{\mathrm{F}}.$$

Taking the infimum over $\mathbf{Y} \in \mathsf{E}$ yields $\tilde{f}(\mathbf{X}) \le \tilde{f}(\mathbf{X}') + \mathsf{L}_f \|\mathbf{X} - \mathbf{X}'\|_{\mathrm{F}}$. Swapping $\mathbf{X}$ and $\mathbf{X}'$ gives the reverse inequality, hence $\tilde{f}$ is $\mathsf{L}_f$-Lipschitz.

For $\mathbf{X} \in \mathsf{E}$, the Lipschitz property of $f$ on $\mathsf{E}$ implies $f(\mathbf{X}) \le f(\mathbf{Y}) + \mathsf{L}_f \|\mathbf{X} - \mathbf{Y}\|_{\mathrm{F}}$ for every $\mathbf{Y} \in \mathsf{E}$. Taking the infimum over $\mathbf{Y}$ gives $\tilde{f}(\mathbf{X}) \ge f(\mathbf{X})$, while choosing $\mathbf{Y} = \mathbf{X}$ gives $\tilde{f}(\mathbf{X}) \le f(\mathbf{X})$. Thus $\tilde{f}(\mathbf{X}) = f(\mathbf{X})$ on $\mathsf{E}$. $\qquad\square$

Leveraging lemma 4, we now prove the announced concentration bound for $f(X)\, \mathbb{1}_{\mathsf{E}}(X)$ when $f$ is only Lipschitz on $\mathsf{E}$.

**Proposition 6.** *Let $\mathsf{E} \subset \mathbb{R}^{d \times n}$ and $f : \mathsf{E} \to \mathbb{R}$ be $\mathsf{L}_f$-Lipschitz. Assume $\|f\|_\infty < \infty$ and that $X$ satisfies H6. Then $f(X)\, \mathbb{1}_{\mathsf{E}}(X) - \mathbb{E}[f(X)\, \mathbb{1}_{\mathsf{E}}(X)]$ is sub-Gaussian with variance proxy*

$$\sigma_{f,\mathsf{E}}^2 \ \lesssim \ \mathbb{P}(X \in \mathsf{E})^2\, \mathsf{L}_f^2 \ + \ \|f\|_\infty^2\, \sigma_{\mathsf{E}}^2 \,,$$

*where for $p \in (0,1)$,*

$$\sigma(p) \ = \ \sqrt{\frac{1 - 2p}{2 \ln\big((1-p)/p\big)}}\,, \qquad \sigma_{\mathsf{E}} = \sigma\big(\mathbb{P}(X \in \mathsf{E})\big).$$

*Proof.* By lemma 4, extend $f$ to $\tilde{f} : \mathbb{R}^{d \times n} \to \mathbb{R}$ with $\mathrm{Lip}(\tilde{f}) = \mathsf{L}_f$ and $\tilde{f}|_{\mathsf{E}} = f$. Define the clipped map

$$g(\mathbf{X}) \ = \ \max\big\{\, \min\{\tilde{f}(\mathbf{X}), \|f\|_\infty\}, -\|f\|_\infty \big\}.$$

Then $g$ is $\mathsf{L}_f$-Lipschitz, $\|g\|_\infty = \|f\|_\infty$, and $g = f$ on $\mathsf{E}$, hence $f(X)\, \mathbb{1}_{\mathsf{E}}(X) = g(X)\, \mathbb{1}_{\mathsf{E}}(X)$ a.s. Write

$$\bar{g}(X) = g(X) - \mathbb{E}[g(X)], \qquad \bar{\mathbb{1}}_{\mathsf{E}}(X) = \mathbb{1}_{\mathsf{E}}(X) - \mathbb{P}(X \in \mathsf{E}), \qquad p := \mathbb{P}(X \in \mathsf{E}).$$

A direct decomposition gives

$$g(X)\,\mathbb{1}_{\mathsf{E}}(X) - \mathbb{E}[g(X)\,\mathbb{1}_{\mathsf{E}}(X)] = \underbrace{\bar{g}(X)\,\bar{\mathbb{1}}_{\mathsf{E}}(X) - \mathrm{Cov}(g(X),\mathbb{1}_{\mathsf{E}}(X))}_{=:W} + p\,\bar{g}(X) + \mathbb{E}[g(X)]\,\bar{\mathbb{1}}_{\mathsf{E}}(X).$$

(15)

Applying Hölder with exponents $(3,3,3)$ to the MGF of the sum in (15) yields

$$\mathbb{E}\big[\exp\big\{s(g\mathbb{1}_{\mathsf{E}} - \mathbb{E}[g\mathbb{1}_{\mathsf{E}}])\big\}\big] \le \mathbb{E}[\exp\{3sW\}]^{1/3}\,\mathbb{E}[\exp\{3s\,p\,\bar{g}\}]^{1/3}\,\mathbb{E}[\exp\{3s\,\mathbb{E}[g]\,\bar{\mathbb{1}}_{\mathsf{E}}\}]^{1/3}. \quad (16)$$

*Two easy sub-Gaussian factors.* Since $X$ is Lipschitz concentrated and $g$ is $\mathsf{L}_f$-Lipschitz, there exists a universal $c > 0$ such that

$$\mathbb{E}[\exp\{3s\,p\,\bar{g}(X)\}]^{1/3} \le \exp\{c\,s^2\,p^2\,\mathsf{L}_f^2\}. \tag{17}$$

Moreover, $\bar{\mathbb{1}}_{\mathsf{E}}(X)$ is a centered Bernoulli random variable with sub-Gaussian proxy $\sigma_{\mathsf{E}} = \sigma(p)$ (see [BM13, Thm. 2.1]), and $|\mathbb{E}[g(X)]| \le \|f\|_\infty$, hence

$$\mathbb{E}[\exp\{3s\,\mathbb{E}[g]\,\bar{\mathbb{1}}_{\mathsf{E}}(X)\}]^{1/3} \le \exp\{c\,s^2\,\|f\|_\infty^2\,\sigma_{\mathsf{E}}^2\}. \tag{18}$$

*The product term $W$ is sub-Gaussian (detailed $\psi_2$ bound).* We prove that $W = \bar{g}\,\bar{\mathbb{1}}_{\mathsf{E}} - \mathrm{Cov}(g,\mathbb{1}_{\mathsf{E}})$ is sub-Gaussian by exhibiting a scale $S > 0$ with $\mathbb{E}\exp\{W^2/S^2\} \le 2$, i.e. $\|W\|_{\psi_2} \le S$.

First, using $(u - v)^2 \le 2u^2 + 2v^2$ and $|\bar{g}| \le |g| + |\mathbb{E}g| \le 2\|f\|_\infty$,

$$W^2 \le 2\,\bar{g}^2\,\bar{\mathbb{1}}_{\mathsf{E}}^2 + 2\,\mathrm{Cov}(g,\mathbb{1}_{\mathsf{E}})^2 \le 8\|f\|_\infty^2\,\bar{\mathbb{1}}_{\mathsf{E}}^2 + 2\,\mathrm{Cov}(g,\mathbb{1}_{\mathsf{E}})^2. \tag{19}$$

Next, by Cauchy–Schwarz,

$$|\mathrm{Cov}(g,\mathbb{1}_{\mathsf{E}})| \le \sqrt{\mathrm{Var}(g)}\,\sqrt{\mathrm{Var}(\mathbb{1}_{\mathsf{E}})}.$$

Since $\bar{g}$ is sub-Gaussian with proxy $\lesssim \mathsf{L}_f$, there exists a universal constant $C_0$ such that $\mathrm{Var}(g) \le C_0\,\mathsf{L}_f^2$; also $\mathrm{Var}(\mathbb{1}_{\mathsf{E}}) = p(1 - p)$. Hence

$$\mathrm{Cov}(g,\mathbb{1}_{\mathsf{E}})^2 \le C_0\,\mathsf{L}_f^2\,p(1 - p). \tag{20}$$

Fix $\alpha \in (0,1]$ and set

$$S^2 := \frac{8\,\|f\|_\infty^2\,\sigma_{\mathsf{E}}^2}{\alpha} \quad\Longrightarrow\quad \frac{8\,\|f\|_\infty^2}{S^2} = \frac{\alpha}{\sigma_{\mathsf{E}}^2}. \tag{21}$$

Using (19)–(21),

$$\mathbb{E}\exp\Big\{\frac{W^2}{S^2}\Big\} \le \mathbb{E}\exp\Big\{\frac{8\|f\|_\infty^2}{S^2}\,\bar{\mathbb{1}}_{\mathsf{E}}^2\Big\}\exp\Big\{\frac{2\,\mathrm{Cov}(g,\mathbb{1}_{\mathsf{E}})^2}{S^2}\Big\}$$
$$= \mathbb{E}\exp\Big\{\alpha\,\frac{\bar{\mathbb{1}}_{\mathsf{E}}^2}{\sigma_{\mathsf{E}}^2}\Big\}\exp\Big\{\frac{\alpha\,\mathrm{Cov}(g,\mathbb{1}_{\mathsf{E}})^2}{4\,\|f\|_\infty^2\,\sigma_{\mathsf{E}}^2}\Big\}. \tag{22}$$

By the definition of the $\psi_2$-norm of $\bar{\mathbb{1}}_{\mathsf{E}}$, $\mathbb{E}\exp\{\bar{\mathbb{1}}_{\mathsf{E}}^2/\sigma_{\mathsf{E}}^2\} \le 2$. For $0 < \alpha \le 1$, Lyapunov's inequality gives

$$\mathbb{E}\exp\Big\{\alpha\,\frac{\bar{\mathbb{1}}_{\mathsf{E}}^2}{\sigma_{\mathsf{E}}^2}\Big\} \le \big(\mathbb{E}\exp\{\bar{\mathbb{1}}_{\mathsf{E}}^2/\sigma_{\mathsf{E}}^2\}\big)^\alpha \le 2^\alpha.$$

Using (20) in the second factor of (22),

$$\exp\Big\{\frac{\alpha\,\mathrm{Cov}(g,\mathbb{1}_{\mathsf{E}})^2}{4\,\|f\|_\infty^2\,\sigma_{\mathsf{E}}^2}\Big\} \le \exp\Big\{\alpha\,C_1\,\frac{\mathsf{L}_f^2\,p(1 - p)}{\|f\|_\infty^2\,\sigma_{\mathsf{E}}^2}\Big\} \tag{23}$$

for some absolute $C_1 > 0$.

Combining (22)–(23) gives

$$\mathbb{E}\exp\Big\{\frac{W^2}{S^2}\Big\} \le 2^\alpha\exp\{\alpha A\}, \qquad A := C_1\,\frac{\mathsf{L}_f^2\,p(1 - p)}{\|f\|_\infty^2\,\sigma_{\mathsf{E}}^2}.$$

Choose
$$\alpha^\star := \Big(1 + \frac{A}{\ln 2}\Big)^{-1} \in (0, 1],$$

so that $2^{\alpha^\star} \exp\{\alpha^\star A\} \le 2$. With $S^2$ as in (21) at $\alpha = \alpha^\star$ we obtain

$$\mathbb{E}\exp\Big\{\frac{W^2}{S^2}\Big\} \le 2, \qquad \text{hence} \qquad \|W\|_{\psi_2}^2 \;\le\; S^2 \;\le\; C\Big(\|f\|_\infty^2\, \sigma_{\mathsf{E}}^2 + \mathsf{L}_f^2\, p(1-p)\Big),$$

for a universal constant $C$ (use $1/\alpha^\star = 1 + A/\ln 2$ and the definition of $A$). By the standard sub-Gaussian MGF bound, there exists a universal $c > 0$ with

$$\mathbb{E}[\exp\{3sW\}]^{1/3} \le \exp\{c\, s^2\, \|W\|_{\psi_2}^2\} \le \exp\{c\, s^2(\|f\|_\infty^2\, \sigma_{\mathsf{E}}^2 + \mathsf{L}_f^2\, p(1-p))\}. \qquad (24)$$

*Conclusion.* Combining (16), (17), (18), and (24),

$$\mathbb{E}\big[\exp\big\{s\big(f(X)\,\mathbb{1}_{\mathsf{E}}(X) - \mathbb{E}[f(X)\,\mathbb{1}_{\mathsf{E}}(X)]\big)\big\}\big] \le \exp\Big\{c\, s^2\Big(p^2\, \mathsf{L}_f^2 + \|f\|_\infty^2\, \sigma_{\mathsf{E}}^2\Big)\Big\},$$

for a universal constant $c > 0$. This is the desired sub-Gaussian MGF bound and proves the claimed variance proxy. $\qquad\qquad\square$

## B.2 Concentration bounds for random quadratic forms

We now study the concentration of random quadratic forms of the type $X_1^\top M(X^-)X_1$, where $M : \mathbb{R}^{d\times n} \to \mathbb{R}^{d\times d}$. Such quantities naturally appear in the proof of Proposition 1. We prove the following lemma.

**Lemma 5.** *Let $X \in \mathbb{R}^{d\times n}$ satisfy **H**6. Then there exists a universal constant $c > 0$ such that for any $M : \mathbb{R}^{d\times n} \to \mathbb{R}^{d\times d}$,*

$$\mathrm{Var}\big(X_1^\top M(X^-)X_1\big) \le \frac{2}{c}\, \mathbb{E}\bigg[\|M(X^-)\|_{\mathrm{F}}^2 + \frac{2}{c}\|M(X^-)\|_{\mathrm{op}}^2\bigg] + \mathrm{Var}\big(\mathrm{tr}\big(\Sigma_X M(X^-)\big)\big)\ .$$

*Proof.* By the law of total variance applied to $(X_1, X^-)$,

$$\begin{aligned}
\mathrm{Var}\big(X_1^\top M(X^-)X_1\big) &= \mathbb{E}\big[\mathrm{Var}\big(X_1^\top M(X^-)X_1 \,\big|\, X^-\big)\big] + \mathrm{Var}\big(\mathbb{E}\big[X_1^\top M(X^-)X_1 \,\big|\, X^-\big]\big) \\
&= \mathbb{E}\big[\mathrm{Var}\big(X_1^\top M(X^-)X_1 \,\big|\, X^-\big)\big] + \mathrm{Var}\big(\mathrm{tr}\big(\Sigma_X M(X^-)\big)\big)\ ,
\end{aligned}$$

where we used $\mathbb{E}[X_1^\top M(X^-)X_1 \mid X^-] = \mathrm{tr}(\Sigma_X M(X^-))$.

It remains to control the first term. Conditionally on $X^-$, the Hanson–Wright inequality (see [LC18, Remark 2.31]) yields, almost surely,

$$\mathbb{P}\big(\big|X_1^\top M(X^-)X_1 - \mathrm{tr}\big(M(X^-)\Sigma_X\big)\big| \ge t \,\big|\, X^-\big) \le 2\exp\Big(-c\min\Big(\frac{t^2}{\|M(X^-)\|_{\mathrm{F}}^2},\, \frac{t}{\|M(X^-)\|_{\mathrm{op}}}\Big)\Big).$$

Writing the conditional variance in integral form,

$$\begin{aligned}
\mathrm{Var}\big(X_1^\top M(X^-)X_1 \,\big|\, X^-\big) &= \int_0^\infty \mathbb{P}\Big(\big|X_1^\top M(X^-)X_1 - \mathrm{tr}\big(\Sigma_X M(X^-)\big)\big| \ge \sqrt{t}\,\Big|\, X^-\Big)\,\mathrm{d}t \\
&\le 2\int_0^\infty \exp\Big(-c\,\frac{t}{\|M(X^-)\|_{\mathrm{F}}^2}\Big)\,\mathrm{d}t + 2\int_0^\infty \exp\Big(-c\,\frac{\sqrt{t}}{\|M(X^-)\|_{\mathrm{op}}}\Big)\,\mathrm{d}t \\
&= \frac{2\|M(X^-)\|_{\mathrm{F}}^2}{c} + \frac{4\|M(X^-)\|_{\mathrm{op}}^2}{c^2}\ ,
\end{aligned}$$

where we used the change of variables $u = c\sqrt{t}/\|M(X^-)\|_{\mathrm{op}}$ in the second integral. Taking expectations in $X^-$ gives

$$\mathbb{E}\big[\mathrm{Var}\big(X_1^\top M(X^-)X_1 \,\big|\, X^-\big)\big] \le \frac{2}{c}\, \mathbb{E}\bigg[\|M(X^-)\|_{\mathrm{F}}^2 + \frac{2}{c}\|M(X^-)\|_{\mathrm{op}}^2\bigg],$$

which, combined with the total-variance decomposition above, completes the proof. $\qquad\square$

In the special case where the map $M : \mathbb{R}^{d\times n} \to \mathbb{R}^{d\times d}$ is Lipschitz on $\mathsf{A}_\eta$ (with $\mathsf{A}_\eta$ defined in **H**2), Lemma 5 yields:

**Proposition 7.** *Let $X \in \mathbb{R}^{d\times n}$ satisfy **H**6 and **H**2. For any functions $M_1 : \mathsf{A}_\eta \to \mathbb{R}^{d\times d}$ and $M_2 : \mathsf{A}_\eta \to \mathbb{R}^{d\times d}$ that are respectively $\mathsf{L}_1$- and $\mathsf{L}_2$-Lipschitz and bounded, and any $\mathbf{B} \in \mathbb{R}^{d\times d}$ with $\|\mathbf{B}\|_{\mathrm{F}} = 1$, we have*

$$\mathrm{Var}\big(X_1^\top M_1(X^-)X_1 \, \mathbb{1}_{\mathsf{A}_\eta}(X)\big) \lesssim d \, \|\Sigma_X\|_{\mathrm{op}}^2 \big\{ \mathsf{L}_1^2 + \|M_1\|_\infty^2 \, (1 + c_X^{-1}) \big\} \;,$$

$$\mathrm{Var}\big(X_1^\top M_1(X^-)\mathbf{B}\, X_1 \, \mathbb{1}_{\mathsf{A}_\eta}(X)\big) \lesssim \|\Sigma_X\|_{\mathrm{op}}^2 \big\{ \mathsf{L}_1^2 + \|M_1\|_\infty^2 \, (1 + c_X^{-1}) \big\} \;,$$

$$\mathrm{Var}\big(X_1^\top M_1(X^-)\mathbf{B}\, M_2(X^-)X_1 \, \mathbb{1}_{\mathsf{A}_\eta}(X)\big) \lesssim \|\Sigma_X\|_{\mathrm{op}}^2 \Big\{ \big(\|M_1\|_\infty \mathsf{L}_2 + \|M_2\|_\infty \mathsf{L}_1\big)^2 + \|M_1\|_\infty^2 \|M_2\|_\infty^2 \, (1 + c_X^{-1}) \Big\} \;,$$

*where $\|M_i\|_\infty = \big\| \|M_i(\cdot)\|_{\mathrm{op}} \big\|_\infty$.*

*Proof.* We treat the three cases in the same way. By Lemma 5 and since $\mathbb{1}_{\mathsf{A}_\eta}(X)$ is $\sigma(X^-)$-measurable,

$$\mathrm{Var}\big(X_1^\top M_1(X^-)X_1 \, \mathbb{1}_{\mathsf{A}_\eta}(X)\big)$$
$$\leq \frac{2}{c} \, \mathbb{E}\bigg[ \|M_1(X^-)\|_{\mathrm{F}}^2 \, \mathbb{1}_{\mathsf{A}_\eta}(X) + \frac{2}{c} \|M_1(X^-)\|_{\mathrm{op}}^2 \, \mathbb{1}_{\mathsf{A}_\eta}(X) \bigg] + \mathrm{Var}\big(\mathrm{tr}\big(\Sigma_X M_1(X^-) \, \mathbb{1}_{\mathsf{A}_\eta}(X)\big)\big)$$
$$\leq \frac{2}{c}\bigg( d \|M_1\|_\infty^2 + \frac{2}{c}\|M_1\|_\infty^2 \bigg) + \mathrm{Var}\big(\mathrm{tr}\big(\Sigma_X M_1(X^-) \, \mathbb{1}_{\mathsf{A}_\eta}(X)\big)\big) \lesssim d \|M_1\|_\infty^2 + \mathrm{Var}\big(\mathrm{tr}\big(\Sigma_X M_1(X^-) \, \mathbb{1}_{\mathsf{A}_\eta}(X)\big)\big).$$

Similarly,

$$\mathrm{Var}\big(X_1^\top M_1(X^-)\mathbf{B}\, X_1 \, \mathbb{1}_{\mathsf{A}_\eta}(X)\big)$$
$$\leq \frac{2}{c} \, \mathbb{E}\bigg[ \|M_1(X^-)\mathbf{B}\|_{\mathrm{F}}^2 \, \mathbb{1}_{\mathsf{A}_\eta}(X) + \frac{2}{c} \|M_1(X^-)\mathbf{B}\|_{\mathrm{op}}^2 \, \mathbb{1}_{\mathsf{A}_\eta}(X) \bigg] + \mathrm{Var}\big(\mathrm{tr}\big(\Sigma_X M_1(X^-)\mathbf{B} \, \mathbb{1}_{\mathsf{A}_\eta}(X)\big)\big)$$
$$\leq \frac{2}{c}\bigg( \|M_1\|_\infty^2 + \frac{2}{c}\|M_1\|_\infty^2 \bigg) + \mathrm{Var}\big(\mathrm{tr}\big(\Sigma_X M_1(X^-)\mathbf{B} \, \mathbb{1}_{\mathsf{A}_\eta}(X)\big)\big) \lesssim \|M_1\|_\infty^2 + \mathrm{Var}\big(\mathrm{tr}\big(\Sigma_X M_1(X^-)\mathbf{B} \, \mathbb{1}_{\mathsf{A}_\eta}(X)\big)\big).$$

Finally,

$$\mathrm{Var}\big(X_1^\top M_1(X^-)\mathbf{B}\, M_2(X^-)X_1 \, \mathbb{1}_{\mathsf{A}_\eta}(X)\big)$$
$$\leq \frac{2}{c} \, \mathbb{E}\bigg[ \|M_1(X^-)\mathbf{B}\, M_2(X^-)\|_{\mathrm{F}}^2 \, \mathbb{1}_{\mathsf{A}_\eta}(X) + \frac{2}{c} \|M_1(X^-)\mathbf{B}\, M_2(X^-)\|_{\mathrm{op}}^2 \, \mathbb{1}_{\mathsf{A}_\eta}(X) \bigg]$$
$$\qquad + \mathrm{Var}\big(\mathrm{tr}\big(\Sigma_X M_1(X^-)\mathbf{B}\, M_2(X^-) \, \mathbb{1}_{\mathsf{A}_\eta}(X)\big)\big)$$
$$\leq \frac{2}{c}\bigg( \|M_1\|_\infty^2 \|M_2\|_\infty^2 + \frac{2}{c}\|M_1\|_\infty^2 \|M_2\|_\infty^2 \bigg) + \mathrm{Var}\big(\mathrm{tr}\big(\Sigma_X M_1(X^-)\mathbf{B}\, M_2(X^-) \, \mathbb{1}_{\mathsf{A}_\eta}(X)\big)\big)$$
$$\lesssim \|M_1\|_\infty^2 \|M_2\|_\infty^2 + \mathrm{Var}\big(\mathrm{tr}\big(\Sigma_X M_1(X^-)\mathbf{B}\, M_2(X^-) \, \mathbb{1}_{\mathsf{A}_\eta}(X)\big)\big).$$

It remains to bound the trace-variance terms. By Cauchy–Schwarz, for all $\mathbf{X}, \mathbf{Y} \in \mathsf{A}_\eta$,

$$\big| \mathrm{tr}\big(\Sigma_X M_1(\mathbf{X}^-)\big) - \mathrm{tr}\big(\Sigma_X M_1(\mathbf{Y}^-)\big)\big| \leq \sqrt{d}\, \|\Sigma_X\|_{\mathrm{op}} \, \mathsf{L}_1 \, \|\mathbf{X} - \mathbf{Y}\|_{\mathrm{F}} \;,$$

$$\big| \mathrm{tr}\big(\Sigma_X M_1(\mathbf{X}^-)\mathbf{B}\big) - \mathrm{tr}\big(\Sigma_X M_1(\mathbf{Y}^-)\mathbf{B}\big)\big| \leq \|\Sigma_X\|_{\mathrm{op}} \, \mathsf{L}_1 \, \|\mathbf{X} - \mathbf{Y}\|_{\mathrm{F}} \;,$$

and

$$\big| \mathrm{tr}\big(\Sigma_X M_1(\mathbf{X}^-)\mathbf{B}\, M_2(\mathbf{X}^-)\big) - \mathrm{tr}\big(\Sigma_X M_1(\mathbf{Y}^-)\mathbf{B}\, M_2(\mathbf{Y}^-)\big)\big|$$
$$\leq \|\Sigma_X\|_{\mathrm{op}}\big(\|M_1\|_\infty \mathsf{L}_2 + \|M_2\|_\infty \mathsf{L}_1\big) \|\mathbf{X} - \mathbf{Y}\|_{\mathrm{F}} \;.$$

Therefore, by Proposition 6 and standard sub-Gaussian variance bounds,

$$\mathrm{Var}\big(\mathrm{tr}\big(\Sigma_X M_1(X^-)\big)\big) \lesssim d \, \|\Sigma_X\|_{\mathrm{op}}^2 \, \mathsf{L}_1^2 + d \, \|M_1\|_\infty^2 + d^2 \, \|\Sigma_X\|_{\mathrm{op}}^2 \, \|M_1\|_\infty^2 \, \sigma_{\mathsf{A}_\eta}^2 \;,$$

$$\mathrm{Var}\big(\mathrm{tr}(\Sigma_X M_1(X^-)\mathbf{B})\big) \lesssim \|\Sigma_X\|_{\mathrm{op}}^2 \, \mathsf{L}_1^2 + \|M_1\|_\infty^2 + d\,\|\Sigma_X\|_{\mathrm{op}}^2 \|M_1\|_\infty^2\, \sigma_{\mathsf{A}_\eta}^2 \,,$$

and

$$\mathrm{Var}\big(\mathrm{tr}(\Sigma_X M_1(X^-)\mathbf{B}\, M_2(X^-))\big) \lesssim \|\Sigma_X\|_{\mathrm{op}}^2 \big(\|M_1\|_\infty \mathsf{L}_2 + \|M_2\|_\infty \mathsf{L}_1\big)^2 + \|M_1\|_\infty^2 \|M_2\|_\infty^2$$
$$+ d\,\|\Sigma_X\|_{\mathrm{op}}^2 \|M_1\|_\infty^2\, \sigma_{\mathsf{A}_\eta}^2 \,.$$

Finally, by **H**2,

$$\sigma_{\mathsf{A}_\eta}^2 = \frac{1 - 2\mathbb{P}(X \in \mathsf{A}_\eta)}{2\log\left(\frac{1-\mathbb{P}(X\in\mathsf{A}_\eta)}{\mathbb{P}(X\in\mathsf{A}_\eta)}\right)} \lesssim \frac{1}{n\,c_X} \,,$$

since $1 - \mathbb{P}(\mathsf{A}_\eta) \lesssim \exp(-c_X n)$. We also note that **H**2 forces $d < n$ (otherwise $C_X$ would be rank-deficient a.s., yielding $\mathbb{P}(X \in \mathsf{A}_\eta) = 0$ for all $\eta > 0$). Plugging the bound on $\sigma_{\mathsf{A}_\eta}^2$ above into the previous displays gives the stated upper bounds. $\qquad\square$

### B.3 A deterministic equivalent for $\mathrm{R}_X(\lambda)$, with arbitrarly small regularization parameter

We first show that the resolvent map is locally Lipschitz, and in particular Lipschitz on $\mathsf{A}_\eta$.

**Lemma 6.** *Let* $\mathbf{X}_1, \mathbf{X}_2 \in \mathbb{R}^{d\times n}$ *and let* $\mathbf{D} \succeq 0$. *Assume that for* $i \in \{1,2\}$, $\lambda_d(C_{\mathbf{X}_i} + \mathbf{D}) \geq \epsilon > 0$.
*Then*

$$\big\|(C_{\mathbf{X}_1} + \mathbf{D})^{-1} - (C_{\mathbf{X}_2} + \mathbf{D})^{-1}\big\|_{\mathrm{F}} \;\leq\; \frac{2}{\sqrt{n}\,\epsilon^3} \,\|\mathbf{X}_1 - \mathbf{X}_2\|_{\mathrm{F}} \,.$$

*Proof.* Using $A^{-1} - B^{-1} = A^{-1}(B-A)B^{-1}$ and $\|AB\|_{\mathrm{F}} \leq \|A\|_{\mathrm{op}}\|B\|_{\mathrm{F}}$,

$$\big\|(C_{\mathbf{X}_1} + \mathbf{D})^{-1} - (C_{\mathbf{X}_2} + \mathbf{D})^{-1}\big\|_{\mathrm{F}} = \big\|(C_{\mathbf{X}_1} + \mathbf{D})^{-1}\big(C_{\mathbf{X}_2} - C_{\mathbf{X}_1}\big)(C_{\mathbf{X}_2} + \mathbf{D})^{-1}\big\|_{\mathrm{F}}$$
$$= \frac{1}{n}\big\|(C_{\mathbf{X}_1} + \mathbf{D})^{-1}\big(\mathbf{X}_1(\mathbf{X}_1 - \mathbf{X}_2)^\top + (\mathbf{X}_1 - \mathbf{X}_2)\mathbf{X}_2^\top\big)(C_{\mathbf{X}_2} + \mathbf{D})^{-1}\big\|_{\mathrm{F}} \,.$$

Using $\|UV^\top\|_{\mathrm{F}} \leq \|U\|_{\mathrm{op}}\|V\|_{\mathrm{F}}$ and the triangle inequality,

$$\cdots \;\leq\; \frac{1}{n}\Big(\|(C_{\mathbf{X}_1} + \mathbf{D})^{-1}\mathbf{X}_1\|_{\mathrm{op}}\, \|(C_{\mathbf{X}_2} + \mathbf{D})^{-1}\|_{\mathrm{op}} + \|(C_{\mathbf{X}_2} + \mathbf{D})^{-1}\mathbf{X}_2\|_{\mathrm{op}}\, \|(C_{\mathbf{X}_1} + \mathbf{D})^{-1}\|_{\mathrm{op}}\Big)\|\mathbf{X}_1 - \mathbf{X}_2\|_{\mathrm{F}}.$$

Since $\lambda_d(C_{\mathbf{X}_i} + \mathbf{D}) \geq \epsilon$, we have $\|(C_{\mathbf{X}_i} + \mathbf{D})^{-1}\|_{\mathrm{op}} \leq \epsilon^{-1}$. Moreover,

$$\frac{1}{\sqrt{n}}\|(C_{\mathbf{X}_i} + \mathbf{D})^{-1}\mathbf{X}_i\|_{\mathrm{op}} = \sqrt{\lambda_d\Big((C_{\mathbf{X}_i} + \mathbf{D})^{-1}\frac{\mathbf{X}_i\mathbf{X}_i^\top}{n}(C_{\mathbf{X}_i} + \mathbf{D})^{-1}\Big)}$$
$$= \sqrt{\lambda_d((C_{\mathbf{X}_i} + \mathbf{D})^{-1}C_{\mathbf{X}_i}(C_{\mathbf{X}_i} + \mathbf{D})^{-1})} \;\leq\; \sqrt{\lambda_d((C_{\mathbf{X}_i} + \mathbf{D})^{-1})} \;\leq\; \epsilon^{-1/2} \,,$$

where we used $C_{\mathbf{X}_i} \preceq C_{\mathbf{X}_i} + \mathbf{D}$. Plugging these bounds into the previous display yields

$$\big\|(C_{\mathbf{X}_1} + \mathbf{D})^{-1} - (C_{\mathbf{X}_2} + \mathbf{D})^{-1}\big\|_{\mathrm{F}} \;\leq\; \frac{2}{\sqrt{n}}\,\epsilon^{-1/2}\,\epsilon^{-1}\,\|\mathbf{X}_1 - \mathbf{X}_2\|_{\mathrm{F}} = \frac{2}{\sqrt{n}\,\epsilon^3}\,\|\mathbf{X}_1 - \mathbf{X}_2\|_{\mathrm{F}},$$

as claimed. $\qquad\square$

**In particular.** On $\mathsf{A}_\eta = \{\mathbf{X} : \lambda_d(C_{\mathbf{X}}) \geq \eta\}$ (take $\mathbf{D} = 0$), the map $\mathbf{X} \mapsto (C_{\mathbf{X}} + \lambda\,\mathrm{I}_d)^{-1}$ is Lipschitz with constant $2/\sqrt{n(\eta + \lambda)^3}$, for all $\lambda \geq 0$.

Define, for any $\mathfrak{b} \in [1, \infty)$ and any matrix $\mathbf{D} \in \mathbb{R}^{d\times d}$,

$$\bar{\mathrm{R}}_X^{\mathfrak{b}}(\mathbf{D}) \;:=\; \left(\tfrac{\Sigma_X}{\mathfrak{b}} + \mathbf{D}\right)^{-1} \,.$$

We provide two choices of the parameter $\mathfrak{b}$ for which $\bar{\mathrm{R}}_X^{\mathfrak{b}}(\mathbf{D})$ is a deterministic equivalent of $\mathrm{R}_X(\mathbf{D})$. Precisely:

**Proposition 8.** *Assume $X$ satisfies **H6** and **H2** for some $\eta > 0$. Let $\mathbf{B} \in \mathbb{R}^{d \times d}$ and let $\mathbf{D} \succeq 0$ be positive semidefinite. Define*

$$\mathfrak{a}^* \;=\; 1 + \frac{1}{n}\,\mathrm{tr}\big(\Sigma_X\,\mathbb{E}\big[\mathrm{R}_X(\mathbf{D})\,\mathbb{1}_{\mathsf{A}_\eta}(X)\big]\big)\,,$$

*and $\mathfrak{b}^*$ be the unique fixed point of*

$$f_{\mathbf{D}} : \; \mathfrak{b} \mapsto 1 + \frac{1}{n}\,\mathrm{tr}\Big(\Sigma_X\,\bar{\mathrm{R}}_X^{\mathfrak{b}}(\mathbf{D})\Big)\,.$$

*Then, for an absolute constant $k > 0$ and all $t > 0$,*

$$\mathbb{P}\bigg(\bigg|\frac{1}{d}\,\mathrm{tr}\big(\mathbf{B}\{\mathrm{R}_X(\mathbf{D})\,\mathbb{1}_{\mathsf{A}_\eta}(X) - \mathbb{E}[\mathrm{R}_X(\mathbf{D})\,\mathbb{1}_{\mathsf{A}_\eta}(X)]\}\big)\bigg| \geq t\bigg) \;\lesssim\; \exp\bigg(-\,k\,\frac{c_X\,(\eta + \lambda_d(\mathbf{D}))^3\,n\,t^2}{\|\mathbf{B}\|_{\mathrm{op}}^2\,(\eta + \lambda_d(\mathbf{D}) + c_X/d)}\bigg)\,.$$

*Furthermore, define the polynomial $Q : \mathbb{R}^5 \to \mathbb{R}$ by*

$$Q(X,Y,Z,U,V) \;=\; (1 + UX + VX)\,(X^3 Z + X^2 + YX^2 Z + YX) \;+\; YX^2 Z \;+\; YX^4 Y^2\,,$$

*and set $q = Q\big(\eta + \lambda_d(\mathbf{D}),\, \lambda_d(\Sigma_X),\, \|\Sigma_X\|_{\mathrm{op}}^{-1},\, c_X^{-1},\, n^{-1}\big)$. Then*

$$\Big\|\mathbb{E}\Big[\big\{\,\mathrm{R}_X(\mathbf{D}) - \bar{\mathrm{R}}_X^{\mathfrak{a}^*}(\mathbf{D})\big\}\,\mathbb{1}_{\mathsf{A}_\eta}(X)\Big]\Big\|_{\mathrm{F}} \;\lesssim\; \frac{q\,\sqrt{d}\,\|\Sigma_X\|_{\mathrm{op}}^3}{n\,\lambda_d(\Sigma_X)\,(\eta + \lambda_d(\mathbf{D}))^6}\,,$$

*and*

$$\Big\|\mathbb{E}\Big[\big\{\,\mathrm{R}_X(\mathbf{D}) - \bar{\mathrm{R}}_X^{\mathfrak{b}^*}(\mathbf{D})\big\}\,\mathbb{1}_{\mathsf{A}_\eta}(X)\Big]\Big\|_{\mathrm{F}} \;\lesssim\; \bigg(1 + \frac{d\,\|\Sigma_X\|_{\mathrm{op}}}{n\,\lambda_d(\Sigma_X)}\bigg)\bigg(\frac{q\,\sqrt{d}\,\|\Sigma_X\|_{\mathrm{op}}^3}{n\,\lambda_d(\Sigma_X)\,(\eta + \lambda_d(\mathbf{D}))^6}$$
$$+ \bigg(\frac{d\,\|\Sigma_X\|_{\mathrm{op}}}{n} + \frac{d^2\,\|\Sigma_X\|_{\mathrm{op}}^2}{(\eta + \lambda_d(\mathbf{D}))\,n^2}\bigg)\,\mathrm{e}^{-c_X n}\bigg)\,.$$

*Remark.* This result generalizes those of [Cho22]. Firstly, under **H2** one may take an arbitrarily small regularization $\mathbf{D} \succeq 0$ and still retain favorable concentration properties for the resolvent (even $\mathbf{D} = 0$), secondly we provide fully explicit bounds which allow to understand deeper the dependancies to all the parameters. Our proof follows [Cho22] closely.

*Proof.* The proof procees in two parts, first we derive the claimed concentration bound for terms of the form $d^{-1}\,\mathrm{tr}\,(\mathbf{B}\,\mathrm{R}_X(\mathbf{D}))$, which follows from concentration of Lipschitz transformations of $X$ (**H6**), as well as Proposition 6. Then we will derive the claimed bias bound, using the Shermann-Morison formula.

**Concentration of $d^{-1}\,\mathrm{tr}\,(\mathbf{B}\,\mathrm{R}_X(\mathbf{D}))\,\mathbb{1}_{\mathsf{A}_\eta}(X)$** We mainly rely on Proposition 6, first note that the map

$$h_{\mathbf{B},\mathbf{D}} : \begin{cases} \mathsf{A}_\eta & \to \mathbb{R} \\ \mathbf{X} & \mapsto \dfrac{1}{d}\,\mathrm{tr}\,(\mathbf{B}\,\mathrm{R}_{\mathbf{X}}(\mathbf{D})) \end{cases}\,,$$

is $2\|\mathbf{B}\|_{\mathrm{op}}(\eta + \lambda_1(\mathbf{D}))^{-3/2}n^{-1/2}d^{-1/2}$-Lipschitz from Lemma 6. Moreover $\|h_{\mathbf{B},\mathbf{D}}\|_\infty \leq \|\mathbf{B}\|_{\mathrm{op}}^2(\eta + \lambda_1(\mathbf{D}))^{-1}$, we have from Proposition 6 that $h_{\mathbf{B},\mathbf{D}}(X)\,\mathbb{1}_{\mathsf{A}_\eta}(X)$ is sub Gaussian, with parameter,

$$\sigma_{h_{\mathbf{B},\mathbf{D}}(X)}^2 \;\lesssim\; \mathbb{P}(X \in \mathsf{A}_\eta)^2\,\frac{\|\mathbf{B}\|_{\mathrm{op}}^2}{(\eta + \lambda_d(\mathbf{D}))^3 nd} + \frac{\|\mathbf{B}\|_{\mathrm{op}}^2\sigma(\mathbb{P}(X \in \mathsf{A}_\eta))^2}{(\eta + \lambda_d(\mathbf{D}))^2}\,,$$

and, remarking that by definition of $\sigma$ given in Proposition 6, since $\eta$ satisfies **H2**, we have,

$$\sigma(\mathbb{P}(X \in \mathsf{E}))^2 \;\lesssim\; \frac{1}{nc_X}\,,$$

which implies that

$$\sigma_{h_{\mathbf{B},\mathbf{D}}}^2 \;\lesssim\; \frac{\|\mathbf{B}\|_{\mathrm{op}}^2}{nd(\eta + \lambda_d(\mathbf{D}))^3} + \frac{\|\mathbf{B}\|_{\mathrm{op}}^2}{c_X n(\eta + \lambda_d(\mathbf{D}))^2} = \frac{\|\mathbf{B}\|_{\mathrm{op}}^2}{c_X n(\eta + \lambda_d(\mathbf{D}))^3}\bigg(\frac{c_X}{d} + (\eta + \lambda_d(\mathbf{D}))\bigg)\,,$$

hence, using the variance bound for sub Gaussian random variable, we have for a universal constant $k$,

$$\mathbb{P}\left(\left|\text{tr}\left(\mathbf{B}\,\mathrm{R}_X(\mathbf{D})\right)\mathbb{1}_{\mathsf{A}_\eta}(X) - \mathbb{E}\left[\text{tr}\left(\mathbf{B}\,\mathrm{R}_X(\mathbf{D})\right)\mathbb{1}_{\mathsf{A}_\eta}(X)\right]\right| \geq t\right) \lesssim \exp\left(-k\frac{t^2 c_X(\eta + \lambda_d(\mathbf{D}))^3 n t^2}{\|\mathbf{B}\|_{\text{op}}^2(\eta + \lambda_d(\mathbf{D} + c_X/d))}\right) ,$$

**First equivalent for** $\mathbb{E}\left[\mathrm{R}_X(\mathbf{D})\mathbb{1}_{\mathsf{A}_\eta}(X)\right]$ Recall the notation $\mathrm{R}_X^-(\mathbf{D}) = R_{X^-}(\mathbf{D})$ where $X^- = [0, X_1, \cdots, X_n]$, we have from the Shermann-Morison formula [SM50],

$$\mathrm{R}_X(\mathbf{D})\mathbb{1}_{\mathsf{A}_\eta}(X) = \left\{\mathrm{R}_X^-(\mathbf{D}) - \frac{1}{n}\frac{\mathrm{R}_X^-(\mathbf{D})X_1 X_1^\top \mathrm{R}_X^-(\mathbf{D})}{1 + n^{-1}X_1^\top \mathrm{R}_X^-(\mathbf{D})X_1}\right\}\mathbb{1}_{\mathsf{A}_\eta}(X) . \tag{25}$$

hence, multiplying both sides by $X_1$, we obtain,

$$\mathrm{R}_X(\mathbf{D})X_1\mathbb{1}_{\mathsf{A}_\eta}(X) = \frac{\mathrm{R}_X^-(\mathbf{D})X_1}{1 + n^{-1}X_1^\top \mathrm{R}_X^-(\mathbf{D})X_1}\mathbb{1}_{\mathsf{A}_\eta}(X) .$$

Denoting $\mathfrak{a}_X = 1 + n^{-1}X_1^\top \mathrm{R}_X^-(\mathbf{D})X_1\mathbb{1}_{\mathsf{A}_\eta}(X)$, we simplify the above expression to,

$$\mathrm{R}_X(\mathbf{D})X_1\mathbb{1}_{\mathsf{A}_\eta}(X) = \frac{\mathrm{R}_X^-(\mathbf{D})X_1}{\mathfrak{a}_X}\mathbb{1}_{\mathsf{A}_\eta}(X) . \tag{26}$$

We now focus on bounding the bias of $\mathrm{R}_X(\mathbf{D})\mathbb{1}_{\mathsf{A}_\eta}(X)$. First, using the identity $\mathbf{A}^{-1} - \mathbf{B}^{-1} = \mathbf{A}^{-1}(\mathbf{B} - \mathbf{A})\mathbf{B}^{-1}$, we have,

$$\mathbb{E}\left[\left\{\mathrm{R}_X(\mathbf{D}) - \bar{\mathrm{R}}_X^{\mathfrak{a}^*}(\mathbf{D})\right\}\mathbb{1}_{\mathsf{A}_\eta}(X)\right] = \mathbb{E}\left[\mathrm{R}_X(\mathbf{D})\left\{\frac{\Sigma_X}{\mathfrak{a}^*} - C_X\right\}\bar{\mathrm{R}}_X^{\mathfrak{a}^*}(\mathbf{D})\mathbb{1}_{\mathsf{A}_\eta}(X)\right]$$

$$= \mathbb{E}\left[\mathrm{R}_X(\mathbf{D})\left\{\frac{\Sigma_X}{\mathfrak{a}^*} - X_1 X_1^\top\right\}\bar{\mathrm{R}}_X^{\mathfrak{a}^*}(\mathbf{D})\mathbb{1}_{\mathsf{A}_\eta}(X)\right]$$

$$= \mathbb{E}\left[\left\{\frac{1}{\mathfrak{a}^*}\mathrm{R}_X(\mathbf{D})\Sigma_X\bar{\mathrm{R}}_X^{\mathfrak{a}^*}(\mathbf{D}) - \frac{1}{\mathfrak{a}_X}\mathrm{R}_X^-(\mathbf{D})X_1 X_1^\top\bar{\mathrm{R}}_X^{\mathfrak{a}^*}(\mathbf{D})\right\}\mathbb{1}_{\mathsf{A}_\eta}(X)\right]$$

where, in the last equality, we have used (26). Further rearranging the terms, we get,

$$\mathbb{E}\left[\left\{\mathrm{R}_X(\mathbf{D}) - \bar{\mathrm{R}}_X^{\mathfrak{a}^*}(\mathbf{D})\right\}\mathbb{1}_{\mathsf{A}_\eta}(X)\right]$$

$$= \mathbb{E}\left[\frac{1}{\mathfrak{a}^*}\{\mathrm{R}_X(\mathbf{D}) - \mathrm{R}_X^-(\mathbf{D})\}\Sigma_X\bar{\mathrm{R}}_X^{\mathfrak{a}^*}(\mathbf{D})\mathbb{1}_{\mathsf{A}_\eta}(X) + \left(\frac{1}{\mathfrak{a}^*} - \frac{1}{\mathfrak{a}_X}\right)\mathrm{R}_X^-(\mathbf{D})X_1 X_1^\top\bar{\mathrm{R}}_X^{\mathfrak{a}^*}(\mathbf{D})\mathbb{1}_{\mathsf{A}_\eta}(X)\right] ,$$

Hence, by applying the triangle inequality to bound the bias, we obtain,

$$\left\|\mathbb{E}\left[\left\{\mathrm{R}_X(\mathbf{D}) - \bar{\mathrm{R}}_X^{\mathfrak{a}^*}(\mathbf{D})\right\}\mathbb{1}_{\mathsf{A}_\eta}(X)\right]\right\|_{\text{F}}$$

$$\leq \left\|\mathbb{E}\left[\{\mathrm{R}_X(\mathbf{D}) - \mathrm{R}_X^-(\mathbf{D})\}\mathbb{1}_{\mathsf{A}_\eta}(X)\right]\frac{\Sigma_X\bar{\mathrm{R}}_X^{\mathfrak{a}^*}(\mathbf{D})}{\mathfrak{a}^*}\right\|_{\text{F}} + \left\|\mathbb{E}\left[\left(\frac{1}{\mathfrak{a}^*} - \frac{1}{\mathfrak{a}_X}\right)\mathrm{R}_X^-(\mathbf{D})X_1 X_1^\top\bar{\mathrm{R}}_X^{\mathfrak{a}^*}(\mathbf{D})\mathbb{1}_{\mathsf{A}_\eta}(X)\right]\right\|_{\text{F}}$$
$$\tag{27}$$

Controlling each term individualy, we first use $\|\mathbf{AB}\|_{\text{F}} \leq \|\mathbf{A}\|_{\text{op}}\|\mathbf{B}\|_{\text{F}}$ to get,

$$\left\|\mathbb{E}\left[\{\mathrm{R}_X(\mathbf{D}) - \mathrm{R}_X^-(\mathbf{D})\}\mathbb{1}_{\mathsf{A}_\eta}(X)\right]\frac{\Sigma_X\bar{\mathrm{R}}_X(\mathbf{D})}{\mathfrak{a}^*}\right\|_{\text{F}} \leq \left\|\mathbb{E}\left[\{\mathrm{R}_X(\mathbf{D}) - \mathrm{R}_X^-(\mathbf{D})\}\mathbb{1}_{\mathsf{A}_\eta}(X)\right]\right\|_{\text{F}}\left\|\frac{\Sigma_X\bar{\mathrm{R}}_X^{\mathfrak{a}}(\mathbf{D})}{\mathfrak{a}^*}\right\|_{\text{op}}$$

$$\leq \left\|\mathbb{E}\left[\{\mathrm{R}_X(\mathbf{D}) - \mathrm{R}_X^-(\mathbf{D})\}\mathbb{1}_{\mathsf{A}_\eta}(X)\right]\right\|_{\text{F}} ,$$

From (26),

$$\left\|\mathbb{E}\left[\{\mathrm{R}_X(\mathbf{D}) - \mathrm{R}_X^-(\mathbf{D})\}\mathbb{1}_{\mathsf{A}_\eta}(X)\right]\right\|_{\text{F}} = \frac{1}{n}\left\|\mathbb{E}\left[\frac{1}{\mathfrak{a}_X}\mathrm{R}_X^-(\mathbf{D})X_1 X_1^\top \mathrm{R}_X^-(\mathbf{D})\mathbb{1}_{\mathsf{A}_\eta}(X)\right]\right\|_{\text{F}} ,$$

In order to easily bound the riht-hand side of the previous inequality, we introduce the Lowner order on symetrix matrices. We say that $\mathbf{A} \preceq \mathbf{B}$ if and only if $\mathbf{B} - \mathbf{A}$ is a PSD matrix, then we have

$\mathfrak{a}_X^{-1} \mathrm{R}_X^-(\mathbf{D}) X_1 X_1^\top \mathrm{R}_X^-(\mathbf{D}) \mathbb{1}_{\mathsf{A}_\eta}(X) \preceq \mathrm{R}_X^-(\mathbf{D}) X_1 X_1^\top \mathrm{R}_X^-(\mathbf{D}) \mathbb{1}_{\mathsf{A}_\eta}(X)$ almost surely. It is clear that this ordering is preserved when averaging the matrices, we get

$$\mathbb{E}\left[\frac{1}{\mathfrak{a}_X} \mathrm{R}_X^-(\mathbf{D}) X_1 X_1^\top \mathrm{R}_X^-(\mathbf{D}) \mathbb{1}_{\mathsf{A}_\eta}(X)\right] \preceq \mathbb{E}\left[\mathrm{R}_X^-(\mathbf{D}) X_1 X_1^\top \mathrm{R}_X^-(\mathbf{D}) \mathbb{1}_{\mathsf{A}_\eta}(X)\right]$$

and, using the fact that the Frobenius norm is non-decreasing w.r.t. the Lowner order on PSD matrices, we deduce,

$$
\begin{aligned}
\left\|\mathbb{E}\left[\{\mathrm{R}_X(\mathbf{D}) - \mathrm{R}_X^-(\mathbf{D})\} \mathbb{1}_{\mathsf{A}_\eta}(X)\right]\right\|_{\mathrm{F}} &= \frac{1}{n}\left\|\mathbb{E}\left[\frac{1}{\mathfrak{a}_X} \mathrm{R}_X^-(\mathbf{D}) X_1 X_1^\top \mathrm{R}_X^-(\mathbf{D}) \mathbb{1}_{\mathsf{A}_\eta}(X)\right]\right\|_{\mathrm{F}} \\
&\leq \frac{1}{n}\left\|\mathbb{E}\left[\mathrm{R}_X^-(\mathbf{D}) X_1 X_1^\top \mathrm{R}_X^-(\mathbf{D}) \mathbb{1}_{\mathsf{A}_\eta}(X)\right]\right\|_{\mathrm{F}} \\
&\leq \frac{1}{n}\left\|\mathbb{E}\left[\mathrm{R}_X^-(\mathbf{D}) \Sigma_X \mathrm{R}_X^-(\mathbf{D}) \mathbb{1}_{\mathsf{A}_\eta}(X)\right]\right\|_{\mathrm{F}} \\
&\leq \frac{\sqrt{d}\|\Sigma_X\|_{\mathrm{op}}}{n(\eta + \lambda_d(\mathbf{D}))^2} .
\end{aligned}
$$

Plugging the previous computations in (27), we get

$$
\left\|\mathbb{E}\left[\left\{\mathrm{R}_X(\mathbf{D}) - \bar{\mathrm{R}}_X^{\mathfrak{a}^*}(\mathbf{D})\right\} \mathbb{1}_{\mathsf{A}_\eta}(X)\right]\right\|_{\mathrm{F}} \tag{28}
$$
$$
\leq \left\|\mathbb{E}\left[\left(\frac{1}{\mathfrak{a}^*} - \frac{1}{\mathfrak{a}_X}\right) \mathrm{R}_X^-(\mathbf{D}) X_1 X_1^\top \bar{\mathrm{R}}_X^{\mathfrak{a}^*}(\mathbf{D}) \mathbb{1}_{\mathsf{A}_\eta}(X)\right]\right\|_{\mathrm{F}} + \frac{\sqrt{d}\|\Sigma_X\|_{\mathrm{op}}}{n(\eta + \lambda_d(\mathbf{D}))^2} .
$$

It remains only to bound the second term in the right hand side of (27), recalling the dual representation of the Frobenius norm,

$$\|\mathbf{A}\|_{\mathrm{F}} = \sup_{\|\mathbf{B}\|_{\mathrm{F}} \leq 1} \mathrm{tr}(\mathbf{B}^\top \mathbf{A}) ,$$

we define for any $\mathbf{B}$ of unit Frobenius norm, the random variable $\zeta_{\mathbf{B},X} = X_1^\top \bar{\mathrm{R}}_X^{\mathfrak{a}^*}(\mathbf{D}) \mathbf{B} \mathrm{R}_X^-(\mathbf{D}) X_1 \mathbb{1}_{\mathsf{A}_\eta}(X)$, we have

$$
\begin{aligned}
\left\|\mathbb{E}\left[\left(\frac{1}{\mathfrak{a}^*} - \frac{1}{\mathfrak{a}_X}\right) \mathrm{R}_X^-(\mathbf{D}) X_1 X_1^\top \bar{\mathrm{R}}_X^{\mathfrak{a}^*}(\mathbf{D}) \mathbb{1}_{\mathsf{A}_\eta}(X)\right]\right\|_{\mathrm{F}} \\
= \sup_{\|\mathbf{B}\|_{\mathrm{F}}=1} \mathbb{E}\left[\mathrm{tr}\left(\mathbf{B}^\top \left(\frac{1}{\mathfrak{a}^*} - \frac{1}{\mathfrak{a}_X}\right) \mathrm{R}_X^-(\mathbf{D}) X_1 X_1^\top \bar{\mathrm{R}}_X^{\mathfrak{a}^*}(\mathbf{D}) \mathbb{1}_{\mathsf{A}_\eta}(X)\right)\right] \\
= \sup_{\|\mathbf{B}\|_{\mathrm{F}}=1} \left|\mathbb{E}\left[\mathrm{tr}\left(\mathbf{B}\left(\frac{1}{\mathfrak{a}^*} - \frac{1}{\mathfrak{a}_X}\right) \mathrm{R}_X^-(\mathbf{D}) X_1 X_1^\top \bar{\mathrm{R}}_X^{\mathfrak{a}^*}(\mathbf{D}) \mathbb{1}_{\mathsf{A}_\eta}(X)\right)\right]\right| \\
= \sup_{\|\mathbf{B}\|_{\mathrm{F}}=1} \left|\mathbb{E}\left[\left(\frac{1}{\mathfrak{a}^*} - \frac{1}{\mathfrak{a}_X}\right) \zeta_{\mathbf{B},X}\right]\right|
\end{aligned}
$$

To conclude the proof, we use proposition 7 to bound the variances of $\mathfrak{a}_X$ as well as $\zeta_{\mathbf{B},X}$, to bound the above term uniformly over all the possible choices of $\mathbf{B}$. Using the triangle inequality, we have,

$$
\left|\mathbb{E}\left[\left(\frac{1}{\mathfrak{a}^*} - \frac{1}{\mathfrak{a}_X}\right) \zeta_{\mathbf{B},X}\right]\right| \leq \left|\left(\frac{1}{\mathfrak{a}^*} - \frac{1}{\mathbb{E}[\mathfrak{a}_X]}\right) \mathbb{E}[\zeta_{\mathbf{B},X}]\right| + \left|\mathbb{E}\left[\left(\frac{1}{\mathbb{E}[\mathfrak{a}_X]} - \frac{1}{\mathfrak{a}_X}\right) \zeta_{\mathbf{B},X}\right]\right| \tag{29}
$$

We further rewrite the first term by remarking that $\mathfrak{a}_X \geq 1$ almost surely, and similarly $\mathfrak{a}^*$, we get:

$$
\left|\left(\frac{1}{\mathfrak{a}^*} - \frac{1}{\mathbb{E}[\mathfrak{a}_X]}\right) \mathbb{E}[\zeta_{\mathbf{B},X}]\right| \leq |\mathfrak{a}^* - \mathbb{E}[\mathfrak{a}_X]| \frac{\mathbb{E}[\zeta_{\mathbf{B},X}]}{\mathfrak{a}^*} .
$$

Now, observe that $|\mathbb{E}[\zeta_{\mathbf{B},X}]|$ may be explicitly controlled by,

$$
\begin{aligned}
|\mathbb{E}[\zeta_{\mathbf{B},X}]/\mathfrak{a}^*| &= \left|\mathbb{E}\left[\mathrm{tr}\left(\frac{\Sigma_X}{\mathfrak{a}^*} \bar{\mathrm{R}}_X^{\mathfrak{a}^*}(\mathbf{D}) \mathbf{B} \mathrm{R}_X^-(\mathbf{D}) \mathbb{1}_{\mathsf{A}_\eta}(X)\right)\right]\right| \\
&\leq \frac{\sqrt{d}}{\eta + \lambda_d(\mathbf{D})} . \tag{30}
\end{aligned}
$$

Where we have used the fact that $\Sigma_X \bar{R}_X^{\mathfrak{a}^*}(\lambda)/\mathfrak{a}^* \preceq I_d$. Secondly, the bias of $\mathfrak{a}_X$ can be bounded using (25) as,

$$
\begin{aligned}
|\mathfrak{a}^* - \mathbb{E}[\mathfrak{a}_X]| &= \left| \frac{1}{n} \operatorname{tr} \left( \Sigma_X \mathbb{E}\left[ R_X^-(\mathbf{D}) - R_X(\mathbf{D}) \right] \right) \right| = \frac{1}{n^2} \operatorname{tr} \left( \Sigma_X \mathbb{E}\left[ \frac{1}{\mathfrak{a}_X} R_X^-(\mathbf{D}) X_1 X_1^\top R_X^-(\mathbf{D}) \right] \right) \\
&\leq \frac{1}{n^2} \operatorname{tr} \left( \Sigma_X \mathbb{E}\left[ R_X^-(\mathbf{D}) X_1 X_1^\top R_X^-(\mathbf{D}) \right] \right) \leq \frac{1}{n^2} \mathbb{E}\left[ \operatorname{tr} \left( (\Sigma_X R_X^-(\mathbf{D}))^2 \right) \right] \\
&\leq \frac{d\|\Sigma_X\|_{\mathrm{op}}^2}{(\eta + \lambda_d(\mathbf{D}))^2 n^2} \; .
\end{aligned}
$$

Which implies the following bound on the first term in (29),

$$
\left| \left( \frac{1}{\mathfrak{a}^*} - \frac{1}{\mathbb{E}[\mathfrak{a}_X]} \right) \mathbb{E}[\zeta_{\mathbf{B},X}] \right| \lesssim \frac{d^{3/2}\|\Sigma_X\|_{\mathrm{op}}^2}{(\eta + \lambda_d(\mathbf{D}))^3 n^2} \; . \tag{31}
$$

Now dealing with the second term in (29), using the Cauchy-Schwarz inequality, as well as $\mathbb{E}[\mathfrak{a}_X] \geq 1$, we have,

$$
\left| \mathbb{E}\left[ \left( \frac{1}{\mathbb{E}[\mathfrak{a}_X]} - \frac{1}{\mathfrak{a}_X} \right) \zeta_{\mathbf{B},X} \right] \right| = \left| \mathbb{E}\left[ (\mathfrak{a}_X - \mathbb{E}[\mathfrak{a}_X]) \frac{\zeta_{\mathbf{B},X}}{\mathfrak{a}_X \mathbb{E}[\mathfrak{a}_X]} \right] \right| \leq \sqrt{\operatorname{Var}(\mathfrak{a}_X) \operatorname{Var}(\zeta_{\mathbf{B},X}/\mathfrak{a}_X)}
$$

We write $\zeta_{\mathbf{B},X}/\mathfrak{a}_X = (\zeta_{\mathbf{B},X} - \mathbb{E}[\zeta_{\mathbf{B},X}])/\mathfrak{a}_X + \mathbb{E}[\zeta_{\mathbf{B},X}]/\mathfrak{a}_X$, and using $\operatorname{Var}(a+b) \leq 2\operatorname{Var}(a) + 2\operatorname{Var}(b)$, we have,

$$
\begin{aligned}
\operatorname{Var}(\zeta_{\mathbf{B},X}/\mathfrak{a}_X) &\leq 2 \operatorname{Var}\left( (\zeta_{\mathbf{B},X} - \mathbb{E}[\zeta_{\mathbf{B},X}])/\mathfrak{a}_X \right) + 2 \operatorname{Var}\left( \mathbb{E}[\zeta_{\mathbf{B},X}]/\mathfrak{a}_X \right) \\
&\leq 2\mathbb{E}\left[ \left( (\zeta_{\mathbf{B},X} - \mathbb{E}[\zeta_{\mathbf{B},X}])/\mathfrak{a}_X \right)^2 \right] + 2\mathbb{E}[\zeta_{\mathbf{B},X}]^2 \operatorname{Var}(\mathfrak{a}_X^{-1}) \\
&\leq 2\mathbb{E}\left[ (\zeta_{\mathbf{B},X} - \mathbb{E}[\zeta_{\mathbf{B},X}])^2 \right] + 2\mathbb{E}[\zeta_{\mathbf{B},X}]^2 \operatorname{Var}(\mathfrak{a}_X^{-1}) \\
&\leq 2 \operatorname{Var}(\zeta_{\mathbf{B},X}) + 2\mathbb{E}[\zeta_{\mathbf{B},X}]^2 \operatorname{Var}(\mathfrak{a}_X^{-1}) \; .
\end{aligned}
$$

Furthermore, $\operatorname{Var}(\mathfrak{a}_X^{-1}) = \inf_m \mathbb{E}\left[ (\mathfrak{a}_X^{-1} - m)^2 \right] \leq \mathbb{E}\left[ (\mathfrak{a}_X^{-1} - \mathbb{E}[\mathfrak{a}_X]^{-1})^2 \right] \leq \operatorname{Var}(\mathfrak{a}_X)$, which follows from $\mathfrak{a}_X \geq 1$ again, we get,

$$
\operatorname{Var}(\zeta_{\mathbf{B},X}/\mathfrak{a}_X) \leq 2 \operatorname{Var}(\zeta_{\mathbf{B},X}) + 2\mathbb{E}[\zeta_{\mathbf{B},X}]^2 \operatorname{Var}(\mathfrak{a}_X) \; ,
$$

which results in,

$$
\mathbb{E}\left[ \left( \frac{1}{\mathbb{E}[\mathfrak{a}_X]} - \frac{1}{\mathfrak{a}_X} \right) \zeta_{\mathbf{B},X} \right]^2 \leq 2 \operatorname{Var}(\mathfrak{a}_X) \operatorname{Var}(\zeta_{\mathbf{B},X}) + 2\mathbb{E}[\zeta_{\mathbf{B},X}]^2 \operatorname{Var}(\mathfrak{a}_X)^2 \tag{32}
$$

We conclude by bounding $\operatorname{Var}(\mathfrak{a}_X)$ and $\operatorname{Var}(\zeta_{\mathbf{B},X})$, using Proposition 7 and the Lipschitz property of $\mathbf{X} \mapsto R_{\mathbf{X}}^-(\mathbf{D})$ on $A_\eta$ (which results from Lemma 6), we have,

$$
\begin{aligned}
\operatorname{Var}(\mathfrak{a}_X) &= \frac{1}{n^2} \operatorname{Var}\left( X_1^\top R_X^-(\mathbf{D}) X_1 \mathbb{1}_{A_\eta}(X) \right) \lesssim \frac{d\|\Sigma_X\|_{\mathrm{op}}^2}{n^2} \left\{ \frac{1}{(\eta + \lambda_d(\mathbf{D}))^3 n} + \frac{1 + c_X^{-1}}{(\eta + \lambda_d(\mathbf{D}))^2} \right\} \\
&= \frac{d\|\Sigma_X\|_{\mathrm{op}}^2}{n^2 (\eta + \lambda_d(\mathbf{D}))^3} \left\{ \frac{1}{n} + \left( 1 + \frac{1}{c_X} \right) (\eta + \lambda_d(\mathbf{D})) \right\} \; ,
\end{aligned}
$$

similarly,

$$
\begin{aligned}
\operatorname{Var}(\zeta_{\mathbf{D},X}) &\lesssim X_1^\top \bar{R}_X^{\mathfrak{a}^*}(\mathbf{D}) \mathbf{B} R_X^-(\mathbf{D}) X_1 \mathbb{1}_{A_\eta}(X) \\
&\lesssim \frac{\|\Sigma_X\|_{\mathrm{op}}^2 \|\bar{R}_X^{\mathfrak{a}^*}(\mathbf{D})\|_{\mathrm{op}}^2}{(\eta + \lambda_d(\mathbf{D}))^3} \left\{ \frac{1}{n} + \left( 1 + \frac{1}{c_X} \right) (\eta + \lambda_d(\mathbf{D})) \right\} \; ,
\end{aligned}
$$

it results from the two previous upper bounds, as well as (30) and (32), that

$$\mathbb{E}\left[\left(\frac{1}{\mathbb{E}\left[\mathfrak{a}_X\right]} - \frac{1}{\mathfrak{a}_X}\right)\zeta_{\mathbf{B},X}\right] \lesssim \frac{\sqrt{d}\|\Sigma_X\|_{\mathrm{op}}^2\|\bar{\mathrm{R}}_X^{\mathfrak{a}^*}(\mathbf{D})\|_{\mathrm{op}}}{n(\eta + \lambda_d(\mathbf{D}))^3}\left\{\frac{1}{n} + \left(1 + \frac{1}{c_X}\right)(\eta + \lambda_d(\mathbf{D}))\right\} \quad (33)$$

$$+ \frac{d\|\Sigma_X\|_{\mathrm{op}}^2\mathbb{E}\left[\zeta_{\mathbf{B},X}\right]}{n^2(\eta + \lambda_d(\mathbf{D}))^3}\left\{\frac{1}{n} + \left(1 + \frac{1}{c_X}(\eta + \lambda_d(\mathbf{D}))\right)\right\}$$

$$\lesssim \frac{\sqrt{d}\|\Sigma_X\|_{\mathrm{op}}^2\|\bar{\mathrm{R}}_X^{\mathfrak{a}^*}(\mathbf{D})\|_{\mathrm{op}}}{n(\eta + \lambda_d(\mathbf{D}))^3}\left\{\frac{1}{n} + \left(1 + \frac{1}{c_X}\right)(\eta + \lambda_d(\mathbf{D}))\right\}$$

$$+ \frac{\mathfrak{a}^* d^{3/2}\|\Sigma_X\|_{\mathrm{op}}^2}{n^2(\eta + \lambda_d(\mathbf{D}))^4}\left\{\frac{1}{n} + \left(1 + \frac{1}{c_X}(\eta + \lambda_d(\mathbf{D}))\right)\right\} .$$

Finally, we bound $\|\bar{\mathrm{R}}_X^{\mathfrak{a}^*}(\mathbf{D})\|_{\mathrm{op}}$ and $\mathfrak{a}^*$ by writing

$$\mathfrak{a}^* = 1 + \frac{1}{n}\,\mathrm{tr}\left(\Sigma_X\mathbb{E}\left[\mathrm{R}_X(\mathbf{D})\mathbb{1}_{\mathsf{A}_\eta}(X)\right]\right) \leq 1 + \frac{d\|\Sigma_X\|_{\mathrm{op}}}{n(\eta + \lambda_d(\mathbf{D}))} ,$$

and

$$\|\bar{\mathrm{R}}_X^{\mathfrak{a}^*}(\mathbf{D})\|_{\mathrm{op}} \leq \left(\frac{\lambda_1(\Sigma_X)}{\mathfrak{a}^*} + \lambda_1(\mathbf{D})\right)^{-1} \leq \frac{\mathfrak{a}}{\lambda_1(\Sigma_X)} \leq \frac{1 + \frac{d}{n}\|\Sigma_X\|_{\mathrm{op}}(\eta + \lambda_d(\mathbf{D}))^{-1}}{\lambda_1(\Sigma_X)} . \quad (34)$$

We conclude from (33),

$$\mathbb{E}\left[\left(\frac{1}{\mathbb{E}\left[\mathfrak{a}_X\right]} - \frac{1}{\mathfrak{a}_X}\right)\zeta_{\mathbf{B},X}\right]^2 \lesssim \frac{\sqrt{d}\|\Sigma_X\|_{\mathrm{op}}^2}{n\lambda_d(\Sigma_X)(\eta + \lambda_d(\mathbf{D}))^3}\left\{\frac{1}{n} + \left(1 + \frac{1}{c_X}\right)(\eta + \lambda_d(\mathbf{D}))\right\} \quad (35)$$

$$+ \frac{d^{3/2}\|\Sigma_X\|_{\mathrm{op}}^3}{n^2\lambda_d(\Sigma_X)(\eta + \lambda_d(\mathbf{D}))^4}\left\{\frac{1}{n} + \left(1 + \frac{1}{c_X}\right)(\eta + \lambda_d(\mathbf{D}))\right\}$$

$$+ \frac{d^{3/2}\|\Sigma_X\|_{\mathrm{op}}^2}{n^2(\eta + \lambda_d(\mathbf{D}))^4}\left\{\frac{1}{n} + \left(1 + \frac{1}{c_X}(\eta + \lambda_d(\mathbf{D}))\right)\right\}$$

$$+ \frac{d^{5/2}\|\Sigma_X\|_{\mathrm{op}}^3}{n^6(\eta + \lambda_d(\mathbf{D}))^5}\left\{\frac{1}{n} + \left(1 + \frac{1}{c_X}(\eta + \lambda_d(\mathbf{D}))\right)\right\} .$$

We slighly simplify the previous upper bound by remarking that **H**2 implies that $d < n$, hence:

$$\frac{d^{5/2}}{n^3} \leq \frac{d^{3/2}}{n^2} \leq \frac{\sqrt{d}}{n} ,$$

Using this in (35), we obtain,

$$\mathbb{E}\left[\left(\frac{1}{\mathbb{E}\left[\mathfrak{a}_X\right]} - \frac{1}{\mathfrak{a}_X}\right)\zeta_{\mathbf{B},X}\right]^2 \lesssim \frac{\sqrt{d}\|\Sigma_X\|_{\mathrm{op}}^3}{n\lambda_d(\Sigma_X)(\eta + \lambda_d(\mathbf{D}))^5}\left\{\frac{(\eta + \lambda_d(\mathbf{D}))^2}{n\|\Sigma_X\|_{\mathrm{op}}} + \left(1 + \frac{1}{c_X}\right)\frac{(\eta + \lambda_d(\mathbf{D}))^3}{\|\Sigma_X\|_{\mathrm{op}}}\right\}$$

$$+ \frac{\sqrt{d}\|\Sigma_X\|_{\mathrm{op}}^3}{n\lambda_d(\Sigma_X)(\eta + \lambda_d(\mathbf{D}))^5}\left\{\frac{\eta + \lambda_d(\mathbf{D})}{n} + \left(1 + \frac{1}{c_X}\right)(\eta + \lambda_d(\mathbf{D}))^2\right\}$$

$$+ \frac{\sqrt{d}\|\Sigma_X\|_{\mathrm{op}}^3}{n\lambda_d(\Sigma_X)(\eta + \lambda_d(\mathbf{D}))^5}\left\{\frac{\lambda_d(\Sigma_X)(\eta + \lambda_d(\mathbf{D}))}{n\|\Sigma_X\|_{\mathrm{op}}} + \left(1 + \frac{1}{c_X}\right)\frac{\lambda_d(\Sigma_X)(\eta + \lambda_d(\mathbf{D}))^2}{\|\Sigma_X\|_{\mathrm{op}}}\right\}$$

$$+ \frac{\sqrt{d}\|\Sigma_X\|_{\mathrm{op}}^3}{n\lambda_d(\Sigma_X)(\eta + \lambda_d(\mathbf{D}))^5}\left\{\frac{\lambda_d(\mathbf{D})}{n} + \left(1 + \frac{1}{c_X}\right)\lambda_d(\Sigma_X)(\eta + \lambda_d(\mathbf{D}))\right\} .$$

Defining the polynomial $P_1$ as:

$$P_1(X,Y,Z) = X^3Z + X^2 + YX^2Z + YX , \quad p_1 = P_1(\eta + \lambda_d(\mathbf{D}), \lambda_d(\Sigma_X), \|\Sigma_X\|_{\mathrm{op}}^{-1}) ,$$

we can rewrite the previous upper bound as,

$$\mathbb{E}\left[\left(\frac{1}{\mathbb{E}\left[\mathfrak{a}_X\right]} - \frac{1}{\mathfrak{a}_X}\right)\zeta_{\mathbf{B},X}\right]^2 \quad (36)$$

$$\lesssim \frac{\sqrt{d}\|\Sigma_X\|_{\mathrm{op}}^3}{n\lambda_d(\Sigma_X)(\eta + \lambda_d(\mathbf{D}))^5}p_1\left(\frac{1}{n(\eta + \lambda_d(\mathbf{D}))} + 1 + \frac{1}{c_X}\right) .$$

Plugging (31) and (36) in (29), we get,

$$\left\|\mathbb{E}\left[\left(\frac{1}{\mathfrak{a}^*} - \frac{1}{\mathfrak{a}_X}\right)\mathrm{R}_X^-(\mathbf{D})X_1X_1^\top\bar{\mathrm{R}}_X^{\mathfrak{a}^*}(\mathbf{D})\mathbb{1}_{\mathsf{A}_\eta}(X)\right]\right\|_{\mathrm{F}}$$

$$\lesssim \frac{d^{3/2}\|\Sigma_X\|_{\mathrm{op}}^2}{(\eta+\lambda_d(\mathbf{D}))^3n^2} + \frac{\sqrt{d}\|\Sigma_X\|_{\mathrm{op}}^3}{n\lambda_d(\Sigma_X)(\eta+\lambda_d(\mathbf{D}))^5}p_1\left(\frac{1}{n(\eta+\lambda_d(\mathbf{D}))}+1+\frac{1}{c_X}\right)$$

$$\lesssim \frac{\sqrt{d}\|\Sigma_X\|_{\mathrm{op}}^3}{n\lambda_d(\Sigma_X)(\eta+\lambda_d(\mathbf{D}))^5}\left\{p_1\left(\frac{1}{n(\eta+\lambda_d(\mathbf{D}))}+1+\frac{1}{c_X}\right)+\frac{\lambda_d(\Sigma_X)(\eta+\lambda_d(\mathbf{D}))^2}{\|\Sigma_X\|_{\mathrm{op}}}\right\}$$

Where, once again, we have used the fact that $d < n$ from **H**2.

Finally, from (28), we have,

$$\left\|\mathbb{E}\left[\left\{\mathrm{R}_X(\mathbf{D}) - \bar{\mathrm{R}}_X^{\mathfrak{a}^*}(\mathbf{D})\right\}\mathbb{1}_{\mathsf{A}_\eta}(X)\right]\right\|_{\mathrm{F}}$$

$$\lesssim \frac{\sqrt{d}\|\Sigma_X\|_{\mathrm{op}}}{(\eta+\lambda_d(\mathbf{D}))^2n} + \frac{\sqrt{d}\|\Sigma_X\|_{\mathrm{op}}^3}{n\lambda_d(\Sigma_X)(\eta+\lambda_d(\mathbf{D}))^5}\left\{p_1\left(\frac{1}{n(\eta+\lambda_d(\mathbf{D}))}+1+\frac{1}{c_X}\right)+\frac{\lambda_d(\Sigma_X)(\eta+\lambda_d(\mathbf{D}))^2}{\|\Sigma_X\|_{\mathrm{op}}}\right\}$$

$$\lesssim \frac{\sqrt{d}\|\Sigma_X\|_{\mathrm{op}}^3}{n\lambda_d(\Sigma_X)(\eta+\lambda_d(\mathbf{D}))^6}\left\{\frac{(\eta+\lambda_d(\mathbf{D}))p_1}{n} + (\eta+\lambda_d(\mathbf{D}))p_1\left(1+\frac{1}{c_X}\right)\right.$$

$$\left. + \frac{\lambda_d(\Sigma_X)(\eta+\lambda_d(\mathbf{D}))^3}{\|\Sigma_X\|_{\mathrm{op}}} + \frac{\lambda_d(\Sigma_X)(\eta+\lambda_d(\mathbf{D})^4)}{\|\Sigma_X\|_{\mathrm{op}}^2}\right\}\ .$$

Defining $Q$ as,

$$Q(X,Y,Z,U,V) = XVP(X,Y,Z) + (1+U)\,XP(X,Y,Z) + YX^3Z + YX^4Y^2\ ,\qquad (37)$$

We have shown that:

$$\left\|\mathbb{E}\left[\left\{\mathrm{R}_X(\mathbf{D}) - \bar{\mathrm{R}}_X^{\mathfrak{a}^*}(\mathbf{D})\right\}\mathbb{1}_{\mathsf{A}_\eta}(X)\right]\right\|_{\mathrm{F}} \le \frac{\sqrt{d}\|\Sigma_X\|_{\mathrm{op}}^3}{n\lambda_d(\Sigma_X)(\eta+\lambda_d(\mathbf{D}))^6}Q(\eta+\lambda_d(\mathbf{D}),\lambda_d(\Sigma_X),\|\Sigma_X\|_{\mathrm{op}}^{-1},c_X^{-1},n^{-1})\ ,$$

$$(38)$$

This concludes the proof for the first deterministic equivalent.

**Second equivalent for $\mathbb{E}\left[\mathrm{R}_X(\mathbf{D})\mathbb{1}_{\mathsf{E}}(X)\right]$**

Now, we show that:

$$\left\|\mathbb{E}\left[\left\{\mathrm{R}_X(\mathbf{D}) - \bar{\mathrm{R}}_X^{\mathfrak{b}^*}(\mathbf{D})\right\}\mathbb{1}_{\mathsf{A}_\eta}(X)\right]\right\|_{\mathrm{F}}$$

$$\lesssim \left(1+\frac{d\|\Sigma_X\|_{\mathrm{op}}}{n\lambda_d(\Sigma_X)}\right)\left(\frac{q\sqrt{d}\|\Sigma_X\|_{\mathrm{op}}^3}{n\lambda_d(\Sigma_X)(\eta+\lambda_d(\mathbf{D}))^6} + \left(\frac{d\|\Sigma_X\|_{\mathrm{op}}}{n} + \frac{d^2\|\Sigma_X\|_{\mathrm{op}}^2}{(\eta+\lambda_d(\mathbf{D}))n^2}\right)\mathrm{e}^{-c_Xn}\right)\ ,$$

where $\mathfrak{b}^*$ is defined as the only positive solution to equation $\mathfrak{b} = 1 + n^{-1}\operatorname{tr}\left(\Sigma_X\bar{\mathrm{R}}_X^{\mathfrak{b}}(\mathbf{D})\right)$. Recall the definition of $f_{\mathbf{D}}$, for any $\mathfrak{b} \in [1,\infty)$,

$$f_{\mathbf{D}}(\mathfrak{b}) = 1 + n^{-1}\operatorname{tr}\left(\Sigma_X\bar{\mathrm{R}}_X^{\mathfrak{b}}(\mathbf{D})\right)\ .$$

For notation simplicity, we introduce $q \in \mathbb{R}$ defined as:

$$q = Q(\eta+\lambda_d(\mathbf{D}),\lambda_d(\Sigma_X),\|\Sigma_X\|_{\mathrm{op}}^{-1},c_X^{-1},n^{-1})\ ,$$

where $Q$ is defined in (37).

First, using (38), we have,

$$\left\|\mathbb{E}\left[\left\{\mathrm{R}_X(\mathbf{D}) - \bar{\mathrm{R}}_X^{\mathfrak{b}^*}(\mathbf{D})\right\}\mathbb{1}_{\mathsf{A}_\eta}(X)\right]\right\|_{\mathrm{F}} \qquad\qquad (39)$$

$$\le \left\|\mathbb{E}\left[\left\{\mathrm{R}_X(\mathbf{D}) - \bar{\mathrm{R}}_X^{\mathfrak{a}^*}(\mathbf{D})\right\}\mathbb{1}_{\mathsf{A}_\eta}(X)\right]\right\|_{\mathrm{F}} + \left\|\mathbb{E}\left[\left\{\bar{\mathrm{R}}_X^{\mathfrak{a}^*}(\mathbf{D}) - \bar{\mathrm{R}}_X^{\mathfrak{b}^*}(\mathbf{D})\right\}\mathbb{1}_{\mathsf{A}_\eta}(X)\right]\right\|_{\mathrm{F}}$$

$$\lesssim \frac{q\sqrt{d}\|\Sigma_X\|_{\mathrm{op}}^3}{n\lambda_d(\Sigma_X)(\eta+\lambda_d(\mathbf{D}))^6} + \left\|\mathbb{E}\left[\left\{\bar{\mathrm{R}}_X^{\mathfrak{a}^*}(\mathbf{D}) - \bar{\mathrm{R}}_X^{\mathfrak{b}^*}(\mathbf{D})\right\}\mathbb{1}_{\mathsf{A}_\eta}(X)\right]\right\|_{\mathrm{F}}\ ,$$

Then, using the identity $\mathbf{A}^{-1} - \mathbf{B}^{-1} = \mathbf{A}^{-1}(\mathbf{B} - \mathbf{A})\mathbf{B}^{-1}$ we deduce

$$\left\| \mathbb{E}\left[ \left\{ \bar{\mathrm{R}}_X^{\mathfrak{a}^*}(\mathbf{D}) - \bar{\mathrm{R}}_X^{\mathfrak{b}^*}(\mathbf{D}) \right\} \mathbb{1}_{\mathsf{A}_\eta}(X) \right] \right\|_{\mathrm{F}} = \left| \frac{1}{\mathfrak{a}^*} - \frac{1}{\mathfrak{b}^*} \right| \mathbb{P}(X \in \mathsf{A}_\eta) \left\| \bar{\mathrm{R}}_X^{\mathfrak{a}^*}(\mathbf{D}) \Sigma_X \bar{\mathrm{R}}_X^{\mathfrak{b}^*}(\mathbf{D}) \right\|_{\mathrm{F}} \tag{40}$$

$$\lesssim |\mathfrak{a}^* - \mathfrak{b}^*| \, \|\bar{\mathrm{R}}_X^{\mathfrak{a}^*}(\mathbf{D})\|_{\mathrm{F}}/\mathfrak{a}^*$$

$$\lesssim |\mathfrak{a}^* - \mathfrak{b}^*| \frac{\sqrt{d}}{\lambda_d(\Sigma_X)} \ ,$$

Furthermore, we remark that $\mathfrak{a}^*$ is almost a fixed point of $f_{\mathbf{D}}$ from the first deterministic equivalent, indeed,

$$|\mathfrak{a}^* - f_{\mathbf{D}}(\mathfrak{a}^*)| = n^{-1} \left| \mathrm{tr}\left( \Sigma_X \left\{ \mathbb{E}\left[ \mathrm{R}_X(\mathbf{D}) \mathbb{1}_{\mathsf{A}_\eta}(X) \right] - \bar{\mathrm{R}}_X^{\mathfrak{a}^*}(\mathbf{D}) \right\} \right) \right|$$

$$\leq \frac{\sqrt{d}\|\Sigma_X\|_{\mathrm{op}}}{n} \left\| \mathbb{E}\left[ \mathrm{R}_X(\mathbf{D}) \mathbb{1}_{\mathsf{A}_\eta}(X) \right] - \bar{\mathrm{R}}_X^{\mathfrak{a}^*}(\mathbf{D}) \right\|_{\mathrm{F}}$$

$$\leq \frac{\sqrt{d}\|\Sigma_X\|_{\mathrm{op}}}{n} \left\| \mathbb{E}\left[ \mathrm{R}_X(\mathbf{D}) - \bar{\mathrm{R}}_X^{\mathfrak{a}^*}(\mathbf{D}) \mathbb{1}_{\mathsf{A}_\eta}(X) \right] \right\|_{\mathrm{F}} + \frac{d\|\Sigma_X\|_{\mathrm{op}} \|\bar{\mathrm{R}}_X^{\mathfrak{a}^*}(\mathbf{D})\|_{\mathrm{op}}}{n} (1 - \mathbb{P}(X \in \mathsf{A}_\eta))$$

Recalling (34), and using **H** 2, we have,

$$\frac{d\|\Sigma_X\|_{\mathrm{op}} \|\bar{\mathrm{R}}_X^{\mathfrak{a}^*}(\mathbf{D})\|_{\mathrm{op}}}{n} (1 - \mathbb{P}(X \in \mathsf{A}_\eta)) \leq \left( \frac{d\|\Sigma_X\|_{\mathrm{op}}}{n} + \frac{d^2\|\Sigma_X\|_{\mathrm{op}}^2}{(\eta + \lambda_d(\mathbf{D}))n^2} \right) \mathrm{e}^{-c_X n} \ ,$$

Now, using equation (38), we have,

$$|\mathfrak{a}^* - f_{\mathbf{D}}(\mathfrak{a}^*)| \lesssim \frac{q\sqrt{d}\|\Sigma_X\|_{\mathrm{op}}^3}{n\lambda_d(\Sigma_X)(\eta + \lambda_d(\mathbf{D}))^6} + \left( \frac{d\|\Sigma_X\|_{\mathrm{op}}}{n} + \frac{d^2\|\Sigma_X\|_{\mathrm{op}}^2}{(\eta + \lambda_d(\mathbf{D}))n^2} \right) \mathrm{e}^{-c_X n} \ .$$

Furthermore, we show that $f_{\mathbf{D}}$ is a contraction mapping around $\mathfrak{b}^*$, indeed we have for any $\mathfrak{b} \in [1, \infty)$,

$$|f_{\mathbf{D}}(\mathfrak{b}) - f_{\mathbf{D}}(\mathfrak{b}^*)| = \frac{1}{n} \left| \mathrm{tr}\left( \Sigma_X \left\{ \bar{\mathrm{R}}_X^{\mathfrak{b}}(\mathbf{D}) - \bar{\mathrm{R}}_X^{\mathfrak{b}^*}(\mathbf{D}) \right\} \right) \right| = \frac{1}{n} \left| \mathrm{tr}\left( \Sigma_X \bar{\mathrm{R}}_X^{\mathfrak{b}}(\mathbf{D}) \Sigma_X \bar{\mathrm{R}}_X^{\mathfrak{b}^*}(\mathbf{D}) \right) \right| \left| \frac{1}{\mathfrak{b}} - \frac{1}{\mathfrak{b}^*} \right|$$

$$= \frac{1}{n} \mathrm{tr}\left( \left\{ \mathrm{I}_d - \mathbf{D}\bar{\mathrm{R}}_X^{\mathfrak{b}}(\mathbf{D}) \right\} \frac{\Sigma_X}{\mathfrak{b}^*} \bar{\mathrm{R}}_X^{\mathfrak{b}^*}(\mathbf{D}) \right) |\mathfrak{b} - \mathfrak{b}^*|$$

$$\leq \frac{1}{n\mathfrak{b}^*} \mathrm{tr}\left( \Sigma_X \bar{\mathrm{R}}_X^{\mathfrak{b}^*}(\mathbf{D}) \right) |\mathfrak{b} - \mathfrak{b}^*|$$

$$\leq \frac{f_{\mathbf{D}}(\mathfrak{b}^*) - 1}{f_{\mathbf{D}}(\mathfrak{b}^*)} |\mathfrak{b} - \mathfrak{b}^*| = \frac{\mathfrak{b}^* - 1}{\mathfrak{b}^*} |\mathfrak{b} - \mathfrak{b}^*| \ ,$$

where in the last line, we have used the fact that $\mathfrak{b}^*$ is the only fixed point of $f_{\mathbf{D}}$. We conclude that $f_{\mathbf{D}}$ is contractive around $\mathfrak{b}^*$. We conclude on the distance between $\mathfrak{a}^*$ and $\mathfrak{b}^*$ by writing,

$$|\mathfrak{a}^* - \mathfrak{b}^*| \leq |\mathfrak{a}^* - f_{\mathbf{D}}(\mathfrak{a}^*)| + |f_{\mathbf{D}}(\mathfrak{a}^*) - f_{\mathbf{D}}(\mathfrak{b}^*)| \leq |\mathfrak{a}^* - f_{\mathbf{D}}(\mathfrak{a}^*)| + \frac{\mathfrak{b}^* - 1}{\mathfrak{b}^*} |\mathfrak{a}^* - \mathfrak{b}^*| \ ,$$

which implies,

$$|\mathfrak{a}^* - \mathfrak{b}^*| \leq \mathfrak{b}^* |\mathfrak{a}^* - f_{\mathbf{D}}(\mathfrak{a}^*)| \ .$$

To conclude the proof, we need to bound $\mathfrak{b}^*$. To this end, write

$$\mathfrak{b}^* = 1 + \frac{1}{n} \mathrm{tr}\left( \Sigma_X \bar{\mathrm{R}}_X^{\mathfrak{b}^*}(\mathbf{D}) \right) \leq 1 + \frac{1}{n} \mathrm{tr}\left( \Sigma_X \bar{\mathrm{R}}_X^1(\mathbf{D}) \right) \leq 1 + \frac{d\|\Sigma_X\|_{\mathrm{op}}}{n\lambda_d(\Sigma_X)} \ ,$$

which followed from $\bar{\mathrm{R}}_X^{\mathfrak{b}^*}(\mathbf{D}) \preceq \bar{\mathrm{R}}_X^1(\mathbf{D})$, we obtain from (40),

$$\left\| \mathbb{E}\left[ \left\{ \bar{\mathrm{R}}_X^{\mathfrak{a}^*}(\mathbf{D}) - \bar{\mathrm{R}}_X^{\mathfrak{b}^*}(\mathbf{D}) \right\} \mathbb{1}_{\mathsf{E}}(X) \right] \right\|_{\mathrm{F}}$$

$$\lesssim \left( 1 + \frac{d\|\Sigma_X\|_{\mathrm{op}}}{n\lambda_d(\Sigma_X)} \right) \left( \frac{q\sqrt{d}\|\Sigma_X\|_{\mathrm{op}}^3}{n\lambda_d(\Sigma_X)(\eta + \lambda_d(\mathbf{D}))^6} + \left( \frac{d\|\Sigma_X\|_{\mathrm{op}}}{n} + \frac{d^2\|\Sigma_X\|_{\mathrm{op}}^2}{(\eta + \lambda_d(\mathbf{D}))n^2} \right) \mathrm{e}^{-c_X n} \right)$$

Finally, from (39), we have,

$$\left\| \mathbb{E}\left[\left\{ R_X(\mathbf{D}) - \bar{R}_X^{\mathfrak{b}^*}(\mathbf{D}) \right\} \mathbb{1}_{\mathsf{E}}(X) \right] \right\|_{\mathrm{F}}$$

$$\lesssim \left( 1 + \frac{d\|\Sigma_X\|_{\mathrm{op}}}{n\lambda_d(\Sigma_X)} \right) \left( \frac{q\sqrt{d}\|\Sigma_X\|_{\mathrm{op}}^3}{n\lambda_d(\Sigma_X)(\eta+\lambda_d(\mathbf{D}))^6} + \left( \frac{d\|\Sigma_X\|_{\mathrm{op}}}{n} + \frac{d^2\|\Sigma_X\|_{\mathrm{op}}^2}{(\eta+\lambda_d(\mathbf{D}))n^2} \right) \mathrm{e}^{-c_X n} \right) .$$

This concludes the proof. □

## B.4 Conclusion on the proof of theorem 1

We leverage proposition 1, to prove that $\hat{\mathcal{E}}_X(\lambda)$ approximates $\mathcal{E}_X(\lambda)$. In all this proof, we set $X \in \mathbb{R}^{d \times n}$ and $\eta > 0$, such that **H** 6 and **H** 2 are satisfied. We first recall,

$$\mathcal{E}_X(\lambda) = \| R_X(\lambda) - \Sigma_X \|_{\mathrm{F}}^2 , \quad \text{for } \lambda > 0 ,$$

and

$$\hat{\mathcal{E}}_X(\lambda) := \frac{1}{d} \operatorname{tr}\left( R_X(\lambda)^2 \right) - \frac{2(1 - d/n)}{\lambda d} \operatorname{tr}\left( R_X(0) \right) \mathbb{1}_{\mathsf{A}_\eta}(X) + \frac{2}{\lambda \mathfrak{b}(\lambda)d} \operatorname{tr}\left( R_X(\lambda) \right) + \frac{1}{d} \operatorname{tr}\left( \Sigma_X^{-2} \right) ,$$

where

$$\mathfrak{b}(\lambda) := \frac{1}{1 - d/n + \lambda/n \operatorname{tr}\left( R_X(\lambda) \right)} .$$

We will write $\Delta\mathcal{E}_X(\lambda) = \hat{\mathcal{E}}_X(\lambda) - \mathcal{E}_X(\lambda)$, and we remark that,

$$\Delta\mathcal{E}_X(\lambda) = -\frac{2(1 - d/n)}{\lambda d} \operatorname{tr}\left( R_X(0) \right) \mathbb{1}_{\mathsf{A}_\eta}(X) + \frac{2}{\lambda \mathfrak{b}(\lambda)d} \operatorname{tr}\left( R_X(\lambda) \right) - \frac{1}{d} \operatorname{tr}\left( \Sigma_X^{-1} R_X(\lambda) \right) .$$

We derive the claimed concentration bound by applying proposition 1, first noting that $\Delta\mathcal{E}_X(\lambda)$ is close to the following quantity with high probability,

$$\overline{\Delta\mathcal{E}_X}(\lambda) = -\frac{2(1 - d/n)}{\lambda d} \operatorname{tr}\left( R_X(0) \right) \mathbb{1}_{\mathsf{A}_\eta}(X) + \frac{2}{\lambda \mathfrak{b}(\lambda)d} \operatorname{tr}\left( R_X(\lambda) \right) \mathbb{1}_{\mathsf{A}_\eta}(X) - \frac{1}{d} \operatorname{tr}\left( \Sigma_X^{-1} R_X(\lambda) \right) \mathbb{1}_{\mathsf{A}_\eta}(X)$$

indeed, we have for all $\epsilon > 0$,

$$\mathbb{P}\left( \left| \Delta\mathcal{E}_X(\lambda) - \overline{\Delta\mathcal{E}_X}(\lambda) \right| \geq \epsilon \right) \leq \mathbb{P}\left( X \notin \mathsf{A}_\eta \right) \lesssim \mathrm{e}^{-c_X n} .$$

Furthermore, we show that each term in $\overline{\Delta\mathcal{E}_X}(\lambda)$ concentrates around its expectation. We have,

$$\left| \overline{\Delta\mathcal{E}_X}(\lambda) - \mathbb{E}\left[ \overline{\Delta\mathcal{E}_X}(\lambda) \right] \right| \leq \frac{2(1 - d/n)}{\lambda d} \left| \operatorname{tr}\left( R_X(0) \right) \mathbb{1}_{\mathsf{A}_\eta}(X) - \mathbb{E}\left[ \operatorname{tr}\left( R_X(0) \right) \mathbb{1}_{\mathsf{A}_\eta}(X) \right] \right|$$

$$+ \frac{2 \operatorname{tr}\left( R_X(\lambda) \right)}{\lambda d} \left| \frac{1}{\mathfrak{b}(\lambda)} - \mathbb{E}\left[ \frac{1}{\mathfrak{b}(\lambda)} \right] \right|$$

$$+ \mathbb{E}\left[ \frac{2}{\lambda \mathfrak{b}(\lambda)d} \right] \left| \operatorname{tr}\left( R_X(\lambda) \right) \mathbb{1}_{\mathsf{A}_\eta}(X) - \mathbb{E}\left[ \operatorname{tr}\left( R_X(\lambda) \right) \mathbb{1}_{\mathsf{A}_\eta}(X) \right] \right|$$

$$+ \left| \frac{1}{d} \operatorname{tr}\left( \Sigma_X^{-1} R_X(\lambda) \right) \mathbb{1}_{\mathsf{A}_\eta}(X) - \mathbb{E}\left[ \frac{1}{d} \operatorname{tr}\left( \Sigma_X^{-1} R_X(\lambda) \right) \mathbb{1}_{\mathsf{A}_\eta}(X) \right] \right|$$

Remarking that $\eta > 0$ implies that $d < n$ hence, $0 \leq 1 - d/n \leq 1$, as well as $R_X(\lambda) \preceq \lambda^{-1} I_d$ and $\mathfrak{b}(\lambda) \geq 1$, we bound the various multiplicative term in the previous inequation as,

$$
\begin{aligned}
\left| \overline{\Delta \mathcal{E}_X}(\lambda) - \mathbb{E}\left[ \overline{\Delta \mathcal{E}_X}(\lambda) \right] \right| &\leq \frac{2}{\lambda d} \left| \operatorname{tr}(R_X(0)) \mathbb{1}_{A_\eta}(X) - \mathbb{E}\left[ \operatorname{tr}(R_X(0)) \mathbb{1}_{A_\eta}(X) \right] \right| \\
&+ 2\lambda \frac{1}{n} \left| \operatorname{tr}(R_X(\lambda) - \mathbb{E}\left[ \operatorname{tr}(R_X(\lambda)) \right]) \right| + \frac{2}{\lambda d} \left| \operatorname{tr}(R_X(\lambda)) - \mathbb{E}\left[ \operatorname{tr}(R_X(\lambda)) \right] \right| \\
&+ \left| \frac{1}{d} \operatorname{tr}\left( \Sigma_X^{-1} R_X(\lambda) \right) - \mathbb{E}\left[ \frac{1}{d} \operatorname{tr}\left( \Sigma_X^{-1} R_X(\lambda) \right) \right] \right| \\
&\leq \frac{2}{\lambda d} \left| \operatorname{tr}(R_X(0)) \mathbb{1}_{A_\eta}(X) - \mathbb{E}\left[ \operatorname{tr}(R_X(0)) \mathbb{1}_{A_\eta}(X) \right] \right| \\
&+ 2 \left\{ \lambda + \frac{1}{\lambda} \right\} \frac{1}{d} \left| \operatorname{tr}\left( R_X(\lambda) \mathbb{1}_{A_\eta}(X) - \mathbb{E}\left[ \operatorname{tr}(R_X(\lambda)) \mathbb{1}_{A_\eta}(X) \right] \right) \right| \\
&+ \frac{1}{d} \left| \operatorname{tr}\left( \Sigma_X^{-1} R_X(\lambda) \right) \mathbb{1}_{A_\eta}(X) - \mathbb{E}\left[ \operatorname{tr}\left( \Sigma_X^{-1} R_X(\lambda) \right) \mathbb{1}_{A_\eta}(X) \right] \right|
\end{aligned}
$$

Hence, from a union bound argument, we have,

$$
\begin{aligned}
\mathbb{P}\left( \left| \overline{\Delta \mathcal{E}_X}(\lambda) - \mathbb{E}\left[ \overline{\Delta \mathcal{E}_X}(\lambda) \right] \right| \geq t \right) &\leq \mathbb{P}\left( \frac{1}{d} \left| \operatorname{tr}\left( R_X(0) \mathbb{1}_{A_\eta}(X) - \mathbb{E}\left[ R_X(0) \mathbb{1}_{A_\eta}(X) \right] \right) \right| \geq \frac{\lambda t}{6} \right) \quad (41) \\
&+ \mathbb{P}\left( \frac{1}{d} \left| \operatorname{tr}\left( R_X(\lambda) \mathbb{1}_{A_\eta}(X) - \mathbb{E}\left[ R_X(\lambda) \mathbb{1}_{A_\eta}(X) \right] \right) \right| \geq \frac{t}{6(\lambda + \lambda^{-1})} \right) \\
&+ \mathbb{P}\left( \frac{1}{d} \left| \operatorname{tr}\left( \Sigma_X^{-1} \{ R_X(\lambda) \mathbb{1}_{A_\eta}(X) - \mathbb{E}\left[ R_X(\lambda) \mathbb{1}_{A_\eta}(X) \right] \} \right) \right| \geq \frac{t}{3} \right) .
\end{aligned}
$$

We now control each of these term, by using the concentration statement of proposition 1 every time. We denote,

$$
\xi_X(t) = \frac{c_X \eta^3 n t^2}{\max\{ \|\Sigma_X\|_{\mathrm{op}}, 1 \}^2 (\eta + c_X/d)} .
$$

Then, Proposition 1 and (41) implies that,

$$
\mathbb{P}\left( \left| \overline{\Delta \mathcal{E}_X}(\lambda) - \mathbb{E}\left[ \overline{\Delta \mathcal{E}_X}(\lambda) \right] \right| \geq t \right) \lesssim e^{-k\xi(\lambda t/6)} + e^{-k\xi(t/(6(\lambda + \lambda^{-1})))} + e^{-k\xi(t/3)} ,
$$

for a universal constant $k > 0$. Now remarking that $\xi(\lambda t/6) \geq \xi(t/(6(\lambda + \lambda^{-1})))$, we have,

$$
\mathbb{P}\left( \left| \overline{\Delta \mathcal{E}_X}(\lambda) - \mathbb{E}\left[ \overline{\Delta \mathcal{E}_X}(\lambda) \right] \right| \geq t \right) \lesssim e^{k'\xi(t \min\{1/(\lambda + \lambda^{-1}), 1\})} ,
$$

for a universal constant $k'$. And, we conclude,

$$
\mathbb{P}\left( \left| \Delta \mathcal{E}_X(\lambda) - \mathbb{E}\left[ \overline{\Delta \mathcal{E}_X}(\lambda) \right] \right| > t \right) \lesssim \exp\left( -k' \frac{c_X \eta^3 n \min\{1, \lambda + \frac{1}{\lambda}\}^2 t^2}{\max\{ \|\Sigma_X\|_{\mathrm{op}}, 1 \}(\eta + c_X/d)} \right) + \exp\left( -c_X n \right) .
$$

$$(42)$$

We now bound $\mathbb{E}\left[\overline{\Delta \mathcal{E}_X}(\lambda)\right]$, to this end, write,

$$\mathbb{E}\left[\overline{\Delta \mathcal{E}_X}(\lambda)\right] = -\frac{2(1-d/n)}{\lambda d}\operatorname{tr}\left(\mathbb{E}\left[\mathrm{R}_X(0)\mathbb{1}_{\mathsf{A}_\eta}(X)\right]\right) + \mathbb{E}\left[\frac{2}{\lambda \mathfrak{b}(\lambda)d}\operatorname{tr}\left(\mathrm{R}_X(\lambda)\right)\mathbb{1}_{\mathsf{A}_\eta}(X)\right] - \frac{1}{d}\operatorname{tr}\left(\Sigma_X^{-1}\mathbb{E}\left[\mathrm{R}_X(\lambda)\mathbb{1}_{\mathsf{A}_\eta}(X)\right]\right)$$

$$= -\frac{2(1-d/n)}{\lambda d}\operatorname{tr}\left(\mathbb{E}\left[\mathrm{R}_X(0)\mathbb{1}_{\mathsf{A}_\eta}(X)\right]\right) + \mathbb{E}\left[\frac{2}{\lambda \mathfrak{b}(\lambda)d}\right]\operatorname{tr}\left(\left[\mathrm{R}_X(\lambda)\mathbb{1}_{\mathsf{A}_\eta}(X)\right]\right) - \frac{1}{d}\operatorname{tr}\left(\Sigma_X^{-1}\mathbb{E}\left[\mathrm{R}_X(\lambda)\mathbb{1}_{\mathsf{A}_\eta}(X)\right]\right)$$

$$+ \frac{2}{\lambda}\operatorname{Cov}\left(\frac{1}{\mathfrak{b}(\lambda)}, \frac{1}{d}\operatorname{tr}\left(\mathrm{R}_X(\lambda)\right)\mathbb{1}_{\mathsf{A}_\eta}(X)\right)$$

$$\lesssim \left\{-\frac{2(1-d/n)}{\lambda d}\operatorname{tr}\left(\bar{\mathrm{R}}_X^{\mathfrak{b}^*(0)}(0)\right) + \frac{2}{\lambda \mathfrak{b}^*(\lambda)d}\operatorname{tr}\left(\bar{\mathrm{R}}_X^{\mathfrak{b}^*(\lambda)}(\lambda)\right) - \frac{1}{d}\operatorname{tr}\left(\Sigma_X^{-1}\bar{\mathrm{R}}_X^{\mathfrak{b}^*(\lambda)}(\lambda)\right)\right\}\mathbb{P}\left(X \in \mathsf{A}_\eta\right)$$

$$+ \frac{2}{\lambda}\operatorname{Cov}\left(\frac{1}{\mathfrak{b}(\lambda)}, \frac{1}{d}\operatorname{tr}\left(\mathrm{R}_X(\lambda)\right)\mathbb{1}_{\mathsf{A}_\eta}(X)\right)$$

$$+ \left(1 + \frac{d\|\Sigma_X\|_{\mathrm{op}}}{n\lambda_1(\Sigma_X)}\right)\left(\frac{q\sqrt{d}\|\Sigma_X\|_{\mathrm{op}}^3}{n\lambda_1(\Sigma_X)\eta^6} + \left(\frac{d\|\Sigma_X\|_{\mathrm{op}}}{n} + \frac{d^2\|\Sigma_X\|_{\mathrm{op}}^2}{\eta n^2}\right)\mathrm{e}^{-c_X n}\right)$$

where the last line followed from applying the second deterministic equivalent presented in Proposition 1.

Recalling the definition of $\mathfrak{b}(\lambda)$, we have,

$$\frac{2}{\lambda}\operatorname{Cov}\left(\frac{1}{\mathfrak{b}(\lambda)}, \frac{1}{d}\operatorname{tr}\left(\mathrm{R}_X(\lambda)\right)\right) = \frac{2\lambda d/n}{\lambda}\operatorname{Cov}\left(\frac{1}{d}\operatorname{tr}\left(\mathrm{R}_X(\lambda)\right), \frac{1}{d}\operatorname{tr}\left(\mathrm{R}_X(\lambda)\right)\right)$$

$$\leq 2\operatorname{Var}\left(\frac{1}{d}\operatorname{tr}\left(\mathrm{R}_X(\lambda)\right)\right),$$

which is controlled using **H** 1, from the fact that $\mathbf{X} \mapsto d^{-1}\operatorname{tr}\left(\mathrm{R}_{\mathbf{X}}(\lambda)\right)$ is $2\lambda^{-3/2}n^{-1/2}d^{1/2}$-lipschitz (from lemma 6), hence **H** 1 ensures that $d^{-1}\operatorname{tr}\left(\mathrm{R}_X(\lambda)\right)$ is sub Gaussian, and has variance bounded as,

$$\operatorname{Var}\left(\frac{1}{d}\operatorname{tr}\left(\mathrm{R}_X(\lambda)\right)\right) \lesssim \frac{1}{\lambda^3 n d},$$

which implies,

$$\mathbb{E}\left[\overline{\Delta \mathcal{E}_X}(\lambda)\right] \lesssim \left\{-\frac{2(1-d/n)}{\lambda d}\operatorname{tr}\left(\bar{\mathrm{R}}_X^{\mathfrak{b}^*(0)}(0)\right) + \frac{2}{\lambda \mathfrak{b}^*(\lambda)d}\operatorname{tr}\left(\bar{\mathrm{R}}_X^{\mathfrak{b}^*(\lambda)}(\lambda)\right) - \frac{1}{d}\operatorname{tr}\left(\Sigma_X^{-1}\bar{\mathrm{R}}_X^{\mathfrak{b}^*(\lambda)}(\lambda)\right)\right\}\mathbb{P}\left(X \in \mathsf{A}_\eta\right)$$

$$+ \left(1 + \frac{d\|\Sigma_X\|_{\mathrm{op}}}{n\lambda_1(\Sigma_X)}\right)\left(\frac{q\sqrt{d}\|\Sigma_X\|_{\mathrm{op}}^3}{n\lambda_1(\Sigma_X)\eta^6} + \left(\frac{d\|\Sigma_X\|_{\mathrm{op}}}{n} + \frac{d^2\|\Sigma_X\|_{\mathrm{op}}^2}{\eta n^2}\right)\mathrm{e}^{-c_X n}\right) + \frac{1}{\lambda^3 n d}.$$

Now, we remark that by definition $\mathfrak{b}^*(0)$ is the unique fixed point of $f_0(\mathfrak{b}) = 1 + \mathfrak{b}d/n$, which gives, $\mathfrak{b}^*(0) = (1-d/n)^{-1}$, hence,

$$\mathbb{E}\left[\overline{\Delta \mathcal{E}_X}(\lambda)\right] \leq \left\{-\frac{2}{\lambda d}\operatorname{tr}\left(\Sigma_X^{-1}\right) + \frac{2}{\lambda \mathfrak{b}^*(\lambda)d}\operatorname{tr}\left(\bar{\mathrm{R}}_X^{\mathfrak{b}^*(\lambda)}(\lambda)\right) - \frac{1}{d}\operatorname{tr}\left(\Sigma_X^{-1}\bar{\mathrm{R}}_X^{\mathfrak{b}^*(\lambda)}(\lambda)\right)\right\}\mathbb{P}\left(X \in \mathsf{A}_\eta\right)$$

$$+ \left(1 + \frac{d\|\Sigma_X\|_{\mathrm{op}}}{n\lambda_1(\Sigma_X)}\right)\left(\frac{q\sqrt{d}\|\Sigma_X\|_{\mathrm{op}}^3}{n\lambda_1(\Sigma_X)\eta^6} + \left(\frac{d\|\Sigma_X\|_{\mathrm{op}}}{n} + \frac{d^2\|\Sigma_X\|_{\mathrm{op}}^2}{\eta n^2}\right)\mathrm{e}^{-c_X n}\right) + \frac{1}{\lambda^3 n d},$$

Finally, using the identity $\mathbf{A}^{-1} - \mathbf{B}^{-1} = \mathbf{A}^{-1}\{\mathbf{B} - \mathbf{A}\}\mathbf{B}^{-1}$, we get,

$$\frac{1}{\mathfrak{b}^*(\lambda)}\bar{\mathrm{R}}_X^{\mathfrak{b}^*(\lambda)}(\lambda) - \Sigma_X^{-1} = \lambda\bar{\mathrm{R}}_X^{\mathfrak{b}^*(\lambda)}(\lambda)\Sigma_X^{-1},$$

we thus conclude on the bias of $\hat{\mathcal{E}}_X(\lambda)$ as,

$$\mathbb{E}\left[\overline{\Delta \mathcal{E}_X}(\lambda)\right] \lesssim \underbrace{\left(1 + \frac{d\|\Sigma_X\|_{\mathrm{op}}}{n\lambda_1(\Sigma_X)}\right)\left(\frac{q\sqrt{d}\|\Sigma_X\|_{\mathrm{op}}^3}{n\lambda_1(\Sigma_X)\eta^6} + \left(\frac{d\|\Sigma_X\|_{\mathrm{op}}}{n} + \frac{d^2\|\Sigma_X\|_{\mathrm{op}}^2}{\eta n^2}\right)\mathrm{e}^{-c_X n}\right) + \frac{1}{\lambda^3 n d}}_{B(n)}.$$

and, merging the previous equation with (42), we get a universal constant $k$, and the function $B(n)$ defined above,

$$\mathbb{P}\left(|\Delta\mathcal{E}_X(\lambda)| > t + B(n)\right) \leq \mathbb{P}\left(|\Delta\mathcal{E}_X(\lambda)| > t + \mathbb{E}\left[\overline{\Delta\mathcal{E}_X}(\lambda)\right]\right) \leq \mathbb{P}\left(\left|\Delta\mathcal{E}_X(\lambda) - \mathbb{E}\left[\overline{\Delta\mathcal{E}_X}(\lambda)\right]\right| \geq t\right)$$
$$\lesssim \exp\left(-k\frac{c_X\eta^3 n\min\{1, \lambda + \frac{1}{\lambda}\}^2 t^2}{\max\{\|\Sigma_X\|_{\text{op}}, 1\}(\eta + c_X/d)\}}\right) + \exp\left(-c_X n\right) .$$

which terminates the proof of theorem 1.

## C   A deterministic equivalent for resolvent matrices of augmented sample covariances

In this section we generalize Proposition 1 to the setting where a non-negligible proportion of the dataset is produced by a data-augmentation scheme. We consider the augmented dataset $[X_1, \ldots, X_n, G_1, \ldots, G_m]$, where the two blocks $[X_1, \ldots, X_n]$ and $[G_1, \ldots, G_m]$ satisfy **H2**–**H 6**. The proof follows the non-augmented case presented in Appendix B.

We begin with the following technical lemma, which will be used to derive concentration for the augmented resolvent matrix $\mathrm{R}_{\mathrm{Aug}}(\mathbf{D})$.

**Lemma 7.** *Assume $X$ and $\nu_X$ satisfy **H6** and **H4**, let $f : \mathbb{R}^{d \times n} \times \mathbb{R}^{d \times m} \to \mathbb{R}$ be a $\mathsf{L}_f$-Lispchitz function on $\mathsf{A}_\eta \times \mathbb{R}^{d \times m}$. Then for any $\mathbf{X}, \mathbf{Y} \in \mathsf{A}_\eta$,*

$$\left| \int f(\mathbf{X}, g) \nu_{\mathbf{X}}^{\otimes m}(\mathrm{d}g) - \int f(\mathbf{Y}, g) \nu_{\mathbf{Y}}^{\otimes m}(\mathrm{d}g) \right| \le \mathsf{L}_f \left( 1 + \sqrt{m} \mathsf{L}_G \right) \|\mathbf{X} - \mathbf{Y}\|_{\mathrm{F}} .$$

*i.e, $\mathbf{X} \mapsto \int f(\mathbf{X}, g) d\nu_{\mathbf{X}}^{\otimes m}(dg)$ is $\mathsf{L}_f(1 + \mathsf{L}_G)$ Lispchitz on $\mathsf{A}_\eta$.*

*Proof.* Using **H1** and **H4**, we have

$$\left| \int f(\mathbf{X}, g) \nu_{\mathbf{X}}^{\otimes m}(\mathrm{d}g) - \int f(\mathbf{Y}, g) \nu_{\mathbf{Y}}^{\otimes m}(\mathrm{d}g) \right|$$

$$\le \left| \int f(\mathbf{X}, g) \nu_{\mathbf{X}}^{\otimes m}(\mathrm{d}g) - \int f(\mathbf{Y}, g) \nu_{\mathbf{X}}^{\otimes m}(\mathrm{d}g) \right| + \left| \int f(\mathbf{Y}, g) \nu_{\mathbf{X}}^{\otimes m}(\mathrm{d}g) - \int f(\mathbf{Y}, g) \nu_{\mathbf{Y}}^{\otimes m}(\mathrm{d}g) \right|$$

$$\le \mathsf{L}_f \|\mathbf{X} - \mathbf{Y}\|_{\mathrm{F}} + \mathsf{L}_f W_1(\nu_{\mathbf{X}}^{\otimes m}, \nu_{\mathbf{Y}}^{\otimes m})$$

$$\le \mathsf{L}_f \|\mathbf{X} - \mathbf{Y}\|_{\mathrm{F}} + \mathsf{L}_f \sqrt{m} \mathsf{L}_G \|\mathbf{X} - \mathbf{Y}\|_{\mathrm{F}}$$

$$= \mathsf{L}_f \left( 1 + \sqrt{m} \mathsf{L}_G \right) \|\mathbf{X} - \mathbf{Y}\|_{\mathrm{F}} .$$

Where the last upper bound followed from **H 4**. $\square$

We further recall the following notation for any positive semi-definite matrix $\mathbf{D} \in \mathbb{R}^{d \times d}$, and two dilation factors $\mathfrak{a}_x$ and $\mathfrak{a}_g$.

$$\mathrm{R}_{\mathrm{Aug}}(\mathbf{D}) := (C_{\mathrm{Aug}} + \mathbf{D})^{-1} = ((1 - \alpha)C_X + \alpha C_G + \mathbf{D})^{-1} ,$$

and,

$$\bar{\mathrm{R}}_{\mathrm{Aug}}^{(\mathfrak{a}_x, \mathfrak{a}_g)}(\mathbf{D}) := \left( \frac{1 - (1 - \beta/\mathfrak{a}_g)\alpha}{\mathfrak{a}_x} \Sigma_X + \frac{\alpha}{\mathfrak{a}_g} \bar{\Lambda}_G + \mathbf{D} \right)^{-1} ,$$

where $\bar{\Lambda}_G = \mathbb{E}[\Lambda_G(X)]$.

We prove the following result,

**Theorem 4.** *Assume that **H 2** to **H6** hold. Let $\mathbf{B} \in \mathbb{R}^{d \times d}$, and let $\mathbf{D}$ be a positive semi-definite matrix. We define $(\mathfrak{a}_x^*, \mathfrak{a}_g^*)$ as,*

$$\mathfrak{a}_x^* = 1 + \frac{1 - (1 - \beta/\mathfrak{a}_g^*)\alpha}{n} \mathrm{tr}\left( \Sigma_X \mathbb{E}\left[ \mathrm{R}_{\mathrm{Aug}}(\mathbf{D}) \mathbb{1}_{\mathsf{A}_\eta}(X) \right] \right) , \quad \mathfrak{a}_g^* = 1 + \frac{\alpha}{m} \mathrm{tr}\left( \mathbb{E}\left[ \{\beta C_X + \Lambda_G(X)\} \mathrm{R}_{\mathrm{Aug}}(\mathbf{D}) \mathbb{1}_{\mathsf{A}_\eta}(X) \right] \right) ,$$

*Then, we have for a universal constant $k$,*

$$\mathbb{P}\left( \left| \frac{1}{d} \mathrm{tr}\left( \mathbf{B} \left\{ \mathrm{R}_{\mathrm{Aug}}(\mathbf{D}) \mathbb{1}_{\mathsf{A}_\eta}(X) - \mathbb{E}\left[ \mathrm{R}_{\mathrm{Aug}}(\mathbf{D}) \mathbb{1}_{\mathsf{A}_\eta}(X) \right] \right\} \right) \right| \ge t \right)$$

$$\le 2 \exp\left( -k \frac{n(\eta + \lambda_d(\mathbf{D}))^3 t^2}{\alpha(1 + \sqrt{m} \mathsf{L}_G)^2/d + (1 - \alpha)\sigma_G^2/d + (\eta + \lambda_d(\mathbf{D}))\sigma_G^2} \right) .$$

*And,*

$$\left\| \mathbb{E}\left[ \left\{ R_{\text{Aug}}(\mathbf{D}) - \bar{R}_{\text{Aug}}^{(\mathfrak{a}_x^*, \mathfrak{a}_g^*)}(\mathbf{D}) \right\} \mathbb{1}_{A_\eta}(X) \right] \right\|_{\text{F}}$$

$$\lesssim \frac{\alpha^5 (\kappa q_1 + q_2) \sqrt{d} \left\{ \sigma_G^2 (\beta^3 \|\Sigma_X\|_{\text{op}}^3 + \kappa^3) + \sigma_X^{12} \|\Sigma_X\|_{\text{op}} \lambda_d(\Sigma_X)^{-1} n^{-1/2} d^{-1} \right\}}{n((1-\alpha)\eta + \lambda_d(\mathbf{D}))^6}$$

$$+ \frac{\alpha\beta(1-\alpha')^{-1} + 1}{(1-\alpha)\eta + \lambda_d(\mathbf{D})} \frac{\alpha\beta\sqrt{d}\|\Sigma_X\|_{\text{op}}}{(\eta + \lambda_d(\mathbf{D}))^{3/2} m} \left\{ \frac{\sqrt{d}}{\sqrt{n+m}}(1 + u(n)) + \frac{1}{\sqrt{n+m}} + \sqrt{\alpha} L_G + (1 + c_X^{-1/2})\sqrt{\eta + \lambda_d(\mathbf{D})} \right\}$$

$$+ \frac{\alpha\beta(1-\alpha')^{-1} + 1}{(1-\alpha)\eta + \lambda_d(\mathbf{D})} \frac{\alpha\sqrt{d}\|\bar{\Lambda}_G\|_{\text{op}}}{(\eta + \lambda_d(\mathbf{D}))^{3/2} m} \left( \frac{\mathbb{E}\left[\|\Lambda_G(X) - \bar{\Lambda}_G\|_{\text{F}}\right] \sqrt{\eta + \lambda_d(\mathbf{D})}}{\|\bar{\Lambda}_G\|_{\text{op}}} + \sqrt{\alpha} L_G + \frac{1 + \sqrt{\eta + \lambda_d(\mathbf{D})}}{\sqrt{n}} \right)$$

$$+ \frac{\mathbb{E}\left[\|\Lambda_G(X) - \bar{\Lambda}_G\|_{\text{F}}\right]}{((1-\alpha)\eta + \lambda_d(\mathbf{D}))^2} + \frac{1}{1-\alpha'} \frac{q_3\sqrt{d}\|\Sigma_X\|_{\text{op}}^3}{n\lambda_d(\Sigma_X)(\eta + \lambda_d(\mathbf{D})/(1-\alpha'))^6}$$

*where $\alpha' = \alpha - \alpha\beta/\mathfrak{a}_g^*$, $q_1, q_2$ and $q_3$ are polynomials in $\eta + \lambda_d(\mathbf{D}), \lambda_d(\Sigma_X), \|\Sigma_X\|_{\text{op}}^{-1}, c_X^{-1}$, and $n^{-1}$.*

*Proof.* Following the proof of proposition 1, we consider only the simpler case of $\sigma_X = 1$ (the general case readily follows by considering $X/\sigma_X$ and $G/\sigma_X$.) we prove the two statements of the theorem independantly. First focusing on the concentration of $d^{-1}\text{tr}(\mathbf{B}\,R_{\text{Aug}}(\mathbf{D}))$.

**Proof of the concentration inequality.**

Let $\mathbf{B} \in \mathbb{R}^{d \times d}$ be any squared matrix, for notation simplicity we will denote $h_{\mathbf{B}} : \mathbf{X}, \mathbf{G} \mapsto d^{-1}\text{tr}(\mathbf{B}\,R_{\mathbf{X} \sqcup \mathbf{G}}(\mathbf{D}))\mathbb{1}_{A_\eta}(\mathbf{X})$ as well as $\tilde{X} = X \sqcup G$ (for $\sqcup$ being the column-wise concatenation operator), so that we simply need to bound the cumulative probality function of $h_{\mathbf{B}}(\tilde{X}) - \mathbb{E}[h_{\mathbf{B}}(\tilde{X})]$. To do so, we first bound its moment generating function, for any scalar $s \in \mathbb{R}$,

$$\mathbb{E}\left[ \exp\left( s\left\{ h_{\mathbf{B}}(\tilde{X}) - \mathbb{E}\left[h_{\mathbf{B}}(\tilde{X})\right] \right\} \right) \right]$$

$$= \mathbb{E}\left[ \exp\left( s\left\{ h_{\mathbf{B}}(\tilde{X}) - \mathbb{E}\left[h_{\mathbf{B}}(\tilde{X}) \mid X\right] \right\} \right) \cdot \exp\left( s\left\{ \mathbb{E}\left[h_{\mathbf{B}}(\tilde{X}) \mid X\right] - \mathbb{E}\left[h_{\mathbf{B}}(\tilde{X})\right] \right\} \right) \right]$$

$$= \mathbb{E}\left[ \mathbb{E}\left[ \exp\left( s\left\{ h_{\mathbf{B}}(\tilde{X}) - \mathbb{E}\left[h_{\mathbf{B}}(\tilde{X}) \mid X\right] \right\} \right) \Big| X\right] \cdot \exp\left( s\left\{ \mathbb{E}\left[h_{\mathbf{B}}(\tilde{X}) \mid X\right] - \mathbb{E}\left[h_{\mathbf{B}}(\tilde{X})\right] \right\} \right) \right]$$

Note that the random function $\mathbf{G} \mapsto h_{\mathbf{B}}(X \sqcup \mathbf{G})$ is almost surely $2\|\mathbf{B}\|_{\text{op}}(n + m)^{-1/2}d^{-1/2}(\eta + \lambda_d(\mathbf{D}))^{-3/2}$-Lispchitz on $A_\eta$ from Lemma 6, hence, relying on the $\sigma_G$-Lipschitz concentration property of $G$ conditionally to $X$, and applying Proposition 6, we get

$$\mathbb{E}\left[ \exp\left( s\left\{ h_{\mathbf{B}}(\tilde{X}) - \mathbb{E}\left[h_{\mathbf{B}}(\tilde{X}) \mid X\right] \right\} \right) \Big| X\right] \leq \exp\left( -cs^2\sigma_G^2 \left\{ \frac{1}{(n+m)d(\eta + \lambda_d(\mathbf{D}))^3} + \|h_{\mathbf{B}}\|_\infty^2 \sigma_{A_\eta}^2 \right\} \right) .$$

Finally, remarking that under **H2**, we have $\sigma_{A_\eta}^2 \lesssim n^{-1}$, as well as using $\|h_{\mathbf{B}}\|_\infty \lesssim (\eta + \lambda_d(\mathbf{D}))^{-1}$, we have,

$$\mathbb{E}\left[ \exp\left( s\left\{ h_{\mathbf{B}}(\tilde{X}) - \mathbb{E}\left[h_{\mathbf{B}}(\tilde{X}) \mid X\right] \right\} \right) \right] \leq \exp\left( -cs^2\sigma_G^2 \left\{ \frac{1}{(n+m)d(\eta + \lambda_d(\mathbf{D}))^3} + \frac{1}{n(\eta + \lambda_d(\mathbf{D}))^2} \right\} \right)$$

$$= \exp\left( -cs^2 \frac{\sigma_G^2}{n(\eta + \lambda_d(\mathbf{D}))^3} \left\{ \frac{1-\alpha}{d} + \eta + \lambda_d(\mathbf{D}) \right\} \right) .$$

Now, we will leverage Proposition 6 to bound the remaining term, writting $\mathbb{E}[h_{\mathbf{B}}(X) \mid X] = g_{\mathbf{B}}(X)\mathbb{1}_{A_\eta}(X)$, where for any $\mathbf{X} \in A_\eta$,

$$g_{\mathbf{B}}(\mathbf{X}) = \int \frac{1}{d}\text{tr}(\mathbf{B}R_{\mathbf{X} \sqcup g}(\mathbf{D}))\, d\nu_{\mathbf{X}}^{\otimes m}(g) .$$

We know from Lemma 7 that $g_{\mathbf{B}}$ is $L_{g_{\mathbf{B}}}$-Lispchitz on $A_\eta$, with $L_{g_{\mathbf{B}}} = 2(1 + \sqrt{m}L_G)(n + m)^{-1/2}d^{-1/2}(\eta + \lambda_d(\mathbf{D}))^{-3/2}$. Hence, using Proposition 6, we prove the existence of a numerical

constant $c$, such the following bound holds,

$$\mathbb{E}\left[\exp\left(s\left\{g_{\mathbf{B}}(X)\mathbb{1}_{\mathsf{A}_\eta}(X) - \mathbb{E}\left[g_{\mathbf{B}}(X)\mathbb{1}_{\mathsf{A}_\eta}(X)\right]\right\}\right)\right] \leq \exp\left(-c_2 s^2 \frac{(1+\sqrt{m}\mathsf{L}_G)^2}{d(n+m)(\eta+\lambda_d(\mathbf{D}))^3}\right),$$

Putting the previous bounds all together, we have shown that the moment generating function of $h_{\mathbf{B}}(X)$ is bounded for a universal constant $c$ by,

$$\mathbb{E}\left[\exp\left(s\left\{h_{\mathbf{B}}(\tilde{X}) - \mathbb{E}\left[h_{\mathbf{B}}(\tilde{X})\right]\right\}\right)\right]$$
$$\leq \exp\left(-cs^2\left\{\frac{(1+\sqrt{m}\mathsf{L}_G)^2}{d(n+m)(\eta+\lambda_d(\mathbf{D}))^3} + \frac{\sigma_G^2}{n(\eta+\lambda_d(\mathbf{D}))^3}\left(\frac{1-\alpha}{d} + \eta + \lambda_d(\mathbf{D})\right)\right\}\right)$$
$$= \exp\left(-cs^2\frac{1}{n(\eta+\lambda_d(\mathbf{D}))^3}\left\{\frac{\alpha}{d}\left(1+\sqrt{m}\mathsf{L}_G\right)^2 + \frac{(1-\alpha)\sigma_X^2}{d} + \sigma_G(\eta+\lambda_d(\mathbf{D}))\right\}\right).$$

Relying on the Chernoff's bound, and usual computations, the claimed concentration bound follows.

**A first equivalent for $\mathbb{E}\left[\mathrm{R}_{\mathrm{Aug}}(\mathbf{D})\mathbb{1}_{\mathsf{A}_\eta}(X)\right]$.**

We now focus on the bias of $\mathrm{R}_{\mathrm{Aug}}(\mathbf{D})\mathbb{1}_{\mathsf{A}_\eta}(X)$. We will show that $\bar{\mathrm{R}}_{\mathrm{Aug}}^{(\mathfrak{a}_x^*,\mathfrak{a}_g^*)}(\mathbf{D})$ is close to $\mathbb{E}\left[\mathrm{R}_{\mathrm{Aug}}(\mathbf{D})\mathbb{1}_{\mathsf{A}_\eta}(X)\right]$.

As a first step, let us notice that conditionally to $X$, $G$ satisfies all the assumptions of Proposition 8, hence for any $\sigma(X)$-measurable matrix $D_X$, $\mathrm{R}_G(D_X)$ admist a deterministic equivalent conditionally to $X$. We will through a slight abuse of notation denote $\bar{\mathrm{R}}_{G|X}^{\mathfrak{a}_g(X)}(D_X)$ the equivalent of $\mathrm{R}_G(D_X)$ conditionally to $X$. It is given by Proposition 8, i.e,

$$\bar{\mathrm{R}}_{G|X}^{\mathfrak{a}_g(X)}(D_X) = \left(\frac{\mathbb{E}\left[C_G \mid X\right]}{\mathfrak{a}_g(X)} + D_X\right)^{-1}\mathbb{1}_{\mathsf{A}_\eta}(X) = \left(\frac{\beta C_X}{\mathfrak{a}_g(X)} + \frac{\Lambda_G(X)}{\mathfrak{a}_g(X)} + D_X\right)^{-1}\mathbb{1}_{\mathsf{A}_\eta}(X),$$

$$\mathfrak{a}_g(X) = 1 + \frac{1}{m}\mathrm{tr}\left(\mathbb{E}\left[C_G \mid X\right]\mathbb{E}\left[R_G(D_X) \mid X\right]\right)$$
$$= 1 + \frac{1}{m}\mathrm{tr}\left(\{\beta C_X + \Lambda_G(X)\}\mathbb{E}\left[R_G(D_X) \mid X\right]\right).$$

Writting for simplicity, $\alpha' = (1 - \beta/\mathfrak{a}_g^*)\alpha$, and $\alpha'(X) = (1 - \beta/\mathfrak{a}_g(X))\alpha$, we will rely on the following upper bound which follows from the triangle inequality,

$$\left\|\mathbb{E}\left[\left\{\mathrm{R}_{\mathrm{Aug}}(\mathbf{D}) - \bar{\mathrm{R}}_{\mathrm{Aug}}^{(\mathfrak{a}_x^*,\mathfrak{a}_g^*)}(\mathbf{D})\right\}\mathbb{1}_{\mathsf{A}_\eta}(X)\right]\right\|_{\mathrm{F}}$$
$$\leq \left\|\mathbb{E}\left[\left\{\mathrm{R}_{\mathrm{Aug}}(\mathbf{D}) - \frac{1}{\alpha}\bar{\mathrm{R}}_{G|X}^{\mathfrak{a}_g(X)}\left(\frac{1-\alpha}{\alpha}C_X + \frac{1}{\alpha}\mathbf{D}\right)\right\}\mathbb{1}_{\mathsf{A}_\eta}(X)\right]\right\|_{\mathrm{F}}$$
$$+ \left\|\mathbb{E}\left[\left\{\frac{1}{\alpha}\bar{\mathrm{R}}_{G|X}^{\mathfrak{a}_g(X)}\left(\frac{1-\alpha}{\alpha}C_X + \frac{1}{\alpha}\mathbf{D}\right) - \frac{1}{1-\alpha'}\mathrm{R}_X\left(\frac{\alpha\bar{\Lambda}_G}{(1-\alpha')\mathfrak{a}_g^*} + \frac{1}{1-\alpha'}\mathbf{D}\right)\right\}\mathbb{1}_{\mathsf{A}_\eta}(X)\right]\right\|_{\mathrm{F}}$$
$$+ \left\|\mathbb{E}\left[\left\{\frac{1}{1-\alpha'}\mathrm{R}_X\left(\frac{\alpha\bar{\Lambda}_G}{(1-\alpha')\mathfrak{a}_g^*} + \frac{1}{1-\alpha'}\mathbf{D}\right) - \bar{\mathrm{R}}_{\mathrm{Aug}}^{(\mathfrak{a}_x^*,\mathfrak{a}_g^*)}(\mathbf{D})\right\}\mathbb{1}_{\mathsf{A}_\eta}(X)\right]\right\|_{\mathrm{F}}$$

and, remark that we can rewrite,

$$\mathrm{R}_{\mathrm{Aug}}(\mathbf{D}) = ((1-\alpha)C_X + \alpha C_G + \mathbf{D})^{-1} = \frac{1}{\alpha}\mathrm{R}_G\left(\frac{(1-\alpha)}{\alpha}C_X + \frac{1}{\alpha}\mathbf{D}\right),$$

$$\frac{1}{\alpha}\bar{\mathrm{R}}_{G|X}^{\mathfrak{a}_g(X)}\left(\frac{1-\alpha}{\alpha}C_X + \frac{1}{\alpha}\mathbf{D}\right)\mathbb{1}_{\mathsf{A}_\eta}(X) = \frac{1}{\alpha}\left(\frac{\mathbb{E}\left[C_G \mid X\right]}{\mathfrak{a}_g(X)} + \frac{1-\alpha}{\alpha}C_X + \mathbf{D}\right)^{-1}\mathbb{1}_{\mathsf{A}_\eta}(X)$$
$$= \left(\left(1 - \left(1 - \frac{\beta}{\mathfrak{a}_g(X)}\right)\alpha\right)C_X + \frac{\alpha\Lambda_G(X)}{\mathfrak{a}_g(X)} + \mathbf{D}\right)^{-1}\mathbb{1}_{\mathsf{A}_\eta}(X)$$
$$= \left((1-\alpha'(X))C_X + \frac{\alpha\Lambda_G(X)}{\mathfrak{a}_g(X)} + \mathbf{D}\right)^{-1}\mathbb{1}_{\mathsf{A}_\eta}(X),$$

$$\frac{1}{1-\alpha'}\,R_X\left(\frac{\alpha\bar{\Lambda}_G}{(1-\alpha')\mathfrak{a}_g^*}+\frac{1}{1-\alpha'}\mathbf{D}\right)\mathbb{1}_{A_\eta}(X)=\left((1-\alpha')C_X+\frac{\alpha\bar{\Lambda}_G}{\mathfrak{a}_g^*}+\mathbf{D}\right)^{-1}\mathbb{1}_{A_\eta}(X)\,,$$

and,

$$\bar{R}_{\text{Aug}}^{(\mathfrak{a}_x^*,\mathfrak{a}_g^*)}(\mathbf{D})\mathbb{1}_{A_\eta}(X)=\frac{1}{1-\alpha'}\bar{R}_X^{\mathfrak{a}_x^*}\left(\frac{\alpha\bar{\Lambda}_G}{(1-\alpha')\mathfrak{a}_g^*}+\frac{1}{1-\alpha'}\mathbf{D}\right)\mathbb{1}_{A_\eta}(X)\,.$$

Using these equalities, we rewrite the previous bound as,

$$\left\|\mathbb{E}\left[\left\{R_{\text{Aug}}(\mathbf{D})-\bar{R}_{\text{Aug}}^{(\mathfrak{a}_x^*,\mathfrak{a}_g^*)}(\mathbf{D})\right\}\mathbb{1}_{A_\eta}(X)\right]\right\|_{\text{F}} \tag{43}$$

$$\leq\left\|\mathbb{E}\left[\left\{\frac{1}{\alpha}R_G\left(\frac{(1-\alpha)C_X}{\alpha}+\frac{1}{\alpha}\mathbf{D}\right)-\frac{1}{\alpha}\bar{R}_{G|X}^{\mathfrak{a}_g(X)}\left(\frac{1-\alpha}{\alpha}C_X+\frac{1}{\alpha}\mathbf{D}\right)\right\}\mathbb{1}_{A_\eta}(X)\right]\right\|_{\text{F}}$$

$$+\left\|\mathbb{E}\left[\left\{\left((1-\alpha'(X))C_X+\frac{\alpha\Lambda_G(X)}{\mathfrak{a}_g(X)}+\mathbf{D}\right)^{-1}-\left((1-\alpha')C_X+\frac{\alpha\bar{\Lambda}_G}{\mathfrak{a}_g}+\mathbf{D}\right)^{-1}\right\}\mathbb{1}_{A_\eta}(X)\right]\right\|_{\text{F}}$$

$$+\left\|\mathbb{E}\left[\left\{\frac{1}{1-\alpha'}R_X\left(\frac{\alpha\bar{\Lambda}_G}{\mathfrak{a}_g^*}+\frac{1}{1-\alpha'}\mathbf{D}\right)-\frac{1}{1-\alpha'}\bar{R}_X^{\mathfrak{a}_x^*}\left(\frac{\alpha\bar{\Lambda}_G}{(1-\alpha')\mathfrak{a}_g^*}+\frac{1}{1-\alpha'}\mathbf{D}\right)\right\}\mathbb{1}_{A_\eta}(X)\right]\right\|_{\text{F}}$$

The first and third terms in (43) are bounded by Proposition 8, whereas the second term is controlled from the fact that $\mathfrak{a}_g(X)$ and $\Lambda_G(X)$ have small deviations. We deal with each of the terms in the right hand side of the previous upper bound one by one, first using the Jensen's inequality,

$$\left\|\mathbb{E}\left[\left\{\frac{1}{\alpha}R_G\left(\frac{(1-\alpha)C_X}{\alpha}+\frac{1}{\alpha}\mathbf{D}\right)-\frac{1}{\alpha}\bar{R}_{G|X}^{\mathfrak{a}_g(X)}\left(\frac{1-\alpha}{\alpha}C_X+\frac{1}{\alpha}\mathbf{D}\right)\right\}\mathbb{1}_{A_\eta}(X)\right]\right\|_{\text{F}}$$

$$\leq\frac{1}{\alpha}\mathbb{E}\left[\left\|\mathbb{E}\left[\left\{R_G\left(\frac{(1-\alpha)C_X}{\alpha}+\frac{1}{\alpha}\mathbf{D}\right)-\bar{R}_G^{\mathfrak{a}_g(X)}\left(\frac{1-\alpha}{\alpha}C_X+\frac{1}{\alpha}\mathbf{D}\right)\right\}\Big|X\right]\right\|_{\text{F}}\mathbb{1}_{A_\eta}(X)\right]$$

Remarking that for all $\mathbf{X}\in A_\eta$, we have $\lambda_d((1-\alpha)/\alpha C_\mathbf{X}+\mathbf{D}/\alpha)\geq\alpha^{-1}((1-\alpha)\eta+\lambda_d(\mathbf{D}))$. We set $\epsilon=(1-\alpha)\eta/\alpha+\lambda_d(\mathbf{D})/\alpha$, we have from the previous remark and using proposition 8,

$$\left\|\mathbb{E}\left[\left\{\frac{1}{\alpha}R_G\left(\frac{(1-\alpha)C_X}{\alpha}+\frac{1}{\alpha}\mathbf{D}\right)-\frac{1}{\alpha}\bar{R}_{G|X}^{\mathfrak{a}_g(X)}\left(\frac{1-\alpha}{\alpha}C_X+\frac{1}{\alpha}\mathbf{D}\right)\right\}\mathbb{1}_{A_\eta}(X)\right]\right\|_{\text{F}}$$

$$\leq\frac{1}{\alpha}\mathbb{E}\left[\left\|\mathbb{E}\left[\left\{R_G\left(\frac{(1-\alpha)C_X}{\alpha}+\frac{1}{\alpha}\mathbf{D}\right)-\bar{R}_{G|X}^{\mathfrak{a}_g(X)}\left(\frac{1-\alpha}{\alpha}C_X+\frac{1}{\alpha}\mathbf{D}\right)\right\}\Big|X\right]\right\|_{\text{F}}\mathbb{1}_{A_\eta}(X)\right]$$

$$\lesssim\frac{\sigma_G^2}{\alpha}\mathbb{E}\left[\left\{Q(\epsilon,\lambda_d(\mathbb{E}\left[C_G\mid X\right]),\|\mathbb{E}\left[C_G\mid X\right]\|_{\text{op}}^{-1},0,n^{-1})\frac{\sqrt{d}\|\mathbb{E}\left[C_G\mid X\right]\|_{\text{op}}^3}{n\lambda_d(\mathbb{E}\left[C_G\mid X\right])\epsilon^6}\right\}\mathbb{1}_{A_\eta}(X)\right]\,,$$

where $Q$ is the polynomial function defined in Proposition 8. In order to integrate the above error over the distribution of $X$, one needs to ensure that this random quantity doesn't blow up. We deal with this, first by using the fact that $Q$ is non-decreasing with respect to it's third entry, and we recall the notation $\kappa$, such that,

$$Sp(\Lambda_G(X))\subset[\kappa^{-1},\kappa]\quad\text{a.s}\,,$$

and using $\|\mathbb{E}\left[C_G\mid X\right]\|_{\text{op}}\geq\inf_\mathbf{X}\|\Lambda_G(\mathbf{X})\|_{\text{op}}\geq\kappa$, we write,

$$Q(\epsilon,\lambda_d(\mathbb{E}\left[C_G\mid X\right]),\|\mathbb{E}\left[C_G\mid X\right]\|_{\text{op}}^{-1},0,n^{-1})\leq Q(\epsilon,\lambda_d(\mathbb{E}\left[C_G\mid X\right]),(\inf_\mathbf{X}\|\Lambda_G(\mathbf{X})\|_{\text{op}})^{-1},0,n^{-1})$$

$$\leq Q(\epsilon,\lambda_d(\mathbb{E}\left[C_G\mid X\right]),\kappa,0,n^{-1})$$

and, remarking that there exists two polwnomials $q_1$ and $q_2$ (polynomials in $\epsilon,\kappa,n^{-1}$) such that,

$$Q(\epsilon,\lambda_d(\mathbb{E}\left[C_G\mid X\right]),\kappa,0,n^{-1})=q_1+\lambda_d(\mathbb{E}\left[C_G\mid X\right])q_2\,,$$

which is trivial from the definition of $Q$ in Proposition 1. We write,

$$\mathbb{E}\left[\left\{Q(\epsilon,\lambda_d(\mathbb{E}\left[C_G\mid X\right]),\|\mathbb{E}\left[C_G\mid X\right]\|_{\text{op}}^{-1},0,n^{-1})\frac{\sqrt{d}\|\mathbb{E}\left[C_G\mid X\right]\|_{\text{op}}^3}{n\lambda_d(\mathbb{E}\left[C_G\mid X\right])\epsilon^6}\right\}\mathbb{1}_{A_\eta}(X)\right]$$

$$=q_1\mathbb{E}\left[q_1\frac{\|\mathbb{E}\left[C_G\mid X\right]\|_{\text{op}}^3}{n\lambda_d(\mathbb{E}\left[C_G\mid X\right])}\mathbb{1}_{A_\eta}(X)\right]+q_2\frac{\sqrt{d}\mathbb{E}\left[\|\mathbb{E}\left[C_G\mid X\right]\|_{\text{op}}^3\mathbb{1}_{A_\eta}(X)\right]}{n\epsilon^6}\,,$$

and, we control the remaining $\lambda_d(\mathbb{E}\left[C_G \mid X\right])$ term by remarking that $\lambda_d(\mathbb{E}\left[C_G \mid X\right]) = \lambda_d(\beta C_X + \Lambda_G(X)) \geq \inf_{\mathbf{X}} \lambda_d(\Lambda_G(\mathbf{X})) \geq \kappa^{-1}$, hence,

$$\mathbb{E}\left[\left\{Q(\epsilon, \lambda_d(\mathbb{E}\left[C_G \mid X\right]), \|\mathbb{E}\left[C_G \mid X\right]\|_{\mathrm{op}}^{-1}, 0, n^{-1}) \frac{\sqrt{d}\|\mathbb{E}\left[C_G \mid X\right]\|_{\mathrm{op}}^3}{n\lambda_d(\mathbb{E}\left[C_G \mid X\right])\epsilon^6}\right\} \mathbb{1}_{\mathsf{A}_\eta}(X)\right]$$

$$= (\kappa q_1 + q_2) \frac{\sqrt{d}\mathbb{E}\left[\|\mathbb{E}\left[C_G \mid X\right]\|_{\mathrm{op}}^3 \mathbb{1}_{\mathsf{A}_\eta}(X)\right]}{n\epsilon^6} ,$$

It remains only to control the term $\mathbb{E}\left[\|\mathbb{E}\left[C_G \mid X\right]\|_{\mathrm{op}}^3 \mathbb{1}_{\mathsf{A}_\eta}(X)\right]$. Recall from **H3** that $\mathbb{E}\left[C_G \mid X\right] = \beta C_X + \Lambda_G(X)$, which thanks to $(a + b)^3 \lesssim a^3 + b^3$ implies,

$$\mathbb{E}\left[\|\mathbb{E}\left[C_G \mid X\right]\|_{\mathrm{op}}^3 \mathbb{1}_{\mathsf{A}_\eta}(X)\right] \lesssim \beta^3 \mathbb{E}\left[\|C_X - \Sigma_X\|_{\mathrm{op}}^3\right] + \|\beta\Sigma_X + \Lambda_G(X)\|_{\mathrm{op}}^3$$

$$\lesssim \beta^3 \mathbb{E}\left[\|C_X - \Sigma_X\|_{\mathrm{op}}^3\right] + \beta^3 \|\Sigma_X\|_{\mathrm{op}}^3 + \kappa^3 .$$

hence,

$$\left\|\mathbb{E}\left[\left\{\frac{1}{\alpha}\mathrm{R}_G\left(\frac{(1-\alpha)C_X}{\alpha} + \frac{1}{\alpha}\mathbf{D}\right) - \frac{1}{\alpha}\bar{\mathrm{R}}_{G|X}^{\mathfrak{a}_g(X)}\left(\frac{1-\alpha}{\alpha}C_X + \frac{1}{\alpha}\mathbf{D}\right)\right\} \mathbb{1}_{\mathsf{A}_\eta}(X)\right]\right\|_{\mathrm{F}} \quad (44)$$

$$\lesssim (\kappa q_1 + q_2) \frac{\sqrt{d}\sigma_G^2(\beta^3 \mathbb{E}\left[\|C_X - \Sigma_X\|_{\mathrm{op}}^3\right] + \beta^3 \|\Sigma_X\|_{\mathrm{op}}^3 + \kappa^3)}{n\alpha\epsilon^6} .$$

It remains only to handle the deviation of $C_X$ in operator norm. To this end, we rely on [Ver18] result 9.2.5, which states that for a universal constant $K$,

$$\mathbb{P}\left(\|C_X - \Sigma_X\|_{\mathrm{op}} \geq K\sigma_X^4 \left(\sqrt{\frac{r + u}{n}} + \frac{r + u}{n}\right) \|\Sigma_X\|_{\mathrm{op}}\right) \leq 2\mathrm{e}^{-u} , \quad r = \frac{\mathrm{tr}\left(\Sigma_X\right)}{\|\Sigma_X\|_{\mathrm{op}}}$$

Note that [Ver18] states this result in the case of $X$ having sub-Gaussian columns, which in our setting is a direct consequence of Definition 1. In particular, we write,

$$\varphi(u) = K\sigma_X^4 \left(\sqrt{\frac{r + u}{n}} + \frac{r + u}{n}\right) \|\Sigma_X\|_{\mathrm{op}} , \quad \text{thus} \quad \varphi'(u) = K\sigma_X^4 \left(\frac{1}{2\sqrt{n(r + u)}} + \frac{1}{n}\right) \|\Sigma_X\|_{\mathrm{op}} .$$

and, by the change of variable $t = \varphi(u)$, we have,

$$\mathbb{E}\left[\|C_X - \Sigma_X\|_{\mathrm{op}}^3\right] = \int_0^\infty \mathbb{P}\big(\|C_X - \Sigma_X\|_{\mathrm{op}}^3 \geq t\big) \, dt = \int_0^\infty \mathbb{P}\big(\|C_X - \Sigma_X\|_{\mathrm{op}} \geq t^{1/3}\big) \, dt$$

$$= \int_0^\infty \mathbb{P}\big(\|C_X - \Sigma_X\|_{\mathrm{op}} \geq \varphi(u)\big) \, 3\,\varphi(u)^2 \, \varphi'(u) \, du$$

$$\leq 2 \int_0^\infty \mathrm{e}^{-u} \, 3\,\varphi(u)^2 \, \varphi'(u) \, du \; \leq \; 6\,\varphi'(0) \int_0^\infty \mathrm{e}^{-u} \, \varphi(u)^2 \, du$$

$$\leq 6\varphi'(0)\frac{K^2\sigma_X^8}{n} \int_0^\infty \mathrm{e}^{-u}(r + u) + 12\varphi'(0)\frac{K^2\sigma_X^8}{n^2} \int_0^\infty \mathrm{e}^{-u}(r + u)^2$$

$$\leq 6K^3\sigma_X^{12}\left(\frac{1}{2\sqrt{nr}} + \frac{1}{n}\right)\left(\frac{r + 1}{n} + \frac{2(r^2 + 2r + 2)}{n^2}\right) ,$$

recalling that $r = \mathrm{tr}\left(\Sigma_X\right)/\|\Sigma_x\|_{\mathrm{op}}$, and noticing $d\lambda_d(\Sigma_X)/\|\Sigma_X\|_{\mathrm{op}} \leq r \leq d$, it results that

$$\mathbb{E}\left[\|C_X - \Sigma_X\|_{\mathrm{op}}^3\right] \lesssim \sigma_X^{12}\left(\frac{\|\Sigma_X\|_{\mathrm{op}}}{\sqrt{n}d\lambda_d(\Sigma_X)} + \frac{1}{n}\right)\left(\frac{d + 1}{n} + \frac{d^2 + d + 2}{n^2}\right)$$

$$\lesssim \sigma_X^{12}\left(\frac{\|\Sigma_X\|_{\mathrm{op}}}{\sqrt{n}d\lambda_d(\Sigma_X)} + \frac{1}{n}\right)$$

where we have used $d < n$, garenteed by **H**2. We plug this into (44), and we get,

$$\left\|\mathbb{E}\left[\left\{\frac{1}{\alpha}R_G\left(\frac{(1-\alpha)C_X}{\alpha}+\frac{1}{\alpha}\mathbf{D}\right)-\frac{1}{\alpha}\bar{R}_{G|X}^{\mathfrak{a}_g(X)}\left(\frac{1-\alpha}{\alpha}C_X+\frac{1}{\alpha}\mathbf{D}\right)\right\}\mathbb{1}_{\mathsf{A}_\eta}(X)\right]\right\|_{\mathrm{F}} \quad (45)$$

$$\lesssim (\kappa q_1+q_2)\left\{\frac{\sqrt{d}\sigma_G^2(\beta^3\|\Sigma_X\|_{\mathrm{op}}^3+\kappa^3)}{n\alpha\epsilon^6}+\frac{\sqrt{d}\sigma_G^{12}}{n\alpha\epsilon^6}\left(\frac{\|\Sigma_X\|_{\mathrm{op}}}{\sqrt{n}d\lambda_d(\Sigma_X)}+\frac{1}{n}\right)\right\}$$

$$\lesssim \frac{(\kappa q_1+q_2)\sqrt{d}\left\{\sigma_G^2(\beta^3\|\Sigma_X\|_{\mathrm{op}}^3+\kappa^3)+\sigma_X^{12}\|\Sigma_X\|_{\mathrm{op}}\lambda_d(\Sigma_X)^{-1}n^{-1/2}d^{-1}\right\}}{n\alpha\epsilon^6}$$

$$\lesssim \frac{\alpha^5(\kappa q_1+q_2)\sqrt{d}\left\{\sigma_G^2(\beta^3\|\Sigma_X\|_{\mathrm{op}}^3+\kappa^3)+\sigma_X^{12}\|\Sigma_X\|_{\mathrm{op}}\lambda_d(\Sigma_X)^{-1}n^{-1/2}d^{-1}\right\}}{n((1-\alpha)\eta+\lambda_d(\mathbf{D}))^6}$$

This concludes our analysis of the first term in (43). We now focus on the second term in Equation (43), which is controlled, provided $\mathbb{E}\left[\|\Lambda_G(X)-\bar{\Lambda}_G\|_{\mathrm{F}}\right]$ is small.

First, we check that,

$$\mathfrak{a}_g(X) = 1 + \frac{1}{m}\operatorname{tr}\left(\{\beta C_X+\Lambda_G(X)\}\mathbb{E}\left[R_G\left((1-\alpha)C_X/\alpha+\mathbf{D}/\alpha\right)\mid X\right]\right)$$

$$= 1 + \frac{1}{m}\operatorname{tr}\left(\{\beta C_X+\Lambda_G(X)\}\mathbb{E}\left[\alpha\left((1-\alpha)C_X+\alpha C_G+\mathbf{D}\right)^{-1}\mid X\right]\right)$$

$$= 1 + \frac{\alpha}{m}\operatorname{tr}\left(\{\beta C_X+\Lambda_G(X)\}\mathbb{E}\left[R_{\mathrm{Aug}}(\mathbf{D})\mid X\right]\right)$$

From this, one can hope that $\mathfrak{a}_g(X)$ concentrates around $\mathfrak{a}_g^* = \mathbb{E}\left[\mathfrak{a}_g(X)\right]$, hence, we write,

$$\left\|\mathbb{E}\left[\left\{\left((1-\alpha'(X))C_X+\frac{\alpha\Lambda_G(X)}{\mathfrak{a}_g(X)}+\mathbf{D}\right)^{-1}-\left((1-\alpha')C_X+\frac{\alpha\bar{\Lambda}_G}{\mathfrak{a}_g^*}+\mathbf{D}\right)^{-1}\right\}\mathbb{1}_{\mathsf{A}_\eta}(X)\right]\right\|_{\mathrm{F}}$$

$$\leq \left\|\mathbb{E}\left[\left\{\left((1-\alpha'(X))C_X+\frac{\alpha\Lambda_G(X)}{\mathfrak{a}_g(X)}+\mathbf{D}\right)^{-1}-\left((1-\alpha')C_X+\frac{\alpha\Lambda_G(X)}{\mathfrak{a}_g(X)}+\mathbf{D}\right)^{-1}\right\}\mathbb{1}_{\mathsf{A}_\eta}(X)\right]\right\|_{\mathrm{F}}$$

$$+ \left\|\mathbb{E}\left[\left\{\left((1-\alpha')C_X+\frac{\alpha\Lambda_G(X)}{\mathfrak{a}_g(X)}+\mathbf{D}\right)^{-1}-\left((1-\alpha')C_X+\frac{\alpha\bar{\Lambda}_G}{\mathfrak{a}_g^*}+\mathbf{D}\right)^{-1}\right\}\mathbb{1}_{\mathsf{A}_\eta}(X)\right]\right\|_{\mathrm{F}}$$

furthermore, relying on the identity $\mathbf{A}^{-1}-\mathbf{B}^{-1}$, we write,

$$\left\|\mathbb{E}\left[\left\{\left((1-\alpha'(X))C_X+\frac{\alpha\Lambda_G(X)}{\mathfrak{a}_g(X)}+\mathbf{D}\right)^{-1}-\left((1-\alpha')C_X+\frac{\alpha\Lambda_G(X)}{\mathfrak{a}_g(X)}+\mathbf{D}\right)^{-1}\right\}\mathbb{1}_{\mathsf{A}_\eta}(X)\right]\right\|_{\mathrm{F}}$$

$$= \left\|\mathbb{E}\left[(\alpha'(X)-\alpha')\left\{\left((1-\alpha'(X))C_X+\frac{\alpha\Lambda_G(X)}{\mathfrak{a}_g(X)}+\mathbf{D}\right)^{-1}C_X\left((1-\alpha')C_X+\frac{\alpha\Lambda_G(X)}{\mathfrak{a}_g(X)}+\mathbf{D}\right)^{-1}\right\}\mathbb{1}_{\mathsf{A}_\eta}(X)\right]\right\|_{\mathrm{F}}$$

$$\leq \mathbb{E}\left[|\alpha'(X)-\alpha'|\left\|\left((1-\alpha'(X))C_X+\frac{\alpha\Lambda_G(X)}{\mathfrak{a}_g(X)}+\mathbf{D}\right)^{-1}C_X\left((1-\alpha')C_X+\frac{\alpha\Lambda_G(X)}{\mathfrak{a}_g(X)}+\mathbf{D}\right)^{-1}\right\|_{\mathrm{F}}\mathbb{1}_{\mathsf{A}_\eta}(X)\right],$$

Now, remarking that

$$\left\|C_X\left((1-\alpha')C_X+\frac{\alpha\Lambda_G(X)}{\mathfrak{a}_g(X)}+\mathbf{D}\right)^{-1}\right\|_{\mathrm{op}}\mathbb{1}_{\mathsf{A}_\eta}(X)\leq\frac{1}{1-\alpha'},$$

and,

$$\left\|\left((1-\alpha'(X))C_X+\alpha\frac{\Lambda_G(X)}{\mathfrak{a}_g(X)}+\mathbf{D}\right)^{-1}\right\|_{\mathrm{op}}\mathbb{1}_{\mathsf{A}_\eta}(X)\leq\frac{1}{(1-\alpha'(X))\eta+\lambda_d(\mathbf{D})}\leq\frac{1}{(1-\alpha)\eta+\lambda_d(\mathbf{D})}.$$

Similarly,

$$\left\| \mathbb{E}\left[ \left\{ \left( (1-\alpha')C_X + \frac{\alpha\Lambda_G(X)}{\mathfrak{a}_g(X)} + \mathbf{D} \right)^{-1} - \left( (1-\alpha')C_X + \frac{\alpha\bar{\Lambda}_G}{\mathfrak{a}_g^*} + \mathbf{D} \right)^{-1} \right\} \mathbb{1}_{\mathsf{A}_\eta}(X) \right] \right\|_{\mathrm{F}}$$

$$\leq \alpha\mathbb{E}\left[ \left| \frac{1}{\mathfrak{a}_g(X)} - \frac{1}{\mathfrak{a}_g^*} \right| \left\| \left( (1-\alpha')C_X + \frac{\alpha\Lambda_G(X)}{\mathfrak{a}_g(X)} + \mathbf{D} \right)^{-1} \Lambda_G(X) \left( (1-\alpha')C_X + \frac{\alpha\bar{\Lambda}_G}{\mathfrak{a}_g^*} + \mathbf{D} \right)^{-1} \right\|_{\mathrm{F}} \mathbb{1}_{\mathsf{A}_\eta}(X) \right]$$

$$+ \frac{\alpha}{\mathfrak{a}_g}\mathbb{E}\left[ \left\| \left( (1-\alpha')C_X + \frac{\alpha\Lambda_G(X)}{\mathfrak{a}_g(X)} + \mathbf{D} \right)^{-1} \left\{ \Lambda_G(X) - \bar{\Lambda}_G \right\} \left( (1-\alpha')C_X + \frac{\alpha\Lambda_G(X)}{\mathfrak{a}_g(X)} + \mathbf{D} \right)^{-1} \right\|_{\mathrm{F}} \right]$$

$$\leq \frac{\mathbb{E}\left[ \left| \mathfrak{a}_g(X) - \mathfrak{a}_g^* \right| \right]}{(1-\alpha)\eta + \lambda_1(\mathbf{D})} + \frac{\mathbb{E}\left[ \|\Lambda_G(X) - \bar{\Lambda}_G\|_{\mathrm{F}} \right]}{((1-\alpha)\eta + \lambda_d(\mathbf{D}))^2} \;,$$

which results in

$$\left\| \mathbb{E}\left[ \left\{ \left( (1-\alpha'(X))C_X + \frac{\alpha\Lambda_G(X)}{\mathfrak{a}_g(X)} + \mathbf{D} \right)^{-1} - \left( (1-\alpha')C_X + \frac{\alpha\Lambda_G(X)}{\mathfrak{a}_g(X)} + \mathbf{D} \right)^{-1} \right\} \mathbb{1}_{\mathsf{A}_\eta}(X) \right] \right\|_{\mathrm{F}} \quad (46)$$

$$\leq \frac{\mathbb{E}\left[ |\alpha'(X) - \alpha'| \right](1-\alpha')^{-1}}{(1-\alpha)\eta + \lambda_d(\mathbf{D})} + \frac{\mathbb{E}\left[ \left| \mathfrak{a}_g(X) - \mathfrak{a}_g^* \right| \right]}{(1-\alpha)\eta + \lambda_1(\mathbf{D})} + \frac{\mathbb{E}\left[ \|\Lambda_G(X) - \bar{\Lambda}_G\|_{\mathrm{F}} \right]}{((1-\alpha)\eta + \lambda_d(\mathbf{D}))^2}$$

$$\leq \frac{\alpha\beta\mathbb{E}\left[ \left| \mathfrak{a}_g(X) - \mathfrak{a}_g^* \right| \right](1-\alpha')^{-1}}{(1-\alpha)\eta + \lambda_d(\mathbf{D})} + \frac{\mathbb{E}\left[ \left| \mathfrak{a}_g(X) - \mathfrak{a}_g^* \right| \right]}{(1-\alpha)\eta + \lambda_1(\mathbf{D})} + \frac{\mathbb{E}\left[ \|\Lambda_G(X) - \bar{\Lambda}_G\|_{\mathrm{F}} \right]}{((1-\alpha)\eta + \lambda_d(\mathbf{D}))^2}$$

$$\leq \frac{\alpha\beta(1-\alpha')^{-1} + 1}{(1-\alpha)\eta + \lambda_d(\mathbf{D})}\mathbb{E}\left[ \left| \mathfrak{a}_g(X) - \mathfrak{a}_g^* \right| \right] + \frac{\mathbb{E}\left[ \|\Lambda_G(X) - \bar{\Lambda}_G\|_{\mathrm{F}} \right]}{((1-\alpha)\eta + \lambda_d(\mathbf{D}))^2} \;.$$

From the previous, one can see that controlling $\mathbb{E}\left[ \|\Lambda_G(X) - \bar{\Lambda}_G\|_{\mathrm{F}} \right]$ and $\mathbb{E}\left[ \left| \mathfrak{a}_g(X) - \mathfrak{a}_g^* \right| \right]$ is sufficient in order to control the second term in decomposition (43). While the first needs to be assumed small, we can show that $\mathbb{E}\left[ |\mathfrak{a}_g(X) - \mathfrak{a}_g| \right]$ is small under quite general conditions, to this end, we write,

$$\left| \mathfrak{a}_g(X) - \mathfrak{a}_g^* \right|$$

$$\leq \frac{\alpha\beta}{m}\left| \mathrm{tr}\left( \mathbb{E}\left[ C_X\,\mathrm{R}_{\mathrm{Aug}}(\mathbf{D})\mathbb{1}_{\mathsf{A}_\eta}(X) \mid X \right] - \mathbb{E}\left[ C_X\,\mathrm{R}_{Aug}(\mathbf{D})\mathbb{1}_{\mathsf{A}_\eta}(X) \right] \right) \right|$$

$$+ \frac{\alpha}{m}\left| \mathrm{tr}\left( \mathbb{E}\left[ \Lambda_G(X)\,\mathrm{R}_{\mathrm{Aug}}(\mathbf{D})\mathbb{1}_{\mathsf{A}_\eta}(X) \mid X \right] - \mathbb{E}\left[ \Lambda_G(X)\,\mathrm{R}_{\mathrm{Aug}}(\mathbf{D})\mathbb{1}_{\mathsf{A}_\eta}(X) \right] \right) \right|$$

$$\leq \frac{\alpha\beta}{mn}\sum_{i=1}^n\left| \mathrm{tr}\left( \mathbb{E}\left[ X_iX_i^\top\,\mathrm{R}_{\mathrm{Aug}}(\mathbf{D})\mathbb{1}_{\mathsf{A}_\eta}(X) \mid X \right] - \mathbb{E}\left[ X_iX_i^\top\,\mathrm{R}_{\mathrm{Aug}}(\mathbf{D})\mathbb{1}_{\mathsf{A}_\eta}(X) \right] \right) \right|$$

$$+ \frac{\alpha}{m}\left| \mathrm{tr}\left( \mathbb{E}\left[ \Lambda_G(X)\,\mathrm{R}_{\mathrm{Aug}}(\mathbf{D})\mathbb{1}_{\mathsf{A}_\eta}(X) \mid X \right] - \mathbb{E}\left[ \Lambda_G(X)\,\mathrm{R}_{\mathrm{Aug}}(\mathbf{D})\mathbb{1}_{\mathsf{A}_\eta}(X) \right] \right) \right| \;.$$

Now remark that the distribution of $\left| \mathrm{tr}\left( \mathbb{E}\left[ X_iX_i^\top\,\mathrm{R}_{\mathrm{Aug}}(\lambda) \mid X \right] - \mathbb{E}\left[ X_iX_i^\top\,\mathrm{R}_{\mathrm{Aug}}(\lambda) \right] \right) \right|$ doesn't depend on $i$, by exchangeability of the columns of $X$ and **H5**, we thus focus only on the term $i = 1$, by writting,

$$\mathbb{E}\left[ \left| \mathfrak{a}_g(X) - \mathfrak{a}_g^* \right| \right] \leq \frac{\alpha\beta}{m}\mathbb{E}\left[ \left| \mathbb{E}\left[ X_1^\top\,\mathrm{R}_{\mathrm{Aug}}(\mathbf{D})X_1\mathbb{1}_{\mathsf{A}_\eta}(X) \mid X \right] - \mathbb{E}\left[ X_1^\top\,\mathrm{R}_{\mathrm{Aug}}(\mathbf{D})X_1\mathbb{1}_{\mathsf{A}_\eta}(X) \right] \right| \right]$$

$$+ \frac{\alpha}{m}\mathbb{E}\left[ \left| \mathrm{tr}\left( \mathbb{E}\left[ \Lambda_G(X)\,\mathrm{R}_{\mathrm{Aug}}(\mathbf{D})\mathbb{1}_{\mathsf{A}_\eta}(X) \mid X \right] - \mathbb{E}\left[ \Lambda_G(X)\,\mathrm{R}_{\mathrm{Aug}}(\mathbf{D})\mathbb{1}_{\mathsf{A}_\eta}(X) \right] \right) \right| \right]$$

$$\leq \frac{\alpha\beta}{m}\mathbb{E}\left[ \left| \mathbb{E}\left[ X_1^\top\,\mathrm{R}_{\mathrm{Aug}}(\mathbf{D})X_1\mathbb{1}_{\mathsf{A}_\eta}(X) \mid X \right] - \mathbb{E}\left[ X_1^\top\,\mathrm{R}_{\mathrm{Aug}}(\mathbf{D})X_1\mathbb{1}_{\mathsf{A}_\eta}(X) \right] \right| \right]$$

$$+ \frac{\alpha}{m}\mathbb{E}\left[ \left| \mathrm{tr}\left( \{\Lambda_G(X) - \bar{\Lambda}_G\}\mathbb{E}\left[ \mathrm{R}_{\mathrm{Aug}}(\mathbf{D}) \mid X \right]\mathbb{1}_{\mathsf{A}_\eta}(X) - \mathbb{E}\left[ \{\Lambda_G(X) - \bar{\Lambda}_G\}\,\mathrm{R}_{\mathrm{Aug}}(\mathbf{D})\mathbb{1}_{\mathsf{A}_\eta}(X) \right] \right) \right| \right]$$

$$+ \frac{\alpha}{m}\mathbb{E}\left[ \left| \mathrm{tr}\left( \bar{\Lambda}_G\{\mathbb{E}\left[ \mathrm{R}_{\mathrm{Aug}}(\mathbf{D}) \mid X \right]\mathbb{1}_{\mathsf{A}_\eta}(X) - \mathbb{E}\left[ \mathrm{R}_{\mathrm{Aug}}(\mathbf{D})\mathbb{1}_{\mathsf{A}_\eta}(X) \right] \} \right) \right| \right]$$

where the last inequality followed from using the triangle inequality. To bound the above quantity, we first focus on the second and third terms, which are notably less technical, it holds from Cauchy-

Schwarz inequality and remarking that $\| \mathrm{R}_{\mathrm{Aug}}(\mathbf{D})\mathbb{1}_{\mathsf{A}_\eta}(X)\|_{\mathrm{F}} \le \sqrt{d}(\eta + \lambda_d(\mathbf{D}))^{-1}$ that

$$\frac{\alpha}{m}\mathbb{E}\left[\left|\mathrm{tr}\left(\{\Lambda_G(X) - \bar{\Lambda}_G\}\mathbb{E}\left[\mathrm{R}_{\mathrm{Aug}}(\mathbf{D}) \mid X\right]\mathbb{1}_{\mathsf{A}_\eta}(X) - \mathbb{E}\left[\{\Lambda_G(X) - \bar{\Lambda}_G\}\mathrm{R}_{\mathrm{Aug}}(\mathbf{D})\mathbb{1}_{\mathsf{A}_\eta}(X)\right]\right)\right|\right]$$
$$\le 2\frac{\alpha\sqrt{d}}{m(\eta + \lambda_d(\mathbf{D}))}\mathbb{E}\left[\|\Lambda_G(X) - \bar{\Lambda}_G\|_{\mathrm{F}}\right] \ .$$

Furthermore, the function $\mathbf{X} \mapsto \mathrm{tr}\left(\bar{\Lambda}_G \int \mathrm{R}_{\mathbf{X}\sqcup g}(\mathbf{D})\right) d\nu_{\mathbf{X}}^{\otimes m}(g)$ is $2\sqrt{d}\|\bar{\Lambda}_G\|_{\mathrm{op}}(1 + \sqrt{m}\mathsf{L}_G)(\eta + \lambda_d(\mathbf{D}))^{-3/2}(n + m)^{-1/2}$-Lipschitz, from lemma 6 and lemma 7. Hence, we have that $\mathrm{tr}\left(\bar{\Lambda}_G\mathbb{E}\left[\mathrm{R}_{\mathrm{Aug}}(\mathbf{D}) \mid X\right]\right)\mathbb{1}_{\mathsf{A}_\eta}$ is sub-Gaussian (which follows from **H**1 and proposition 6), and we have,

$$\frac{\alpha}{m}\mathbb{E}\left[\left|\mathrm{tr}\left(\bar{\Lambda}_G\{\mathbb{E}\left[\mathrm{R}_{\mathrm{Aug}}(\mathbf{D}) \mid X\right]\mathbb{1}_{\mathsf{A}_\eta}(X) - \mathbb{E}\left[\mathrm{R}_{\mathrm{Aug}}(\mathbf{D})\mathbb{1}_{\mathsf{A}_\eta}(X)\right]\}\right)\right|\right]$$
$$\le \frac{\alpha}{m}\sqrt{\mathrm{Var}(\mathrm{tr}\left(\bar{\Lambda}_G\mathbb{E}\left[\mathrm{R}_{\mathrm{Aug}}(\mathbf{D}) \mid X\right]\right)\mathbb{1}_{\mathsf{A}_\eta}(X))}$$
$$\lesssim \frac{\alpha}{m}\left(\frac{\sqrt{d}\|\bar{\Lambda}_G\|_{\mathrm{op}}(1 + \sqrt{m}\mathsf{L}_G)}{\sqrt{(\eta + \lambda_d(\mathbf{D}))^3(n + m)}} + \frac{d\|\bar{\Lambda}_G\|_{\mathrm{op}}}{\eta + \lambda_d(\mathbf{D})}\sigma_{\mathsf{A}_\eta}\right) \ ,$$

Recalling that $\sigma_{\mathsf{A}_\eta} \lesssim n^{-1}$ from **H**2, and using that $d < n$ (which also follows from **H**2), we simplify the previous bound

$$\frac{\alpha}{m}\mathbb{E}\left[\left|\mathrm{tr}\left(\bar{\Lambda}_G\{\mathbb{E}\left[\mathrm{R}_{\mathrm{Aug}}(\mathbf{D}) \mid X\right]\mathbb{1}_{\mathsf{A}_\eta}(X) - \mathbb{E}\left[\mathrm{R}_{\mathrm{Aug}}(\mathbf{D})\mathbb{1}_{\mathsf{A}_\eta}(X)\right]\}\right)\right|\right]$$
$$\lesssim \frac{\alpha\sqrt{d}\|\bar{\Lambda}_G\|_{\mathrm{op}}}{(\eta + \lambda_d(\mathbf{D}))^{3/2}m}\left(\frac{1 + \sqrt{m}\mathsf{L}_G}{\sqrt{n + m}} + \sqrt{\frac{\eta + \lambda_d(\mathbf{D})}{n}}\right) \lesssim \frac{\alpha\sqrt{d}\|\bar{\Lambda}_G\|_{\mathrm{op}}}{(\eta + \lambda_d(\mathbf{D}))^{3/2}m}\left(\sqrt{\alpha}\mathsf{L}_G + \frac{1 + \sqrt{\eta + \lambda_d(\mathbf{D})}}{\sqrt{n}}\right)$$

Plugging the previous calculation back into (C), we find,

$$\mathbb{E}\left[\left|\mathfrak{a}_g(X) - \mathfrak{a}_g^*\right|\right] \lesssim \frac{\alpha\beta}{m}\mathbb{E}\left[\left|\mathbb{E}\left[X_1^\top \mathrm{R}_{\mathrm{Aug}}(\mathbf{D})X_1\mathbb{1}_{\mathsf{A}_\eta}(X) \mid X\right] - \mathbb{E}\left[X_1^\top \mathrm{R}_{\mathrm{Aug}}(\mathbf{D})X_1\mathbb{1}_{\mathsf{A}_\eta}(X)\right]\right|\right] \quad (47)$$
$$+ \frac{\alpha\sqrt{d}\|\bar{\Lambda}_G\|_{\mathrm{op}}}{(\eta + \lambda_d(\mathbf{D}))^{3/2}m}\left(\frac{\mathbb{E}\left[\|\Lambda_G(X) - \bar{\Lambda}_G\|_{\mathrm{F}}\right]\sqrt{\eta + \lambda_d(\mathbf{D})}}{\|\bar{\Lambda}_G\|_{\mathrm{op}}} + \sqrt{\alpha}\mathsf{L}_G + \frac{1 + \sqrt{\eta + \lambda_d(\mathbf{D})}}{\sqrt{n}}\right)$$

It remains only to bound the expected deviation of $\mathbb{E}\left[X_1^\top \mathrm{R}_{\mathrm{Aug}}(\mathbf{D})\mathbb{1}_{\mathsf{A}_\eta}(X) \mid X\right]$. Using the Shermann-morisson's formula, we first write,

$$X_1^\top \mathrm{R}_{\mathrm{Aug}}(\mathbf{D})X_1\mathbb{1}_{\mathsf{A}_\eta}(X) = \left\{X_1^\top \mathrm{R}_{X^-\sqcup G}(\mathbf{D})X_1 - \frac{1}{n + m}\frac{X_1 \mathrm{R}_{X^-\sqcup G}(\mathbf{D})X_1 X_1^\top \mathrm{R}_{X^-\sqcup G}(\lambda)X_1}{1 + (n + m)^{-1}X_1^\top \mathrm{R}_{X^-\sqcup G}(\mathbf{D})X_1}\right\}\mathbb{1}_{\mathsf{A}_\eta}(X)$$
$$= \frac{X_1 \mathrm{R}_{X^-\sqcup G}(\mathbf{D})X_1}{1 + (n + m)^{-1}X_1^\top \mathrm{R}_{X^-\sqcup G}(\mathbf{D})X_1}\mathbb{1}_{\mathsf{A}_\eta}(X)$$
$$= \left\{(n + m) - \frac{(n + m)}{1 + (n + m)^{-1}X_1^\top \mathrm{R}_{X^-\sqcup G}(\mathbf{D})X_1}\right\}\mathbb{1}_{\mathsf{A}_\eta}(X) \ ,$$

hence, writting $f : x \mapsto (n + m)/(1 + (n + m)^{-1}x)$ (note that $f$ is 1-Lipschitz), we have,

$$\mathbb{E}\left[X_1^\top \mathrm{R}_{\mathrm{Aug}}(\mathbf{D})X_1\mathbb{1}_{\mathsf{A}_\eta}(X) \mid X\right] = (n + m)\mathbb{P}\left(X \in \mathsf{A}_\eta\right) - \mathbb{E}\left[f(X_1^\top \mathrm{R}_{X^-\sqcup G}(\mathbf{D})X_1)\mathbb{1}_{\mathsf{A}_\eta}(X) \mid X\right]$$

which allows to rewrite,

$$
\mathbb{E}\left[X_1^\top R_{\mathrm{Aug}}(\mathbf{D})X_1 \mathbb{1}_{\mathsf{A}_\eta}(X) \mid X\right] = \left\{(n+m) - \int f(X_1^\top R_{X^-\sqcup g}(\mathbf{D})X_1)d\nu_X^{\otimes m}(g)\right\}\mathbb{1}_{\mathsf{A}_\eta}(X)
$$

$$
= (n+m)\mathbb{1}_{\mathsf{A}_\eta}(X) - \int f(X_1^\top R_{X^-\sqcup g}(\mathbf{D})X_1)\mathbb{1}_{\mathsf{A}_\eta}(X)d\nu_{X^-}^{\otimes m}(g)
$$

$$
+ \int f(X_1^\top R_{X^-\sqcup g}(\mathbf{D})X_1)\mathbb{1}_{\mathsf{A}_\eta}(X)d\{\nu_{X^-}^{\otimes m} - \nu_X^{\otimes m}\}(g)
$$

$$
= (n+m)\mathbb{1}_{\mathsf{A}_\eta}(X) - f\left(\int X_1^\top R_{X^-\sqcup g}(\mathbf{D})X_1\mathbb{1}_{\mathsf{A}_\eta}(X)d\nu_{X^-}^{\otimes m}(g)\right)
$$

$$
- \left(\int f(X_1^\top R_{X^-\sqcup g}(\mathbf{D})X_1)\mathbb{1}_{\mathsf{A}_\eta}(X)d\nu_{X^-}^{\otimes m}(g) - f\left(\int X_1^\top R_{X^-\sqcup g}(\mathbf{D})X_1 d\nu_{X^-}^{\otimes m}(g)\right)\right)
$$

$$
+ \int f(X_1^\top R_{X^-\sqcup g}(\mathbf{D})X_1)\mathbb{1}_{\mathsf{A}_\eta}(X)d\{\nu_{X^-}^{\otimes m} - \nu_X^{\otimes m}\}(g)\,,
$$

which ensures,

$$
\mathbb{E}\left[\left|\mathbb{E}\left[X_1^\top R_{\mathrm{Aug}}(\lambda)X_1\mathbb{1}_{\mathsf{A}_\eta}(X) \mid X\right] - \mathbb{E}\left[X_1^\top R_{\mathrm{Aug}}(\lambda)X_1\mathbb{1}_{\mathsf{A}_\eta}(X)\right]\right|\right] \tag{48}
$$

$$
\lesssim \mathbb{E}\left[\left|f\left(\int X_1^\top R_{X^-\sqcup g}(\mathbf{D})X_1 d\nu_{X^-}^{\otimes m}(g)\right)\mathbb{1}_{\mathsf{A}_\eta}(X) - \mathbb{E}\left[f\left(\int X_1^\top R_{X^-\sqcup g}(\mathbf{D})X_1 d\nu_{X^-}^{\otimes m}(g)\right)\mathbb{1}_{\mathsf{A}_\eta}(X)\right]\right|\right]
$$

$$
+ \mathbb{E}\left[\left|\int f(X_1^\top R_{X^-\sqcup g}(\mathbf{D})X_1)\mathbb{1}_{\mathsf{A}_\eta}(X)d\nu_{X^-}^{\otimes m}(g) - f\left(\int X_1^\top R_{X^-\sqcup g}(\mathbf{D})X_1 d\nu_{X^-}^{\otimes m}(g)\right)\mathbb{1}_{\mathsf{A}_\eta}(X)\right|\right]
$$

$$
+ \mathbb{E}\left[\left|\int f(X_1^\top R_{X^-\sqcup g}(\mathbf{D})X_1)\mathbb{1}_{\mathsf{A}_\eta}(X)d\{\nu_{X^-}^{\otimes m} - \nu_X^{\otimes m}\}(g)\right|\right]\,,
$$

and we once again bound each term in the previous upper bound (48), starting with the last term, we notice that the function $\mathbf{G} \mapsto f(X_1^\top R_{X^-\sqcup\mathbf{G}}(\mathbf{D})\mathbb{1}_{\mathsf{A}_\eta}(X)X_1)$ is $2X_1^\top X_1(\eta + \lambda_d(\mathbf{D}))^{-3/2}(n+m)^{-1/2}$ from Lemma 6 (almsot surely). Hence, using **H5**, we have,

$$
\mathbb{E}\left[\left|\int f(X_1^\top R_{X^-\sqcup g}(\mathbf{D})X_1)\mathbb{1}_{\mathsf{A}_\eta}(X)d\{\nu_{X^-}^{\otimes m} - \nu_X^{\otimes m}\}(g)\right|\right] \lesssim \frac{\mathbb{E}\left[X_1 X_1^\top\right]}{(\eta + \lambda_d(\mathbf{D}))^{3/2}(n+m)^{1/2}}u(n)
$$

$$
= \frac{\operatorname{tr}(\Sigma_X)}{(\eta + \lambda_d(\mathbf{D}))^{3/2}(n+m)^{1/2}}u(n)\,.
$$

Furthermore, using the Jensen's inequality, we have,

$$
\mathbb{E}\left[\left|\int f(X_1^\top R_{X^-\sqcup g}(\mathbf{D})X_1)d\nu_{X^-}^{\otimes m}(g) - f\left(\int X_1^\top R_{X^-\sqcup g}(\mathbf{D})X_1 d\nu_{X^-}^{\otimes m}(g)\right)\right|\mathbb{1}_{\mathsf{A}_\eta}(X)\right]
$$

$$
\leq \mathbb{E}\left[\int\left|f(X_1^\top R_{X^-\sqcup g}(\mathbf{D})X_1) - f\left(\int X_1^\top R_{X^-\sqcup g}(\mathbf{D})X_1 d\nu_{X^-}^{\otimes m}(g)\right)\right|d\nu_{X^-}^{\otimes m}(g)\mathbb{1}_{\mathsf{A}_\eta}(X)\right]
$$

$$
\leq \mathbb{E}\left[\int\left|\operatorname{tr}\left(X_1 X_1^\top\left\{R_{X^-\sqcup g}(\mathbf{D}) - \int R_{X^-\sqcup g}(\mathbf{D})d\nu_{X^-}^{\otimes m}(g)\right\}\right)\right|d\nu_{X^-}^{\otimes m}(g)\mathbb{1}_{\mathsf{A}_\eta}(X)\right].
$$

Relying on the $\sigma_X$-Lipschitz concentration property of $\nu_{X^-}^{\otimes m}$, we can bound the previous term using the fact that $\mathbf{G} \mapsto \operatorname{tr}\left(X_1 X_1^\top R_{X^-\sqcup\mathbf{G}}(\mathbf{D})\right)$ is Lipschitz (from Lemma 6). We get,

$$
\mathbb{E}\left[\left|\int f(X_1^\top R_{X^-\sqcup g}(\mathbf{D})X_1)d\nu_{X^-}^{\otimes m}(g) - f\left(\int X_1^\top R_{X^-\sqcup g}(\mathbf{D})X_1 d\nu_{X^-}^{\otimes m}(g)\right)\right|\mathbb{1}_{\mathsf{A}_\eta}(X)\right]
$$

$$
\lesssim \frac{\mathbb{E}\left[X_1^\top X_1\right]}{(\eta + \lambda_d(\mathbf{D}))^{3/2}(n+m)^{1/2}} = \frac{\operatorname{tr}(\Sigma_X)}{(\eta + \lambda_d(\mathbf{D}))^{3/2}(n+m)^{1/2}}
$$

Now, focusing on the first term in (48), we write using the Jensen's inequality as well as leveraging the Lipschitz property of $f$,

$$\mathbb{E}\left[\left|f\left(\int X_1^\top \mathrm{R}_{X^-\sqcup g}(\mathbf{D})\,X_1\,d\nu_{X^-}^{\otimes m}(g)\right)\mathbb{1}_{\mathsf{A}_\eta}(X) - \mathbb{E}\left[f\left(\int X_1^\top \mathrm{R}_{X^-\sqcup g}(\mathbf{D})\,X_1\,d\nu_{X^-}^{\otimes m}(g)\right)\mathbb{1}_{\mathsf{A}_\eta}(X)\right]\right|\right]$$

$$\leq \mathbb{E}\left[\left|f\left(\int X_1^\top \mathrm{R}_{X^-\sqcup g}(\mathbf{D})\,X_1\,d\nu_{X^-}^{\otimes m}(g)\right)\mathbb{1}_{\mathsf{A}_\eta}(X) - f\left(\mathbb{E}\left[\int X_1^\top \mathrm{R}_{X^-\sqcup g}(\mathbf{D})\,X_1\,d\nu_{X^-}^{\otimes m}(g)\right]\right)\mathbb{1}_{\mathsf{A}_\eta}(X)\right|\right]$$

$$+ \mathbb{E}\left[\left|f\left(\mathbb{E}\left[\int X_1^\top \mathrm{R}_{X^-\sqcup g}(\mathbf{D})\,X_1\,d\nu_{X^-}^{\otimes m}(g)\right]\right)\mathbb{1}_{\mathsf{A}_\eta}(X) - \mathbb{E}\left[f\left(\int X_1^\top \mathrm{R}_{X^-\sqcup g}(\mathbf{D})\,X_1\,d\nu_{X^-}^{\otimes m}(g)\right)\mathbb{1}_{\mathsf{A}_\eta}(X)\right]\right|\right]$$

$$\leq \mathbb{E}\left[\left|\int X_1^\top \mathrm{R}_{X^-\sqcup g}(\mathbf{D})\,X_1\,d\nu_{X^-}^{\otimes m}(g)\mathbb{1}_{\mathsf{A}_\eta}(X) - \mathbb{E}\left[\int X_1^\top \mathrm{R}_{X^-\sqcup g}(\mathbf{D})\,X_1\,d\nu_{X^-}^{\otimes m}(g)\mathbb{1}_{\mathsf{A}_\eta}(X)\right]\right|\right]$$

$$+ \mathbb{E}\left[\left|\mathbb{E}\left[\int X_1^\top \mathrm{R}_{X^-\sqcup g}(\mathbf{D})\,X_1\,d\nu_{X^-}^{\otimes m}(g)\mathbb{1}_{\mathsf{A}_\eta}(X)\right] - \int X_1^\top \mathrm{R}_{X^-\sqcup g}(\mathbf{D})\,X_1\,d\nu_{X^-}^{\otimes m}(g)\mathbb{1}_{\mathsf{A}_\eta}(X)\right|\right]$$

$$\lesssim \sqrt{\mathrm{Var}\left(X_1^\top\int \mathrm{R}_{X^-\sqcup g}(\mathbf{D})\mathrm{d}\nu_{X^-}^{\otimes m}(g)\mathbb{1}_{\mathsf{A}_\eta}(X)X_1\right)}$$

Now, we remark that $\int \mathrm{R}_{X^-\sqcup g}(\mathbf{D})\mathrm{d}\nu_{X^-}^{\otimes m}(g)\mathbb{1}_{\mathsf{A}_\eta}(X)$ is $\sigma(X^-)$ measureable, our Proposition 7 applies, and we get,

$$\mathrm{Var}\left(X_1^\top\int \mathrm{R}_{X^-\sqcup g}(\mathbf{D})\mathrm{d}\nu_{X^-}^{\otimes m}(g)\mathbb{1}_{\mathsf{A}_\eta}(X)X_1\right) \lesssim d\|\Sigma_X\|_{\mathrm{op}}^2\left\{\frac{(1+\sqrt{m}\mathrm{L}_G)^2}{(\eta+\lambda_d(\mathbf{D}))^3(n+m)} + \frac{1+c_X^{-1}}{(\eta+\lambda_d(\mathbf{D}))^2}\right\}$$

Which implies,

$$\mathbb{E}\left[\left|f\left(\int X_1^\top \mathrm{R}_{X^-\sqcup g}(\mathbf{D})\,X_1\,d\nu_{X^-}^{\otimes m}(g)\right)\mathbb{1}_{\mathsf{A}_\eta}(X) - \mathbb{E}\left[f\left(\int X_1^\top \mathrm{R}_{X^-\sqcup g}(\mathbf{D})\,X_1\,d\nu_{X^-}^{\otimes m}(g)\right)\mathbb{1}_{\mathsf{A}_\eta}(X)\right]\right|\right]$$

$$\lesssim \sqrt{d}\|\Sigma_X\|_{\mathrm{op}}\left\{\frac{(1+\sqrt{m}\mathrm{L}_G)}{(\eta+\lambda_d(\mathbf{D}))^{3/2}\sqrt{n+m}} + \frac{1+c_X^{-1/2}}{(\eta+\lambda_d(\mathbf{D}))}\right\}$$

Putting all these bounds together, and plugging them back in (48), we find that,

$$\mathbb{E}\left[\left|\mathbb{E}\left[X_1^\top \mathrm{R}_{\mathrm{Aug}}(\mathbf{D})X_1\mathbb{1}_{\mathsf{A}_\eta}(X)\mid X\right] - \mathbb{E}\left[X_1^\top \mathrm{R}_{\mathrm{Aug}}(\mathbf{D})X_1\mathbb{1}_{\mathsf{A}_\eta}(X)\right]\right|\right]$$

$$\leq \sqrt{d}\|\Sigma_X\|_{\mathrm{op}}\left\{\frac{(1+\sqrt{m}\mathrm{L}_G)}{(\eta+\lambda_d(\mathbf{D}))^{3/2}\sqrt{n+m}} + \frac{1+c_X^{-1/2}}{(\eta+\lambda_d(\mathbf{D}))}\right\}$$

$$+ \frac{\mathrm{tr}\,(\Sigma_X)}{(\eta+\lambda_d(\mathbf{D}))^{3/2}(n+m)^{1/2}}(1+u(n))$$

$$\leq \frac{\sqrt{d}\|\Sigma_X\|_{\mathrm{op}}}{(\eta+\lambda_d(\mathbf{D}))^{3/2}}\left\{\frac{\sqrt{d}}{\sqrt{n+m}}(1+u(n)) + \frac{1}{\sqrt{n+m}} + \sqrt{\alpha}\mathrm{L}_G + (1+c_X^{-1/2})\sqrt{\eta+\lambda_d(\mathbf{D})}\right\}$$

We conclude on the second term in (43) by plugging the above bound into (47),

$$\mathbb{E}\left[|\mathfrak{a}_g(X) - \mathfrak{a}_x^*|\right] \lesssim \frac{\alpha\beta\sqrt{d}\|\Sigma_X\|_{\mathrm{op}}}{(\eta+\lambda_d(\mathbf{D}))^{3/2}m}\left\{\frac{\sqrt{d}}{\sqrt{n+m}}(1+u(n)) + \frac{1}{\sqrt{n+m}} + \sqrt{\alpha}\mathrm{L}_G + (1+c_X^{-1/2})\sqrt{\eta+\lambda_d(\mathbf{D})}\right\}$$

$$+ \frac{\alpha\sqrt{d}\|\bar{\Lambda}_G\|_{\mathrm{op}}}{(\eta+\lambda_d(\mathbf{D}))^{3/2}m}\left(\frac{\mathbb{E}\left[\|\Lambda_G(X) - \bar{\Lambda}_G\|_{\mathrm{F}}\right]\sqrt{\eta+\lambda_d(\mathbf{D})}}{\|\bar{\Lambda}_G\|_{\mathrm{op}}} + \sqrt{\alpha}\mathrm{L}_G + \frac{1+\sqrt{\eta+\lambda_d(\mathbf{D})}}{\sqrt{n}}\right)$$

Hence, using (46), we have,

$$
\left\| \mathbb{E}\left[ \left\{ \left( (1-\alpha'(X))C_X + \frac{\alpha \Lambda_G(X)}{\mathfrak{a}_g(X)} + \mathbf{D} \right)^{-1} - \left( (1-\alpha')C_X + \frac{\alpha \Lambda_G(X)}{\mathfrak{a}_g(X)} + \mathbf{D} \right)^{-1} \right\} \mathbb{1}_{\mathrm{A}_\eta}(X) \right] \right\|_{\mathrm{F}} \tag{49}
$$
$$
\lesssim \frac{\alpha\beta(1-\alpha')^{-1}+1}{(1-\alpha)\eta+\lambda_d(\mathbf{D})} \frac{\alpha\beta\sqrt{d}\|\Sigma_X\|_{\mathrm{op}}}{(\eta+\lambda_d(\mathbf{D}))^{3/2}m} \left\{ \frac{\sqrt{d}}{\sqrt{n+m}}(1+u(n)) + \frac{1}{\sqrt{n+m}} + \sqrt{\alpha}\mathsf{L}_G + (1+c_X^{-1/2})\sqrt{\eta+\lambda_d(\mathbf{D})} \right\}
$$
$$
+ \frac{\alpha\beta(1-\alpha')^{-1}+1}{(1-\alpha)\eta+\lambda_d(\mathbf{D})} \frac{\alpha\sqrt{d}\|\bar{\Lambda}_G\|_{\mathrm{op}}}{(\eta+\lambda_d(\mathbf{D}))^{3/2}m} \left( \frac{\mathbb{E}\left[\|\Lambda_G(X)-\bar{\Lambda}_G\|_{\mathrm{F}}\right]\sqrt{\eta+\lambda_d(\mathbf{D})}}{\|\bar{\Lambda}_G\|_{\mathrm{op}}} + \sqrt{\alpha}\mathsf{L}_G + \frac{1+\sqrt{\eta+\lambda_d(\mathbf{D})}}{\sqrt{n}} \right)
$$
$$
+ \frac{\mathbb{E}\left[\|\Lambda_G(X)-\bar{\Lambda}_G\|_{\mathrm{F}}\right]}{((1-\alpha)\eta+\lambda_d(\mathbf{D}))^2}
$$

Which conclude our analysis of the second term in (43).

Finally, we turn to the third and final term in (43), which is controlled using Proposition 8. First note that $\lambda_d(\alpha\bar{\Lambda}_G/\mathfrak{a}_g^* + \mathbf{D}/(1-\alpha')) \geq \lambda_d(\mathbf{D})/(1-\alpha')$, hence Proposition 8 ensures that there exists a constant $q_3$ that depends polynomially on $\cdots$, such that,

$$
\left\| \mathbb{E}\left[ \left\{ \frac{1}{1-\alpha'}\mathrm{R}_X\left( \frac{\alpha\bar{\Lambda}_G}{\mathfrak{a}_g^*} + \frac{1}{1-\alpha'}\mathbf{D} \right) - \frac{1}{1-\alpha'}\bar{\mathrm{R}}_X^{\mathfrak{a}_x^*}\left( \frac{\alpha\bar{\Lambda}_G}{(1-\alpha')\mathfrak{a}_g^*} + \frac{1}{1-\alpha'}\mathbf{D} \right) \right\} \mathbb{1}_{\mathrm{A}_\eta}(X) \right] \right\|_{\mathrm{F}} \tag{50}
$$
$$
\lesssim \frac{1}{1-\alpha'} \frac{q_3\sqrt{d}\|\Sigma_X\|_{\mathrm{op}}^3}{n\lambda_d(\Sigma_X)(\eta+\lambda_d(\mathbf{D})/(1-\alpha'))^6}
$$

Now putting our computations all together, in particular plugging (45), (49) and (50), in (43) we have shown,

$$
\left\| \mathbb{E}\left[ \left\{ \mathrm{R}_{\mathrm{Aug}}(\mathbf{D}) - \bar{\mathrm{R}}_{\mathrm{Aug}}^{(\mathfrak{a}_x^*,\mathfrak{a}_g^*)}(\mathbf{D}) \right\} \mathbb{1}_{\mathrm{A}_\eta}(X) \right] \right\|_{\mathrm{F}}
$$
$$
\lesssim \frac{\alpha^5(\kappa q_1+q_2)\sqrt{d}\left\{ \sigma_G^2(\beta^3\|\Sigma_X\|_{\mathrm{op}}^3+\kappa^3) + \sigma_X^{12}\|\Sigma_X\|_{\mathrm{op}}\lambda_d(\Sigma_X)^{-1}n^{-1/2}d^{-1} \right\}}{n((1-\alpha)\eta+\lambda_d(\mathbf{D}))^6}
$$
$$
+ \frac{\alpha\beta(1-\alpha')^{-1}+1}{(1-\alpha)\eta+\lambda_d(\mathbf{D})} \frac{\alpha\beta\sqrt{d}\|\Sigma_X\|_{\mathrm{op}}}{(\eta+\lambda_d(\mathbf{D}))^{3/2}m} \left\{ \frac{\sqrt{d}}{\sqrt{n+m}}(1+u(n)) + \frac{1}{\sqrt{n+m}} + \sqrt{\alpha}\mathsf{L}_G + (1+c_X^{-1/2})\sqrt{\eta+\lambda_d(\mathbf{D})} \right\}
$$
$$
+ \frac{\alpha\beta(1-\alpha')^{-1}+1}{(1-\alpha)\eta+\lambda_d(\mathbf{D})} \frac{\alpha\sqrt{d}\|\bar{\Lambda}_G\|_{\mathrm{op}}}{(\eta+\lambda_d(\mathbf{D}))^{3/2}m} \left( \frac{\mathbb{E}\left[\|\Lambda_G(X)-\bar{\Lambda}_G\|_{\mathrm{F}}\right]\sqrt{\eta+\lambda_d(\mathbf{D})}}{\|\bar{\Lambda}_G\|_{\mathrm{op}}} + \sqrt{\alpha}\mathsf{L}_G + \frac{1+\sqrt{\eta+\lambda_d(\mathbf{D})}}{\sqrt{n}} \right)
$$
$$
+ \frac{\mathbb{E}\left[\|\Lambda_G(X)-\bar{\Lambda}_G\|_{\mathrm{F}}\right]}{((1-\alpha)\eta+\lambda_d(\mathbf{D}))^2} + \frac{1}{1-\alpha'} \frac{q_3\sqrt{d}\|\Sigma_X\|_{\mathrm{op}}^3}{n\lambda_d(\Sigma_X)(\eta+\lambda_d(\mathbf{D})/(1-\alpha'))^6}
$$

To simplify the above upper bound, we use the fact that $1 - \alpha' \geq 1 - \alpha$, $\alpha \leq 1$ as well as $d < n$, which yields,

$$
\left\| \mathbb{E}\left[ \left\{ \mathrm{R}_{\mathrm{Aug}}(\mathbf{D}) - \bar{\mathrm{R}}_{\mathrm{Aug}}^{(\mathfrak{a}_x^*, \mathfrak{a}_g^*)}(\mathbf{D}) \right\} \mathbb{1}_{\mathsf{A}_\eta}(X) \right] \right\|_{\mathrm{F}}
$$

$$
\lesssim \frac{\alpha^5(\kappa q_1 + q_2)\sqrt{d}\left\{ \sigma_G^2(\beta^3 \|\Sigma_X\|_{\mathrm{op}}^3 + \kappa^3) + \sigma_X^{12}\|\Sigma_X\|_{\mathrm{op}}\lambda_d(\Sigma_X)^{-1} n^{-1/2} d^{-1} \right\}}{n((1-\alpha)\eta + \lambda_d(\mathbf{D}))^6}
$$

$$
+ \frac{\alpha\beta(1-\alpha)^{-1} + 1}{(1-\alpha)\eta + \lambda_d(\mathbf{D})} \frac{\alpha\beta\sqrt{d}\|\Sigma_X\|_{\mathrm{op}}}{(\eta + \lambda_d(\mathbf{D}))^{3/2}m} \left\{ (1 + u(n)) + \sqrt{\alpha}\mathrm{L}_G + (1 + c_X^{-1/2})\sqrt{\eta + \lambda_d(\mathbf{D})} \right\}
$$

$$
+ \frac{\alpha\beta(1-\alpha)^{-1} + 1}{(1-\alpha)\eta + \lambda_d(\mathbf{D})} \frac{\alpha\sqrt{d}\|\bar{\Lambda}_G\|_{\mathrm{op}}}{(\eta + \lambda_d(\mathbf{D}))^{3/2}m} \left( \frac{\mathbb{E}\left[\|\Lambda_G(X) - \bar{\Lambda}_G\|_{\mathrm{F}}\right]\sqrt{\eta + \lambda_d(\mathbf{D})}}{\|\bar{\Lambda}_G\|_{\mathrm{op}}} + \sqrt{\alpha}\mathrm{L}_G + \frac{1 + \sqrt{\eta + \lambda_d(\mathbf{D})}}{\sqrt{n}} \right)
$$

$$
+ \frac{\mathbb{E}\left[\|\Lambda_G(X) - \bar{\Lambda}_G\|_{\mathrm{F}}\right]}{((1-\alpha)\eta + \lambda_d(\mathbf{D}))^2} + \frac{(1-\alpha)^5 q_3 \sqrt{d}\|\Sigma_X\|_{\mathrm{op}}^3}{n\lambda_d(\Sigma_X)((1-\alpha)\eta + \lambda_d(\mathbf{D}))^6}
$$

$$
\lesssim \frac{\sqrt{d}\sigma_G^2(\beta^3\|\Sigma_X\|_{\mathrm{op}}^3 + \kappa^3)}{n\lambda_d(\Sigma_X)((1-\alpha)\eta + \lambda_d(\mathbf{D}))^6}(\kappa q_1 + q_2) \left\{ \lambda_d(\Sigma_X) + \frac{\sigma_X^{10}\|\Sigma_X\|_{\mathrm{op}}}{d(\beta^3\|\Sigma_X\|_{\mathrm{op}}^3 + \kappa^3)\sqrt{n}} \right\}
$$

$$
+ \left( \frac{\alpha\beta}{1-\alpha} + 1 \right) \frac{\alpha\beta\sqrt{d}(\|\Sigma_X\|_{\mathrm{op}} + q_3\|\bar{\Lambda}_G\|_{\mathrm{op}})}{((1-\alpha)\eta + \lambda_d(\mathbf{D}))^{5/2}m} \left\{ (1 + u(n)) + \sqrt{\alpha}\mathrm{L}_G + (1 + c_X^{-1/2})\sqrt{\eta + \lambda_d(\mathbf{D})} \right\}
$$

$$
+ \left( \frac{\alpha\beta}{1-\alpha} + 1 \right) \left( \frac{1}{((1-\alpha)\eta + \lambda_d(\mathbf{D}))^2} + \frac{\alpha\beta\sqrt{d}}{((1-\alpha)\eta + \lambda_d(\mathbf{D}))^{5/2}m} \right) \mathbb{E}\left[\|\Lambda_G(X) - \bar{\Lambda}_G\|_{\mathrm{F}}\right]
$$

$\square$

# D  Proof of theorem 2

This section of this Appendix details the proof of theorem 2. First recall the definition of $\hat{\mathcal{E}}_{\mathrm{Aug}}(\lambda)$, for all $\lambda > 0$:

$$
\begin{aligned}
\Phi_1(\mathbf{X}) &= \frac{(1 - d/n)}{d} \operatorname{tr}\left(\mathrm{R}_{\mathbf{X}}(0)\left(\frac{\alpha\Lambda_G(\mathbf{X})}{\mathfrak{a}_g(\mathbf{X})} + \lambda\,\mathrm{I}_d\right)^{-1}\right)\mathbb{1}_{\mathrm{A}_\eta}(\mathbf{X}) , \\
\Phi_2(\mathbf{X}) &= \frac{1 - (1 - \beta/\mathfrak{a}_g(\mathbf{X}))\alpha}{d\mathfrak{a}_x(\mathbf{X})} \operatorname{tr}\left(\bar{\mathrm{R}}_{G|X}^{(\mathfrak{a}_g(\mathbf{X}))}(\lambda,\mathbf{X})\left(\frac{\alpha\Lambda_G(\mathbf{X})}{\mathfrak{a}_g(\mathbf{X})} + \lambda\,\mathrm{I}_d\right)^{-1}\right) ,
\end{aligned}
\tag{51}
$$

Where we have used the three notations,

$$
\begin{aligned}
\mathfrak{a}_x(\mathbf{X}) &= 1 + \frac{1 - (1 - \beta/\mathfrak{a}_g(X))\alpha}{n} X_1^\top \int \mathrm{R}_{\mathbf{X}^-\sqcup g}\, d\nu_{\mathbf{X}}^{\otimes m}(g) X_1 , \\
\mathfrak{a}_g(\mathbf{X}) &= 1 + \frac{\alpha}{m} \operatorname{tr}\left(\{\beta C_X + \Lambda_G(\mathbf{X})\}\int \mathrm{R}_{\mathbf{X}\sqcup g}(\lambda) d\nu_{\mathbf{X}}^{\otimes m}(g)\right) ,
\end{aligned}
\tag{52}
$$

and, for any $\mathfrak{a} \geq 1$,

$$
\bar{\mathrm{R}}_{G|X}^{(\mathfrak{a})}(\lambda,\mathbf{X}) := \left((1 - \alpha)\,C_X + \frac{\alpha\Lambda_G(X) + \alpha\beta C_X}{\mathfrak{a}} + \lambda\,\mathrm{I}_d\right)^{-1} .
$$

Finally, we set,

$$
\hat{\mathcal{E}}_{\mathrm{Aug}}(\lambda) := \frac{1}{d} \operatorname{tr}\left(\mathrm{R}_{\mathrm{Aug}}(\lambda)^2\right) - 2(\Phi_1(X) - \Phi_2(X)) + \frac{1}{d} \operatorname{tr}\left(\Sigma_X^{-2}\right) ,
$$

Firstly, in appendix D.1 we detail the concentration of $\mathfrak{a}_x(X)$ and $\mathfrak{a}_g(X)$ defined in (52). Secondly, in appendix D.2, we show that $\Phi_1(X)$ and $\Phi_2(X)$ (defined in (51)) essencially have sub-Exponential tail, we provide an upper bound on their sub-Exponential norm. We then conclude on the proof of theorem 2 in the last part of the Appendix.

## D.1  Concentration of $\mathfrak{a}_g(X)$ and $\mathfrak{a}_x(X)$

**Proposition 9.** *Assume that $X$ and $G$ satisfy assumptions **H1** to **H2**. Let $\mathfrak{a}_g(X)$ and $\mathfrak{a}_x(X)$ defined as in (52), we set,*

$$
\zeta_x(t) := \min\left\{\frac{\lambda^2 t^2}{(1 - \alpha)}, \lambda t, \frac{\lambda^3 t^2}{\sigma_G^2(\sqrt{\alpha}\mathrm{L}_G + 1/\sqrt{n + m})^2}, \zeta_x\left(\frac{\lambda t}{\beta\|\Sigma_X\|_{\mathrm{op}}}\right)\right\}
$$

$$
\zeta_g(t) = \min\left\{\frac{\lambda^2 t^2}{\beta^2}, \frac{\lambda t}{\beta}, \frac{\lambda^3(n + m)t^2}{\beta^2(\sigma_G + u(n))^2}, \frac{\sqrt{\lambda^3}t}{\beta(\sigma_G + u(n))}, \frac{\lambda^3(n + m)^2 t^2}{\alpha^2\left(\mathrm{L}_\Lambda/\sqrt{\lambda} + \sqrt{\alpha}\kappa\mathrm{L}_G + \kappa/\sqrt{n + m}\right)^2} + \frac{\ln(n)}{n + m}\right\}
$$

*as well as,*

$$
\begin{aligned}
\delta_g &:= \frac{\alpha\beta}{m}\left(\frac{4(\sigma_G + u(n))\operatorname{tr}(\Sigma_X)}{\sqrt{\lambda^3(n + m)}} + \frac{(1 + \mathrm{L}_G)}{\sqrt{\lambda^3(n + m)}}\right) , \\
\delta_x &:= \delta_g + \frac{\|\Sigma_X\|_{\mathrm{op}}}{\sqrt{n}}\left(\frac{2u(n)}{\lambda^{3/2}} + \frac{\operatorname{tr}(\Sigma_X)}{\lambda(n + m)}\right) .
\end{aligned}
$$

*then the following holds for a universal constant $c > 0$,*

$$
\mathbb{P}\left(\left|\mathfrak{a}_g(X) - \mathfrak{a}_g^*\right| \geq t + \delta_x\right) \lesssim \exp\left(-c(n + m)\zeta_x(t)\right) ,
$$

*and,*

$$
\mathbb{P}\left(\left|\mathfrak{a}_x(X) - \mathfrak{a}_x^*\right| \geq t + \delta_g\right) \lesssim \exp\left(-c(n + m)\zeta_g(t)\right) .
$$

*Proof.* We first recall that $\lambda > 0$ and from (52),

$$\mathfrak{a}_g(X) = 1 + \frac{\alpha\beta}{m} \operatorname{tr}\left(C_X \mathbb{E}\left[\mathrm{R}_{\mathrm{Aug}}(\lambda) \mid X\right]\right) + \frac{\alpha}{m} \operatorname{tr}\left(\Lambda_G(X)\mathbb{E}\left[\mathrm{R}_{\mathrm{Aug}}(\lambda) \mid X\right]\right)$$

and from Theorem 4,

$$\mathfrak{a}_g^* = 1 + \frac{\alpha\beta}{m} \operatorname{tr}\left(\mathbb{E}\left[C_X \mathrm{R}_{\mathrm{Aug}}(\lambda)\right]\right) + \frac{\alpha}{m} \operatorname{tr}\left(\mathbb{E}\left[\Lambda_G(X) \mathrm{R}_{\mathrm{Aug}}(\lambda)\right]\right)$$

we can thus write,

$$\left|\mathfrak{a}_g(X) - \mathfrak{a}_g^*\right| \tag{53}$$

$$\leq \frac{\alpha\beta}{m}\left|\operatorname{tr}\left(\mathbb{E}\left[C_X \mathrm{R}_{\mathrm{Aug}}(\lambda) \mid X\right] - \mathbb{E}\left[C_X \mathrm{R}_{Aug}(\lambda)\right]\right)\right| + \frac{\alpha}{m}\left|\operatorname{tr}\left(\mathbb{E}\left[\Lambda_G(X) \mathrm{R}_{\mathrm{Aug}}(\lambda) \mid X\right] - \mathbb{E}\left[\Lambda_G(X) \mathrm{R}_{\mathrm{Aug}}(\lambda)\right]\right)\right|$$

$$\leq \frac{\alpha\beta}{mn}\sum_{i=1}^{n}\left|\operatorname{tr}\left(\mathbb{E}\left[X_i X_i^\top \mathrm{R}_{\mathrm{Aug}}(\lambda) \mid X\right] - \mathbb{E}\left[X_i X_i^\top \mathrm{R}_{\mathrm{Aug}}(\lambda)\right]\right)\right|$$

$$+ \frac{\alpha}{m}\left|\operatorname{tr}\left(\mathbb{E}\left[\Lambda_G(X) \mathrm{R}_{\mathrm{Aug}}(\lambda) \mid X\right] - \mathbb{E}\left[\Lambda_G(X) \mathrm{R}_{\mathrm{Aug}}(\lambda)\right]\right)\right| .$$

Now remark that the distribution of $\left|\operatorname{tr}\left(\mathbb{E}\left[X_i X_i^\top \mathrm{R}_{\mathrm{Aug}}(\lambda) \mid X\right] - \mathbb{E}\left[X_i X_i^\top \mathrm{R}_{\mathrm{Aug}}(\lambda)\right]\right)\right|$ doesn't depend on $i$, by exchangeability of the columns of $X$, we thus focus only on the term $i = 1$. Using the Shermann-morisson's formula, we have,

$$X_1^\top \mathrm{R}_{\mathrm{Aug}}(\lambda)X_1 = X_1^\top \mathrm{R}_{X^- \sqcup G}(\lambda)X_1 - \frac{1}{n+m}\frac{X_1 \mathrm{R}_{X^- \sqcup G}(\lambda)X_1 X_1^\top \mathrm{R}_{X^- \sqcup G}(\lambda)X_1}{1 + (n+m)^{-1}X_1^\top \mathrm{R}_{X^- \sqcup G} X_1}$$

$$= \frac{X_1 \mathrm{R}_{X^- \sqcup G} X_1}{1 + (n+m)^{-1}X_1^\top \mathrm{R}_{X^- \sqcup G}(\lambda)X_1}$$

$$= (n+m) - \frac{(n+m)}{1 + (n+m)^{-1}X_1^\top \mathrm{R}_{X^- \sqcup G}(\lambda)X_1} ,$$

hence, writting $f : x \mapsto (n+m)/(1 + (n+m)^{-1}x)$ (note that $f$ is 1-Lipschitz), we have,

$$\mathbb{E}\left[X_1^\top \mathrm{R}_{\mathrm{Aug}}(\lambda)X_1 \mid X\right] = (n+m) - \mathbb{E}\left[f(X_1^\top \mathrm{R}_{X^- \sqcup G}(\lambda)X_1) \mid X\right]$$

In order to derive the concentration of $\mathbb{E}\left[X_1^\top \mathrm{R}_{\mathrm{Aug}}(\lambda)X_1 \mid X\right]$, we mostly rely on the use of the Hanson-Wright inequality, which applies to quadratic forms of the shape $X_1^\top M(X^-)X_1$ with $M(X^-)$ being a $\sigma(X^-)$ measureable random matrix. To this end, we show that $\mathbb{E}\left[X_1^\top \mathrm{R}_{\mathrm{Aug}}(\lambda)X_1 \mid X\right]$ is close to being of this form. We have,

$$\mathbb{E}\left[X_1^\top \mathrm{R}_{\mathrm{Aug}}(\lambda)X_1 \mid X\right] = (n+m) - \int f(X_1^\top \mathrm{R}_{X^- \sqcup g}(\lambda)X_1)d\nu_X^{\otimes m}(g)$$

$$= (n+m) - \int f(X_1^\top \mathrm{R}_{X^- \sqcup g}(\lambda)X_1)d\nu_{X^-}^{\otimes m}(g)$$

$$+ \int f(X_1^\top \mathrm{R}_{X^- \sqcup g}(\lambda)X_1)d\{\nu_{X^-}^{\otimes m} - \nu_X^{\otimes m}\}(g)$$

$$= (n+m) - f\left(\int X_1^\top \mathrm{R}_{X^- \sqcup g}(\lambda)X_1 d\nu_{X^-}^{\otimes m}(g)\right)$$

$$- \left(\int f(X_1^\top \mathrm{R}_{X^- \sqcup g}(\lambda)X_1)d\nu_{X^-}^{\otimes m}(g) - f\left(\int X_1^\top \mathrm{R}_{X^- \sqcup g}(\lambda)X_1 d\nu_{X^-}^{\otimes m}(g)\right)\right)$$

$$+ \int f(X_1^\top \mathrm{R}_{X^- \sqcup g}(\lambda)X_1)d\{\nu_{X^-}^{\otimes m} - \nu_X^{\otimes m}\}(g) ,$$

Similarly, we write,

$$\mathbb{E}\left[X_1^\top \mathrm{R}_{\tilde{X}}(\lambda)X_1\right] = (n+m) - f\left(\mathbb{E}\left[\int X_1^\top \mathrm{R}_{X^- \sqcup g}(\lambda)X_1 d\nu_{X^-}^{\otimes m}(g)\right]\right)$$

$$- \left(\mathbb{E}\left[\int f(X_1^\top \mathrm{R}_{X^- \sqcup g}(\lambda)X_1)d\nu_{X^-}^{\otimes m}(g)\right] - f\left(\mathbb{E}\left[\int X_1^\top \mathrm{R}_{X^- \sqcup g}(\lambda)X_1 d\nu_{X^-}^{\otimes m}(g)\right]\right)\right)$$

$$+ \mathbb{E}\left[\int f(X_1^\top \mathrm{R}_{X^- \sqcup g}(\lambda)X_1)d\{\nu_{X^-}^{\otimes m} - \nu_X^{\otimes m}\}(g)\right] ,$$

which ensures,

$$\left| \mathbb{E}\left[ X_1^\top \mathrm{R}_{\mathrm{Aug}}(\lambda) X_1 \mid X \right] - \mathbb{E}\left[ X_1^\top \mathrm{R}_{\mathrm{Aug}}(\lambda) X_1 \right] \right|$$

$$\leq \left| f\left( \int X_1^\top \mathrm{R}_{X^- \sqcup g}(\lambda) X_1 d\nu_{X^-}^{\otimes m}(g) \right) - f\left( \mathbb{E}\left[ \int X_1^\top \mathrm{R}_{X^- \sqcup g}(\lambda) X_1 d\nu_{X^-}^{\otimes m}(g) \right] \right) \right|$$

$$+ \left| \int f(X_1^\top \mathrm{R}_{X^- \sqcup g}(\lambda) X_1) d\nu_{X^-}^{\otimes m}(g) - f\left( \int X_1^\top \mathrm{R}_{X^- \sqcup g}(\lambda) X_1 d\nu_{X^-}^{\otimes m}(g) \right) \right|$$

$$+ \left| \mathbb{E}\left[ \int f(X_1^\top \mathrm{R}_{X^- \sqcup g}(\lambda) X_1) d\nu_{X^-}^{\otimes m}(g) \right] - f\left( \mathbb{E}\left[ \int X_1^\top \mathrm{R}_{X^- \sqcup g}(\lambda) X_1 d\nu_{X^-}^{\otimes m}(g) \right] \right) \right|$$

$$+ \left| \int f(X_1^\top \mathrm{R}_{X^- \sqcup g}(\lambda) X_1) d\{\nu_{X^-}^{\otimes m} - \nu_X^{\otimes m}\}(g) \right| + \left| \mathbb{E}\left[ \int f(X_1^\top \mathrm{R}_{X^- \sqcup g}(\lambda) X_1) d\{\nu_{X^-}^{\otimes m} - \nu_X^{\otimes m}\}(g) \right] \right| ,$$

Now, using the Jensen's inequality, as well as the 1-Lispchtiz property of $f$, the previous equation implies,

$$\left| \mathbb{E}\left[ X_1^\top \mathrm{R}_{\mathrm{Aug}}(\lambda) X_1 \mid X \right] - \mathbb{E}\left[ X_1^\top \mathrm{R}_{\mathrm{Aug}}(\lambda) X_1 \right] \right|$$

$$\leq \left| X_1^\top \int \mathrm{R}_{X^- \sqcup g}(\lambda) d\nu_{X^-}^{\otimes m}(g) X_1 - \mathrm{tr}\left( \Sigma_X \int \mathrm{R}_{X^- \sqcup g}(\lambda) d\nu_{X^-}^{\otimes m}(g) \right) \right|$$

$$+ \int \left| \mathrm{tr}\left( X_1 X_1^\top \left\{ \mathrm{R}_{X^- \sqcup g}(\lambda) - \int \mathrm{R}_{X^- \sqcup g}(\lambda) d\nu_{X^-}^{\otimes m}(g) \right\} \right) \right| d\nu_{X^-}^{\otimes m}(g)$$

$$+ \mathbb{E}\left[ \int \left| \mathrm{tr}\left( X_1 X_1^\top \left\{ \mathrm{R}_{X^- \sqcup g}(\lambda) - \mathbb{E}\left[ \int \mathrm{R}_{X^- \sqcup g}(\lambda) d\nu_{X^-}^{\otimes m}(g) \right] \right\} \right) \right| d\nu_{X^-}^{\otimes m}(g) \right]$$

$$+ \left| \int f(X_1^\top \mathrm{R}_{X^- \sqcup g}(\lambda) X_1) d\{\nu_{X^-}^{\otimes m} - \nu_X^{\otimes m}\}(g) \right| + \left| \mathbb{E}\left[ \int f(X_1^\top \mathrm{R}_{X^- \sqcup g}(\lambda) X_1) d\{\nu_{X^-}^{\otimes m} - \nu_X^{\otimes m}\}(g) \right] \right| .$$

To bound the above, first remark that the map $g : \mathbf{X} \sqcup \mathbf{G} \mapsto \mathrm{tr}\left( X_1 X_1^\top \mathrm{R}_{\mathbf{X}^- \sqcup \mathbf{G}} \right)$ is $2\|X_1\|_2^2 \lambda^{-3/2}(n+m)^{-1/2}$ conditionally on $X_1$ (as a consequence of lemma 6), we have from **H**3, that $X_1^\top \mathrm{R}_{X^- \sqcup g}(\lambda) X_1$ is sub-Gaussian conditionally to $X$, for $g \sim \nu_{X^-}^{\otimes m}$, and from the moment bounds for sub-Gaussian random variables,

$$\int \left| \mathrm{tr}\left( X_1 X_1^\top \left\{ \mathrm{R}_{X^- \sqcup g}(\lambda) - \int \mathrm{R}_{X^- \sqcup g}(\lambda) d\nu_{X^-}^{\otimes m}(g) \right\} \right) \right| d\nu_{X^-}^{\otimes m}(g)$$

$$\leq \sqrt{\int \left( \mathrm{tr}\left( X_1 X_1^\top \left\{ \mathrm{R}_{X^- \sqcup g}(\lambda) - \int \mathrm{R}_{X^- \sqcup g}(\lambda) d\nu_{X^-}^{\otimes m}(g) \right\} \right) \right)^2 d\nu_{X^-}^{\otimes m}(g)}$$

$$\leq \frac{2\sigma_G X_1^\top X_1}{\sqrt{\lambda^3(n+m)}} ,$$

Similarly, and using a triangle inequality, we have,

$$\mathbb{E}\left[ \int \left| \mathrm{tr}\left( X_1 X_1^\top \left\{ \mathrm{R}_{X^- \sqcup g}(\lambda) - \mathbb{E}\left[ \int \mathrm{R}_{X^- \sqcup g}(\lambda) d\nu_{X^-}^{\otimes m}(g) \right] \right\} \right) \right| d\nu_{X^-}^{\otimes m}(g) \right]$$

$$\leq \mathbb{E}\left[ \int \left| \mathrm{tr}\left( X_1 X_1^\top \left\{ \int \mathrm{R}_{X^- \sqcup g}(\lambda) d\nu_{X^-}^{\otimes m}(g) - \mathbb{E}\left[ \int \mathrm{R}_{X^- \sqcup g}(\lambda) d\nu_{X^-}^{\otimes m}(g) \right] \right\} \right) \right| d\nu_{X^-}^{\otimes m}(g) \right]$$

$$+ \frac{2\sigma_G \mathbb{E}\left[ X_1 X_1^\top \right]}{\sqrt{\lambda^3(n+m)}}$$

and, using Lemma 7, we have that and the variance bound for sub-Gaussian random variables, we have,

$$\mathbb{E}\left[ \int \left| \mathrm{tr}\left( X_1 X_1^\top \left\{ \mathrm{R}_{X^- \sqcup g}(\lambda) - \mathbb{E}\left[ \int \mathrm{R}_{X^- \sqcup g}(\lambda) d\nu_{X^-}^{\otimes m}(g) \right] \right\} \right) \right| d\nu_{X^-}^{\otimes m}(g) \right]$$

$$\leq \frac{2(1 + \sqrt{m}\mathrm{L}_G) + 2\sigma_G \mathbb{E}\left[ X_1^\top X_1 \right]}{\sqrt{\lambda^3(n+m)}} = \frac{2(1 + \sqrt{m}\mathrm{L}_G) + 2\sigma_G \mathrm{tr}\left( \Sigma_X \right)}{\sqrt{\lambda^3(n+m)}}$$

Furthermore, using **H4**, and recalling that $\mathbf{G} \mapsto f(X_1^\top \mathrm{R}_{X \sqcup \mathbf{G}}(\lambda)X_1)$ is $2\|X_1\|_2^2 \lambda^{-3/2}(n+m)^{-1/2}$-Lispchitz, the two final terms are bounded as,

$$\left|\int f(X_1^\top \mathrm{R}_{X^-\sqcup g}(\lambda)X_1)d\{\nu_{X^-}^{\otimes m} - \nu_X^{\otimes m}\}(g)\right| \leq 2\|X_1\|_2^2 \lambda^{-3/2}(n+m)^{-1/2}W_1(\nu_X^{\otimes m}, \nu_{X^-}^{\otimes m})$$

$$\leq 2\|X_1\|_2^2 \lambda^{-3/2}(n+m)^{-1/2}\sqrt{m}u(n),$$

and,

$$\mathbb{E}\left[\left|\int f(X_1^\top \mathrm{R}_{X^-\sqcup g}(\lambda)X_1)d\{\nu_{X^-}^{\otimes m} - \nu_X^{\otimes m}\}(g)\right|\right] \leq 2\,\mathrm{tr}\,(\Sigma_X)\,\lambda^{-3/2}\alpha^{1/2}u(n),$$

Merging all these together, we have shown,

$$\left|\mathbb{E}\left[X_1^\top \mathrm{R}_{\mathrm{Aug}}(\lambda)X_1 \mid X\right] - \mathbb{E}\left[X_1^\top \mathrm{R}_{\mathrm{Aug}}(\lambda)X_1\right]\right|$$

$$\leq \left|\int X_1^\top \mathrm{R}_{X^-\sqcup g}(\lambda)X_1 d\nu_{X^-}^{\otimes m}(g) - \mathbb{E}\left[\int X_1^\top \mathrm{R}_{X^-\sqcup g}(\lambda)X_1 d\nu_{X^-}^{\otimes m}(g)\right]\right|$$

$$+ (\sigma_G + \alpha u(n))\frac{2X_1^\top X_1}{\sqrt{\lambda^3(n+m)}} + (\sigma_G + u(n))\frac{2\,\mathrm{tr}\,(\Sigma_X)}{\sqrt{\lambda^3(n+m)}} + \frac{2(1+\mathrm{L}_G)}{\sqrt{\lambda^3(n+m)}}$$

$$\leq \left|X_1^\top \int \mathrm{R}_{X^-\sqcup g}(\lambda)d\nu_{X^-}^{\otimes m}(g)X_1 - \mathrm{tr}\left(\Sigma_X \int \mathrm{R}_{X^-\sqcup g}(\lambda)X_1 d\nu_{X^-}^{\otimes m}(g)\right)\right|$$

$$+ \frac{2(\sigma_G + u(n))}{\sqrt{\lambda^3(n+m)}}\left|X_1^\top X_1 - \mathrm{tr}\,(\Sigma_X)\right| + \frac{4(\sigma_G + u(n))\,\mathrm{tr}\,(\Sigma_X)}{\sqrt{\lambda^3(n+m)}} + \frac{2(1+\mathrm{L}_G)}{\sqrt{\lambda^3(n+m)}}$$

Finally, putting back the previous upper bound in (53), we have,

$$\left|\mathfrak{a}_g(X) - \mathfrak{a}_g^*\right| \leq \frac{\alpha\beta}{nm}\sum_{i=1}^n \left|\mathrm{tr}\left(\mathbb{E}\left[X_iX_i^\top \mathrm{R}_{\tilde{X}}(\lambda) \mid X\right] - \mathbb{E}\left[X_iX_i^\top \mathrm{R}_{\tilde{X}}(\lambda)\right]\right)\right|$$

$$+ \frac{2\alpha\beta(\sigma_G + u(n))}{\sqrt{\lambda^3(n+m)}mn}\sum_{i=1}^n \left|X_iX_i^\top - \mathrm{tr}\,(\Sigma_X)\right| + \frac{4\alpha\beta(\sigma_G + u(n))\,\mathrm{tr}\,(\Sigma_X) + 2\alpha\beta(1+\mathrm{L}_G)}{\sqrt{\lambda^3(n+m)}m}$$

$$+ \frac{\alpha}{m}\left|\mathrm{tr}\left(\mathbb{E}\left[\Lambda_G(X)\mathrm{R}_{\mathrm{Aug}}(\lambda) \mid X\right] - \mathbb{E}\left[\Lambda_G(X)\mathrm{R}_{\mathrm{Aug}}(\lambda)\right]\right)\right|,$$

Applying a union bound, we have,

$$\mathbb{P}\left(\left|\mathfrak{a}_g(X) - \mathfrak{a}_g^*\right| \geq t + \frac{\alpha\beta}{m}\left(\frac{4(\sigma_G + u(n))\,\mathrm{tr}\,(\Sigma_X)}{\sqrt{\lambda^3(n+m)}} + \frac{(1+\mathrm{L}_G)}{\sqrt{\lambda^3(n+m)}}\right)\right) \tag{54}$$

$$\leq n\mathbb{P}\left(\left|X_1^\top \int \mathrm{R}_{X^-\sqcup g}(\lambda)d\nu_{X^-}^{\otimes m}(g)X_1 - \mathrm{tr}\left(\Sigma_X \int \mathrm{R}_{X^-\sqcup g}(\lambda)X_1 d\nu_{X^-}^{\otimes m}(g)\right)\right| \geq \frac{mt}{3\alpha\beta}\right)$$

$$+ n\mathbb{P}\left(\left|X_1X_1^\top - \mathrm{tr}\,(\Sigma_X)\right| \geq \frac{\sqrt{\lambda^3(n+m)}mt}{6\alpha\beta(\sigma_G + u(n))}\right)$$

$$+ \mathbb{P}\left(\left|\mathrm{tr}\left(\Lambda_G(X)\mathbb{E}\left[\mathrm{R}_{\mathrm{Aug}}(\lambda) \mid X\right]\right) - \mathbb{E}\left[\mathrm{tr}\left(\Lambda_G(X)\mathrm{R}_{\mathrm{Aug}}(\lambda)\right)\right]\right| \geq \frac{mt}{3\alpha}\right).$$

We now bound each term that appears in the left side of the previous equation, beginning with the third term, remark that the function $g : \mathbf{X} \mapsto \mathrm{tr}\left(\Lambda_G(\mathbf{X})\int \mathrm{R}_{\mathbf{X}\sqcup g}(\lambda)d\nu_{\mathbf{X}^{\otimes m}}(g)\right)$ is Lipschitz, and,

$$\mathbb{P}\left(\left|\mathrm{tr}\left(\mathbb{E}\left[\Lambda_G(X)\mathrm{R}_{\mathrm{Aug}}(\lambda) \mid X\right] - \mathbb{E}\left[\Lambda_G(X)\mathrm{R}_{\mathrm{Aug}}(\lambda)\right]\right)\right| \geq \frac{mt}{3\alpha}\right) = \mathbb{P}\left(\left|g(X) - \mathbb{E}\left[g(X)\right]\right| \geq \frac{mt}{3\alpha}\right)$$

indeed, writting for $\mathbf{X}, \mathbf{Y} \in \mathbb{R}^{d \times n}$,

$$
\begin{aligned}
|g(\mathbf{X}) - g(\mathbf{Y})| &\leq \left| \operatorname{tr}\left( \{\Lambda_G(\mathbf{X}) - \Lambda_G(\mathbf{Y})\} \int \mathrm{R}_{\mathbf{X} \sqcup g}(\lambda) \, d\nu_{\mathbf{X}^{\otimes m}}(g) \right) \right| \\
&\quad + \left| \operatorname{tr}\left( \Lambda_G(\mathbf{Y}) \left\{ \int \mathrm{R}_{\mathbf{X} \sqcup g}(\lambda) \, d\nu_{\mathbf{X}^{\otimes m}}(g) - \int \mathrm{R}_{\mathbf{Y} \sqcup g}(\lambda) \, d\nu_{\mathbf{Y}^{\otimes m}}(g) \right\} \right) \right| \\
&\leq \left( \frac{\mathrm{L}_\Lambda \sqrt{d}}{\lambda} + \frac{2\|\Lambda_G(\mathbf{Y})\|_{\mathrm{op}}(1 + \sqrt{m}\mathrm{L}_G)\sqrt{d}}{\lambda^{3/2}\sqrt{(n+m)}} \right) \|\mathbf{X} - \mathbf{Y}\|_{\mathrm{F}} \\
&\leq \left( \frac{\mathrm{L}_\Lambda \sqrt{d}}{\lambda} + \frac{2\kappa(1 + \sqrt{m}\mathrm{L}_G)\sqrt{d}}{\lambda^{3/2}\sqrt{(n+m)}} \right) \|\mathbf{X} - \mathbf{Y}\|_{\mathrm{F}} .
\end{aligned}
$$

where the last bounds were derived by using **H**4, Lemma 7, and the fact that $\|\Lambda_G(\mathbf{Y})\|_{\mathrm{op}} \leq \kappa$ (as well as the fact that $\mathbf{X} \sqcup \mathbf{G} \mapsto \mathrm{R}_{\mathbf{X} \sqcup \mathbf{G}}(\lambda)$ is $2\lambda^{-3/2}(n+m)^{-1/2}$-Lispchitz). Furthermore, note,

$$
\begin{aligned}
\frac{3\alpha}{m} \left( \frac{\mathrm{L}_\Lambda \sqrt{d}}{\lambda} + \frac{2\kappa(1 + \sqrt{m}\mathrm{L}_G)\sqrt{d}}{\lambda^{3/2}\sqrt{(n+m)}} \right) &\leq \frac{3}{\sqrt{\lambda^3(m+n)}} \left( \frac{\mathrm{L}_\Lambda}{\sqrt{\lambda}} + \frac{2\kappa(1 + \sqrt{m}\mathrm{L}_G)}{\sqrt{n+m}} \right) \\
&\lesssim \frac{1}{\sqrt{\lambda^3(m+n)}} \left( \frac{\mathrm{L}_\Lambda}{\sqrt{\lambda}} + \sqrt{\alpha}\kappa\mathrm{L}_G + \frac{\kappa}{\sqrt{n+m}} \right) ,
\end{aligned}
$$

Hence, the third term in (54) is bounded by applying **H**1, we get for a universal constant $k$,

$$
\mathbb{P}\left( \left| \operatorname{tr}\left( \mathbb{E}\left[ \Lambda_G(X) \mathrm{R}_{\mathrm{Aug}}(\lambda) \mid X \right] - \mathbb{E}\left[ \Lambda_G(X) \mathrm{R}_{\mathrm{Aug}}(\lambda) \right] \right) \right| \geq \frac{mt}{3\alpha} \right) \leq 2\exp\left( -k \frac{\lambda^3(n+m)^3 t^2}{\left( \mathrm{L}_\Lambda/\sqrt{\lambda} + \sqrt{\alpha}\kappa\mathrm{L}_G + \kappa/\sqrt{n+m} \right)^2} \right) .
$$

We now focus on the first term in (54), by using the Hanson-Wright inequality, we have for a universal constant $k$,

$$
\begin{aligned}
&\mathbb{P}\left( \left| X_1^\top \int \mathrm{R}_{X^- \sqcup g}(\lambda) d\nu_{X^-}^{\otimes m}(g) X_1 - \operatorname{tr}\left( \Sigma_X \int \mathrm{R}_{X^- \sqcup g}(\lambda) X_1 d\nu_{X^-}^{\otimes m}(g) \right) \right| \geq \frac{mt}{3\alpha\beta} \right) \\
&\leq 2\exp\left( -k \min\left\{ \frac{\lambda^2(n+m)^2 t^2}{d\beta^2}, \frac{\lambda(n+m)t}{\beta} \right\} \right) \leq 2\exp\left( -k(n+m)\min\left\{ \frac{\lambda^2 t^2}{\beta^2}, \frac{\lambda t}{\beta} \right\} \right) ,
\end{aligned}
$$

where we have used the fact that $m/\alpha = (n+m)$, as well as $\|\mathrm{R}_{\mathbf{X} \sqcup \mathbf{G}}(\lambda)\|_{\mathrm{op}} \leq \lambda^{-1}$. Similarly for the second term in (54),

$$
\begin{aligned}
&\mathbb{P}\left( \left| X_1 X_1^\top - \operatorname{tr}\left( \Sigma_X \right) \right| \geq \frac{\sqrt{\lambda^3(n+m)}mt}{6\alpha\beta(\sigma_G + u(n))} \right) \leq 2\exp\left( -c \min\left\{ \frac{\lambda^3(n+m)^3 t^2}{\beta(\sigma_G + u(n))^2 d}, \frac{\sqrt{\lambda^3(n+m)^3}t}{\beta(\sigma_G + u(n))} \right\} \right) \\
&\leq 2\exp\left( -c(n+m)\min\left\{ \frac{\lambda^3(n+m)t^2}{\beta^2(\sigma_G + u(n))^2}, \frac{\sqrt{\lambda^3(n+m)}t}{\beta(\sigma_G + u(n))} \right\} \right) .
\end{aligned}
$$

We conclude, by merging the three previous bounds in Equation (54), it holds for a universal constant $k > 0$,

$$
\begin{aligned}
&\mathbb{P}\left( \left| \mathfrak{a}_g(X) - \mathfrak{a}_g^* \right| \geq t + \frac{\alpha\beta}{m} \left( \frac{4(\sigma_G + u(n))\operatorname{tr}(\Sigma_X)}{\sqrt{\lambda^3(n+m)}} + \frac{(1 + \mathrm{L}_G)}{\sqrt{\lambda^3(n+m)}} \right) \right) \\
&\leq 2n\exp\left( -k(n+m)\min\left\{ \frac{\lambda^2 t^2}{\beta^2}, \frac{\lambda t}{\beta} \right\} \right) + 2n\exp\left( -k(n+m)\min\left\{ \frac{\lambda^3(n+m)t^2}{\beta^2(\sigma_G + u(n))^2}, \frac{\sqrt{\lambda^3(n+m)}t}{\beta(\sigma_G + u(n))} \right\} \right) \\
&\quad + 2\exp\left( -k(n+m)\frac{\lambda^3(n+m)^2 t^2}{\alpha^2 \left( \mathrm{L}_\Lambda/\sqrt{\lambda} + \sqrt{\alpha}\kappa\mathrm{L}_G + \kappa/\sqrt{n+m} \right)^2} \right)
\end{aligned}
$$

Thus, only keeping the dominant term, define

$$\zeta_g(t) = \min\left\{\frac{\lambda^2 t^2}{\beta^2}, \frac{\lambda t}{\beta}, \frac{\lambda^3(n+m)t^2}{\beta^2(\sigma_G + u(n))^2}, \frac{\sqrt{\lambda^3}t}{\beta(\sigma_G + u(n))}, \frac{\lambda^3(n+m)^2 t^2}{\alpha^2\left(L_\Lambda/\sqrt{\lambda} + \sqrt{\alpha}\kappa L_G + \kappa/\sqrt{n+m}\right)^2} + \frac{\ln(n)}{n+m}\right\}$$
$$- \frac{\ln(n)}{n+m},$$

we have shown, for a universal constant $c$,

$$\mathbb{P}\left(\left|\mathfrak{a}_g(X) - \mathfrak{a}_g^*\right| \geq t + \frac{\alpha\beta}{m}\left(\frac{4(\sigma_G + u(n))\operatorname{tr}(\Sigma_X)}{\sqrt{\lambda^3(n+m)}} + \frac{(1+L_G)}{\sqrt{\lambda^3(n+m)}}\right)\right) \leq 6\mathbb{P}\left(-c(n+m)\zeta_g(t)\right), \tag{55}$$

We now turn to the concentration of $\mathfrak{a}_x(X)$, we have from (7) and the triangle inequality,

$$\left|\mathfrak{a}_x(X) - \mathfrak{a}_x^*\right| \leq \left|\frac{1 - (1 - \beta/\mathfrak{a}_g(X))\alpha}{n}\operatorname{tr}\left(\{X_1 X_1^\top - \Sigma_X\}\int R_{X^- \sqcup g}(\lambda)d\nu_{\mathbf{X}^-}^{\otimes m}(g)\right)\right|$$

$$+ \left|\frac{1 - (1 - \beta/\mathfrak{a}_g(X))}{n}\operatorname{tr}\left(\Sigma_X\left\{\int R_{X^- \sqcup g}(\lambda)d\nu_{X^-}^{\otimes m}(g) - \mathbb{E}[R_{\text{Aug}}(\lambda)]\right\}\right)\right|$$

$$+ \beta\alpha\left|\frac{1}{\mathfrak{a}_g(X)} - \frac{1}{\mathfrak{a}_g^*}\right|\frac{1}{n}\operatorname{tr}(\Sigma_X\mathbb{E}[R_{\text{Aug}}(\lambda)])$$

$$\leq \left|\frac{1-\alpha}{n}\operatorname{tr}\left(\{X_1 X_1^\top - \Sigma_X\}\int R_{X^- \sqcup g}(\lambda)d\nu_{\mathbf{X}^-}^{\otimes m}(g)\right)\right|$$

$$+ \left|\frac{1-\alpha}{n}\operatorname{tr}\left(\Sigma_X\left\{\int R_{X^- \sqcup g}(\lambda)d\nu_{X^-}^{\otimes m}(g) - \mathbb{E}\left[\int R_{X^- \sqcup g}(\lambda)d\nu_{X^-}^{\otimes m}(g)\right]\right\}\right)\right|$$

$$+ \left|\frac{1-\alpha}{n}\operatorname{tr}\left(\Sigma_X\left\{\mathbb{E}\left[\int R_{X^- \sqcup g}(\lambda)d\nu_{X^-}^{\otimes m}(g)\right] - \mathbb{E}[R_{\text{Aug}}(\lambda)]\right\}\right)\right|$$

$$+ \frac{\beta\alpha\|\Sigma_X\|_{\text{op}}d}{n\lambda}\left|\mathfrak{a}_g(X) - \mathfrak{a}_g^*\right|$$

where we have used the fact that $\mathfrak{a}_g(X) \geq \mathfrak{a}_g^*$, and $\mathfrak{a}_g^* \geq 1$, as well as the Cauchy-Schwarz inequality. Similarly as previously, we bound the deviation probability of each term independantly then use a union bound argument to conclude. First,

$$\mathbb{P}\left(\left|\frac{1-\alpha}{n}\operatorname{tr}\left(\{X_1 X_1^\top - \Sigma_X\}\int R_{X^- \sqcup g}(\lambda)d\nu_{X^-}^{\otimes m}(g)\right)\right| \geq t\right)$$

$$= \mathbb{E}\left[\mathbb{P}\left(\left|\operatorname{tr}\left(\{X_1 X_1^\top - \Sigma_X\}\int R_{X^- \sqcup g}(\lambda)d\nu_{X^-}^{\otimes m}(g)\right)\right| \geq \frac{nt}{1-\alpha}\bigg| X^-\right)\right]$$

$$\leq 2\exp\left(-k\min\left\{\frac{\lambda^2 n^2 t^2}{(1-\alpha)^2 d}, \frac{\lambda n t}{(1-\alpha)}\right\}\right)$$

$$\leq 2\exp\left(-k(n+m)\min\left\{\frac{\lambda^2 t^2}{1-\alpha}, \lambda t\right\}\right),$$

which followed from the Hanson-Wright inequality. The second term is controlled by remarking that,

$$\mathbb{P}\left(\left|\frac{1-\alpha}{n}\operatorname{tr}\left(\Sigma_X\left\{\int R_{X^- \sqcup g}(\lambda)d\nu_{X^-}^{\otimes m}(g) - \mathbb{E}\left[\int R_{X^- \sqcup g}(\lambda)d\nu_{X^-}^{\otimes m}(g)\right]\right\}\right)\right| \geq t\right)$$

$$= \mathbb{P}\left(\left|g(X^-) - \mathbb{E}[g(X^-)]\right| \geq (n+m)t\right)$$

where $g : \mathbf{X} \mapsto \operatorname{tr}\left(\Sigma_X \int R_{\mathbf{X} \sqcup g}(\lambda)d\nu_{\mathbf{X}}^{\otimes m}(g)\right)$ is $2\sqrt{d}\|\Sigma_X\|_{\text{op}}(1 + \sqrt{m}L_G)\lambda^{-3/2}(n+m)^{-1/2}$-Lispchitz, and so does $\mathbf{X} \mapsto g(X^-)$ by composition of Lispchitz maps. It resutls from **H**1,

$$\mathbb{P}\left(\left|\frac{1-\alpha}{n}\operatorname{tr}\left(\Sigma_X\left\{\int R_{X^- \sqcup g}(\lambda)d\nu_{X^-}^{\otimes m}(g) - \mathbb{E}\left[\int R_{X^- \sqcup g}(\lambda)d\nu_{X^-}^{\otimes m}(g)\right]\right\}\right)\right| \geq t\right)$$

$$\leq 2\exp\left(-k\frac{(n+m)^3\lambda^3 t^2}{d\|\Sigma_X\|_{\text{op}}^2(1 + \sqrt{m}L_G)^2}\right) \leq 2\exp\left(-k\frac{(n+m)\lambda^3 t^2}{\|\Sigma_X\|_{\text{op}}^2(\sqrt{\alpha}L_G + 1/\sqrt{n+m})^2}\right)$$

The third term is bounded by using the Caucy-Schwarz inequality,

$$\left| \frac{1-\alpha}{n} \operatorname{tr}\left( \Sigma_X \left\{ \mathbb{E}\left[ \int R_{X^-\sqcup g}(\lambda) d\nu_{X^-}^{\otimes m}(g) \right] - \mathbb{E}\left[ R_{\mathrm{Aug}}(\lambda) \right] \right\} \right) \right|$$

$$\leq \frac{(1-\alpha)\|\Sigma_X\|_{\mathrm{op}}\sqrt{d}}{n} \left\| \mathbb{E}\left[ \int R_{X^-\sqcup g}(\lambda) d\nu_{X^-}^{\otimes m}(g) \right] - \mathbb{E}\left[ R_{\mathrm{Aug}}(\lambda) \right] \right\|_{\mathrm{F}}$$

and, using the shermann-Morisson's formula, it results,

$$\left\| \mathbb{E}\left[ \int R_{X^-\sqcup g}(\lambda) d\nu_{X^-}^{\otimes m}(g) \right] - \mathbb{E}\left[ R_{\mathrm{Aug}}(\lambda) \right] \right\|_{\mathrm{F}}$$

$$\leq \left\| \mathbb{E}\left[ \int R_{X^-\sqcup g}(\lambda) d\{\nu_{X^-}^{\otimes m} - \nu_X^{\otimes m}\}(g) \right] \right\|_{\mathrm{F}} + \left\| \mathbb{E}\left[ \int \left\{ R_{X^-\sqcup g}(\lambda) - R_{X\sqcup g}(\lambda) \right\} d\nu_X^{\otimes m}(g) \right] \right\|_{\mathrm{F}}$$

$$= \left\| \mathbb{E}\left[ \int R_{X^-\sqcup g}(\lambda) d\{\nu_{X^-}^{\otimes m} - \nu_X^{\otimes m}\}(g) \right] \right\|_{\mathrm{F}} + \frac{1}{n+m}\left\| \mathbb{E}\left[ \int \frac{R_{X^-\sqcup g}(\lambda)X_1 X_1^\top R_{X^{-1}\sqcup g}(\lambda)}{1+(n+m)^{-1}X_1^\top R_{X^-\sqcup g}(\lambda)X_1} d\nu_X^{\otimes m}(g) \right] \right\|_{\mathrm{F}}$$

remarking that, for the Lowner order $\preceq$, we have,

$$\frac{R_{X^-\sqcup g}(\lambda)X_1 X_1^\top R_{X^{-1}\sqcup g}(\lambda)}{1+(n+m)^{-1}X_1^\top R_{X^-\sqcup g}(\lambda)X_1} \preceq R_{X^-\sqcup g}(\lambda)X_1 X_1^\top R_{X^{-1}\sqcup g}(\lambda) ,$$

further using the facts that the Lowner order is preserved when integrating over the distribution of the random matrices, and that the Forbenius norm is increasing for the Lowner order on PSd matrices, we have,

$$\left\| \mathbb{E}\left[ \int \frac{R_{X^-\sqcup g}(\lambda)X_1 X_1^\top R_{X^{-1}\sqcup g}(\lambda)}{1+(n+m)^{-1}X_1^\top R_{X^-\sqcup g}(\lambda)X_1} d\nu_X^{\otimes m}(g) \right] \right\| \leq \left\| \mathbb{E}\left[ \int R_{X^-\sqcup g}(\lambda)X_1 X_1^\top R_{X^{-1}\sqcup g}(\lambda) d\nu_X^{\otimes m}(g) \right] \right\|$$

$$\leq \mathbb{E}\left[ \int \| R_{X^-\sqcup g}(\lambda)X_1 \|_2^2 d\nu_X^{\otimes m}(g) \right]$$

$$\leq \mathbb{E}\left[ \frac{\|X_1\|_2^2}{\lambda} \right]$$

$$\leq \frac{\operatorname{tr}(\Sigma_X)}{\lambda} ,$$

Thus,

$$\left\| \mathbb{E}\left[ \int R_{X^-\sqcup g}(\lambda) d\nu_{X^-}^{\otimes m}(g) \right] - \mathbb{E}\left[ R_{\mathrm{Aug}}(\lambda) \right] \right\|_{\mathrm{F}} \leq \left\| \mathbb{E}\left[ \int R_{X^-\sqcup g}(\lambda) d\{\nu_{X^-}^{\otimes m} - \nu_X^{\otimes m}\}(g) \right] \right\|_{\mathrm{F}} + \frac{\operatorname{tr}(\Sigma_X)}{\lambda(n+m)} ,$$

Finally, using the dual representation of the Frobeniusn norm, **H5**, and the Lispchitz property of $g \mapsto R_{X^-\sqcup g}(\lambda)$ we have,

$$\left\| \mathbb{E}\left[ \int R_{X^-\sqcup g}(\lambda) d\{\nu_{X^-}^{\otimes m} - \nu_X^{\otimes m}\}(g) \right] \right\|_{\mathrm{F}} = \sup_{\|\mathbf{B}\|_{\mathrm{F}}=1} \mathbb{E}\left[ \int \operatorname{tr}\left( \mathbf{B}\, R_{X^-\sqcup g}(\lambda) \right) d\{\nu_{X^-}^{\otimes m} - \nu_X^{\otimes m}\}(g) \right]$$

$$\leq \sup_{\|\mathbf{B}\|_{\mathrm{F}}=1} \frac{2\mathbb{E}\left[ W_1(\nu_X^{\otimes m}, \nu_{X^-}^{\otimes m}) \right]}{\lambda^{3/2}(n+m)^{1/2}}$$

$$\leq \frac{2\sqrt{m}u(n)}{\lambda^{3/2}(n+m)^{1/2}} = \frac{2\sqrt{\alpha}u(n)}{\lambda^{3/2}}$$

we conclude on the third term by,

$$\left| \frac{1-\alpha}{n} \operatorname{tr}\left( \Sigma_X \left\{ \mathbb{E}\left[ \int R_{X^-\sqcup g}(\lambda) d\nu_{X^-}^{\otimes m}(g) \right] - \mathbb{E}\left[ R_{\mathrm{Aug}}(\lambda) \right] \right\} \right) \right|$$

$$\leq \frac{(1-\alpha)\|\Sigma_X\|_{\mathrm{op}}\sqrt{d}}{n} \left\| \mathbb{E}\left[ \int R_{X^-\sqcup g}(\lambda) d\nu_{X^-}^{\otimes m}(g) \right] - \mathbb{E}\left[ R_{\mathrm{Aug}}(\lambda) \right] \right\|_{\mathrm{F}}$$

$$\leq \frac{(1-\alpha)\|\Sigma_X\|_{\mathrm{op}}\sqrt{d}}{n} \left( \frac{2u(n)}{\lambda^{3/2}} + \frac{\operatorname{tr}(\Sigma_X)}{\lambda(n+m)} \right)$$

$$\leq \frac{\|\Sigma_X\|_{\mathrm{op}}}{\sqrt{n}} \left( \frac{2\sqrt{\alpha}u(n)}{\lambda^{3/2}} + \frac{\operatorname{tr}(\Sigma_X)}{\lambda(n+m)} \right)$$

finally, the deviation probability of $\mathfrak{a}_g(X)$ that appears in (55) was already controlled in the first part of the proof, we thus conclude throuh a union bound argument that,

$$
\mathbb{P}\left(\left|\mathfrak{a}_x(X) - \mathfrak{a}_x^*\right| \geq t + \frac{\|\Sigma_X\|_{\mathrm{op}}}{\sqrt{n}}\left(\frac{2\sqrt{\alpha}u(n)}{\lambda^{3/2}} + \frac{\mathrm{tr}\,(\Sigma_X)}{\lambda(n+m)}\right) + \delta_g\right)
$$

$$
\leq \mathbb{P}\left(\left|\frac{1-\alpha}{n}\,\mathrm{tr}\left(\left\{X_1 X_1^\top - \Sigma_X\right\}\int \mathrm{R}_{X^-\sqcup g}(\lambda)d\nu_{X^-}^{\otimes m}(g)\right)\right| \geq \frac{t}{3}\right)
$$

$$
+ \mathbb{P}\left(\left|\frac{1-\alpha}{n}\,\mathrm{tr}\left(\Sigma_X\left\{\int \mathrm{R}_{X^-\sqcup g}(\lambda)d\nu_{X^-}^{\otimes m}(g) - \mathbb{E}\left[\int \mathrm{R}_{X^-\sqcup g}(\lambda)d\nu_{X^-}^{\otimes m}(g)\right]\right\}\right)\right| \geq \frac{t}{3}\right)
$$

$$
+ \mathbb{P}\left(\frac{\beta\alpha\|\Sigma_X\|_{\mathrm{op}}d}{n\lambda}\left|\mathfrak{a}_g(X) - \mathfrak{a}_g^*\right| \geq \frac{t}{3} + \delta_g\right)
$$

$$
\leq 2\exp\left(-c(n+m)\min\left\{\frac{\lambda^2 t^2}{(1-\alpha)}, \lambda t\right\}\right) + 2\exp\left(-k\frac{(n+m)\lambda^3 t^2}{\sigma_G^2(\sqrt{\alpha}\mathrm{L}_G + 1/\sqrt{n+m})^2}\right)
$$

$$
+ 6\exp\left(-c(n+m)\zeta_g\left(\frac{n\lambda t}{\beta\alpha\|\Sigma_X\|_{\mathrm{op}}d}\right)\right)
$$

$$
\leq 2\exp\left(-c(n+m)\min\left\{\frac{\lambda^2 t^2}{(1-\alpha)}, \lambda t\right\}\right) + 2\exp\left(-k\frac{(n+m)\lambda^3 t^2}{\sigma_G^2(\sqrt{\alpha}\mathrm{L}_G + 1/\sqrt{n+m})^2}\right)
$$

$$
+ 6\exp\left(-c(n+m)\zeta_g\left(\frac{\lambda t}{\beta\|\Sigma_X\|_{\mathrm{op}}}\right)\right)
$$

defining,

$$
\zeta_x(t) := \min\left\{\frac{\lambda^2 t^2}{(1-\alpha)}, \lambda t, \frac{\lambda^3 t^2}{\sigma_G^2(\sqrt{\alpha}\mathrm{L}_G + 1/\sqrt{n+m})^2}, \zeta_x\left(\frac{\lambda t}{\beta\|\Sigma_X\|_{\mathrm{op}}}\right)\right\}
$$

the claim follows.  $\qquad\square$

## D.2 Proof of Theorem 2

This section is dedicated to the proof of Theorem 2. To this end, we define $\Delta\mathcal{E}_{\mathrm{Aug}}(\lambda) = \mathcal{E}_{\mathrm{Aug}}(\lambda) - \hat{\mathcal{E}}_{\mathrm{Aug}}(\lambda)$, and we notice that

$$
\Delta\mathcal{E}_{\mathrm{Aug}}(\lambda) = 2\left\{-\frac{1}{d}tr\left(\Sigma_X^{-1}\,\mathrm{R}_{\mathrm{Aug}}(\lambda)\right) + (\Phi_1(X) - \Phi_2(X))\right\}\,, \tag{56}
$$

where $\Phi_1$ and $\Phi_2$ are defined in (8). The proof of Theorem 2 goes in several step that we hereby describe. In the first step (Appendix D.2.1), we provide tight concentration bounds for $\Phi_1$ and $\Phi_2$. In a second step (Appendix D.2.2), we shall bound $\mathbb{E}\left[\Delta\mathcal{E}_{\mathrm{Aug}}(\lambda)\right]$. Finally, in the third and last step (Appendix D.2.3), we deduce the concentration property of $\Delta\mathcal{E}_{\mathrm{Aug}}(\lambda)$

### D.2.1 Concentration of $\Phi_1$ and $\Phi_2$

First recall the definitions of $\Phi_1(X)$ and $\Phi_2(X)$ from (8),

$$
\Phi_1(\mathbf{X}) = \frac{(1-d/n)}{d}\,\mathrm{tr}\left(\mathrm{R}_{\mathbf{X}}(0)\left(\frac{\alpha\Lambda_G(\mathbf{X})}{\mathfrak{a}_g(\mathbf{X})} + \lambda\,\mathrm{I}_d\right)^{-1}\right)\mathbb{1}_{\mathrm{A}_\eta}(\mathbf{X})\,,
$$

$$
\Phi_2(\mathbf{X}) = \frac{1 - (1 - \beta/\mathfrak{a}_g(\mathbf{X}))\alpha}{d\mathfrak{a}_x(\mathbf{X})}\,\mathrm{tr}\left(\bar{\mathrm{R}}_{G|X}^{(\mathfrak{a}_g(\mathbf{X}))}(\lambda, \mathbf{X})\left(\frac{\alpha\Lambda_G(\mathbf{X})}{\mathfrak{a}_g(\mathbf{X})} + \lambda\,\mathrm{I}_d\right)^{-1}\right)\,,
$$

We begin by introducing the auxilary functions $\Psi_1, \Psi_2$ defined as follows,

$$
\forall \mathbf{X} \in \mathbb{R}^{d\times n}\,, \qquad
\begin{aligned}
\Psi_1(\mathbf{X}) &= \frac{1 - (d/n)}{d}\,\mathrm{tr}\left(\mathrm{R}_{\mathbf{X}}(0)\left(\frac{\alpha\Lambda_G(\mathbf{X})}{\mathfrak{a}_g^*} + \lambda\,\mathrm{I}_d\right)^{-1}\right)\mathbb{1}_{\mathrm{A}_\eta}(\mathbf{X})\,,\\[2mm]
\Psi_2(\mathbf{X}) &= \frac{1 - (1 - \beta/\mathfrak{a}_g^*)\alpha}{d\mathfrak{a}_x^*}\,\mathrm{tr}\left(\bar{\mathrm{R}}_G^{(\mathfrak{a}_g^*)}(\lambda)\left(\frac{\alpha\Lambda_G(\mathbf{X})}{\mathfrak{a}_g^*} + \lambda\,\mathrm{I}_d\right)^{-1}\right)\,,
\end{aligned}
\tag{57}
$$

where $(\mathfrak{a}_x^*, \mathfrak{a}_g^*)$ were defined in Theorem 4.

This first part of the proof consists in showing that $\Phi_1(X)$ and $\Phi_2(X)$ are respectively close to $\Psi_1(X)$ and $\Psi_2(X)$, and then showing that the functions $\Psi_1$ and $\Psi_2$ are Lipschitz, which will results in sub-Gaussian concentration bounds from **H**1.

**Concentration bounds for $\Phi_1(X) - \Psi_1(X)$ and $\Phi_2(X) - \Psi_2(X)$**

We now show that $\Phi_1(X) - \Psi_1(X)$ has sub-exponential tail, to do so, write the following almost sure decomposition,

$$
\begin{aligned}
&|\Phi_1(X) - \Psi_1(X)| \\
&= \frac{1 - (d/n)}{d} \operatorname{tr}\left(R_X(0)\right) \operatorname{tr}\left(R_X(0)\left\{\left(\frac{\alpha\Lambda_G(X)}{\mathfrak{a}_g(X)} + \lambda \operatorname{I}_d\right)^{-1} - \left(\frac{\alpha\Lambda_G(X)}{\mathfrak{a}_g^*} + \lambda \operatorname{I}_d\right)^{-1}\right\}\right) \mathbb{1}_{\mathsf{A}_\eta}(X) \\
&= \left|\frac{1}{\mathfrak{a}_g(X)} - \frac{1}{\mathfrak{a}_g^*}\right| \frac{\alpha(1 - (d/n))}{d} \operatorname{tr}\left(R_X(0)\left(\frac{\alpha\Lambda_G(X)}{\mathfrak{a}_g(X)} + \lambda \operatorname{I}_d\right)^{-1} \Lambda_G(X)\left(\frac{\alpha\Lambda_G(X)}{\mathfrak{a}_g^*} + \lambda \operatorname{I}_d\right)^{-1}\right) \mathbb{1}_{\mathsf{A}_\eta}(X) \\
&\leq \left|\mathfrak{a}_g(X) - \mathfrak{a}_g^*\right| \frac{\alpha(1 - (d/n))}{\mathfrak{a}_g(X)\eta\lambda} \left\|\frac{\alpha\Lambda_G(X)}{\mathfrak{a}_g^*}\left(\frac{\alpha\Lambda_G(X)}{\mathfrak{a}_g^*} + \lambda \operatorname{I}_d\right)^{-1}\right\|_{\mathrm{op}} \\
&\leq \left|\mathfrak{a}_g(X) - \mathfrak{a}_g^*\right| \frac{\alpha}{\eta\lambda}
\end{aligned}
$$

This implies,

$$
\mathbb{P}\left(|\Phi_1(X) - \Psi_1(X)| \geq t\right) \leq \mathbb{P}\left(\left|\mathfrak{a}_g(X) - \mathfrak{a}_g^*\right| \geq \frac{\eta\lambda t}{\alpha}\right) \,,
$$

Leveraging Proposition 9, the previous implies in a straightwordard way that,

$$
\mathbb{P}\left(|\Phi_1(X) - \Psi_1(X)| \geq t + \frac{\alpha\delta_g}{\eta\lambda}\right) \leq \mathbb{P}\left(\left|\mathfrak{a}_g(X) - \mathfrak{a}_g^*\right| \geq \frac{\eta\lambda t}{\alpha} + \delta_g\right) \lesssim \exp\left(-c(n + m)\zeta_g\left(\frac{\eta\lambda t}{\alpha}\right)\right) \,,
\tag{58}
$$

where $\zeta_g$ was defined in (9).

Similarly, we write for $\Phi_2(X) - \Psi_2(X)$,

$$
\begin{aligned}
|\Phi_2(X) - \Psi_2(X)| \leq{} & \left|\frac{1}{\mathfrak{a}_g^*} - \frac{1}{\mathfrak{a}_g(X)}\right| \frac{\beta\alpha}{d\mathfrak{a}_x(X)} \operatorname{tr}\left(\bar{R}_{G|X}^{(\mathfrak{a}_g(X))}(\lambda, X)\left(\frac{\alpha\Lambda_G(X)}{\mathfrak{a}_g(X)} + \lambda \operatorname{I}_d\right)^{-1}\right) \\
& + \left|\frac{1}{\mathfrak{a}_x^*} - \frac{1}{\mathfrak{a}_x(X)}\right| \frac{1 - \alpha'}{d} \operatorname{tr}\left(\bar{R}_{G|X}^{(\mathfrak{a}_g(X))}(\lambda, X)\left(\frac{\alpha\Lambda_G(X)}{\mathfrak{a}_g(X)} + \lambda \operatorname{I}_d\right)^{-1}\right) \\
& + \left|\frac{1 - \alpha'}{d\mathfrak{a}_x^*} \operatorname{tr}\left(\left\{\bar{R}_{G|X}^{(\mathfrak{a}_g(X))}(\lambda, X) - \bar{R}_{G|X}^{(\mathfrak{a}_g^*)}(\lambda, X)\right\}\left(\frac{\alpha\Lambda_G(X)}{\mathfrak{a}_g(X)} + \lambda \operatorname{I}_d\right)^{-1}\right)\right| \\
& + \left|\frac{1 - \alpha'}{d\mathfrak{a}_x^*} \operatorname{tr}\left(\bar{R}_{G|X}^{(\mathfrak{a}_g^*)}(\lambda, X)\left\{\left(\frac{\alpha\Lambda_G(X)}{\mathfrak{a}_g(X)} + \lambda \operatorname{I}_d\right)^{-1} - \left(\frac{\alpha\Lambda_G(X)}{\mathfrak{a}_g^*} + \lambda \operatorname{I}_d\right)^{-1}\right\}\right)\right|
\end{aligned}
$$

Recalling that $\bar{\mathrm{R}}_{G|X}^{(\mathfrak{a}_g^*)}(\lambda, X)$ was defined in (6), we further get, from $\mathbf{A}^{-1} - \mathbf{B}^{-1} = \mathbf{A}^{-1}(\mathbf{B} - \mathbf{A})\mathbf{B}^{-1}$, that,

$$|\Phi_2(X) - \Psi_2(X)|$$

$$\leq \left| \frac{1}{\mathfrak{a}_g^*} - \frac{1}{\mathfrak{a}_g(X)} \right| \frac{\beta\alpha}{d\mathfrak{a}_x(X)} \operatorname{tr}\left( \bar{\mathrm{R}}_{G|X}^{(\mathfrak{a}_g(X))}(\lambda, X) \left( \frac{\alpha\Lambda_G(X)}{\mathfrak{a}_g(X)} + \lambda\,\mathrm{I}_d \right)^{-1} \right)$$

$$+ \left| \frac{1}{\mathfrak{a}_x^*} - \frac{1}{\mathfrak{a}_x(X)} \right| \frac{1 - \alpha'}{d} \operatorname{tr}\left( \bar{\mathrm{R}}_{G|X}^{(\mathfrak{a}_g(X))}(\lambda, X) \left( \frac{\alpha\Lambda_G(X)}{\mathfrak{a}_g(X)} + \lambda\,\mathrm{I}_d \right)^{-1} \right)$$

$$+ \left| \frac{1}{\mathfrak{a}_g(X)} - \frac{1}{\mathfrak{a}_g^*} \right| \frac{1 - \alpha'}{d\mathfrak{a}_x^*} \operatorname{tr}\left( \left\{ \bar{\mathrm{R}}_{G|X}^{(\mathfrak{a}_g(X))}(\lambda, X)\,(\alpha\beta C_X + \alpha\Lambda_G(X))\,\bar{\mathrm{R}}_{G|X}^{(\mathfrak{a}_g^*)}(\lambda, X) \right\} \left( \frac{\alpha\Lambda_G(X)}{\mathfrak{a}_g(X)} + \lambda\,\mathrm{I}_d \right)^{-1} \right)$$

$$+ \left| \frac{1}{\mathfrak{a}_g(X)} - \frac{1}{\mathfrak{a}_g^*} \right| \frac{1 - \alpha'}{d\mathfrak{a}_x^*} \operatorname{tr}\left( \bar{\mathrm{R}}_{G|X}^{(\mathfrak{a}_g^*)}(\lambda, X) \left\{ \left( \frac{\alpha\Lambda_G(X)}{\mathfrak{a}_g(X)} + \lambda\,\mathrm{I}_d \right)^{-1} \alpha\Lambda_G(X) \left( \frac{\alpha\Lambda_G(X)}{\mathfrak{a}_g^*} + \lambda\,\mathrm{I}_d \right)^{-1} \right\} \right)$$

and, applying Cauchy-Schwarz inequality, as well as using that all dilation factors are greater than 1, we get,

$$|\Phi_2(X) - \Psi_2(X)| \leq \left|\mathfrak{a}_g^* - \mathfrak{a}_g(X)\right| \frac{\beta\alpha}{\lambda^2}$$

$$+ \left|\mathfrak{a}_x^* - \mathfrak{a}_x(X)\right| \frac{1 - \alpha'}{\lambda^2}$$

$$+ \left|\mathfrak{a}_g(X) - \mathfrak{a}_g^*\right| \frac{1 - \alpha'}{\lambda^2} \left\| \frac{\alpha\beta C_X + \alpha\Lambda_G(X)}{\mathfrak{a}_g^*} \bar{\mathrm{R}}_{G|X}^{(\mathfrak{a}_g^*)}(\lambda, X) \right\|_{\mathrm{op}}$$

$$+ \left|\mathfrak{a}_g(X) - \mathfrak{a}_g^*\right| \frac{1 - \alpha'}{\lambda^2} \left\| \frac{\alpha\Lambda_G(X)}{\mathfrak{a}_g^*} \left( \frac{\alpha\Lambda_G(X)}{\mathfrak{a}_g^*} + \lambda\,\mathrm{I}_d \right)^{-1} \right\|_{\mathrm{op}}$$

$$\leq \frac{\beta\alpha + 2(1 - \alpha')}{\lambda^2} \left|\mathfrak{a}_g(X) - \mathfrak{a}_g^*\right| + \frac{1 - \alpha'}{\lambda^2} |\mathfrak{a}_x(X) - \mathfrak{a}_x^*|$$

$$\leq \frac{1}{\lambda^2} |\mathfrak{a}_g(X) - \mathfrak{a}_g^*| + \frac{1}{\lambda^2} |\mathfrak{a}_x(X) - \mathfrak{a}_x^*| \,.$$

It results, from a unoin bound argument, that,

$$\mathbb{P}\left( |\Phi_2(X) - \Psi_2(X)| \geq t + \frac{\delta_x + \delta_g}{\lambda^2} \right) \leq \mathbb{P}\left( |\mathfrak{a}_g(X) - \mathfrak{a}_g^*| \geq \lambda^2 t + \delta_g \right) + \mathbb{P}\left( |\mathfrak{a}_x(X) - \mathfrak{a}_x^*| \geq \lambda^2 t + \delta_x \right)$$

$$\lesssim \exp\left( -c(n + m) \min\left\{ \zeta_x(\lambda^2 t), \zeta_g(\lambda^2 t) \right\} \right) \,,$$

where the last line (as well as the definitions of $\delta_x$, $\delta_g$, $\zeta_x$ and $\zeta_g$) followed from Proposition 9.

**Concentration bounds for $\Psi_1(X)$ and $\Psi_2(X)$**

As previously discussed, the concentration of $\Psi_1(X)$ and $\Psi_2(X)$ follows from the Lipschitz properties of $\Psi_1$ and $\Psi_2$, as well as **H**1. Starting by the concentration of $\Psi_1(X)$, we recall that $d < n$ from **H**2 and we write for any $\mathbf{X}, \mathbf{Y} \in \mathsf{A}_\eta$,

$$|\Psi_1(\mathbf{X}) - \Psi_1(\mathbf{Y})| \leq \left| \frac{1}{d} \operatorname{tr}\left( \{R_\mathbf{X}(0) - R_\mathbf{Y}(0)\} \left( \frac{\alpha\Lambda_G(\mathbf{X})}{\mathfrak{a}_g^*} + \lambda\,\mathrm{I}_d \right)^{-1} \right) \right|$$

$$+ \left| \frac{1}{d} \operatorname{tr}\left( R_\mathbf{Y}(0) \left\{ \left( \frac{\alpha\Lambda_G(\mathbf{X})}{\mathfrak{a}_g^*} + \lambda\,\mathrm{I}_d \right)^{-1} - \left( \frac{\alpha\Lambda_G(\mathbf{Y})}{\mathfrak{a}_g^*} + \lambda\,\mathrm{I}_d \right)^{-1} \right\} \right) \right|$$

$$\leq \left| \frac{1}{d} \operatorname{tr}\left( \{R_\mathbf{X}(0) - R_\mathbf{Y}(0)\} \left( \frac{\alpha\Lambda_G(\mathbf{X})}{\mathfrak{a}_g^*} + \lambda\,\mathrm{I}_d \right)^{-1} \right) \right|$$

$$+ \frac{\alpha}{\mathfrak{a}_g^*} \left| \frac{1}{d} \operatorname{tr}\left( R_\mathbf{Y}(0) \left( \frac{\alpha\Lambda_G(\mathbf{X})}{\mathfrak{a}_g^*} + \lambda\,\mathrm{I}_d \right)^{-1} \left\{ \Lambda_G(\mathbf{X}) - \Lambda_G(\mathbf{Y}) \right\} \left( \frac{\alpha\Lambda_G(\mathbf{Y})}{\mathfrak{a}_g^*} + \lambda\,\mathrm{I}_d \right)^{-1} \right) \right|$$

Using the Cauchy-Schwarz inequality, as well as **H4**, we get

$$|\Psi_1(\mathbf{X}) - \Psi_1(\mathbf{Y})| \leq \frac{1}{\lambda} \frac{1}{\sqrt{d}} \|R_{\mathbf{X}}(0) - R_{\mathbf{Y}}(0)\|_{\mathrm{F}}$$

$$+ \frac{\alpha}{\eta \lambda^2 \sqrt{d}} \|\Lambda_G(\mathbf{X}) - \Lambda_G(\mathbf{Y})\|_{\mathrm{F}}$$

$$\leq \underbrace{\left( \frac{2}{\lambda \eta^{3/2} \sqrt{dn}} + \frac{\alpha \mathrm{L}_\Lambda}{\eta \lambda^2 \sqrt{d}} \right)}_{\mathrm{L}_{\Psi_1}} \|\mathbf{X} - \mathbf{Y}\|_{\mathrm{F}}.$$

Where, we used Lemma 6 and **H4** in the last bound. We have proved that $\Psi_1$ is $\mathrm{L}_{\Psi_1}$-Lispchitz on $\mathsf{A}_\eta$, and $\|\Psi_{1|\mathsf{A}_\eta}\|_\infty \leq \eta^{-1}\lambda^{-1}$, we have from Proposition 6 that $\Psi_1(X)$ is $\sigma_1$-sub-Gaussian, with

$$\sigma_1 \lesssim \mathrm{L}_{\Psi_1}^2 + \|\Psi_{1|\mathsf{A}_\eta}\|_\infty^2 \sigma_{\mathsf{A}_\eta}^2 \lesssim \frac{1}{\lambda \eta^{3/2} \sqrt{dn}} + \frac{\alpha \mathrm{L}_\Lambda}{\eta \lambda^2 \sqrt{d}} ,$$

Hence, there exists a constant $k > 0$ such that,

$$\mathbb{P}\left(|\Psi_1(X) - \mathbb{E}\left[\Psi_1(X)\right]| \geq t\right) \leq 2 \exp\left(-k \frac{t^2}{(\lambda^{-1}\eta^{-3/2}/\sqrt{nd} + \mathrm{L}_\Lambda \eta^{-1}\lambda^{-2}/\sqrt{d})^2}\right) , \quad (60)$$

Similarly for $\Psi_2(X)$, we write for any $\mathbf{X}, \mathbf{Y} \in \mathbb{R}^{d \times n}$,

$$|\Psi_2(\mathbf{X}) - \Psi_2(\mathbf{Y})| \leq \frac{1}{\mathfrak{a}_x^* d} \mathrm{tr}\left( \left\{ \bar{\mathrm{R}}_{G|X}^{(\mathfrak{a}_g^*)}(\lambda, \mathbf{X}) - \bar{\mathrm{R}}_{G|X}^{(\mathfrak{a}_g^*)}(\lambda, \mathbf{Y}) \right\} \left( \frac{\alpha \Lambda_G(\mathbf{X})}{\mathfrak{a}_g^*} + \lambda \mathrm{I}_d \right)^{-1} \right)$$

$$+ \frac{1}{\mathfrak{a}_x^* d} \left| \mathrm{tr}\left( \bar{\mathrm{R}}_{G|X}^{(\mathfrak{a}_g^*)}(\lambda, \mathbf{Y}) \left\{ \left( \frac{\alpha \Lambda_G(\mathbf{X})}{\mathfrak{a}_g^*} + \lambda \mathrm{I}_d \right)^{-1} - \left( \frac{\alpha \Lambda_G(\mathbf{Y})}{\mathfrak{a}_g^*} + \lambda \mathrm{I}_d \right)^{-1} \right\} \right) \right|$$

$$= \frac{1}{\mathfrak{a}_x^* d} \mathrm{tr}\left( \left\{ \bar{\mathrm{R}}_{G|X}^{(\mathfrak{a}_g^*)}(\lambda, \mathbf{X}) - \bar{\mathrm{R}}_{G|X}^{(\mathfrak{a}_g^*)}(\lambda, \mathbf{Y}) \right\} \left( \frac{\alpha \Lambda_G(\mathbf{X})}{\mathfrak{a}_g^*} + \lambda \mathrm{I}_d \right)^{-1} \right)$$

$$+ \frac{\alpha}{\mathfrak{a}_g^* \mathfrak{a}_x^* d} \left| \mathrm{tr}\left( \bar{\mathrm{R}}_{G|X}^{(\mathfrak{a}_g^*)}(\lambda, \mathbf{Y}) \left( \frac{\alpha \Lambda_G(\mathbf{X})}{\mathfrak{a}_g^*} + \lambda \mathrm{I}_d \right)^{-1} \{\Lambda_G(\mathbf{X}) - \Lambda_G(\mathbf{Y})\} \left( \frac{\alpha \Lambda_G(\mathbf{Y})}{\mathfrak{a}_g^*} + \lambda \mathrm{I}_d \right)^{-1} \right) \right|$$

and, using the Cauchy-Scharz inequality, we get,

$$|\Psi_2(\mathbf{X}) - \Psi_2(\mathbf{Y})| \leq \frac{1}{\mathfrak{a}_x^* \lambda \sqrt{d}} \|\bar{\mathrm{R}}_{G|X}^{(\mathfrak{a}_g^*)}(\lambda, \mathbf{X}) - \bar{\mathrm{R}}_{G|X}^{(\mathfrak{a}_g^*)}(\lambda, \mathbf{Y})\|_{\mathrm{F}}$$

$$+ \frac{\alpha}{\mathfrak{a}_g^* \mathfrak{a}_x^* \lambda^2 \eta \sqrt{d}} \|\Lambda_G(\mathbf{X}) - \Lambda_G(\mathbf{Y})\|_{\mathrm{F}}$$

$$\leq \underbrace{\frac{2}{\mathfrak{a}_x^* \lambda \eta^{3/2} \sqrt{d(n+m)}} + \frac{\alpha \mathrm{L}_\Lambda}{\mathfrak{a}_g^* \mathfrak{a}_x^* \lambda^2 \eta \sqrt{d}}}_{= \mathrm{L}_{\Psi_2}} \|\mathbf{X} - \mathbf{Y}\|_{\mathrm{F}} .$$

where we used the Lipschitz property of $\mathbf{X} \mapsto \bar{\mathrm{R}}_{G|X}^{(\mathfrak{a}_g^*)}(\lambda, \mathbf{X})$, which follows from Lemma 6. Remark further that,

$$\mathrm{L}_{\Psi_2} \lesssim \mathrm{L}_{\Psi_1} \lesssim \frac{1}{\lambda \eta^{3/2} \sqrt{d(n+m)}} + \frac{\mathrm{L}_\Lambda}{\eta \lambda^2 \sqrt{d}} ,$$

thus, we get from **H1**, the existence of a constant $k$ such that,

$$\mathbb{P}\left(|\Psi_2(X) - \mathbb{E}\left[\Psi_2(X)\right]| \geq t\right) \leq 2 \exp\left(-k \frac{t^2}{(\lambda^{-1}\eta^{-3/2}/\sqrt{d(n+m)} + \mathrm{L}_\Lambda \eta^{-1}\lambda^{-2}/\sqrt{d})^2}\right) ,$$

$$(61)$$

### D.2.2 An upper bound on the asymptotic bias of $\hat{\mathcal{E}}_{\text{Aug}}(\lambda)$

Let us denote for sake of notational simplicity,

$$
\begin{aligned}
\delta_{\text{Aug}} &= \frac{\sqrt{d}\sigma_G^2(\beta^3\|\Sigma_X\|_{\text{op}}^3 + \kappa^3)}{n\lambda_1(\Sigma_X)((1-\alpha)\eta+\lambda)^6}(\kappa q_1 + q_2)\left\{\lambda_1(\Sigma_X) + \frac{1\|\Sigma_X\|_{\text{op}}}{d(\beta^3\|\Sigma_X\|_{\text{op}}^3+\kappa^3)\sqrt{n}}\right\} \\
&\quad + \left(\frac{\alpha\beta}{1-\alpha}+1\right)\frac{\alpha\beta\sqrt{d}(\|\Sigma_X\|_{\text{op}}+q_3\|\bar{\Lambda}_G\|_{\text{op}})}{((1-\alpha)\eta+\lambda)^{5/2}m}\left\{(1+u(n))+\sqrt{\alpha}\,\text{L}_G + (1+c_X^{-1/2})\sqrt{\eta+\lambda}\right\} \\
&\quad + \left(\frac{\alpha\beta}{1-\alpha}+1\right)\left(\frac{1}{((1-\alpha)\eta+\lambda)^2}+\frac{\alpha\beta\sqrt{d}}{((1-\alpha)\eta+\lambda)^{5/2}m}\right)\mathbb{E}\left[\|\Lambda_G(X)-\bar{\Lambda}_G\|_{\text{F}}\right] \\
&\leq \frac{\sqrt{d}\sigma_G^2\|\Sigma_X\|_{\text{op}}(\|\Sigma_X\|_{\text{op}}^3+\kappa^3)}{n\lambda_1(\Sigma_X)\lambda^6}(\kappa q_1 + q_2)\left\{\frac{\lambda_1(\Sigma_X)}{\|\Sigma_X\|_{\text{op}}}+\frac{1}{\kappa^3}\right\} \\
&\quad + \frac{\sqrt{d}\|\Sigma_X\|_{\text{op}}+q_3\|\bar{\Lambda}_G\|_{\text{op}}}{(1-\alpha)\lambda^{5/2}m}\left\{(1+u(n))+\sqrt{\alpha}\,\text{L}_G + (1+c_X^{-1/2})\sqrt{\eta+\lambda}\right\} \\
&\quad + \left(\lambda^{1/2}+\frac{\sqrt{d}}{m}\right)\frac{\mathbb{E}\left[\|\Lambda_G(X)-\bar{\Lambda}_G\|_{\text{F}}\right]}{(1-\alpha)\lambda^{5/2}}
\end{aligned}
\tag{62}
$$

such that it holds from Theorem 4,

$$
\left\|\mathbb{E}\left[\left\{\text{R}_{\text{Aug}}(\lambda)-\bar{\text{R}}_{\text{Aug}}^{(\mathfrak{a}_x^*,\mathfrak{a}_g^*)}(\lambda)\right\}\right]\right\|_{\text{F}} \lesssim \delta_{\text{Aug}} .
\tag{63}
$$

Let us first recall from (56),

$$
\Delta\mathcal{E}_{\text{Aug}}(\lambda) = 2\left\{-\frac{1}{d}\text{tr}\left(\Sigma_X^{-1}\text{R}_{\text{Aug}}(\lambda)\right)+(\Phi_1(X)-\Phi_2(X))\right\} ,
$$

We have shown in Appendix D.2.1 that $\Phi_1(X)$ and $\Phi_2(X)$ respectively concentrate around $\mathbb{E}[\Psi_1(X)]$ and $\mathbb{E}[\Psi_2(X)]$. In this section, we derive an upper bound for the aboslute value of

$$
\mathbb{E}\left[\Delta\mathcal{E}_{\text{Aug}}^{\Psi}(\lambda)\right] = 2\left\{-\frac{1}{d}\text{tr}\left(\Sigma_X^{-1}\mathbb{E}[\text{R}_{\text{Aug}}(\lambda)]\right)+\mathbb{E}[\Psi_1(X)]-\mathbb{E}[\Psi_2(X)]\right\} ,
$$

First relying on (63), we write,

$$
\left|\mathbb{E}\left[\Delta\mathcal{E}_{\text{Aug}}^{\Psi}(\lambda)\right]\right| \lesssim \left|-\frac{1}{d}\text{tr}\left(\Sigma_X^{-1}\bar{\text{R}}_{\text{Aug}}^{(\mathfrak{a}_x^*,\mathfrak{a}_g^*)}(\lambda)\right)+\mathbb{E}[\Psi_1(X)]-\mathbb{E}[\Psi_2(X)]\right| + \frac{\delta_{\text{Aug}}}{\lambda_1(\Sigma_X)\sqrt{d}} , \tag{64}
$$

and, we remark that in the case where $\Sigma_X$ and $\bar{\Lambda}_G$ commute, then the first term in the right-hand side can 'linearize', in the general case, we use the notation $[\mathbf{A},\mathbf{B}] = \mathbf{A}\mathbf{B}-\mathbf{B}\mathbf{A}$ for the commutator of two matrices, and we write,

$$
\begin{aligned}
\frac{1}{d}\text{tr}\left(\Sigma_X^{-1}\bar{\text{R}}_{\text{Aug}}^{(\mathfrak{a}_x^*,\mathfrak{a}_g^*)}(\lambda)\right) &= \frac{1}{d}\text{tr}\left(\bar{\text{R}}_{\text{Aug}}^{(\mathfrak{a}_x^*,\mathfrak{a}_g^*)}(\lambda)\left(\frac{\alpha\,\bar{\Lambda}_G}{\mathfrak{a}_g^*}+\lambda\,\text{I}_d\right)\left(\frac{\alpha\,\bar{\Lambda}_G}{\mathfrak{a}_g^*}+\lambda\,\text{I}_d\right)^{-1}\Sigma_X^{-1}\right) \\
&= \frac{1}{d}\text{tr}\left(\bar{\text{R}}_{\text{Aug}}^{(\mathfrak{a}_x^*,\mathfrak{a}_g^*)}(\lambda)\left(\frac{\alpha\bar{\Lambda}_G}{\mathfrak{a}_g^*}+\lambda\,\text{I}_d\right)\Sigma_X^{-1}\left(\frac{\alpha\bar{\Lambda}_G}{\mathfrak{a}_g^*}+\lambda\,\text{I}_d\right)^{-1}\right) \\
&\quad - \frac{1}{d}\text{tr}\left(\bar{\text{R}}_{\text{Aug}}^{(\mathfrak{a}_x^*,\mathfrak{a}_g^*)}(\lambda)\left(\frac{\alpha\bar{\Lambda}_G}{\mathfrak{a}_g^*}+\lambda\right)\left[\left(\frac{\alpha\bar{\Lambda}_G}{\mathfrak{a}_g^*}+\lambda\,\text{I}_d\right)^{-1},\Sigma_X^{-1}\right]\right) .
\end{aligned}
\tag{65}
$$

We now bound the second term and provide a new expression for the first one.

Using the identity $[\mathbf{A}^{-1}, \mathbf{B}^{-1}] = \mathbf{A}^{-1}\mathbf{B}^{-1}[\mathbf{A}, \mathbf{B}]\mathbf{B}^{-1}\mathbf{A}^{-1}$, the Cauchy-Schwarz inequality, and $\|\cdot\|_{\mathrm{F}} \leq \sqrt{d}\,\|\cdot\|_{\mathrm{op}}$, we can bound the term involving the commutator as

$$
\left| \frac{1}{d} \operatorname{tr} \left( \bar{\mathrm{R}}_{\mathrm{Aug}}^{(\mathfrak{a}_x^*, \mathfrak{a}_g^*)}(\lambda) \left( \frac{\alpha \bar{\Lambda}_G}{\mathfrak{a}_g^*} + \lambda \mathrm{I}_d \right) \left[ \left( \frac{\alpha \bar{\Lambda}_G}{\mathfrak{a}_g^*} + \lambda \mathrm{I}_d \right)^{-1}, \Sigma_X^{-1} \right] \right) \right|
$$

$$
= \left| \frac{1}{d} \operatorname{tr} \left( \bar{\mathrm{R}}_{\mathrm{Aug}}^{(\mathfrak{a}_x^*, \mathfrak{a}_g^*)}(\lambda) \Sigma_X^{-1} \left[ \frac{\alpha \bar{\Lambda}_G}{\mathfrak{a}_g^*} + \lambda \mathrm{I}_d, \Sigma_X \right] \Sigma_X^{-1} \left( \frac{\alpha \bar{\Lambda}_G}{\mathfrak{a}_g^*} + \lambda \mathrm{I}_d \right)^{-1} \right) \right|
$$

$$
= \alpha \left| \frac{1}{d \mathfrak{a}_g^*} \operatorname{tr} \left( \bar{\mathrm{R}}_{\mathrm{Aug}}^{(\mathfrak{a}_x^*, \mathfrak{a}_g^*)}(\lambda) \Sigma_X^{-1} \left[ \bar{\Lambda}_G, \Sigma_X \right] \Sigma_X^{-1} \left( \frac{\alpha \bar{\Lambda}_G}{\mathfrak{a}_g^*} + \lambda \mathrm{I}_d \right)^{-1} \right) \right|
$$

$$
\leq \alpha \frac{\left\| \Sigma_X^{-1} \left( \frac{\alpha \bar{\Lambda}_G}{\mathfrak{a}_g^*} + \lambda \mathrm{I}_d \right)^{-1} \bar{\mathrm{R}}_{\mathrm{Aug}}^{(\mathfrak{a}_x^*, \mathfrak{a}_g^*)}(\lambda) \Sigma_X^{-1} \right\|_{\mathrm{op}}}{\sqrt{d}\,\mathfrak{a}_g^*} \|[\bar{\Lambda}_G, \Sigma_X]\|_{\mathrm{F}} \lesssim \frac{\|\Sigma_X\|_{\mathrm{op}}^2}{\sqrt{d}\lambda^2} \|[\bar{\Lambda}_G, \Sigma_X]\|_{\mathrm{F}} ,
$$

$$(66)$$

Furthermore, using $\mathbf{A}^{-1} - \mathbf{B}^{-1} = \mathbf{A}^{-1}(\mathbf{B} - \mathbf{A})\mathbf{B}^{-1}$ we get,

$$
\bar{\mathrm{R}}_{\mathrm{Aug}}^{(\mathfrak{a}_x^*, \mathfrak{a}_g^*)}(\lambda) \left( \frac{\alpha \Lambda_G}{\mathfrak{a}_g^*} + \lambda \mathrm{I}_d \right) \Sigma_X^{-1} = \bar{\mathrm{R}}_{\mathrm{Aug}}^{(\mathfrak{a}_x^*, \mathfrak{a}_g^*)}(\lambda) \left( \frac{\alpha \Lambda_G}{\mathfrak{a}_g^*} + \lambda \mathrm{I}_d \right) \left( (1 - (1 - \beta/\mathfrak{a}_g^*)\alpha) \frac{\Sigma_X}{\mathfrak{a}_x^*} \right)^{-1} \frac{1 - (1 - \beta/\mathfrak{a}_g^*)\alpha}{\mathfrak{a}_x^*}
$$

$$
= \Sigma_X^{-1} - \frac{(1 - (1 - \beta/\mathfrak{a}_g^*)\alpha)}{\mathfrak{a}_x^*} \bar{\mathrm{R}}_{\mathrm{Aug}}^{(\mathfrak{a}_x^*, \mathfrak{a}_g^*)}(\lambda) . \qquad (67)
$$

Plugging (66)-(67) in (65), we get,

$$
\frac{1}{d} \operatorname{tr} \left( \Sigma_X^{-1} \bar{\mathrm{R}}_{\mathrm{Aug}}^{(\mathfrak{a}_x^*, \mathfrak{a}_g^*)}(\lambda) \right) \lesssim \frac{1}{d} \operatorname{tr} \left( \Sigma_X^{-1} \left( \frac{\bar{\Lambda}_G}{\mathfrak{a}_g^*} + \lambda \mathrm{I}_d \right)^{-1} \right) - \frac{1 - (1 - \beta/\mathfrak{a}_g^*)\alpha}{\mathfrak{a}_x^*} \operatorname{tr} \left( \bar{\mathrm{R}}_{\mathrm{Aug}}^{(\mathfrak{a}_x^*, \mathfrak{a}_g^*)}(\lambda) \left( \frac{\bar{\Lambda}_G}{\mathfrak{a}_g^*} + \lambda \mathrm{I}_d \right)^{-1} \right)
$$

$$
+ \frac{\|\Sigma_X\|_{\mathrm{op}}^2}{\sqrt{d}\lambda^2} \|[\bar{\Lambda}_G, \Sigma_X]\|_{\mathrm{F}}
$$

Plugging the previous equation in (64), we get,

$$
\left| \mathbb{E} \left[ \Delta \mathcal{E}_{\mathrm{Aug}}^{\Psi}(\lambda) \right] \right| \lesssim \left| \frac{1}{d} \operatorname{tr} \left( \Sigma_X^{-1} \left( \frac{\bar{\Lambda}_G}{\mathfrak{a}_g^*} + \lambda \mathrm{I}_d \right)^{-1} \right) - \mathbb{E} \left[ \Psi_1(X) \right] \right| \qquad (68)
$$

$$
+ \left| \mathbb{E} \left[ \Psi_2(X) \right] - \frac{1 - (1 - \beta/\mathfrak{a}_g^*)\alpha}{\mathfrak{a}_x^*} \operatorname{tr} \left( \bar{\mathrm{R}}_{\mathrm{Aug}}^{(\mathfrak{a}_x^*, \mathfrak{a}_g^*)}(\lambda) \left( \frac{\bar{\Lambda}_G}{\mathfrak{a}_g^*} + \lambda \mathrm{I}_d \right)^{-1} \right) \right|
$$

$$
+ \frac{\|\Sigma_X\|_{\mathrm{op}}^2}{\sqrt{d}\lambda^2} \|[\bar{\Lambda}_G, \Sigma_X]\|_{\mathrm{F}} + \frac{\delta_{\mathrm{Aug}}}{\lambda_1(\Sigma_X)\sqrt{d}} ,
$$

We thus simply need to bound the biases of $\Psi_1(X)$ and $\Psi_2(X)$, for notation simplicity again, we introduce the notations,

$$
\overline{\Psi}_1 = \frac{1}{d} \operatorname{tr} \left( \Sigma_X^{-1} \left( \frac{\bar{\Lambda}_G}{\mathfrak{a}_g^*} + \lambda \mathrm{I}_d \right)^{-1} \right) \qquad (69)
$$

$$
\overline{\Psi}_2 = \frac{1 - (1 - \beta/\mathfrak{a}_g^*)\alpha}{\mathfrak{a}_x^*} \operatorname{tr} \left( \bar{\mathrm{R}}_{\mathrm{Aug}}^{(\mathfrak{a}_x^*, \mathfrak{a}_g^*)}(\lambda) \left( \frac{\bar{\Lambda}_G}{\mathfrak{a}_g^*} + \lambda \mathrm{I}_d \right)^{-1} \right)
$$

Then, we have by definition of $\Psi_1(X)$ in (57)

$$
\mathbb{E} \left[ \Psi_1(X) \right] = \frac{1 - (d/n)}{d} \mathbb{E} \left[ \operatorname{tr} \left( R_X(0) \left\{ \left( \frac{\alpha \Lambda_G(X)}{\mathfrak{a}_g^*} + \lambda \mathrm{I}_d \right)^{-1} - \left( \frac{\alpha \bar{\Lambda}_G}{\mathfrak{a}_g^*} + \lambda \mathrm{I}_d \right)^{-1} \right\} \right) \mathbb{1}_{\mathsf{A}_\eta}(X) \right]
$$

$$
+ \frac{1 - (d/n)}{d} \operatorname{tr} \left( \mathbb{E} \left[ \left\{ R_X(0) - \frac{\Sigma_X^{-1}}{1 - (d/n)} \right\} \mathbb{1}_{\mathsf{A}_\eta}(X) \right] \left( \frac{\alpha \bar{\Lambda}_G}{\mathfrak{a}_g^*} + \lambda \mathrm{I}_d \right)^{-1} \right)
$$

$$
+ \overline{\Psi}_1 \mathbb{P}(X \in \mathsf{A}_\eta) .
$$

Thus, thanks to the triangle inequality,

$$\left|\mathbb{E}\left[\Psi_1(X)\right] - \bar{\Psi}_1\right| \leq \left|\frac{1}{d}\,\mathbb{E}\left[\text{tr}\left(R_X(0)\left\{\left(\frac{\alpha\Lambda_G(X)}{\mathfrak{a}_g^*} + \lambda\,\mathrm{I}_d\right)^{-1} - \left(\frac{\alpha\bar{\Lambda}_G}{\mathfrak{a}_g^*} + \lambda\,\mathrm{I}_d\right)^{-1}\right\}\right)\mathbb{1}_{\mathsf{A}_\eta}(X)\right]\right|$$

$$+ \left|\frac{1}{d}\,\text{tr}\left(\mathbb{E}\left[\left\{R_X(0) - \frac{\Sigma_X^{-1}}{1 - (d/n)}\right\}\mathbb{1}_{\mathsf{A}_\eta}(X)\right]\left(\frac{\alpha\bar{\Lambda}_G}{\mathfrak{a}_g^*} + \lambda\,\mathrm{I}_d\right)^{-1}\right)\right|$$

$$+ \bar{\Psi}_1(1 - \mathbb{P}(\mathsf{A}_\eta))\,.$$

furthermore, using the Cauchy-Schwarz inequality and the Jensen's inequality, we get

$$\left|\mathbb{E}\left[\Psi_1(X)\right] - \bar{\Psi}_1\right| \leq \frac{1}{\eta\sqrt{d}}\mathbb{E}\left[\left\|\left(\frac{\alpha\Lambda_G(X)}{\mathfrak{a}_g^*} + \lambda\,\mathrm{I}_d\right)^{-1} - \left(\frac{\alpha\bar{\Lambda}_G}{\mathfrak{a}_g^*} + \lambda\,\mathrm{I}_d\right)^{-1}\right\|_{\mathrm{F}}\mathbb{1}_{\mathsf{A}_\eta}(X)\right] \quad (70)$$

$$+ \frac{1}{\lambda\sqrt{d}}\left\|\mathbb{E}\left[\left\{R_X(0) - \frac{\Sigma_X^{-1}}{1 - (d/n)}\right\}\mathbb{1}_{\mathsf{A}_\eta}(X)\right]\right\|_{\mathrm{F}}$$

$$+ (1 - \mathbb{P}(\mathsf{A}_\eta))\bar{\Psi}_1\,.$$

Using $\mathbf{A}^{-1} - \mathbf{B}^{-1} = \mathbf{A}^{-1}\{\mathbf{B} - \mathbf{A}\}\mathbf{B}^{-1}$, we can bound the first term in (70),

$$\mathbb{E}\left[\left\|\left(\frac{\alpha\Lambda_G(X)}{\mathfrak{a}_g^*} + \lambda\,\mathrm{I}_d\right)^{-1} - \left(\frac{\alpha\bar{\Lambda}_G}{\mathfrak{a}_g^*} + \lambda\,\mathrm{I}_d\right)^{-1}\right\|_{\mathrm{F}}\mathbb{1}_{\mathsf{A}_\eta}(X)\right] \leq \frac{\alpha}{\lambda^2}\mathbb{E}\left[\|\Lambda_G(X) - \bar{\Lambda}_G\|_{\mathrm{F}}\right]\,,$$
$$(71)$$

the second term in (70) is controlled by applying Proposition 8, remark that $(1 - (d/n))^{-1}$ is the fixed point of $\mathfrak{b} \mapsto 1 + \mathfrak{b}(d/n)$ (which implies that $(1 - (d/n))^{-1}\Sigma_X^{-1} = \bar{\mathrm{R}}_X^{\mathfrak{b}^*}(0)$), the second term is bounded as

$$\left\|\mathbb{E}\left[R_X(0) - \frac{\Sigma_X^{-1}}{1 - (d/n)}\mathbb{1}_{\mathsf{A}_\eta}(X)\right]\right\|_{\mathrm{F}} \quad (72)$$

$$\lesssim \left(1 + \frac{\|\Sigma_X\|_{\mathrm{op}}}{\lambda_1(\Sigma_X)}\right)\left(\frac{q\sqrt{d}\|\Sigma_X\|_{\mathrm{op}}^3}{n\lambda_1(\Sigma_X)\eta^6} + \left(\|\Sigma_X\|_{\mathrm{op}} + \frac{\|\Sigma_X\|_{\mathrm{op}}^2}{\eta}\right)e^{-c_X n}\right)\,,$$

for $q$ being a polynomial function in $\eta + \lambda_1(\mathbf{D}), \lambda_1(\Sigma_X), \|\Sigma_X\|_{\mathrm{op}}^{-1}, c_X^{-1}$, and $n^{-1}$.

Finally, the last term in (70) is controlled thanks to **H**2, prcesiely, it holds,

$$(1 - \mathbb{P}(\mathsf{A}_\eta))\bar{\Psi}_1 \leq \frac{\|\Sigma_X^{-1}\|_{\mathrm{op}}}{\lambda}e^{-c_X n} = \frac{1}{\lambda_1(\Sigma_X)\lambda}e^{-c_X n}\,. \quad (73)$$

Putting (71), (72) and (73) together in (70), we get,

$$\left|\mathbb{E}\left[\Psi_1(X)\right] - \bar{\Psi}_1\right| \lesssim \frac{1}{\lambda\sqrt{d}}\left(1 + \frac{\|\Sigma_X\|_{\mathrm{op}}}{\lambda_1(\Sigma_X)}\right)\left(\frac{q\sqrt{d}\|\Sigma_X\|_{\mathrm{op}}^3}{n\lambda_1(\Sigma_X)\eta^6} + \left(\frac{1}{\lambda_1(\Sigma_X)\lambda} + \|\Sigma_X\|_{\mathrm{op}} + \frac{\|\Sigma_X\|_{\mathrm{op}}^2}{\eta}\right)e^{-c_X n}\right)$$

$$+ \frac{\alpha}{\eta\lambda^2\sqrt{d}}\mathbb{E}\left[\|\Lambda_G(X) - \bar{\Lambda}_G\|_{\mathrm{F}}\right] \quad (74)$$

$$\lesssim \left(1 + \frac{\lambda_1(\Sigma_X)}{\|\Sigma_X\|_{\mathrm{op}}}\right)\frac{q\|\Sigma_X\|_{\mathrm{op}}^4}{n\lambda_1(\Sigma_X)^2\lambda\eta^6}$$

$$+ \left(1 + \frac{\lambda_1(\Sigma_X)}{\|\Sigma_X\|_{\mathrm{op}}}\right)\left(\frac{1}{\|\Sigma_X\|_{\mathrm{op}}^2} + \frac{\lambda_1(\Sigma_X)\min\{\lambda,\eta\}}{\|\Sigma_X\|_{\mathrm{op}}} + \lambda_1(\Sigma_X)\right)\frac{\|\Sigma_X\|_{\mathrm{op}}^3 e^{c_X n}}{\lambda\lambda_1(\Sigma_X)^2\min\{\lambda,\eta\}\sqrt{d}}$$

$$+ \frac{\alpha}{\eta\lambda^2\sqrt{d}}\mathbb{E}\left[\|\Lambda_G(X) - \bar{\Lambda}_G\|_{\mathrm{F}}\right]\,.$$

We now turn to the bias of $\Psi_2(X)$, recalling (57), and (69), we have,

$$\Psi_2(X) - \bar{\Psi}_2 = \frac{1-\alpha'}{\mathfrak{a}_x^* d} \operatorname{tr}\left(\bar{R}_{G|X}^{(\mathfrak{a}_g^*)}(\lambda, X)\left\{\left(\frac{\alpha\Lambda_G(X)}{\mathfrak{a}_g^*} + \lambda I_d\right)^{-1} - \left(\frac{\alpha\bar{\Lambda}_G}{\mathfrak{a}_g^*} + \lambda I_d\right)^{-1}\right\}\right)$$

$$+ \frac{1-\alpha'}{\mathfrak{a}_x^* d} \operatorname{tr}\left(\left\{\bar{R}_{G|X}^{(\mathfrak{a}_g^*)}(\lambda, X) - \bar{R}_{\mathrm{Aug}}^{(\mathfrak{a}_x^*, \mathfrak{a}_g^*)}\right\}\left(\frac{\alpha\bar{\Lambda}_G}{\mathfrak{a}_g^*} + \lambda I_d\right)^{-1}\right)$$

thus, using the triangle inequality yields,

$$\left|\mathbb{E}\left[\Psi_2(X)\right] - \bar{\Psi}_2\right| = \frac{1-\alpha'}{\mathfrak{a}_x^* d}\mathbb{E}\left[\left|\operatorname{tr}\left(\bar{R}_{G|X}^{(\mathfrak{a}_g^*)}(\lambda, X)\left\{\left(\frac{\alpha\Lambda_G(X)}{\mathfrak{a}_g^*} + \lambda I_d\right)^{-1} - \left(\frac{\alpha\bar{\Lambda}_G}{\mathfrak{a}_g^*} + \lambda I_d\right)^{-1}\right\}\right)\right|\right]$$

$$+ \frac{1-\alpha'}{\mathfrak{a}_x^* d}\left|\operatorname{tr}\left(\left\{\mathbb{E}\left[\bar{R}_{G|X}^{(\mathfrak{a}_g^*)}(\lambda, X)\right] - \bar{R}_{\mathrm{Aug}}^{(\mathfrak{a}_x^*, \mathfrak{a}_g^*)}(\lambda)\right\}\left(\frac{\alpha\bar{\Lambda}_G}{\mathfrak{a}_g^*} + \lambda I_d\right)^{-1}\right)\right|,$$

and, further using the Cauchy-Scharz inequality, as well as $1 - \alpha' \leq 1$, we get,

$$\left|\mathbb{E}\left[\Psi_2(X)\right] - \bar{\Psi}_2\right| \leq \frac{1}{\lambda\sqrt{d}}\mathbb{E}\left[\left\|\left(\frac{\alpha\Lambda_G(X)}{\mathfrak{a}_g^*} + \lambda I_d\right)^{-1} - \left(\frac{\alpha\bar{\Lambda}_G}{\mathfrak{a}_g^*} + \lambda I_d\right)^{-1}\right\|_{\mathrm{F}}\right]$$

$$+ \frac{1}{\lambda\sqrt{d}}\left\|\mathbb{E}\left[\bar{R}_{G|X}^{(\mathfrak{a}_g^*)}(\lambda, X)\right] - \bar{R}_{\mathrm{Aug}}^{(\mathfrak{a}_x^*, \mathfrak{a}_g^*)}(\lambda)\right\|_{\mathrm{F}}$$

$$\leq \frac{1}{\lambda^3\sqrt{d}}\mathbb{E}\left[\|\Lambda_G(X) - \bar{\Lambda}_G\|_{\mathrm{F}}\right] + \frac{1}{\lambda\sqrt{d}}\left\|\mathbb{E}\left[\bar{R}_{G|X}^{(\mathfrak{a}_g^*)}(\lambda, X)\right] - \bar{R}_{\mathrm{Aug}}^{(\mathfrak{a}_x^*, \mathfrak{a}_g^*)}(\lambda)\right\|_{\mathrm{F}}.$$

Now, using that $\bar{R}_{G|X}^{(\mathfrak{a}_g^*)}(\lambda, X) = (1-\alpha')^{-1}R_X\left(\alpha\Lambda_G(X)/((1-\alpha')\mathfrak{b}_g^*) + \lambda/(1-\alpha')I_d\right)$, we write,

$$\left|\mathbb{E}\left[\Psi_2(X)\right] - \bar{\Psi}_2\right|$$

$$\leq \frac{1}{\lambda^3\sqrt{d}}\mathbb{E}\left[\|\Lambda_G(X) - \bar{\Lambda}_G\|_{\mathrm{F}}\right]$$

$$+ \frac{1}{\lambda\sqrt{d}}\left\|\mathbb{E}\left[\left((1-\alpha')C_X + \frac{\alpha\Lambda_G(X)}{\mathfrak{a}_g^*} + \lambda I_d\right)^{-1} - \left((1-\alpha')C_X + \frac{\alpha\Lambda_G(X)}{\mathfrak{a}_g^*} + \lambda I_d\right)^{-1}\right]\right\|_{\mathrm{F}}$$

$$+ \frac{1}{(1-\alpha')\lambda\sqrt{d}}\left\|\mathbb{E}\left[R_X\left(\frac{\alpha\bar{\Lambda}_G}{(1-\alpha')\mathfrak{a}_g^*} + \frac{\lambda}{1-\alpha'}I_d\right)\right] - \bar{R}_X^{(\mathfrak{a}_x^*)}\left(\frac{\alpha\bar{\Lambda}_G}{(1-\alpha')\mathfrak{a}_g^*} + \frac{\lambda}{1-\alpha'}I_d\right)\right\|_{\mathrm{F}}$$

$$\lesssim \frac{1}{\lambda^3\sqrt{d}}\mathbb{E}\left[\|\Lambda_G(X) - \bar{\Lambda}_G\|_{\mathrm{F}}\right] + \frac{1}{(1-\alpha')\lambda\sqrt{d}}\left\|\mathbb{E}\left[R_X\left(\mathbf{D}\right)\right] - \bar{R}_X^{(\mathfrak{a}_x^*)}\left(\mathbf{D}\right)\right\|_{\mathrm{F}}$$

Where, we have used the notation,

$$\mathbf{D} = \frac{\alpha\bar{\Lambda}_G}{(1-\alpha')\mathfrak{a}_g^*} + \frac{\lambda}{1-\alpha'}I_d,$$

Finally, using Proposition 8, we get,

$$\left|\mathbb{E}\left[\Psi_2(X)\right] - \bar{\Psi}_2\right| \tag{75}$$

$$\lesssim \frac{1}{(1-\alpha')\lambda\sqrt{d}}\frac{q\sqrt{d}\|\Sigma_X\|_{\mathrm{op}}^3}{n\lambda_1(\Sigma_X)(\eta + \lambda_1(\mathbf{D}))^6} + \frac{1}{\lambda^3\sqrt{d}}\mathbb{E}\left[\|\Lambda_G(X) - \bar{\Lambda}_G\|_{\mathrm{F}}\right].$$

And, remarking that $\lambda_1(\mathbf{D}) \geq \lambda/(1-\alpha')$, we finally get,

$$\left|\mathbb{E}\left[\Psi_2(X)\right] - \bar{\Psi}_2\right| \lesssim \frac{q\|\Sigma_X\|_{\mathrm{op}}^3}{n\lambda_1(\Sigma_X)\lambda((1-\alpha')\eta + \lambda)^6} + \frac{1}{\lambda^3\sqrt{d}}\mathbb{E}\left[\|\Lambda_G(X) - \bar{\Lambda}_G\|_{\mathrm{F}}\right]. \tag{76}$$

### D.2.3 Conclusion on the proof of Theorem 2

To conclude on the proof of Theorem 2, we plug (68), we first show that $\Phi_1(X)$ concentrates around $\bar{\Psi}_1$ (resp. $\Phi_2(X)$ around $\bar{\Psi}_2$) with a bias of order $\delta_{\text{Aug}}$ (resp. $\delta_{\text{Aug}}$). Introduce the following notations,

$$
\begin{aligned}
\delta_{\Psi_1} := {}& \left(1 + \frac{\lambda_1(\Sigma_X)}{\|\Sigma_X\|_{\text{op}}}\right) \frac{q\|\Sigma_X\|_{\text{op}}^4}{n\lambda_1(\Sigma_X)^2 \lambda \eta^6} \\
& + \left(1 + \frac{\lambda_1(\Sigma_X)}{\|\Sigma_X\|_{\text{op}}}\right) \left(\frac{1}{\|\Sigma_X\|_{\text{op}}^2} + \frac{\lambda_1(\Sigma_X)\min\{\lambda,\eta\}}{\|\Sigma_X\|_{\text{op}}} + \lambda_1(\Sigma_X)\right) \frac{\|\Sigma_X\|_{\text{op}}^3 \mathrm{e}^{c_X n}}{\lambda \lambda_1(\Sigma_X)^2 \min\{\lambda,\eta\}\sqrt{d}} \\
& + \frac{\alpha}{\eta \lambda^2 \sqrt{d}} \mathbb{E}\left[\|\Lambda_G(X) - \bar{\Lambda}_G\|_{\text{F}}\right] \\
\delta_{\Psi_2} := {}& \frac{q\|\Sigma_X\|_{\text{op}}^3}{n\lambda_1(\Sigma_X)\lambda((1-\alpha')\eta + \lambda)^6} + \frac{1}{\lambda^3\sqrt{d}} \mathbb{E}\left[\|\Lambda_G(X) - \bar{\Lambda}_G\|_{\text{F}}\right]
\end{aligned}
$$

and, we make the preliminary remark that, for some $C_1$, $C_2$ and $C_3$ independant of $n$, $d$, $m$, and that depend polynomially on $\lambda_1(\Sigma_X)$, $\|\Sigma_X\|_{\text{op}}$ and $\min\{\lambda,\eta\}$, we have,

$$
\begin{aligned}
\delta_{\Psi_1} + \delta_{\Psi_2} &\lesssim \frac{C_1\|\Sigma_X\|_{\text{op}}^4}{n\lambda_1(\Sigma_X)^2\min\{\lambda,\eta\}^7} + \frac{C_2\|\Sigma_X\|_{\text{op}}^3 \mathrm{e}^{-c_X n}}{\lambda_1(\Sigma_X)^2\min\{\lambda,\eta\}^2\sqrt{d}} + \frac{\mathbb{E}\left[\|\Lambda_G(X) - \bar{\Lambda}_G\|_{\text{F}}\right]}{\min\{\eta,\lambda\}^3\sqrt{d}} \quad (77) \\
&\lesssim \frac{C_3(1 + c_X^{-1})\|\Sigma_X\|_{\text{op}}^4}{n\lambda_1(\Sigma_X)^2\min\{\lambda,\eta\}^7} + \frac{\mathbb{E}\left[\|\Lambda_G(X) - \bar{\Lambda}_G\|_{\text{F}}\right]}{\min\{\eta,\lambda\}^3\sqrt{d}}
\end{aligned}
$$

From there, we bound $\mathbb{E}\left[\Delta\mathcal{E}_{\text{Aug}}^\Psi(\lambda)\right]$ as,

$$
\begin{aligned}
\left|\mathbb{E}\left[\Delta\mathcal{E}_{\text{Aug}}^\Psi(\lambda)\right]\right| \lesssim {}& \left|\bar{\Psi}_1 - \mathbb{E}[\Phi_1(X)]\right| + \left|\bar{\Psi}_1 - \mathbb{E}[\Phi_1(X)]\right| \\
& + \frac{\|\Sigma_X\|_{\text{op}}^2}{\sqrt{d}\lambda^2}\|[\bar{\Lambda}_G, \Sigma_X]\|_{\text{F}} + \frac{\delta_{\text{Aug}}}{\lambda_1(\Sigma_X)\sqrt{d}} \\
\lesssim {}& \delta_{\Psi_1} + \delta_{\Psi_2} + \frac{\|\Sigma_X\|_{\text{op}}^2}{\sqrt{d}\lambda^2}\|[\bar{\Lambda}_G, \Sigma_X]\|_{\text{F}} + \frac{\delta_{\text{Aug}}}{\lambda_1(\Sigma_X)\sqrt{d}} \\
\lesssim {}& \frac{C_3(1 + c_X^{-1})\|\Sigma_X\|_{\text{op}}^4}{n\lambda_1(\Sigma_X)^2\min\{\lambda,\eta\}^7} + \frac{\mathbb{E}\left[\|\Lambda_G(X) - \bar{\Lambda}_G\|_{\text{F}}\right]}{\min\{\eta,\lambda\}^3\sqrt{d}} + + \frac{\|\Sigma_X\|_{\text{op}}^2}{\sqrt{d}\lambda^2}\|[\bar{\Lambda}_G, \Sigma_X]\|_{\text{F}} + \frac{\delta_{\text{Aug}}}{\lambda_1(\Sigma_X)\sqrt{d}}
\end{aligned}
$$

Where the first inequality followed from (68), the second from (74) and (76), and the last one followed from (77).

Now, recalling (62), we have,

$$
\begin{aligned}
\frac{\delta_{\text{Aug}}}{\lambda_1(\Sigma_X)\sqrt{d}} \lesssim {}& \frac{\sigma_G^2\|\Sigma_X\|_{\text{op}}\kappa(\|\Sigma_X\|_{\text{op}}^3 + \kappa^3)}{n\lambda_1(\Sigma_X)\lambda^6}\left(q_1 + \frac{q_2}{\kappa}\right)\left\{\frac{\lambda_1(\Sigma_X)}{\|\Sigma_X\|_{\text{op}}} + \frac{1}{\kappa^3}\right\} \\
& + \frac{\|\Sigma_X\|_{\text{op}} + q_3\|\bar{\Lambda}_G\|_{\text{op}}}{(1-\alpha)\lambda_1(\Sigma_X)\lambda^{5/2}m}\left\{(1 + u(n)) + \sqrt{\alpha}\,\mathrm{L}_G + (1 + c_X^{-1/2})\sqrt{\eta + \lambda}\right\} \\
& + \left(\lambda^{1/2} + \frac{\sqrt{d}}{m}\right)\frac{\mathbb{E}\left[\|\Lambda_G(X) - \bar{\Lambda}_G\|\text{F}\right]}{(1-\alpha)\lambda_1(\Sigma_X)\lambda^{5/2}\sqrt{d}}
\end{aligned}
$$

It results that for constants $C_4, C_5$ independant of $n$, $d$, $m$, and that depend polynomially on $\lambda_1(\Sigma_X)$, $\|\Sigma_X\|_{\text{op}}$, $\kappa$, $m/n$, $u(n)$, $\mathrm{L}_G$ and $\min\{\lambda,\eta\}$, we have,

$$
\left|\mathbb{E}\left[\Delta\mathcal{E}_{\text{Aug}}^\Psi(\lambda)\right]\right| \le C_4 \frac{(1 + \sigma_G^2)(1 + c_X^{-1})(\|\Sigma_X\|_{\text{op}}^4\kappa + \|\Sigma_X\|_{\text{op}}\kappa^4)}{(1-\alpha)n\lambda_1(\Sigma_X)^2\min\{\lambda,\eta\}^7} + C_5 \frac{\mathbb{E}\left[\|\Lambda_G(X) - \bar{\Lambda}_G\|_{\text{F}}\right]}{\min\{\eta,\lambda\}^3\sqrt{d}} + \frac{\|\Sigma_X\|_{\text{op}}^2}{\sqrt{d}\lambda^2}\|[\bar{\Lambda}_G, \Sigma_X]\|_{\text{F}}
$$

Writting $\delta_{\text{Total}} = \delta_{\Psi_1} + \delta_{\Psi_2} + \|\Sigma_X\|_{\text{op}}^2 d^{-1/2}\lambda^{-2}\|[\bar{\Lambda}_G, \Sigma_X]\|_{\text{F}} + \delta_{\text{Aug}}$, we write from a union bound,

$$\mathbb{P}\left(|\Delta_{\text{Aug}}(\lambda)| \geq t + K\delta_{\text{Total}}\right) \leq \mathbb{P}\left(\left|\Delta\mathcal{E}_{\text{Aug}}(\lambda) - \mathbb{E}\left[\Delta\mathcal{E}_{\text{Aug}}^{\Psi}(\lambda)\right]\right| \geq t\right)$$

$$\leq \mathbb{P}\left(|\Phi_1(X) - \mathbb{E}[\Psi(X)]| \geq \frac{t}{2}\right) + \mathbb{P}\left(|\Phi_2(X) - \mathbb{E}[\Psi_2(X)]| \geq \frac{t}{2}\right)$$

$$\leq \mathbb{P}\left(|\Phi_1(X) - \Psi_1(X)| \geq \frac{t}{4}\right) + \mathbb{P}\left(|\Psi_1(X) - \mathbb{E}[\Psi_1(X)]| \geq \frac{t}{4}\right)$$

$$+ \mathbb{P}\left(|\Phi_2(X) - \Psi_2(X)| \geq \frac{t}{4}\right) + \mathbb{P}\left(|\Psi_2(X) - \mathbb{E}[\Psi_2(X)]| \geq \frac{t}{4}\right)$$

$$\lesssim \exp\left(-k(n+m)\min\left\{\zeta_x(\lambda^2 t), \zeta_g(\lambda^2 t), \zeta_g\left(\frac{\eta\lambda t}{\alpha}\right)\right\}\right)$$

$$+ \exp\left(-k\frac{t^2}{(\lambda^{-1}\eta^{-3/2}/\sqrt{d(n+m)} + \mathrm{L}_\Lambda\eta^{-1}\lambda^{-2}/\sqrt{d})^2}\right)$$

$$\leq \exp\left(-k(n+m)\min\left\{\zeta_x(\lambda^2 t), \zeta_g(\lambda^2 t), \zeta_g\left(\frac{\eta\lambda t}{\alpha}\right)\right\}\right)$$

$$+ \exp\left(-k\frac{\eta^3\lambda^2 dt^2}{(\mathrm{L}_\Lambda\sqrt{\eta} + 1/\sqrt{(n+m)})^2}\right)$$

Where the last inequality followed from (58), (59), (60) and (61). The previous holds for large enough $K$ and small enough $k$ that are both universal constants.

To conclude the proof, it remains only to simplify the quantities $\delta_{\text{Total}}$ and $\zeta_x$, $\zeta_g$, in order to match the statement of Theorem 2. We start by simplifying the exponent in the concentration statement. First, defining $\varepsilon = \min\{\lambda, \eta, 1\}$, we remark that,

$$\exp\left(-k(n+m)\min\left\{\zeta_x(\lambda^2 t), \zeta_g(\lambda^2 t), \zeta_g\left(\frac{\eta\lambda t}{\alpha}\right)\right\}\right) \leq \exp\left(-k(n+m)\min\left\{\zeta_x(\varepsilon^2 t), \zeta_g(\varepsilon^2 t)\right\}\right)$$

We recall the definition of $\zeta_x(t)$ and $\zeta_g(t)$ from Proposition 9,

$$\zeta_x(t) := \min\left\{\frac{\lambda^2 t^2}{(1-\alpha)}, \lambda t, \frac{\lambda^3 t^2}{\sigma_G^2(\sqrt{\alpha}\mathrm{L}_G + 1/\sqrt{n+m})^2}, \zeta_g\left(\frac{\lambda t}{\beta\|\Sigma_X\|_{\text{op}}}\right)\right\}$$

$$\zeta_g(t) = \min\left\{\frac{\lambda^2 t^2}{\beta^2}, \frac{\lambda t}{\beta}, \frac{\lambda^3(n+m)t^2}{\beta^2(\sigma_G + u(n))^2}, \frac{\sqrt{\lambda^4}t}{\beta(\sigma_G + u(n))}, \frac{\lambda^3(n+m)^2 t^2}{\alpha^2\left(\mathrm{L}_\Lambda + \sqrt{\alpha}\sqrt{\lambda}\kappa\mathrm{L}_G + \kappa\sqrt{\lambda}/\sqrt{n+m}\right)^2} + \frac{\ln(n)}{n+m}\right\}$$

Note that in the worst case scenariom we have $(n+m) = 1$, hence we write,

$$\zeta_g(t) \geq \min\left\{\frac{\varepsilon^2 t^2}{\beta^2}, \frac{\varepsilon t}{\beta}, \frac{\varepsilon^3(n+m)t^2}{\beta^2(\sigma_G + u(n))^2}, \frac{\varepsilon^2 t}{\beta(\sigma_G + u(n))}, \frac{\varepsilon^3(n+m)^2 t^2}{\alpha^2\left(\mathrm{L}_\Lambda + \sqrt{\alpha}\sqrt{\lambda}\kappa\mathrm{L}_G + \kappa\sqrt{\lambda}/\sqrt{n+m}\right)^2} + \frac{\ln(n)}{n+m}\right\}$$

$$\geq \varepsilon^3 \min\left\{\frac{t^2}{\beta^2\varepsilon}, \frac{t}{\beta\varepsilon^2}, \frac{t^2}{\beta^2(\sigma_G + u(n))^2}, \frac{t}{\beta\varepsilon(\sigma_G + u(n))}, \frac{t^2}{\alpha^2\left(\mathrm{L}_\Lambda + \sqrt{\alpha}\sqrt{\lambda}\kappa\mathrm{L}_G + \kappa\sqrt{\lambda}\right)^2}\right\}$$

and

$$\zeta_x(t) \geq \varepsilon^3 \min\left\{\frac{t^2}{\varepsilon(1-\alpha)}, \frac{t}{\varepsilon^2}, \frac{t^2}{\sigma_G^2(\sqrt{\alpha}\mathrm{L}_G + 1)^2}, \zeta_g\left(\frac{\varepsilon t}{\beta\|\Sigma_X\|_{\text{op}}}\right)\right\}$$

We From there, we define,

$$\xi_{1,x} = \varepsilon^2 \qquad \xi_{2,x} = \max\{\varepsilon(1-\alpha), \sigma_G^2(\sqrt{\alpha}L_G + 1)^2\}$$

and,

$$\xi_{1,g} = \max\{\beta^2\varepsilon, \beta\varepsilon(\sigma_G + u(n))\} \qquad \xi_{2,g} = \max\{\beta^2\varepsilon, \beta^2(\sigma_G + u(n))^2, \alpha^2\left(L_\Lambda + \sqrt{\alpha}\sqrt{\lambda}\kappa L_G + \kappa\sqrt{\lambda}\right)^2\}$$

This ensures that,

$$\zeta_g(\varepsilon^2 t) \geq \varepsilon^7 \min\left\{\frac{t}{\varepsilon^2\xi_{1,g}}, \frac{t^2}{\xi_{2,g}}\right\} \geq \varepsilon^9 \min\left\{\frac{t}{\varepsilon^4\xi_{1,g}}, \frac{t^2}{\varepsilon^2\xi_{2,g}}\right\}$$

and,

$$\zeta_x(\varepsilon^2 t) \geq \varepsilon^3 \min\left\{\frac{\varepsilon^2 t}{\xi_{1,g}}, \frac{\varepsilon^4 t^2}{\xi_{2,g}}, \frac{\varepsilon^3 t}{\beta\|\Sigma_X\|_{op}\xi_{1,g}}, \frac{\varepsilon^6 t^2}{\beta^2\|\Sigma_X\|_{op}^2\xi_{2,g}}\right\}$$

$$\geq \varepsilon^9 \min\left\{\frac{t}{\varepsilon^4\xi_{1,g}}, \frac{t^2}{\varepsilon^2\xi_{2,g}}, \frac{t}{\beta\|\Sigma_X\|_{op}\varepsilon^3\xi_{1,g}}, \frac{t^2}{\beta^2\|\Sigma_X\|_{op}^2\xi_{2,g}}\right\}$$

Defining $\rho_1$ and $\rho_2$ such that,

$$\rho_1 = \max\left\{\varepsilon^4\xi_{1,g}, \beta\|\Sigma_X\|_{op}\varepsilon^3\xi_{1,g}, \varepsilon^4\xi_{1,g}\right\} \qquad \rho_2 = \max\left\{\varepsilon^2\xi_{2,g}, \beta^2\|\Sigma_X\|_{op}^2\xi_{2,g}, \varepsilon^2\xi_{2,g}\right\} \tag{78}$$

we have shown,

$$\exp\left(-k(n+m)\min\left\{\zeta_x(\lambda^2 t), \zeta_g(\lambda^2 t), \zeta_g\left(\frac{\eta\lambda t}{\alpha}\right)\right\}\right) \leq \exp\left(-k(n+m)\min\left\{\zeta_x(\varepsilon^2 t), \zeta_g(\varepsilon^2 t)\right\}\right)$$

$$\lesssim n e^{-k(n+m)t^2/\rho_2} + n e^{-k(n+m)t/\rho_1}$$

