# OpenReview forum: "Non-Asymptotic Analysis Of Data Augmentation For Precision Matrix Estimation"
_NeurIPS.cc/2025/Conference — NeurIPS 2025 spotlight_

### Official Review · Reviewer_awgT · 2025-06-19

**Clarity:** 4
**Significance:** 3
**Originality:** 3
**Rating:** 4
**Confidence:** 3

**Summary:**

The paper considers the problem of estimating the inverse covariance matrix in a high-dimensional setting. Suppose that $X_1,\dots,X_n$ are i.i.d. copies of some random vector $X\in \mathbb{R}^d$, where $n$ and $d$ are of the same order. The true covariance matrix is $\Sigma = \mathbb{E}XX^\top$ and the goal is to estimate $\Sigma^{-1}$ from the samples. Since the empirical estimator $\hat{\Sigma} = (1/n) \sum_i X_i X_i^\top$ could be close to being singular, the paper considers the resolvent $R(\lambda) = (\hat{\Sigma} + \lambda I)^{-1}$ and studies the error $E(\lambda) = (1/d)\Vert R(\lambda) - \Sigma\Vert_F^2$.

It proposes an estimator for $E(\lambda)$ and, under certain assumptions (such as Lipschitz concentration) on the distribution, derives a concentration inequality result for the estimator. This enables the use of the estimator as a surrogate objective for tuning $\lambda$.

It also considers an estimator based on data augmentation. Let $g_1, \dots, g_m$ be artificial samples generated from an auxiliary distribution depending on $X$. The augmented resolvent is $R_{aug}(\lambda) = ((n+m)^{-1}(\sum X_i X_i^\top + \sum g_i g_i^\top)^{-1} + \lambda I)^{-1}$ with the associated error $E_{aug}(\lambda) = (1/d)\Vert R_{aug}(\lambda) - \Sigma\Vert_F^2$. Under assumptions on the auxiliary distribution, the paper develops an estimator $E_{aug}(\lambda)$ and proves a concentration bound.

**Questions:**

1. Proposition 1: What is B? Why is the right-hand side of the inequality independent of B? If I rescale B up by a large factor, the tail probability should be much smaller (depending on the factor)?
2. Can authors comment on the dependence on eta in their results? Delta_X(lambda) has a dependence of 1/eta^6 and the tail has a dependence of eta^3, which do not look good for small eta.
3. Can eta depend on n or d? For instance, can eta be at the order of 1/sqrt(n)?

Minor points:
- Line 16: reads better by changing it to “of a random mean-zero vector from i.i.d. Samples”
- Line 79: isn’t E a measurable subset of E^{d x k}?
- Equation (3): “tr” in “tr(R_X(lambda))” should be in upright font
- Line 228: what is A?
- Line 247: add a space before “we also”
- Line 291: what is Equation (66)? It is bad for the main text to point to something in the appendix in a theorem statement.

**Ethical Concerns:**

["NO or VERY MINOR ethics concerns only"]

**Final Justification:**

I think the paper makes meaningful progress and is technically strong; however, there is considerable room for improvement in the presentation, so I cannot recommend a firm “accept".

**Limitations:**

Yes

**Paper Formatting Concerns:**

NIL

**Quality:**

3

**Strengths And Weaknesses:**

Strengths: These are interesting results to find a surrogate to optimal lambda in the resolvent.

Weaknesses: It does not seem that authors have placed a lot of effort to optimize the bounds; the bounds are not trivial but they seem largely based on the recent progress in random matrix theory. (The dependence on some parameters look bad; see below)

---

> ### Author Rebuttal · Authors · 2025-07-31
>
> We thank the reviewer for their valuable feedback and insightful questions. We appreciate the recognition of the novelty of our results and the opportunity to clarify and improve the presentation and technical depth of our paper. We respond to each point below.
>
> **_It does not seem that authors have placed a lot of effort to optimize the bounds_**
>
> Regarding the optimality of the bounds, we would like to emphasize that considerable attention was paid to ensuring that our bounds remain fully explicit and tight
> —particularly with respect to the exponents involving key parameters such as $\lambda$, $\eta$, $\|\Sigma_X\|_{\mathrm{op}}$, and $\uplambda_1(\Sigma_X)$.
> While minor refinements may be possible in regards of the complexity and length of the proof,
> we believe that our bounds already reflect the inherent difficulty of the problem and do not leave any straightforward improvements unaddressed.
>
> Since we introduced the quantities $C_1$ and $C_2$ in the statements of theorems 1 and 2, we only highlight the worst-case contributions of each parameter involved.
> In that regard, we recognize that they might appear as suboptimal bounds. This point may not have been sufficiently clear in the original submission, and we will include a discussion clarifying this in the revised version of the paper.
>
> **_Proposition 1: What is B? Why is the right-hand side of the inequality independent of B? If I rescale B up by a large factor, the tail probability should be much smaller (depending on the factor)?_**
>
> In our setting, $B$ is a deterministic matrix of appropriate dimensions. As the reviewer correctly notes, rescaling $B$ should influence the tail probability. The corrected inequality reads:
>
> $$
> \mathbb{P}\left( \left| d^{-1} \operatorname{tr}\left( B \left\\{ R_X(\lambda) \mathbf\{1\}\_\{A\_\eta\}(X) - \mathbb{E}\left[R_X(\lambda) \mathbf\{1\}\_\{A\_\eta\}(X)\right] \right\\} \right) \right|  \ge t \right) \lesssim \exp\left(-c \\|B\\|_{\mathrm{op}}^2 (\eta + \lambda)^3 \sigma_X^2 n d t^2 \right)
> $$
>
> We thank the reviewer for bringing this typographical error to our attention and will correct the statement in the revised manuscript.
>
> **_Can authors comment on the dependence on eta in their results? $\Delta_X(\lambda)$ has a dependence of $1/\eta^6$ and the tail has a dependence of $\eta^3$, which do not look good for small eta._**
>
> Regarding the dependence on $\eta$, we acknowledge that terms such as $1/\eta^6$ and $\eta^3$ may appear unfavorable for small $\eta$.
> However, this behavior is consistent with the role $\eta$ plays in our framework:
> it serves as a threshold parameter that can be interpreted as a form of forced regularization in the unfavorable event where $C_X$ is not well conditioned.
>
> Provided the covariance matrix $\Sigma_X$ is well-conditioned (i.e., $\uplambda_1(\Sigma_X)$ is bounded below independantly of $n$ and $d$),
> $\eta$ can be chosen proportionally to $\uplambda_1(\Sigma_X)$ and thus stays safely bounded away from zero.
> This scenario is addressed in Appendix A.1.
> Moreover, our bounds align with known results in random matrix theory, such as those in "Quantitative Deterministic Equivalents of Sample Covariance Matrices with a General Dependence Structure" by C. Chouard,
> which also feature cubic dependence on regularization parameters in tail bounds, as well as comparable asymptotic guarentees for the bias bound.
>
> **_Can eta depend on n or d? For instance, can eta be at the order of $1/\sqrt{n}$?_**
>
> The answer is yes.
> Most parameters in our bounds depend (explicitly or implicitly) on the underlying distribution of $X$, which may itself depend on $n$ or $d$.
> In standard settings, assuming $\uplambda_1(\Sigma_X) \ge \epsilon > 0$, $\eta$ can remain independent of $n$, or bounded w.r.t. $n$.
> If instead $\uplambda_1(\Sigma_X)$ scales as $1/\sqrt{n}$, then $\eta$ would also scale accordingly.
> In this setting our bounds would necessarily degrade, and the control over estimation error would deteriorate, and our method would not be applicable for hyperparameter tuning.
> However, a user can easily avoid this scenario by either removing multicolinearity in his features, or by applying a PCA step before using our method (say by projecting the data on a subspace of dimension $\lfloor 0.9 \times d \rfloor$,
> thus removing the directions of very low variance and ensuring that $\uplambda_1(\Sigma_X)$ stays bounded away from $0$).
>
> **_Line 79: isn’t E a measurable subset of $E^{d \times k}$?_**
>
> It is a measurable subset of $\mathbb{R}^{d \times k}$.
>
> **_Line 228: what is A?_**
>
> A measurable subset of $\mathbb{R}^{d \times k}$. In fact, it should be replaced by 'E' to homogenize our notations throughout the paper, as 'A' is used only once to refer to a subset of $\mathbb{R}^{d \times k}$, this will be changed in the revised version of the paper and we thank the reviewer for pointing this out.
>
> **_Line 291: what is Equation (66)? It is bad for the main text to point to something in the appendix in a theorem statement._**
>
> Equation (66) is located in the appendix and provides the detailed expressions of $C_1$ and $C_2$.
> While referencing an appendix from a theorem is not ideal, it seems to us that in machine learning conferences this is relatively commonly accepted due to page limit. In addition, we tried to avoid writing the full upper bounds on $\Delta_X(\lambda)$ and $\Delta_{Aug}(\lambda)$ as they are quite long, and would have made the statement unnecessarily complex.
> However, if space allow and the reviewer thinks that it is preferable, we will add their expression in the final version of our paper.
>
>
> Once again, we thank the reviewer for their helpful comments and careful reading of our manuscript. We believe that these clarifications will improve the presentation and highlight the rigor of our results. We hope the reviewer will find these revisions satisfactory and would kindly consider updating their evaluation to reflect these improvements.

---

> > ### Comment · Reviewer_awgT · 2025-08-02
> >
> > Thanks for the clarifications. Regarding the presentation, it might be a good idea to consider a special case for the main body and leave the general form to the appendix.

---

### Official Review · Reviewer_HgXn · 2025-06-29

**Clarity:** 1
**Significance:** 3
**Originality:** 2
**Rating:** 5
**Confidence:** 4

**Summary:**

The objective of this paper is to derive non-asymptotic guarantees for estimating the precision matrix (i.e., the inverse of the covariance matrix). The quality of the estimation is assessed using the Frobenius norm between the estimator and the true precision matrix.

In the first part of the paper, the precision matrix is estimated using a ridge-like estimator of the form $(C_X + \lambda I)^{-1}$, and the associated risk, which depends on the regularization parameter $\lambda$, is estimated. It is shown that this risk estimator increasingly approximates the true risk. Consequently, the parameter $\lambda$ can be optimized—provided the estimated risk has a computable gradient—in order to improve the precision matrix estimation.

The second part of the paper investigates the effect of augmenting the data matrix $X$ with additional samples (e.g., Gaussian isotropic noise or dropout-based augmentations), and then applying the same estimation and optimization procedure to this enriched dataset.

**Questions:**

You can responds on the weakness(soudness) bullet points to improve your rating.

**Ethical Concerns:**

["NO or VERY MINOR ethics concerns only"]

**Final Justification:**

The author has provided clear and convincing responses to my questions and has made significant improvements to the manuscript (such as simplifying assumptions and adding a paragraph on the case $\gamma > 1$). Given the author’s thorough approach and the strong theoretical contribution of the paper, I am pleased to revise my recommendation to acceptance.

**Limitations:**

yes

**Quality:**

2

**Strengths And Weaknesses:**

**Strengths**

* The paper provides a solid theoretical foundation, offering non-asymptotic results for the inverse of a random matrix—a notably challenging problem.
* The use of deterministic equivalents and the companion Stieltjes transform reflects a modern approach in non-asymptotic random matrix theory.
* Table 1 presents a broad set of data augmentation (DA) examples, and the stated assumptions are satisfied.
* The proposed method includes a practical component, enabling gradient-based optimization of the estimated risk, which is fully computable.

**Weaknesses**

* The choice of the Frobenius norm as the error metric is not sufficiently justified. For instance, in some applications it might be more relevant to focus on estimation accuracy along the leading eigen-directions of $\Sigma$.
* The assumption $\gamma = d/n < 1$ (required for Assumption H2) appears inconsistent with the introduction’s emphasis on the high-dimensional regime and the double descent phenomenon.
* The assumptions in Section 2 are overly complex. Assumption H1, in particular, includes conditions both on the entire matrix $X$ and on its columns. These could be simplified by focusing only on column-level assumptions (e.g., i.i.d. Gaussian or bounded entries). Definition 1 seems unnecessary for the main narrative and could be moved to the appendix.
* The term $\Delta_\lambda$, intended to represent a bound on the bias, is not clearly defined. Its formulation includes the same quantity in both the constant and the expression, making it difficult for the reader to identify the key parameters. Moreover, the homogeneity of the expression is not respected.
* Section 3 is promising, but somewhat disappointing. While it hints at the potential to replace the regularization term $+\lambda I$ with a data augmentation strategy that serves a similar role, this idea is not fully proved in the upper bound and the number of augmented samples is not optimized. Although the paper alludes to the implicit regularization of DA, it does not concretely leverage this concept.

**Writing advices**

* The manuscript may include an excess of theoretical detail (e.g., Proposition 1 and the subsequent discussion) without sufficient commentary or illustrative examples to aid understanding.
* The choice $\lambda_1 < \cdots < \lambda_d$ is unconventional (and somewhat inexacte due to multiplicity).

**Typos (no response needed)**

* $\mathrm{tr}$ instead of $tr$ in Equation (3).
*  $:=$ is not consistently used for definitions.
* $\bar\Lambda$ not defined.

**Summary**

In conclusion, while the paper is built on a solid mathematical foundation and uses modern tools, its overall narrative lacks clarity, which makes it difficult to follow. Nonetheless, the approach and results are compelling. I encourage the authors to clarify their exposition and would be open to revising my evaluation.

After rebuttal period: borderline reject -> accept

---

> ### Author Rebuttal · Authors · 2025-07-31
>
> We sincerely thank the reviewer for the thoughtful and constructive feedback. We are especially grateful for the recognition of the theoretical foundations of our work. Below, we provide detailed responses to the concerns and suggestions raised.
>
> **_The choice of the Frobenius norm as the error metric is not sufficiently justified. For instance, in some applications it might be more relevant to focus on estimation accuracy along the leading eigen-directions of $\Sigma$._**
>
> The reviewer raises a valid point. First, we would like to emphasize that our results, being stated in the Frobenius norm, do imply bounds on quantities of the form
> $|u^\top (\Sigma_X^{-1} - (C_{\text{Aug}} + \lambda I)^{-1}) u|$
> for any fixed vector $u$. In particular, this includes cases where $u$ is an eigenvector of $\Sigma_X$ associated with a large eigenvalue. However, we agree that such bounds would be looser than those obtained by directly analyzing the quantity of interest without relying on norm-based inequalities.
>
> Although this application is indeed feasible, it falls outside the scope of our paper, which focuses on estimation errors that are direction-agnostic. This choice is motivated by the fact that Frobenius-norm-based errors are widely used in applications involving global precision matrix estimation, such as regularization, control of model complexity, risk management (see [1]).
>
> We believe that including an in-depth analysis of directional errors would significantly complicate an already technically dense paper. Therefore, we prefer to leave this direction for future work or a separate study dedicated to directional error analysis.
>
> **_The assumption $\gamma = d/n < 1$ (required for Assumption H2) appears inconsistent with the introduction’s emphasis on the high-dimensional regime and the double descent phenomenon._**
>
> It is a common misconception to think that the condition $\gamma < 1$ is inconsistent with our emphasis on high-dimensional settings. In fact, many practical datasets satisfy $\dim > 100$ while still having $0 \ll \gamma < 1$. Even under this regime, estimating $\Sigma_X^{-1}$ remains a difficult problem—this is well documented in the literature on the Marchenko–Pastur theorem   (see e.g. [2]).
>
> Moreover, our method can be adapted to the case $\gamma > 1$ modifying  the considered estimator. As discussed in the proof sketch of Theorem 1, the assumption $\gamma < 1$ is specifically used to approximate $d^{-1} \operatorname{tr}(\Sigma_X^{-1})$ by $(1 - \gamma)\, d^{-1} \operatorname{tr}(C_X^{-1})$. For $\gamma > 1$, a similar approximation holds via the dual formulation:
> $$
> d^{-1} \operatorname{tr}(\Sigma_X^{-1}) \approx \left(1 - \frac{1}{\gamma}\right) n^{-1} \operatorname{tr}\left((d^{-1} X^\top X)^{-1}\right),
> $$
> i.e., using the Gram matrix rather than the sample covariance matrix. This duality between the $\gamma < 1$ and $\gamma > 1$ regimes is well-established in random matrix theory, as it is known that the eigenvalue distribution of the covariance and the Gram matrix differ only by a Dirac mass in $0$ (see, e.g., [3])
> The case $\gamma = 1$, however, remains more delicate and is not directly addressed by current tools.
>
> While it would certainly be interesting to include the full analysis corresponding to the $\gamma > 1$ regime, doing so would significantly increase the length and complexity of an already notation-heavy paper. Since clarity and focus are among our priorities, and as this was also a concern raised by the reviewers, we suggest to only add a single paragraph describing how our results generalize in the case $\gamma > 1$.
>
> **_The assumptions in Section 2 are overly complex. Assumption H1, in particular, includes conditions both on the entire matrix $X$ and on its columns. These could be simplified by focusing only on column-level assumptions (e.g., i.i.d. Gaussian or bounded entries). Definition 1 seems unnecessary for the main narrative and could be moved to the appendix._**
>
> Based on your feedback and suggestions, we propose the following modifications to improve the clarity of our paper:
>
> * In the main text, we will restrict our presentation to the case of Gaussian (or sub-Gaussian) data and replace Assumption H2 with the simpler requirement that the eigenvalues of $\Sigma_X$ are bounded away from zero. As noted in the submitted version, this assumption is sufficient to apply our results. This simplification will allow us to eliminate some of the more technical notation—such as Definition 1 and Assumption H1—from the main body of the paper.
>
> * The more general framework, along with the full set of assumptions and technical details, will be moved to the appendix. We will also make explicit how the simplified setting presented in the main text is a special case of our broader results.
>
> If the reviewer agrees that this modification would enhance the clarity and readability of the manuscript, we will implement it in the final version of our paper.
>
> **_The term $\Delta_X(\lambda)$, intended to represent a bound on the bias, is not clearly defined. Its formulation includes the same quantity in both the constant and the expression, making it difficult for the reader to identify the key parameters. Moreover, the homogeneity of the expression is not respected._**
>
> First, note that all our bounds are explicit, and a fully detailed expression for $\Delta_X(\lambda)$ is provided in App. C. We would also like to clarify our motivation for the form given in the main text, and in particular, explain why we introduced the constants $C_1$ and $C_2$:
> they are designed to simplify our expressions by encapsulating repetitive and technical terms such as $\|\Sigma_X\|_{\mathrm{op}}^{-1}$, $c_X^{-1}$, and $\eta + \lambda$.
> In all relevant random matrix models, these quantities are upper bounded.
> Moreover, the constants $C_1$ and $C_2$ are non-increasing functions of these quantities, making our upper bounds conservative in a meaningful way.
> In summary, Theorems 1 and 2 provide worst-case bounds where $C_1$ and $C_2$ remain bounded for all relevant models. We recognize that this point may not have been sufficiently clear in the original submission, and we will include a discussion clarifying this in the revised version of the paper.
>
> **_[...] While it hints at the potential to replace the regularization term $+ \lambda I$ with a data augmentation strategy that serves a similar role, this idea is not fully proved [...]_**
>
> **_[...] this idea is not fully proved in the upper bound and the number of augmented samples is not optimized. [...]_**
>
> We would like to highlight that implicit regularization is already a well-known impact of data augmentation (see e.g. [4] or [5].).
> The aim of our paper was only to revisit this effect from a quantitative perspective, allowing the users to 1. compare two data augmentation schemes and 2. tune the data augmentation hyperparameters, which can be done by minimizing $\hat{\mathcal{E}}_{Aug}(\lambda)$ with respect to the scheme's hyperparameters.
>
> In this regard, Our Figures 1.a, 1.b, 1.c show that $\|\Sigma_X^{-1} - (C_{\text{Aug}} + \lambda I)^{-1}\|_{\mathrm{Frob}}$ can be reliably estimated using our results.
> Optimizing the proportion of artificial data is a natural application that we explicitly mention in the paper.
> From a practical perspective, this can be done using a simple binary search or a gradient descent procedure.
>
> **_The choice $\uplambda_1(\Sigma) < \ldots < \uplambda_d(\Sigma)$ is unconventional_**
>
> As noted in the notation section on page 2, we follow the convention $\uplambda_1(\Sigma_X) \leq \ldots \leq \uplambda_d(\Sigma_X)$; therefore, the two quantities mentioned are indeed distinct. While the reverse ordering is perhaps more common in the machine learning literature, our choice is standard in parts of the probability literature and is not a typographical error, see for instance [6] or [7].
> That being said, if the reviewer finds this convention to be detrimental to the clarity of the paper, we are of course open to changing it to align with the standards of the machine learning community.
>
> Once again, we thank the reviewer for their time and valuable insights. We hope that the clarifications and proposed revisions demonstrate our commitment to improving both the technical quality and the accessibility of the paper. If the reviewer finds our responses satisfactory, we would be grateful if they would consider increasing their rating of the submission accordingly.
>
> **References**
>
> [1] "An Overview on the Estimation of Large Covariance and Precision Matrices" by J. Fan et al.
> [2] "Spectral analysis of large dimensional random matrices" by Z. Bai & J. Silverstein
> [3] "Anisotropic local laws for random matrices" by A. Knowles et al.
> [4] "Training with noise is equivalent to Tikhonov regularization" by C.M. Bishop
> [5] "The good, the bad and the ugly sides of data augmentation: An implicit spectral regularization perspective" by C. Lin et al.
> [6] "Large Complex Correlated Wishart Matrices: Fluctuations and Asymptotic Independence at the Edges" by W. Hachem et al.
> [7] "On the Distribution of Eigenvalues of GUE and its Minors at Fixed Index" by T. Tao.

---

> > ### Comment · Reviewer_HgXn · 2025-08-06
> > **response to rebuttal**
> >
> > The author has provided clear and convincing responses to my questions and has made significant improvements to the manuscript (such as simplifying assumptions and adding a paragraph on the case \$\gamma > 1\$). Given the author’s thorough approach and the strong theoretical contribution of the paper, I am pleased to revise my recommendation to acceptance.

---

### Official Review · Reviewer_y2zf · 2025-07-02

**Clarity:** 2
**Significance:** 2
**Originality:** 3
**Rating:** 5
**Confidence:** 4

**Summary:**

The paper analyses common methods for estimating the inverse covariance of a matrix in high-dimensional settings. In particular it develops data-dependent estimates for the squared error of linear shrinkage and data augmentation estimators, and provides guarantees for these data-dependent estimates. It uses these empirical estimates to help tune certain hyper parameters used in covariance matrix estimation.

**Questions:**

1. Can you discuss the weaker convergences guarantees guarantees you promise when one relaxes H4?
2. Can you perform an asymptotic analysis of your guarantees under some simplifying assumptions as a sanity check for your readers?
3. Could you discuss the growth of your error in estimating the squared-error with respect to d and n?

**Ethical Concerns:**

["NO or VERY MINOR ethics concerns only"]

**Final Justification:**

The rebuttal has addressed my qualms but my new score is heavily conditional on a significant improvement in the paper’s presentation and writing.

**Limitations:**

Many have been discussed by the authors, I have discussed some other limitations that inform my decision to lean towards rejecting the paper above.

**Quality:**

3

**Strengths And Weaknesses:**

Strengths:
1. The paper addresses the important problem of estimating inverse covariance matrices from a theoretical perspective.
2. The paper focuses on the important subproblem of estimating the squared error of both kinds of inverse covariance estimators
3. The paper establishes theoretical guarantees for these squared error estimators
4. The paper’s brief empirical experiments help strengthen their claim about the reliability of their empirical estimators

Weaknesses:
1. Part ii of H3 severely limits the applicability of their result, and the work does not discuss how their results would change if parts of the statement were relaxed (such as the conditioning of Lambda_G(X))
2. The Wasserstein distance part of H4 seems quite strong, it would be nice if the paper elaborated on the “weaker convergence guarantees” they mention below it.
3. Most important: This is my biggest qualm. The theoretical results are not well thought out. Their expression uses both the operator norm of Sigma_X and lambda_1(Sigma_X), which are literally the same thing unless I am gravely mistaken. Further, the dependence of the error in squared-error estimation on d and n is not clear since the operator norm of a matrix can grow with d and n. This is further exacerbated by the missing expressions for C_1 and C_2 and no discussion about their growth with d and n. I am leaning reject primarily because the theory is not properly addressed, which is supposed to be the main strength of the paper in my eyes.
4. While the paper is about non-asymptotic guarantees, it is important to have a sanity check where they recover some familiar or believable asymptotic guarantees under perhaps additional distributional assumptions on X.
5. While experiments were perhaps not the main focus of this paper when it was being written, I think a paper addressing an estimation problem and submitted to an ML conference with claims about its utility in choosing hyperparameters should have more extensive experiments across multiple datasets. Unlike some theory papers, this paper’s theory is less about conceptual insight and more about estimating bounds. I think such papers need to have extensive experiments.

---

> ### Author Rebuttal · Authors · 2025-07-31
>
> We sincerely thank the reviewer for taking the time to engage critically and constructively with our submission. We appreciate the recognition of the strengths of the paper, particularly about the quality of the theoretical contributions, and we value the concerns raised. We address each of the reviewer’s points in detail below.
>
> **_Part ii of H3 severely limits the applicability of their result, and the work does not discuss how their results would change if parts of the statement were relaxed (such as the conditioning of $\Lambda_G(X)$)._**
>
> We respectfully disagree with the reviewer’s comment. The decomposition of $\mathbb{E}[C_X \mid X]$ assumed in H2-(ii) is always satisfied (at least for $\beta = 0$), and the conditions on the conditioning, as well as smoothness of $\Lambda_G(X)$ holds for a wide range of common data augmentations, as illustrated in Table 1 and discussed in detail in Appendix A.2. In fact, our formulation of H2-(ii) was specifically motivated by these data augmentation techniques, and we expect that any reasonable augmentation scheme amenable to theoretical analysis will satisfy this assumption. That said, we remain open to revisiting this point should the reviewer be aware of a common data augmentation method that does not satisfy point (ii) of H3.
>
>  **_The Wasserstein distance part of H4 seems quite strong; it would be nice if the paper elaborated on the “weaker convergence guarantees” they mention below it._**
> **_Can you discuss the weaker convergence guarantees you promise when one relaxes H4?_**
>
> We acknowledge that Assumption H4 is relatively restrictive. However, it is satisfied under additional assumptions on the distribution of $X$ (e.g., boundedness of the data), as discussed in detail in Appendix A.2, to which we refer the reviewer for further clarification.
>
> While it is indeed possible to relax this assumption by adapting our proof strategies, e.g., by requiring only local Lipschitz continuity along with stronger assumptions on the data distribution, this would introduce considerable technical complexity and result in looser bounds. We chose not to pursue this direction for two main reasons. First, incorporating such generalizations would significantly increase the length of the paper, which already includes 56 pages of supplementary material. Second, it would further complicate the presentation of our results, which some reviewers already found difficult to digest. In fact, one reviewer even suggested that we simplify the assumptions in the main text and relegate the general framework to the appendix.
>
> For these reasons, we fear that including such an extension would be lost in the supplementary material and would not substantially benefit the clarity or accessibility of the paper. We therefore leave this extension for future work.
>
>  **_[...] uses both the operator norm of $\Sigma_X$ and $\uplambda_1(\Sigma_X)$, which are literally the same thing [...]_**
>
> As noted in the notation section on page 2, we follow the convention $\uplambda_1(\Sigma_X) \leq \ldots \leq \uplambda_d(\Sigma_X)$; therefore, the two quantities mentioned are indeed distinct. While the reverse ordering is perhaps more common in the machine learning literature, our choice is standard in parts of the probability literature and is not a typographical error,
> see for instance [1] or [2].
> That being said, if the reviewer finds this convention to be detrimental to the clarity of the paper, we are of course open to changing it to align with the standards of the machine learning community.
>
> **_[...] the dependence of the error in squared-error estimation on d and n is not clear since the operator norm of a matrix can grow with d and n. This is further exacerbated by the missing expressions for $C_1$ and $C_2$ and no discussion about their growth with d and n._**
>
> While it is true that all quantities involved in these terms depend on $d$ and $n$, it is common in asymptotic random matrix theory to assume that they are bounded. If one is interested in the detailed expressions of these terms, they can be found in Appendix C, page 58, for $C_1$ and $C_2$, and Appendix D, page 74, for $\tilde{C}_1$ and $\tilde{C}_2$.
> Otherwise, assuming the previous greatly simplifies the asymptotic analysis of our bounds and results in $\Delta_X(\lambda) = O(\sqrt{d}/n + 1/(\lambda^3 nd))$.
>
> More precisely, the constants $C_1$ and $C_2$ are designed to simplify our expressions by encapsulating repetitive and technical terms such as $\|\Sigma_X\|_{\mathrm{op}}^{-1}$, $c_X^{-1}$, and $\eta + \lambda$. As previously discussed it is standard to treat these latter quantities as upper bounded. Moreover, the constants $C_1$ and $C_2$ are non-increasing functions of these quantities, making our upper bounds conservative in a meaningful way.
> In summary, Theorems 1 and 2 provide worst-case bounds where $C_1$ and $C_2$ remain bounded for all relevant models. We recognize that this point may not have been sufficiently clear in the original submission, and we will include a discussion clarifying this in the revised version of the paper.
>
> **_While the paper is about non-asymptotic guarantees, it is important to have a sanity check where they recover some familiar or believable asymptotic guarantees under perhaps additional distributional assumptions on $X$._**
>
> To our knowledge, no existing results directly compare to Theorems 1 and 2. However, we did compare the non-asymptotic guarantees derived for our deterministic equivalents, particularly Proposition 1, with results from the asymptotic literature. In particular, our bounds are consistent with those established in [3].
>
> **_[...] I think a paper addressing an estimation problem and submitted to an ML conference with claims about its utility in choosing hyperparameters should have more extensive experiments across multiple datasets. [...]_**
>
> In response to your feedback, we have also conducted additional experiments on CIFAR-10 and CIFAR-100. These results exhibit similar behavior and, in particular, further validate our theoretical findings. For reference, we include the sample tables below, as linking to external figures is not allowed this year. Tables 1 and 2 further validate our methodology for hyperparameter selection. Table 1 gives $(\hat{\lambda} - \lambda^\*)/ \lambda^\*$, where $\lambda^\* = \arg\min \mathcal{E}_X(\lambda)$ and $\hat{\lambda} = \arg\min \hat{\mathcal{E}}_X(\lambda)$, for various values of $\gamma = d/n$, on CIFAR-10 and CIFAR-100 (both in grayscale):
>
> | $\gamma = d/n$ | CIFAR-10 | CIFAR-100 |
> |----------------|----------|------------|
> | 0.15 |  7.13 % |  0.82 % |
> | 0.30 |  0.16 % |  0.13 % |
> | 0.45 |  0.10 % |  0.09 % |
> | 0.60 |  0.08 % |  0.07 % |
> | 0.75 |  0.07 % |  0.06 % |
> | 0.90 |  0.06 % |  0.06 % |
>
> Similarly, Table 2 gives $(\hat{\alpha} - \alpha^\*)/ \alpha^\*$, in the case of the fixed Gaussian TDA with $\Lambda = \sigma I$ (presented in table 1 of the paper), where $\hat{\alpha}$ has been estimated by minimizing $\hat{\mathcal{E}}\_{Aug}(0)$, for various values of $\sigma$, and for fixed $\gamma = 0.85$.
>
> | $\sigma$ | CIFAR-10 | CIFAR-100 |
> |----------|----------|-------------|
> | 1.0      | 0.64 %   | -0.06 %     |
> | 0.9      | 0.59 %   | -3.06 %     |
> | 0.8      | 0.65 %   | 0.85 %      |
> | 0.7      | 1.24 %   | -1.26 %     |
> | 0.6      | -0.78 %  | -1.72 %     |
> | 0.5      | -0.25 %  | 0.71 %      |
>
> In addition, we would like to clarify the reasoning behind our dataset choices and highlight certain limitations. Our primary objective was to evaluate the accuracy of our estimators $\hat{\mathcal{E}}\_X(\lambda)$ and $\hat{\mathcal{E}}\_{\text{Aug}}(\lambda)$. To do this meaningfully, we either needed access to the true covariance matrix of the data distribution (which is not possible) or a highly accurate empirical approximation. This required a high-dimensional dataset (dimension $>$ 100) where the ratio of dimension to sample size was sufficiently small (less than 1/100), ensuring the sample covariance matrix is a reliable proxy for the true $\Sigma_X$. This requirement is motivated by the Marchenko-Pastur. To the best of our knowledge, very few publicly available datasets satisfy these conditions, and we were unable to identify higher-dimensional alternatives to MNIST or CIFAR that fit these stringent criteria.
>
> That being said, we believe that our expanded numerical experiments offer a compelling proof of concept for the estimators presented in Theorems 1 and 2. This level of empirical support aligns with what is typically expected in theoretical contributions such as ours, a point the reviewer also acknowledged as a strength of the paper. To the best of our knowledge, we are the first to provide computable non-asymptotic bounds for precision matrix estimation using data augmentation, while also shedding light on the key limitations of this widely used technique. We view this work as an important step toward better understanding and tuning of data augmentation in modern machine learning. Moreover, the technical difficulty of the proof methods developed in the paper is, in our view, nontrivial. This is  reflected in the length and complexity of the supplemental material, which spans 56 pages of detailed appendices.
>
> Once again, we are grateful to the reviewer for their insightful comments. If the reviewer finds our clarifications and the new experimental results satisfactory, we would respectfully ask them to consider improving their score, as we believe the revisions and explanations provided significantly strengthen the clarity, completeness, and overall contribution of the paper.
>
> **References**
>
> [1] "Large Complex Correlated Wishart Matrices: Fluctuations and Asymptotic Independence at the Edges" by W. Hachem et al.
> [2] "On the Distribution of Eigenvalues of GUE and its Minors at Fixed Index" by T. Tao.
> [3] "Quantitative Deterministic Equivalent of Sample Covariance Matrices with a General Dependence Structure" by C. Chouard.
> \end{itemize}

---

> > ### Comment · Area_Chair_2XA8 · 2025-08-07
> >
> > Dear Reviewer y2zf,
> >
> > This is a gentle reminder that the extended Author–Reviewer Discussion period will conclude on **August 8 (AoE)**.
> >
> > At your earliest convenience, please read the authors' rebuttal, and actively participate in the discussion. Regardless of whether your original concerns have been fully addressed, we kindly ask that you:
> >
> > - Clearly state if your concerns have been resolved, or
> > - Specify what aspects remain unaddressed.
> >
> > Even if you believe no response is needed, please communicate this with the authors. **Staying silent is not acceptable.**
> >
> > Please note that the Mandatory Acknowledgement should only be submitted **after**:
> > 1. Reading the authors' rebuttal,
> > 2. Engaging in the discussion with the authors (and optionally other reviewers or the AC), and
> > 3. Completing your **Final Justification** and updating your score accordingly.
> >
> > Please do **not** click the "Mandatory Acknowledgement" button until all of the above are completed.
> > Reviewers who submit the acknowledgement without meaningful discussion will be flagged as non-participating under the Responsible Reviewing policy.
> > If I have a **strong** feeling that a fellow reviewer has not properly participated, I may flag the review as **insufficient**.
> > The program chairs will use these flags to determine whether a reviewer's own submissions may be desk rejected.
> >
> > Thank you for your thoughtful and timely contributions to the NeurIPS 2025 review process.
> >
> > Best regards,
> > Your AC,

---

> ### Comment · Reviewer_y2zf · 2025-08-09
> **Response to rebuttal**
>
> My apologies for the delay. Due to the delay, I regret to be depriving the authors the opportunity to respond to my comment, but hopefully this is not a huge inconvenience since I am raising my score to a 5.
>
> I agree that weaknesses 1 and 2 are not that important, and I am mostly convinced by the authors’ response.
>
> I would want experiments beyond CIFAR to make this paper’s experimental claims compelling. I don’t think extensive and technically challenging computations alone rise to the level of a NeurIPS paper without providing either some important conceptual insight or some compelling experiments. However, this only has a minor bearing on my score for the paper since I know that this opinion is not universal.
>
> I see now that lambda_1 is in fact the smallest eigenvalue. I don’t see why you have used this convention - it seems needlessly confusing. This has no significant bearing on my score, but I recommend that you use standard notation for readability and clarity.
>
> Most important: I think the key reason that the operator norm doesn’t need to grow with n is that the dimension d also grows with n in the high dimensional regime that the paper works in. In particular, the paper’s distributional assumptions along with the high dimensional regime allow the operator norm to be bounded. I think this is quite important to emphasize for readers who are not experts in ML theory for the high dimensional regime. This fact is clear after some quick thought, but it is very different from what one sees in classical regimes.
>
> I am now convinced by your rebuttal enough to raise my score to a 5, but I strongly implore you to consider refactoring your presentation to make these sorts of points clear. I think your paper has a lot of scope for improvement in writing. Please consider investing time in this paper before moving on to other projects - I suggest sitting with a theoretically inclined ML researcher not in your field and asking them for detailed feedback on the writing before submitting a camera ready version.

---

> > ### Author Response · Authors · 2025-08-09
> >
> > Thank you for engaging in the discussion and for providing feedback that will help improve our submission. We will incorporate your observations into the final version of the manuscript. In addition, we are committed to restructuring our presentation in light of your comments and those of the other reviewers.
> >
> > We sincerely appreciate the time you have taken to evaluate our paper, and we are pleased that you were able to give it a positive assessment.

---

### Official Review · Reviewer_1hLH · 2025-07-09

**Clarity:** 3
**Significance:** 3
**Originality:** 3
**Rating:** 4
**Confidence:** 3

**Summary:**

This paper presents a theoretical and empirical study of inverse covariance (precision matrix) estimation in high-dimensional settings. It compares two classes of estimators: (1) standard linear shrinkage estimators and (2) estimators based on data augmentation (DA), including both transformative and generative DA schemes. The authors develop non-asymptotic concentration bounds for the quadratic error of these estimators, leveraging advanced tools from random matrix theory, including novel deterministic equivalents. They also provide practical estimation formulas and validate them using experiments on synthetic and real datasets (e.g., MNIST).

**Questions:**

Any results on more complex datasets, besides the simple MNIST?

**Ethical Concerns:**

["NO or VERY MINOR ethics concerns only"]

**Final Justification:**

The authors' rebuttal responses clarified some questions. I updated my rating accordingly.

**Limitations:**

No, the authors showed that the proposed method works on high-dimension settings, but the experiments were conducted on a very simple dataset (MNIST).

**Paper Formatting Concerns:**

The format looks fine.

**Quality:**

2

**Strengths And Weaknesses:**

Strengths：
1.	The paper makes a substantial contribution by offering a non-asymptotic framework for analyzing precision matrix estimation with DA, an area where prior work is sparse.

2.	In this paper, the author gives a comprehensive and detailed prove to support the proposed claim.

3.	Numerical results support the claims, demonstrating the tightness of the theoretical bounds.

Weaknesses：

1.	The theorical contribution is substantial, but the definitions, lemma, theorem are too many resulting certain sections are notation-heavy. Is it possible to simplify some statement and make it more reader-friendly.
2.	This paper is more about theory analysis and lack of practical demonstration. The experiment part need to consolidate. MNIST dataset is too simple to verify the theory is  consistent with real world scenario. This paper focus on high dimension data but MNIST data (28*28) is low dimension in reality. And also the experiment part lack comparison with prior key method.
3.	While the theory is well-documented, the experimental section lacks of code and implementation details. Hard to reproduce and understand the connection between theory and practice if lack of code.

---

> ### Author Rebuttal · Authors · 2025-07-31
>
> We thank the reviewer for their thoughtful feedback and for recognizing the importance of the contribution of our work, we address the reviewer's feedbacks and question below:
>
> **_The theoretical contribution is substantial, but the definitions, lemma, theorem are too many resulting certain sections are notation-heavy. Is it possible to simplify some statement and make it more reader-friendly._**
>
> We agree that some parts of the paper may appear notation-heavy. However, we would like to emphasize that this level of formalism is essential to ensure the rigor and generality of our theoretical results. That said, we acknowledge that the complexity may hinder readability for some audiences and may not always be necessary to convey the core narrative of the paper.
>
> If the reviewer believes it would improve accessibility, we are open to implementing the following modifications:
>
> * In the main text, we could restrict our presentation to the case of Gaussian (or sub-Gaussian) data and replace assumption H2 with the simpler condition that the eigenvalues of $\Sigma_X$ are bounded away from zero. As already mentioned in the submitted version, this setting suffices to apply our results (see Appendix A.1 for details). Making this change would allow us to remove some of the heavier notational machinery—such as Definition 1 and H1—from the main document.
> * We could then postpone the more general framework to the appendix and explicitly show how the simplified setting discussed in the main text is a special case of the broader results.
>
> Unfortunately, the statements of Theorems 1 and 2 cannot be easily simplified without weakening our bounds on the estimation error. In this regard, we would like to clarify our motivation for introducing the quantities $C_1$ and $C_2$: they are designed to simplify our expressions by encapsulating repetitive and technical terms such as $\|\Sigma_X\|_{\mathrm{op}}^{-1}$, $c_X^{-1}$, and $\eta + \lambda$.
> In all relevant random matrix models, these quantities are upper bounded.
> Moreover, the constants $C_1$ and $C_2$ are non-increasing functions of these quantities, making our upper bounds conservative in a meaningful way.
> In summary, Theorems 1 and 2 provide worst-case bounds where $C_1$ and $C_2$ remain bounded for all relevant models. We recognize that this point may not have been sufficiently clear in the original submission, and we will include a discussion clarifying this in the revised version of the paper.
>
> **_This paper is more about theory analysis and lack of practical demonstration.
> The experiment part need to consolidate. MNIST dataset is too simple to verify the theory is consistent with real world scenario.
> This paper focus on high dimension data but MNIST data (28*28) is low dimension in reality. And also the experiment part lack comparison with prior key method._**
>
> **_Any results on more complex datasets, besides the simple MNIST?_**
>
> We acknowledge that the MNIST dataset is relatively simple, as the reviewer pointed out. However, we would like to emphasize that it is still widely used as a conventional benchmark in many recent studies,
> see for instance [1] or [2].
>
> In response to your feedback, we have also conducted additional experiments on CIFAR-10 and CIFAR-100. These results exhibit similar behavior and, in particular, further validate our theoretical findings in a larger-scale setting. For reference, we include the sample tables below
> , as linking to external figures is not allowed this year.
> Table 1 gives $(\hat{\lambda} - \lambda^\*) / \lambda^\*$, where $\lambda^\* = \arg\min \mathcal{E}_X(\lambda)$, $\hat{\lambda} = \arg\min \hat{\mathcal{E}}_X(\lambda)$, for various $\gamma = d/n$, on CIFAR-10 and CIFAR-100 (both in grayscale):
>
> | $\gamma = d/n$ | CIFAR-10 | CIFAR-100 |
> |----------------|----------|------------|
> | 0.15 |  7.13   % |  0.82  %|
> | 0.30 |  0.16   % |  0.13  %|
> | 0.45 |  0.10   % |  0.09  %|
> | 0.60 |  0.08   % |  0.07  %|
> | 0.75 |  0.07   % |  0.06  %|
> | 0.90 |  0.06   % |  0.06  %|
>
> Similarly, we provide a table for the relative error of estimating the optimal proportion of artificial data, in the case of the fixed Gaussian TDA with $\Lambda = \sigma I$ (presented in table 1 of the paper), we display $(\hat{\alpha} - \alpha^\*)/ \alpha^\*$, where $\hat{\alpha}$ has been estimated by minimizing $\hat{\mathcal{E}}_{Aug}(0)$, for various values of $\sigma$, and for fixed $\gamma = 0.85$.
>
> | $\sigma$ | CIFAR-10 | CIFAR-100 |
> |----------|-----------|------------|
> | 1   | 0.64   % | -0.06  %|
> | 0.9 | 0.59   % | -3.06  %|
> | 0.8 | 0.65   % | 0.85  %|
> | 0.7 | 1.24   % | -1.26  %|
> | 0.6 | -0.78  % | -1.72  %|
> | 0.5 | -0.25  % | 0.71  %|
>
> In addition, we would like to clarify the reasoning behind our dataset choices and highlight certain limitations. Our primary objective was to evaluate the accuracy of our estimators $\hat{\mathcal{E}}\_X(\lambda)$ and $\hat{\mathcal{E}}\_{\text{Aug}}(\lambda)$.
> To do this meaningfully, we either needed access to the true covariance matrix of the data distribution (which is not possible) or a highly accurate empirical approximation. This required a high-dimensional dataset (dimension > 100) where the ratio of dimension to sample size was sufficiently small (less than 1/100), ensuring the sample covariance matrix is a reliable proxy for the true $\Sigma_X$. This requirement is motivated by the Marchenko-Pastur theorem, which explains why such a ratio is crucial. To the best of our knowledge, very few publicly available datasets satisfy these conditions, and we were unable to identify higher-dimensional alternatives to MNIST or CIFAR that fit these stringent criteria.
>
> That being said, we believe that our expanded numerical experiments offer a compelling proof of concept and empirically validate the estimators presented in Theorems 1 and 2. This level of empirical support aligns with what is typically expected in theoretical contributions such as ours, a point the reviewer also acknowledged as a strength of the paper. To the best of our knowledge, we are the first to provide computable non-asymptotic bounds for precision matrix estimation using data augmentation, while also shedding light on the key limitations of this widely used technique. We view this work as an important step toward better understanding and tuning of data augmentation in modern machine learning. Moreover, the technical difficulty of the proof methods developed in the paper is, in our view, nontrivial. This is  reflected in the length and complexity of the supplemental material, which spans 56 pages of detailed appendices.
>
> **_While the theory is well-documented, the experimental section lacks of code and implementation details. Hard to reproduce and understand the connection between theory and practice if lack of code._**
>
> Since our experiments were relatively small-scale, as the reviewer noted, we did not initially deem it necessary to include the code. However, in response to the reviewer’s request, we will make the code available in the revised version of our paper.
>
> We thank the reviewer again for their valuable comments and constructive suggestions. If the reviewer finds our clarifications and proposed revisions satisfactory, we would kindly ask them to consider updating their evaluation to reflect these improvements.
>
> **References**
>
> [1] "Efficient Data-Dependent Random Projection for Least Square Regressions" by J. Sturges et al.
> [2] "Maximizing the Potential of Synthetic Data: Insights From Random Matrix Theory" by A. El Firdoussi et al.

---

> > ### Comment · Area_Chair_2XA8 · 2025-08-07
> >
> > Dear Reviewer 1hLH,
> >
> > This is a gentle reminder that the extended Author–Reviewer Discussion period will conclude on **August 8 (AoE)**.
> >
> > At your earliest convenience, please read the authors' rebuttal, and actively participate in the discussion. Regardless of whether your original concerns have been fully addressed, we kindly ask that you:
> >
> > - Clearly state if your concerns have been resolved, or
> > - Specify what aspects remain unaddressed.
> >
> > Even if you believe no response is needed, please communicate this with the authors. **Staying silent is not acceptable.**
> >
> > Please note that the Mandatory Acknowledgement should only be submitted **after**:
> > 1. Reading the authors' rebuttal,
> > 2. Engaging in the discussion with the authors (and optionally other reviewers or the AC), and
> > 3. Completing your **Final Justification** and updating your score accordingly.
> >
> > Please do **not** click the "Mandatory Acknowledgement" button until all of the above are completed.
> > Reviewers who submit the acknowledgement without meaningful discussion will be flagged as non-participating under the Responsible Reviewing policy.
> > If I have a **strong** feeling that a fellow reviewer has not properly participated, I may flag the review as **insufficient**.
> > The program chairs will use these flags to determine whether a reviewer's own submissions may be desk rejected.
> >
> > Thank you for your thoughtful and timely contributions to the NeurIPS 2025 review process.
> >
> > Best regards,
> > Your AC,

---

### Decision · Program_Chairs · 2025-09-17

**Decision:**

Accept (spotlight)

**Comment:**

This paper presents a theoretical (as well as a bit empirical) study of inverse covariance (precision matrix) estimation in high-dimensional settings.
The authors compare standard linear shrinkage estimators and estimators based on data augmentation.
The major technical contribution is the derivation of non-asymptotic concentration bounds on the quadratic estimation error, achieved using advanced random matrix theory techniques, as well as novel deterministic equivalents.

The paper is well-organized and clearly written. During the rebuttal period, the authors responded thoroughly to the reviewers' concerns, added further experimental results to support their theoretical findings, and clarified several technical aspects of the manuscript.
As a result, all reviewers are now satisfied and recognize the strength and relevance of the contribution.

Overall, this is a strong paper that makes meaningful theoretical and empirical advances in high-dimensional precision matrix estimation.
I thus recommend acceptance.